# On Statistical Rates of Conditional Diffusion Transformers: Approximation, Estimation and Minimax Optimality

**Jerry Yao-Chieh Hu**[*†]    **Weimin Wu**[*†]    **Yi-Chen Lee**[*‡]    **Yu-Chao Huang**[*‡]

**Minshuo Chen**[†]    **Han Liu**[†]

[†]Northwestern University    [‡]National Taiwan University

{jhu,wwm}@u.northwestern.edu, b10202055@ntu.edu.tw
yuchaohuang@g.ntu.edu.tw, {minshuo.chen,hanliu}@northwestern.edu

## Abstract

We investigate the approximation and estimation rates of conditional diffusion transformers (DiTs) with classifier-free guidance. We present a comprehensive analysis for "in-context" conditional DiTs under various common assumptions: generic and strong Hölder, linear latent (subspace), and Lipschitz score function assumptions. Importantly, we establish minimax optimality of DiTs by leveraging score function regularity. Specifically, we discretize the input domains into infinitesimal grids and then perform term-by-term Taylor expansions on the conditional diffusion score function under the Hölder smooth data assumption. This enables fine-grained use of transformers' universal approximation through a more detailed piecewise constant approximation, and hence obtains tighter bounds. Additionally, we extend our analysis to latent settings. Our findings establish statistical limits for DiTs, and offer practical guidance toward more efficient and accurate designs.

## 1 Introduction

We investigate the approximation and estimation rates of conditional Diffusion Transformers (DiTs) with classifier-free guidance. Specifically, we derive score approximation, score estimation, and distribution estimation guarantees for "in-context" conditional DiTs introduced by Peebles and Xie (2023). We provide a comprehensive analysis under various data and score function assumptions, including generic and strong Hölder, linear latent (subspace), and Lipschitz score function assumptions. Moreover, we show that the analysis of both conditional DiTs and their latent variants lead to the first known minimax optimality of unconditional DiTs under the strong Hölder data assumption.

Transformer-based conditional diffusion models are at the forefront of generative AI due to their success as scalable and flexible backbones for image (Zhao et al., 2024; Wu et al., 2024a; Bao et al., 2023; Batzolis et al., 2021) and video generation (Liu et al., 2024; Ni et al., 2023; Saharia et al., 2022; Voleti et al., 2022). However, the theoretical understanding of conditional DiTs remains limited. On the one hand, while prior work (Hu et al., 2024) reports approximation and estimation rates of DiTs using the established universality of transformers (Yun et al., 2020), their results are not tight and are limited to unconditional diffusion. On the other hand, existing theoretical works on conditional diffusion models only focus on ReLU networks (Fu et al., 2024a; Yuan et al., 2024), model-free settings (Ye et al., 2024; Guo et al., 2024) or generative sampling process (Dinh et al., 2023), without considering the transformer architectures. This work addresses this gap by providing a timely analysis of the statistical limits of both conditional and unconditional DiTs.

In this work, we present a comprehensive analysis of conditional DiT and its latent setting under four common data and score function assumptions. We also establish the minimax optimality of unconditional DiT and its latent version by deriving the tight distribution estimation error bounds. Our techniques include two key parts: (i) Discretizing the input domains into infinitesimal grids. (ii) On each grid, performing term-by-term Taylor expansions on the conditional diffusion score function under generic and stronger Hölder smooth data assumptions, motivated by the local diffused

---

[*]Equal contribution. Full version and future updates are available on arXiv.

Table 1: **Summary of Theoretical Results.** The initial data is $d_x$-dimensional, and the condition is $d_y$-dimensional. Furthermore, $d$ is the feature dimensions of transformer network function class (Definition 2.2). For latent DiT, the latent variable is $d_0$-dimensional. $\sigma_t^2 = 1 - e^{-t}$ is the denoising scheduler. The sample size is $n$, and $0 < \epsilon < 1$ represents the precision parameter. While we report asymptotics for large $d_x, d_0$, we reintroduce the $n$ dependence in the estimation results to emphasize sample complexity convergence. Lastly, we adopt standard $\mathcal{O}(\cdot), \Omega(\cdot), \Theta(\cdot)$ to omit constant factors and $\widetilde{\mathcal{O}}(\cdot)$ to omit logarithmic factors.

| Assumption | Score Approximation | Score Estimation | Dist. Estimation (Total Variation Distance) | Minimax Optimality |
|---|---|---|---|---|
| Generic Hölder Smooth Data Dist. (Sections 3.1 and 3.3) | $\mathcal{O}((\log(\frac{1}{\epsilon}))^{d_x}/\sigma_t^4)$ | $n^{-\Theta(1/d)} \cdot (\log n)^{\mathcal{O}(d_x)}$ | $n^{-\Theta(1/d)} \cdot (\log n)^{\mathcal{O}(d_x)}$ | ✘ |
| Strong Hölder Smooth Data Dist. (Sections 3.2 and 3.3) | $\widetilde{\mathcal{O}}(\epsilon^2/\sigma_t^2)$ | $n^{-\Theta(1/d)} \cdot (\log n)^{\mathcal{O}(d_x)}$ | $n^{-\Theta(1/d)} \cdot (\log n)^{\mathcal{O}(d_x)}$ | ✔ |
| Latent Subspace + Generic Hölder Smooth Data Dist. (Appendix A) | $\mathcal{O}((\log(\frac{1}{\epsilon}))^{d_0}/\sigma_t^4)$ | $n^{-\Theta(1/d_0)} \cdot (\log n)^{\mathcal{O}(d_0)}$ | $n^{-\Theta(1/d_0)} \cdot (\log n)^{\mathcal{O}(d_0)}$ | ✘ |
| Latent Subspace + Stronger Hölder Smooth Data Dist. (Appendix A) | $\widetilde{\mathcal{O}}(\epsilon^2/\sigma_t^2)$ | $n^{-\Theta(1/d_0)} \cdot (\log n)^{\mathcal{O}(d_x)}$ | $n^{-\Theta(1/d_0)} \cdot (\log n)^{\mathcal{O}(d_0)}$ | ✔ |

polynomial analysis (Fu et al., 2024a; Oko et al., 2023). These techniques leverage the nice regularity of the score function imposed by the Hölder smoothness data assumptions and hence enable fine-grained use of transformers' universal approximation (Kajitsuka and Sato, 2024; Yun et al., 2020) through a more detailed piecewise constant approximation. Consequently, we obtain tighter bounds.

**Contributions.** We summarize the theoretical results in Table 1. Our contributions are threefold:

- **Score Approximation.** We characterize the approximation limit of matching the conditional DiT score function with a transformer-based score estimator. The approximation results explain the expressiveness of conditional DiT and its latent version, and guide the score network's structural configuration for practical implementations (Theorems 3.1, 3.2 and A.1). The results also show that the latent version achieves a better approximation for the score function.

- **Score and Distribution Estimation.** We study the score and distribution estimation of conditional DiTs in practical training scenarios. Specifically, we provide a sample complexity bound for score estimation (Theorems 3.3 and D.3), using norm-based covering number bound of transformer architecture. Additionally, we show that the learned score estimator can recover the initial data distribution in both conditional DiT and its latent setting (Theorems 3.4 and A.2).

- **Minimax Optimal Estimator.** We extend our analysis to unconditional DiT and investigate whether the generated data distribution achieves the minimax optimality in the total variation distance. Specifically, we show that the upper bounds on the distribution estimation error match the lower bounds under stronger Hölder smooth data distribution (Theorem 3.5 and Remark A.3).

**Organization.** Section 2 presents preliminaries and the problem setup. Section 3 presents the results of conditional DiTs. Appendix A presents the results of latent conditional DiTs. Appendix C.1 presents related works' discussions. The appendix contains an extended and improved version of (Hu et al., 2024) on conditional DiTs (Appendix E), additional results, and detailed proofs.

**Notations.** The index set $\{1, ..., I\}$ is denoted by $[I]$, where $I \in \mathbb{N}^+$. We denote (column) vectors by lower case letters, and matrices by upper case letters. Let $a[i]$ denote the $i$-th component of vector $a$. Let $A_{ij}$ denotes the $(i, j)$-th entry of matrix $A$. $\|x\|, \|x\|_1$ and $\|x\|_\infty$ denote the Euclidean norm, 1-norm, and infinite norm. $\|W\|_2$ and $\|W\|_F$ denote the spectral norm and Frobenius norm. Lastly, we adopt standard $\mathcal{O}(\cdot), \Omega(\cdot), \Theta(\cdot)$ to omit constant factors and $\widetilde{\mathcal{O}}(\cdot)$ to omit logarithmic factors.

## 2 BACKGROUND AND PRELIMINARIES

In this section, we provide a high-level overview of the conditional diffusion model with classifier-free guidance in Section 2.1 and conditional Diffusion Transformer (DiT) networks in Section 2.2.

### 2.1 CONDITIONAL DIFFUSION MODEL WITH CLASSIFIER-FREE GUIDANCE

**Forward and Backward Conditional Diffusion Process.** In the *forward* process, conditional diffusion models gradually add noise to the original data $x_0 \in \mathbb{R}^{d_x}$. Give a condition $y \in \mathbb{R}^{d_y}$, and $x_0 \sim P_0(\cdot|y)$. Let $x_t$ denote the noisy data at the timestamp $t$, with marginal distribution and density as $P_t(\cdot|y)$ and $p_t(\cdot|y)$. The conditional distribution $P_t(x_t|y)$ follows $N(\alpha_t x_0, \sigma_t^2 I_{d_x})$, where

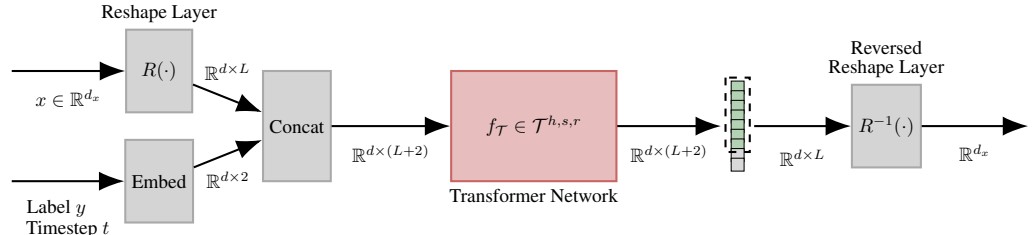

Figure 1: **Conditional DiT Network Architecture.** The architecture consists of a reshape layer $R(\cdot)$, a reversed reshape layer $R^{-1}(\cdot)$, and the embedding layers for label $y$ and timestep $t$. The embeddings of $y$ and $t$ are concatenated with input sequences and then processed by a transformer network $f_{\mathcal{T}} \in \mathcal{T}^{h,s,r}$.

$\alpha_t = e^{-t/2}$, $\sigma_t^2 = 1 - e^{-t}$, and $w(t) > 0$ is a nondecreasing weighting function. In practice, the forward process terminates at a large enough $T$ such that $P_T$ is close to $N(0, I_{d_x})$. In the *backward* process, we obtain $x_t^{\leftarrow}$ by reversing the forward process. The generation of $x_t^{\leftarrow}$ depends on the score function $\nabla \log p_t(\cdot|y)$. See Appendix F.1 for the details. In below, when the context is clear, we suppress the notation dependence of $x_t$ on the time step $t$.

**Classifier-Free Guidance.** Classifier-free guidance (Ho and Salimans, 2022) is the standard workhorse for training condition diffusion models. It approximates both conditional and unconditional score functions using neural networks $s_W$ with parameters $W$. It uses the following loss function:

$$\ell(x_0, y; s_W) = \int_{t_0}^{T} \frac{1}{T - t_0} \mathbb{E}_{\tau, x_t \sim N(\alpha_t x_0, \sigma_t^2 I_{d_x})} \left[ \| s_W(x_t, \tau y, t) - \nabla_{x_t} \log \phi_t(x_t|x_0) \|_2^2 \right] \mathrm{d}t,$$

where $\nabla_{x_t} \log \phi_t(x_t|x_0) = -(x_t - \alpha_t x_0)/\sigma_t^2$, $t_0$ is a small cutoff to stabilize training[1]. $\tau = \emptyset$ denotes the unconditional version, $\tau = \mathrm{id}$ denotes the conditional version, and $P(\tau = \emptyset) = P(\tau = \mathrm{id}) = 0.5$. To train $s_W$, we select $n$ i.i.d. samples $\{x_{0,i}, y_i\}_{i=1}^n$, where $x_{0,i} \sim P_0(\cdot|y_i)$. We use

$$\widehat{\mathcal{L}}(s_W) := \frac{1}{n} \sum_{i=1}^{n} \ell(x_{0,i}, y_i; s_W), \tag{2.1}$$

as the empirical loss. In addition, we denote population loss as $\mathcal{L}(s_W)$. See Appendix F.2 for details.

## 2.2 Conditional Diffusion Transformer Networks

We use a transformer network as a score estimator $s_W$. Our notation follows (Hu et al., 2024).

**Transformer Block.** Let $f^{(\mathrm{SA})} : \mathbb{R}^{d \times L} \to \mathbb{R}^{d \times L}$ denote the self-attention layer. Let $h$ and $s$ denote the number of heads and hidden dimension in the self-attention layer, and then we have

$$f^{(\mathrm{SA})}(Z) := Z + \sum_{i=1}^{h} W_O^i(W_V^i Z) \operatorname{Softmax} \left[ (W_K^i Z)^\top (W_Q^i Z) \right], \tag{2.2}$$

where $W_V^i, W_K^i, W_Q^i \in \mathbb{R}^{s \times d}$, and $W_O^i \in \mathbb{R}^{d \times s}$ are the weight matrices. Next, we define the feed-forward layer with MLP dimension $r$:

$$f^{(\mathrm{FF})}(Z) := Z + W_2 \mathrm{ReLU}(W_1 Z + b_1) + b_2, \tag{2.3}$$

where $W^{(1)} \in \mathbb{R}^{r \times d}$ and $W^{(2)} \in \mathbb{R}^{d \times r}$ are weight matrices, and $b^{(1)} \in \mathbb{R}^r$, and $b^{(2)} \in \mathbb{R}^d$ are bias.

**Definition 2.1** (Transformer Block). We define a transformer block of $h$-head, $s$-hidden dimension, $r$-MLP dimension, and with positional encoding $E \in \mathbb{R}^{d \times L}$ as

$$f^{h,s,r}(Z) := f^{(\mathrm{FF})} \left( f^{(\mathrm{SA})}(Z + E) \right) : \mathbb{R}^{d \times L} \mapsto \mathbb{R}^{d \times L}.$$

Now, we define the transformer networks as compositions of transformer blocks.

---

[1]$t_0$ is the early stopping time to prevent the score function from blowing up (Fu et al., 2024a; Chen et al., 2023c; Dhariwal and Nichol, 2021; Song et al., 2021).

**Definition 2.2** (Transformer Network Function Class). Let $\mathcal{T}^{h,s,r}$ denote the transformer network function class where each function $\tau \in \mathcal{T}^{h,s,r}$ is a composition of transformer blocks $f^{h,s,r}$, i.e.,

$$\mathcal{T}^{h,s,r} := \{\tau : \mathbb{R}^{d \times L} \mapsto \mathbb{R}^{d \times L} \mid \tau = f^{h,s,r} \circ \cdots \circ f^{h,s,r}\}.$$

**Conditional Diffusion Transformer (DiT).** Let $f \in \mathcal{T}^{h,s,r}$ be a transformer network, and $(x, y, t) \in \mathbb{R}^{d_x} \times \mathbb{R}^{d_y} \times [t_0, T]$ be the input data. We follow the "in-context conditioning" conditional DiT network in (Peebles and Xie, 2023) as in Figure 1. The following reshape layer converts a vector input $x \in \mathbb{R}^{d_x}$ into the sequential matrix input format $Z \in \mathbb{R}^{d \times L}$ for transformer with $d_x = d \cdot L$.

**Definition 2.3** (DiT Reshape Layer $R(\cdot)$). Let $R(\cdot) : \mathbb{R}^{d_x} \to \mathbb{R}^{d \times L}$ be a reshape layer that transforms the $d_x$-dimensional input into a $d \times L$ matrix. Specifically, for any $d_x = i \times i$ image input, $R(\cdot)$ converts it into a sequence representation with feature dimension $d := p^2$ (where $p \geq 2$) and sequence length $L := (i/p)^2$. Besides, we define the corresponding reverse reshape (flatten) layer $R^{-1}(\cdot) : \mathbb{R}^{d \times L} \to \mathbb{R}^{d_x}$ as the inverse of $R(\cdot)$. By $d_x = dL$, $R, R^{-1}$ are associative w.r.t. their input.

We define the following transformer network function class with the reshape layer. To simplify, we define $W_{KQ} := (W_K)^\top W_Q$ and $W_{OV} := W_O W_V$.

**Definition 2.4** (Transformer Network Function Class with Reshape Layer $\mathcal{T}_R^{h,s,r}$). The transformer network class with reshape layer $\mathcal{T}_R^{h,s,r}(C_\mathcal{T}, C_{KQ}^{2,\infty}, C_{KQ}, C_{OV}^{2,\infty}, C_{OV}, C_E, C_F^{2,\infty}, C_F, L_\mathcal{T})$ satisfies:

- $\mathcal{T}_R^{h,s,r} := \{R^{-1} \circ f_\mathcal{T} \circ R : \mathbb{R}^{d_x} \to \mathbb{R}^{d_x} \mid f_\mathcal{T} \in \mathcal{T}^{h,s,r}\}$;
- Transformer network output bound: $\sup_Z \|f_\mathcal{T}(Z)\|_2 \leq C_\mathcal{T}$;
- Parameter bound in $\mathcal{F}^{(\mathrm{FF})}$: $\max\{\|W_1\|_{2,\infty}, \|W_2\|_{2,\infty}\} \leq C_F^{2,\infty}$, $\max\{\|W_1\|_2, \|W_2\|_2\} \leq C_F^2$;
- Parameter bound in $\mathcal{F}^{(\mathrm{SA})}$: $\|W_{KQ}\|_2 \leq C_{KQ}$, $\|W_{OV}\|_2 \leq C_{OV}$, $\|W_{KQ}\|_{2,\infty} \leq C_{KQ}^{2,\infty}$, $\|W_{OV}\|_{2,\infty} \leq C_{OV}^{2,\infty}$, $\|E^\top\|_{2,\infty} \leq C_E$, where $2, \infty$-norm follows $\|\cdot\|_{2,\infty} := \max_{j \in [L]} \|Z_{:j}\|_2$;
- Lipschitz of $f_\mathcal{T} \in \mathcal{T}^{h,s,r}$: $\|f_\mathcal{T}(Z_1) - f_\mathcal{T}(Z_2)\|_F \leq L_\mathcal{T}\|Z_1 - Z_2\|_F$, for any $Z_1, Z_2 \in \mathbb{R}^{d \times L}$.

These norm bounds are critical to quantify the interplay between model, performance and data.

## 3 STATISTICAL LIMITS OF CONDITIONAL DITS

In this section, we present a refined decomposition scheme for the fine-grained analysis of score approximation, score estimation, and distribution estimation in conditional DiT. Our analysis considers two assumptions on initial data distributions: (i) a generic Hölder smooth data assumption (Section 3.1 for approximation, and Section 3.3 for estimation), (ii) a stronger Hölder smooth data assumption (Section 3.2 for approximation, and Section 3.3 for estimation). This new scheme leads to tighter bounds, including the minimax optimality of the unconditional DiT score estimator.

### 3.1 SCORE APPROXIMATION: GENERIC HÖLDER SMOOTH DATA DISTRIBUTIONS

We present a fine-grained piecewise approximation using transformers to approximate the conditional score function under the Hölder smoothness assumption on the initial data (Fu et al., 2024b). At its core, we introduce a score function decomposition scheme with term-by-term tractability.

We first introduce the definition of Hölder space and Hölder ball following (Fu et al., 2024b).

**Definition 3.1** (Hölder Space). Let $\alpha \in \mathbb{Z}_+^d$, and let $\beta = k_1 + \gamma$ denote the smoothness parameter, where $k_1 = \lfloor \beta \rfloor$ and $\gamma \in [0, 1)$. For a function $f : \mathbb{R}^d \to \mathbb{R}$, the Hölder space $\mathcal{H}^\beta(\mathbb{R}^d)$ is defined as the set of $\alpha$-differentiable functions satisfying: $\mathcal{H}^\beta(\mathbb{R}^d) := \{f : \mathbb{R}^d \to \mathbb{R} \mid \|f\|_{\mathcal{H}^\beta(\mathbb{R}^d)} < \infty\}$, where the Hölder norm $\|f\|_{\mathcal{H}^\beta(\mathbb{R}^d)}$ satisfies:

$$\|f\|_{\mathcal{H}^\beta(\mathbb{R}^d)} := \sum_{\|\alpha\|_1 < k_1} \sup_x |\partial^\alpha f(x)| + \max_{\alpha: \|\alpha\|_1 = k_1} \sup_{x \neq x'} \frac{|\partial^\alpha f(x) - \partial^\alpha f(x')|}{\|x - x'\|_\infty^\gamma}.$$

We also define the Hölder ball of radius $B$: $\mathcal{H}^\beta(\mathbb{R}^d, B) := \{f : \mathbb{R}^d \to \mathbb{R} \mid \|f\|_{\mathcal{H}^\beta(\mathbb{R}^d)} < B\}$.

Let $x_0 \in \mathbb{R}^{d_x}$ denote the initial data, and $y \in [0, 1]^{d_y}$ the conditional label. With Definition 3.1, we state the first assumption on the conditional distribution of initial data $x_0$.

**Assumption 3.1** (Generic Hölder Smooth Data). The conditional density function $p_0(x_0|y)$ is defined on the domain $\mathbb{R}^{d_x} \times [0,1]^{d_y}$ and belongs to Hölder ball of radius $B > 0$ for Hölder index $\beta > 0$, denoted by $p_0(x_0|y) \in \mathcal{H}^{\beta}(\mathbb{R}^{d_x} \times [0,1]^{d_y}, B)$ (see Definition 3.1 for precise definition.) Also, for any $y \in [0,1]^{d_y}$, there exist positive constants $C_1, C_2$ such that $p_0(x_0|y) \leq C_1 \exp\left(-C_2\|x_0\|_2^2/2\right)$.

**Remark 3.1.** The Hölder continuity assumption captures various smoothness levels in the conditional density function. The light-tail condition relaxes the bounded support assumption in (Oko et al., 2023). Moreover, Assumption 3.1 only applies to the initial conditional distribution and imposes no constraints on the induced conditional score function. This is far less restrictive than the Lipschitz score condition in prior works (Yuan et al., 2024; Lee et al., 2023; Chen et al., 2022).

In our work, we aim to approximate the conditional score function $\nabla \log p_t(x_t|y)$ using transformer architectures. Hu et al. (2024) analyze the unconditional DiTs based on the established universality of transformers (Yun et al., 2020). These theories discretize the input and output domains into infinitesimal grids and employ piecewise constant approximations to construct universal approximators with controllable errors. However, such methods do not yield tight bounds for DiT architectures (Hu et al., 2024). To combat this, we build on the key observation by Fu et al. (2024a)[2]:

$$p_t(x_t|y) = \int_{\mathbb{R}^{d_x}} \frac{\mathrm{d}x_0}{\sigma_t^{d_x}(2\pi)^{d_x/2}} \cdot \underbrace{p_0(x_0|y)}_{\approx k_1\text{-order Taylor polynomial}} \cdot \underbrace{\exp\left(-\frac{\|\alpha_t x_0 - x_t\|^2}{2\sigma_t^2}\right)}_{\approx k_2\text{-order Taylor polynomial}}. \qquad (3.1)$$

A term-by-term Taylor expansion of the above conditional distribution under Assumption 3.1 enables a more fine-grained analysis. As a result, we propose a *fine-grained version* of *piecewise constant approximation* for conditional DiTs, allowing transformers to approximate the conditional score function with tighter error bounds. In particular, we utilize a refined transformer universal approximation modified from (Kajitsuka and Sato, 2024) (see Appendix G.1 for details).

Our score approximation procedure has two stages: first, we construct a score approximator by incorporating the approximation of $p_t$ and $\nabla p_t$ using a Taylor expansion, then use transformers to approximate the score approximator. These lead to provably tight estimation results in Section 3.3.

We state our main result of score approximation using transformers under Assumption 3.1 as follows:

**Theorem 3.1** (Conditional Score Approximation under Assumption 3.1). Assume Assumption 3.1. For any precision parameter $0 < \epsilon < 1$ and smoothness parameter $\beta > 0$, let $\epsilon \leq \mathcal{O}(N^{-\beta})$ for some $N \in \mathbb{N}$. Let $C_\alpha, C_\sigma > 0$ be some absolute constants. For any $y \in [0,1]^{d_y}$ and $t \in [N^{-C_\sigma}, C_\alpha \log N]$, there exists a $\mathcal{T}_{\text{score}}(x,y,t) \in \mathcal{T}_R^{h,s,r}$ such that

$$\int_{\mathbb{R}^{d_x}} \|\mathcal{T}_{\text{score}}(x,y,t) - \nabla \log p_t(x|y)\|_2^2 \cdot p_t(x|y) \, \mathrm{d}x = \mathcal{O}\left(\frac{B^2}{\sigma_t^4} \cdot N^{-\beta} \cdot (\log N)^{d_x + \frac{\beta}{2} + 1}\right).$$

Notably, for $\epsilon = \mathcal{O}(N^{-\beta})$, the approximation error has the upper bound $\mathcal{O}((\log(\frac{1}{\epsilon}))^{d_x}/\sigma_t^4)$. The parameter bounds for the transformer network class are as follows:

$$C_{KQ}, C_{KQ}^{2,\infty} = O(N^{4\beta d + 2\beta}(\log N)^{4d+2}); \quad C_{OV}, C_{OV}^{2,\infty} = O(N^{-\beta});$$

$$C_F, C_F^{2,\infty} = O(N^\beta (\log N)^{\frac{d_x + \beta + 3}{2}}); \quad C_E = O(1); \quad C_{\mathcal{T}} = O(\sqrt{\log N}/\sigma_t^2),$$

where $O(\cdot)$ hides all polynomial factors depending on $d_x, d, L, \beta, C_1, C_2$.

**Remark 3.2.** $N$ is the resolution of the input domain discretization. We remark that domain discretization is essential for utilizing the local smoothness of functions under Hölder assumptions. Furthermore, $C_\sigma$ and $C_\alpha$ control the stability cutoff and early stopping time, respectively.

*Proof Sketch.* Recall that $\nabla \log p_t(x|y) = \frac{\nabla p_t(x|y)}{p_t(x|y)}$. Our proof follows three steps:

**Step 1: Smooth Local Approximations.** A $k_1$-th order and a $k_2$-th order Taylor expansion for $p_0(x|y)$ and $\exp(\cdot)$ yield two explicit functions $f_1(x,y,t)$ and $f_2(x,y,t)$ that approximate $p_t(x|y)$

---

[2]Recall that $p_t(x_t|y) = \int_{\mathbb{R}^{d_x}} p(x_0|y) p_t(x_t|x_0) \, \mathrm{d}x_0$ with $P_t(\cdot|y) \sim N(\alpha_t x_0, \sigma_t I_{d_x})$. In below, when the context is clear, we suppress the notation dependence of $x_t$ on the time step $t$.

and $\nabla p_t(x|y)$ in Lemma H.3 and Lemma H.4, respectively. Both $f_1(x, y, t)$ and $f_2(x, y, t)$ inherit the Hölder smoothness. Altogether, this gives score approximator by $\nabla \log p_t(x|y) = \frac{\nabla p_t(x|y)}{p_t(x|y)}$

**Step 2: Transformer Approximation on a Bounded Domain.** We leverage the universal approximation capabilities of transformers to approximate the score approximator on a bounded domain.

**Step 3: Extension to the Full Space via Sub-Gaussianity.** We extend the bounded-domain approximation to full space using the target density's sub-Gaussian tails. Gaussian tail bounds cap the error outside the domain and maintain the overall approximation accuracy.

**Error Matching.** The overall error includes $\mathrm{Error}_{\mathrm{Taylor}}$ and $\mathrm{Error}_{\mathcal{T}}$. Given a fixed discretization resolution $N$, $\mathrm{Error}_{\mathrm{Taylor}}$ remains fixed. However, the approximation error bound of the transformer can be an arbitrary value. We align $\mathrm{Error}_{\mathcal{T}}$ and $\mathrm{Error}_{\mathrm{Taylor}}$ to optimize the final results.

Please see Appendix H for a detailed proof. □

**Remark 3.3** (Approximation Rate). Given a fixed resolution $N$, the approximation error scales inversely with the smoothness $\beta$. As the smoothness increases, we get a tighter approximation error.

**Remark 3.4** (Comparing with Existing Works). Fu et al. (2024a) provide approximation rates for conditional diffusion models using ReLU networks. We are the first to establish approximation error bounds with transformer networks. Additionally, Oko et al. (2023) establish approximation rates under a compactness condition on the input data. We mitigate this compactness requirement by applying a Hölder smoothness assumption to control approximation error outside a compact domain.

### 3.2 Score Approximation: Stronger Hölder Smooth Data Distributions

Next, we study the conditional DiT score approximation problem using our score decomposition scheme under the stronger Hölder smoothness assumption from Fu et al. (2024b, Assumption 3.3).

**Assumption 3.2** (Stronger Hölder Smooth Data). Let function $f \in \mathcal{H}^\beta(\mathbb{R}^{d_x} \times [0, 1]^{d_y}, B)$. Given a constant radius $B$, positive constants $C$ and $C_2$, we assume the conditional density function $p(x_0|y) = \exp\left(-C_2\|x_0\|_2^2/2\right) \cdot f(x_0, y)$ and $f(x_0, y) \geq C$ for all $(x_0, y) \in \mathbb{R}^{d_x} \times [0, 1]^{d_y}$.

Assumption 3.2 imposes stronger assumption than Assumption 3.1 and induces a refined conditional score function decomposition. Explicitly, by Lemma I.1, $\nabla \log p_t(x|y)$ becomes:

$$\nabla \log p_t(x|y) = \frac{-C_2 x}{\alpha_t^2 + C_2 \sigma_t^2} + \frac{\nabla h(x, y, t)}{h(x, y, t)}, \tag{3.2}$$

where $h(x, y, t) := \int_{\mathbb{R}^{d_x}} \frac{f(x_0, y)}{\widehat{\sigma}_t^{d_x}(2\pi)^{d_x/2}} \exp\left(-\frac{\|x_0 - \widehat{\alpha}_t x\|^2}{2\widehat{\sigma}_t^2}\right) dx_0$, $\widehat{\sigma}_t = \frac{\sigma_t}{\sqrt{\alpha_t^2 + C_2\sigma_t^2}}$, and $\widehat{\alpha}_t = \frac{\alpha_t}{\alpha_t^2 + C_2\sigma_t^2}$.

We highlight that (3.2) leads to a tighter approximation error compared with Theorem 3.1. Intuitively, Assumption 3.2 imposes a lower bound on the conditional density function and hence implies in better regularity of the score function. In contrast, under Assumption 3.1, the score function lacks such regularity and may explode when $p_t$ is small. These low-density regions act as holes in the data support. They cause the score function to diverge near the boundary of these holes. To combat this, an implication of (3.2) is handy — $h$ is bounded from zero, ensuring that the score function remains well-behaved across the entire data domain. To elaborate more, two technical remarks are in order.

**Remark 3.5** (Linearity). The first term on the RHS of (3.2) is linear in $x$. This makes part of $\nabla \log p_t(x|y)$ a *linear* function of $x$, enabling easy approximation with a tighter bound.

**Remark 3.6** (Tightened Approximation Induced by $h$'s Lower Bound). Moreover, the introduction of $h$ tightens the approximation error due to the lower bound imposed by Assumption 3.2 (i.e., $f(x, y) \geq C$). The second term on the RHS of (3.2) mirrors the form $\nabla \log p_t(x|y) = \frac{\nabla p_t(x|y)}{p_t(x|y)}$ by replacing $p$ with $h$. In the analysis of Section 3.1, especially in Step 1 of the proof (resembling $f_1, f_2$ to approximate $\nabla p_t(x|y)$), we have to impose a threshold on the denominator of $\frac{\nabla p_t(x|y)}{p_t(x|y)}$ to prevent score explosion under Assumption 3.1. This threshold introduces additional approximation error (Lemma H.6). Assumption 3.2 remedies this by ensuring a lower bound on $p_t(x|y)$ through the minimum values of $f(x, y)$ and $\exp(-C_2\|x\|_2^2/2)$ within the compact domain after discretization. Setting this lower bound eliminates the need for a threshold and improves the approximation.

Consequently, decomposition (3.2) improves our approximation result from Section 3.1. We state our main result of score approximation using transformers under Assumption 3.2 as follows:

**Theorem 3.2** (Conditional Score Approximation under Assumption 3.2, Informal Version of Theorem I.1). Assume Assumption 3.2. For any precision parameter $0 < \epsilon < 1$ and smoothness parameter $\beta > 0$, let $\epsilon \leq \mathcal{O}(N^{-\beta})$ for some $N \in \mathbb{N}$. Let $C_\alpha, C_\sigma > 0$ be some absolute constants. For any $y \in [0,1]^{d_y}$ and $t \in [N^{-C_\sigma}, C_\alpha \log N]$, there exists a $\mathcal{T}_{\text{score}}(x, y, t) \in \mathcal{T}_R^{h,s,r}$ such that

$$\int_{\mathbb{R}^{d_x}} \|\mathcal{T}_{\text{score}}(x, y, t) - \nabla \log p_t(x|y)\|_2^2 \cdot p_t(x|y)\mathrm{d}x = \mathcal{O}\Big(\frac{B^2}{\sigma_t^2} \cdot N^{-2\beta} \cdot (\log N)^{\beta+1}\Big).$$

Notably, for $\epsilon = \mathcal{O}(N^{-\beta})$, the approximation error has the upper bound $\widetilde{\mathcal{O}}(\epsilon^2/\sigma_t^2)$.

Intrinsically, $N$ is proportional to the size of the transformer network $\mathcal{T}_{\text{score}}$ (see Theorem I.1). Hence, the precision parameter $\epsilon$ is inversely proportional to the size of $\mathcal{T}_{\text{score}}$. Therefore, Theorem 3.2 specifies the required configuration (e.g., size) of $\mathcal{T}_{\text{score}}$ for a desired score approximation accuracy.

*Proof Sketch.* The proof closely follows Theorem 3.1, but uses a different conditional score function decomposition in the form of (3.2). We highlight this key distinction in Lemma I.1.

Please see Appendix I for a detailed proof, and see Theorem I.1 for the formal version. □

**Remark 3.7** (Comparing with Theorem 3.1). Let $\widetilde{\mathcal{O}}(\cdot)$ hide the terms about $t_0, \log t_0, \log n$. In Theorem 3.2, the approximation rate $\widetilde{\mathcal{O}}(N^{-2\beta})$ is faster than that of Theorem 3.1, i.e., $\widetilde{\mathcal{O}}(N^{-\beta})$.

## 3.3 SCORE ESTIMATION AND DISTRIBUTION ESTIMATION OF CONDITIONAL DITs

Next, we study score and distribution estimations based on the two score approximation results for two different data assumptions: Theorems 3.1 and 3.2. Let $\widehat{s}$ denote the trained score estimator.

**Score Estimation.** Building on our approximation results from Sections 3.1 and 3.2, the next objective is to evaluate the performance of the score estimator $\widehat{s}$ trained with a set of finite samples by optimizing the empirical loss (2.1). To quantify this, we introduce the score estimation risk.

**Definition 3.2** (Conditional Score Risk). Given a score estimator $\widehat{s}$, we define risk as the expectation of the squared $\ell_2$ difference between the score estimator and the ground truth with respect to $(x_t, y, t)$:

$$\mathcal{R}(\widehat{s}) := \int_{t_0}^{T} \frac{1}{T - t_0} \mathbb{E}_{x_t, y} \|\widehat{s}(x_t, y, t) - \nabla \log p_t(x_t|y)\|_2^2 \mathrm{d}t.$$

Given a set of i.i.d sample $\{x_i, y_i\}_{i \in [n]}$, direct computation of $\mathbb{E}_{\{x_i, y_i\}_{i \in [n]}}[\mathcal{R}(\widehat{s})]$ is infeasible due to the absence of access to the joint distribution $P(x_t, y)$. To address this, we: (i) Decompose the risk into estimation and approximation errors, (ii) Bound the estimation error using the covering number of transformers, and (iii) Bound the approximation error using Theorem 3.1 and Theorem 3.2.

**Theorem 3.3** (Conditional Score Estimation with Transformer). Consider $y \in [0,1]^{d_y}$ and $t \in [t_0, T]$ with $t_0 = N^{-C_\sigma}$ and $T = C_\alpha \log N$, where $C_\sigma, C_\alpha > 0$ are absolute constants such that $t_0 < 1$. Furthermore, let $\nu_1 = 16\beta d + 12\beta$ and $\nu_2 = 20d_x + 4\beta + 18$.
• Assume Assumption 3.1. Then, it holds

$$\mathbb{E}_{\{x_i, y_i\}_{i=1}^n}[\mathcal{R}(\widehat{s})] = \mathcal{O}\left(n^{-\frac{\beta}{\nu_1 + C_\sigma + 3\beta}} \cdot (\log n)^{\nu_2}\right).$$

• Assume Assumption 3.2. Then, it holds

$$\mathbb{E}_{\{x_i, y_i\}_{i=1}^n}[\mathcal{R}(\widehat{s})] = \mathcal{O}\left(n^{-\frac{2\beta}{\nu_1 + 6\beta}} \cdot (\log n)^{\nu_2}\right).$$

The upper bounds in Theorem 3.3 arises from the fundamental bias-variance trade-off. A key aspect of the risk analysis comes from treating $n$ i.i.d. training samples as random variables drawn from the same distribution. Bounding this expectation relies on the log-covering number of transformer network class, characterized by $\nu_1, \nu_2$ and $\nu_3$. Specifically, the difference between the risk computed from the actual sample and computed from the closest element in the cover remains small. Controlling

this gap ensures that the overall risk bound holds. In particular, A larger log-covering number corresponds to a more expressive model, leading to higher variance in the learned estimator. One the other hand, the bias term arises from the approximation error from Theorem 3.1 and Theorem 3.2. Reducing the approximation error requires increasing network capacity, while expanding the function class raises the log-covering number. The overall risk bound follows from balancing these two effects.

*Proof.* Please see Appendix J.2 for detailed proofs. □

**Remark 3.8** (Sample Complexity Bounds). To obtain $\epsilon$-error in terms of score estimation, we have the sample complexity $\widetilde{\mathcal{O}}(\epsilon^{-(\nu_1+C_\sigma+3\beta)/\beta})$ under Assumption 3.1 and $\widetilde{\mathcal{O}}(\epsilon^{-(\nu_3+6\beta)/2\beta})$ under Assumption 3.2. Here $\widetilde{\mathcal{O}}(\cdot)$ ignores the terms about $t_0$, $\log t_0$ and $\log n$. The Hölder data smoothness degree $\beta$ affects the sample complexity. This indicates that the regularity of the initial data distribution determines the complexity of score estimation.

**Distribution Estimation.** Next, we study the distributional estimation capability of the trained conditional score network $s(x, y, t)$ by analyzing the total variation distance between the estimated and true distributions. Our strategy uses a three-part decomposition: (i) the total variation between the true distributions at timestamps 0 and $t_0$, (ii) the total variation between the true distribution at $t_0$ and the reverse process distribution using the true score function, and (iii) the total variation between the reverse process distributions using the true and estimated score functions at $t_0$.

**Theorem 3.4** (Conditional Distribution Estimation). For $y \in [0, 1]^{d_y}$, let $\widehat{P}_{t_0}(\cdot|y)$ denote *estimated* conditional distributions at $t_0$. Recall that $P_0(\cdot|y)$ is the conditional distribution of initial data $x_0$ given $y$ and $t \in [N^{-C_\sigma}, C_\alpha \log N]$. Assume $\mathrm{KL}\left(P_0(\cdot|y) \mid N(0, I)\right) \leq c$ for some constant $c < \infty$.
- Assume Assumption 3.1. With $\nu_1$ and $\nu_2$, specified in Theorem 3.3, it holds

$$\mathbb{E}_{\{x_i,y_i\}_{i=1}^n}\left[\mathbb{E}_y\left[\mathrm{TV}\left(\widehat{P}_{t_0}(\cdot|y), P_0(\cdot|y)\right)\right]\right] = \mathcal{O}\left(n^{-\omega}(\log n)^{\frac{\nu_2}{2}+\frac{3}{2}}\right),$$

where $\omega = \min\left\{\frac{C_\sigma}{2(\nu_1+C_\sigma+3\beta)}, \frac{C_\alpha}{\nu_1+C_\sigma+3\beta}, \frac{\beta}{2(\nu_1+C_\sigma+3\beta)}\right\}$.
- Assume Assumption 3.2. With $\nu_1$ and $\nu_2$, specified in Theorem 3.3, for all $x \in \mathbb{R}^{d_x}$, it holds

$$\mathbb{E}_{\{x_i,y_i\}_{i=1}^n}\left[\mathbb{E}_y\left[\mathrm{TV}\left(\widehat{P}_{t_0}(\cdot|y), P_0(\cdot|y)\right)\right]\right] = \mathcal{O}\left(n^{-\phi}(\log n)^{\frac{\nu_2}{2}+\frac{3}{2}}\right),$$

where $\phi = \min\left\{\frac{C_\sigma}{2(\nu_1+6\beta)}, \frac{C_\alpha}{\nu_1+6\beta}, \frac{\beta}{\nu_1+6\beta}\right\}$.

*Proof.* Please see Appendix J.4 for a detailed proof. □

## 3.4 MINIMAX OPTIMAL ESTIMATION OF UNCONDITIONAL DiTs

We are now ready to present our minimax optimality result for unconditional DiTs. Recall the minimax optimal rate for distribution estimation under the strong Hölder assumption (Assumption 3.2).

**Lemma 3.1** (Proposition 4.3 of (Fu et al., 2024b) and (Yang and Barron, 1999)). For a fixed constant $C_2$ and a Hölder index $\beta > 0$. We consider the task of estimating a probability distribution $P(x)$ with its density function defined within the following function space

$$\mathcal{P} = \left\{p(x) = f(x)\exp\left(-C_2\|x\|_2^2\right) : f(x) \in \mathcal{H}^\beta(\mathbb{R}^{d_x}, B), f(x) \geq C \geq 0\right\},$$

Given $n$ i.i.d data $\{x_i\}_{i=1}^n$, we have $\inf_{\widehat{\mu}} \sup_{p\in\mathcal{P}} \mathbb{E}_{\{x_i\}_{i=1}^n}[\mathrm{TV}(\widehat{\mu}, \mathrm{P})] \geq \Omega(n^{-\frac{\beta}{d_x+2\beta}})$. Here, the estimator $\widehat{\mu}$ ranges over all possible estimators constructed from the data.

Setting $d_y = 0$, we show unconditional DiTs match the minimax optimal rate under specific conditions.

**Theorem 3.5** (Minimax Optimality of DiTs). Recall Theorem 3.4. The minimax optimality of unconditional diffusion transformer holds under $C_\sigma = 2C_\alpha = 2\beta$ and $16\beta(d + 1) = d_x$.

*Proof.* Please see Appendix J.5 for detailed proofs. □

**Remark 3.9.** Since we do not impose any assumptions on condition $y$ (in contrast to data $x$), Theorem 3.5 establishes only the minimax optimality of unconditional DiTs by setting $d_y = 0$. We leave the minimax optimality of conditional DiTs for future works.

**Remark 3.10** (Comparing with Existing Works)**.** Oko et al. (2023) analyze the ReLU network and provide the near minimax optimal estimation rates in both the total variation distance and Wasserstein distance of order one. Fu et al. (2024b) utilize the ReLU network and provides the minimax optimality for distribution in total variation as well. Our results offer the first and exact minimax optimal guarantee for unconditional DiTs in distribution estimation.

## 4 DISCUSSION AND CONCLUSION

We investigate the approximation and estimation rates of conditional DiT and its latent setting. We focus on the "in-context" conditional DiT setting presented by Peebles and Xie (2023), and conduct a comprehensive analysis under various common data conditions (Section 3 for generic and strong Hölder smooth data, Appendix A for data with intrinsic latent subspace).

Interestingly, we establish the minimax optimality of the unconditional DiTs' estimation by reducing our analysis of conditional DiTs to the unconditional setting (Section 3.4 and Remark A.3). Our key techniques include a well-designed score decomposition scheme (Section 3.1). These enable a finer use of transformers' universal approximation, compared to the prior statistical rates of DiTs derived from the universal approximation results in (Yun et al., 2020) by Hu et al. (2024).

Consequently, we provide three extensions in the appendix:

- In Appendix A, we extend the results from Section 3.2 by assuming the input data has an intrinsic lower-dimensional representation. Importantly, we establish the minimax optimality of estimation for such latent DiTs.

- In Appendix D, we expand Appendix A and extend our well-designed score decomposition scheme from Section 3 to the latent conditional DiT. Notably, we also obtain provably tight rate, i.e., for distribution estimation under Assumption 3.2 (Remark A.3).

- In Appendix E, we extend the analysis of (Hu et al., 2024) to the conditional DiT setting and provide an improved version. In particular, we analyze conditional latent DiTs under the following three assumptions from (Hu et al., 2024) and obtained sharper rates:

  – Low-Dimensional Linear Latent Space Data (Assumption A.1)
  – Lipschitz Score Function (Assumption E.2)
  – Light Tail Data Distribution (Assumption E.3)

  In detail, we use a modified universal approximation of the single-layer self-attention transformers (modified from (Kajitsuka and Sato, 2024)) to avoid the need for dense layers required in (Yun et al., 2020). This refinement results in tighter error bounds for both score and distribution estimation. Consequently, our sample complexity error bounds avoid the gigantic double exponential term $2^{(1/\epsilon)^{2L}}$ reported by Hu et al. (2024), and obtain sharper rates than those of (Hu et al., 2024).

ACKNOWLEDGMENTS

JH would like to thank Sitan Chen, Mimi Gallagher, Sara Sanchez, Dino Feng and Andrew Chen for enlightening discussions; Wei-Po Wang, Zhao Song, Jui-Hui Chung, David Liu, Yibo Wen, Maojiang Su, and Hude Liu for collaborations on related topics; and the Red Maple Family for support. The authors would like to thank the anonymous reviewers and program chairs for constructive comments. They also thank Maojiang Su, Ning Zhu, Mingcheng Lu, and Sophia Pi for pointing out typos.

JH is partially supported by the Walter P. Murphy Fellowship. HL is partially supported by NIH R01LM1372201, AbbVie and Dolby. This research was supported in part through the computational resources and staff contributions provided for the Quest high performance computing facility at Northwestern University which is jointly supported by the Office of the Provost, the Office for Research, and Northwestern University Information Technology. The content is solely the responsibility of the authors and does not necessarily represent the official views of the funding agencies.

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

# Appendix

# A    LATENT CONDITIONAL DITS

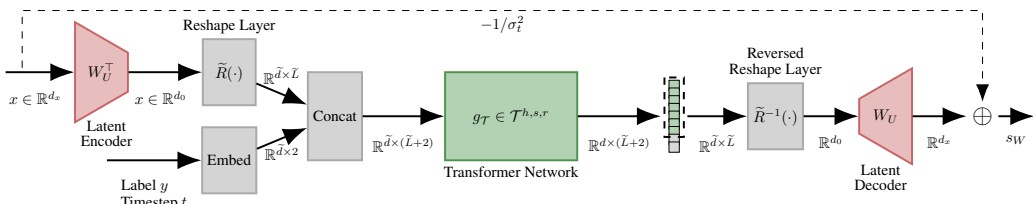

Figure 2: **Network Architecture of Latent Conditional DiT.** The overall architecture consists of linear layer of encoder and decoder $W_U^\top$ and $W_U$ that transform input $x \in \mathbb{R}^{d_x}$ into linear latent space $\mathbb{R}^{d_0}$, reshaping layer $\widetilde{R}(\cdot)$ and $\widetilde{R}^{-1}(\cdot)$, embedding layer for label $y$ and timestep $t$. The embedding concatenates with input sequences and processes by the adapted transformer network $\mathcal{T}_{\widetilde{R}}^{h,s,r} = \widetilde{R}^{-1} \circ g_{\mathcal{T}} \circ f^{(\mathrm{FF})} \circ \widetilde{R}$.

In this section, we extend the results from Section 3 by considering the latent conditional DiTs. Specifically, we assume the raw input $x \in \mathbb{R}^{d_x}$ has an intrinsic lower-dimensional representation.

**Assumption A.1** (Low-Dimensional Linear Latent Space). Initial data $x$ has a latent representation via $x = Uh$, where $U \in \mathbb{R}^{d_x \times d_0}$ is an unknown matrix with orthonormal columns. The latent variable $h \in \mathbb{R}^{d_0}$ follows the distribution $P_h$ with a density function $p_h$.

**Remark A.1.** "Linear Latent Space" means that each entry of a given latent vector is a linear combination of the corresponding input, i.e., $x = Uh$. This is also known as the "low-dimensional data" assumption in literature (Hu et al., 2024; Chen et al., 2023c). This assumption is fundamental in dimensionality reduction techniques for capturing the intrinsic lower-dimensional structure of data.

**Score Decomposition and Model Architecture.** To derive approximation and estimation results, we extend the techniques and network architecture presented in Section 3 to latent diffusion by considering the "low-dimensional linear subpace". Under Assumption A.1, we decompose the score:

$$\nabla \log p_t(x|y) = U(\ \underbrace{\sigma_t^2 \nabla \log p_t^h(U^\top x|y) + U^\top x}_{:=q(U^\top x, y, t):\ \mathbb{R}^{d_0} \times \mathbb{R}^{d_y} \times [t_0, T] \to \mathbb{R}^{d_0}}\ )/\sigma_t^2 - \underbrace{x/\sigma_t^2}_{\text{residual connection}}, \qquad (\text{A.1})$$

following Hu et al. (2024); Chen et al. (2023c) (see Lemma D.1). Based on this decomposition, we construct the model architecture in Figure 2. The network detail for approximate (A.1) are as follow: a transformer $g_{\mathcal{T}}(W_U^\top x, y, t) \in \mathcal{T}_{\widetilde{R}}^{h,s,r}$ to approximate $q(U^\top x, y, t)$, a latent encoder $W_U^\top \in \mathbb{R}^{d_0 \times d_x}$ and decoder $W_U \in \mathbb{R}^{d_x \times d_0}$ to approximate $U^\top \in \mathbb{R}^{d_0 \times d_x}$ and $U \in \mathbb{R}^{d_x \times d_0}$, and a residual connection to approximate $-x/\sigma_t^2$. Importantly, $d_0$ is the latent dimension.

For latent diffusion, we follow the standard setting by Peebles and Xie (2023). For each input $x \in \mathbb{R}^{d_x}$ and corresponding label $y \in \mathbb{R}^{d_y}$, we use a transformer network to obtain a score estimator $s_W \in \mathbb{R}^{d_x}$. The key differences from Section 3 are as follows: First, we apply a latent encoder $W_U^\top \in \mathbb{R}^{d_0 \times d_x}$ to map the raw data $x \in \mathbb{R}^{d_x}$ into a low-dimensional representation $h := W_U^\top x \in \mathbb{R}^{d_0}$, where $d_0 \leq d_x$. Second, we reshape $h \in \mathbb{R}^{d_0}$ into a sequence $H \in \mathbb{R}^{\widetilde{d} \times \widetilde{L}}$ using a layer $\widetilde{R}(\cdot) : \mathbb{R}^{d_0} \to \mathbb{R}^{\widetilde{d} \times \widetilde{L}}$, with $d_0 = \widetilde{d} \cdot \widetilde{L}$. Note that, by $d_0 \leq d_x$, $\widetilde{d} \leq d$, and $\widetilde{L} \leq L$. Third, we pass $H \in \mathbb{R}^{\widetilde{d} \times \widetilde{L}}$ through the transformer $g_{\mathcal{T}}$. Lastly, We then obtain the predicted score $s_W \in \mathbb{R}^{d_x}$ by applying the inverse reshape layer $\widetilde{R}^{-1}(\cdot) : \mathbb{R}^{\widetilde{d} \times \widetilde{L}} \to \mathbb{R}^{d_0}$, followed by the latent decoder $W_U : \mathbb{R}^{d_0} \to \mathbb{R}^{d_x}$.

For our analysis, we study the cases under both the generic and strong Hölder smoothness assumptions on latent representation $z \in \mathbb{R}^{d_0}$. Specifically, we assume the "latent" data is $\beta_0$-Hölder smooth with radius $B_0$ following Assumptions 3.1 and 3.2. We extend both approximation and estimation results from Section 3 to latent diffusion and establish the minimax optimality of latent conditional DiTs.

**Score Approximation.** We now present the approximation rates for latent score function under both generic and stronger Hölder data assumptions. Let $h := W_U^\top x \in \mathbb{R}^{d_0}$ and $\bar{h} := U^\top x \in \mathbb{R}^{d_0}$ be the estimated and ground truth (according to Assumption A.1) latent representations, respectively.

**Theorem A.1** (Score Approximation of Latent Conditional DiTs, Informal Version of Theorems D.1 and D.2). Assume $d_0 = \Omega(\frac{\log N}{\log \log N})$. For any precision $0 < \epsilon < 1$ and smoothness $\beta_0 > 0$, let $\epsilon \leq \mathcal{O}(N^{-\beta_0})$ for some $N \in \mathbb{N}$. Let $C_\alpha, C_\sigma > 0$ be some absolute constant. For any $y \in [0,1]^{d_y}$ and $t \in [N^{-C_\sigma}, C_\alpha \log N]$, there exists a $\mathcal{T}_{\text{score}}(x, y, t) \in \mathcal{T}_{\widetilde{R}}^{h,s,r}$ such that

- Under Assumption 3.1, we have

$$\int_{\mathbb{R}^{d_0}} \left\| \mathcal{T}_{\text{score}}(\overline{h}, y, t) - \nabla \log p_t^h(\overline{h}|y) \right\|_2^2 \cdot p_t^h(\overline{h}|y) \mathrm{d}\overline{h} = \mathcal{O}\left( \frac{B_0^2}{\sigma_t^4} \cdot N^{-\beta_0} \cdot (\log N)^{d_0 + \frac{\beta_0}{2} + 1} \right).$$

  Notably, for $\epsilon = \mathcal{O}(N^{-\beta_0})$, the approximation error has the upper bound $\mathcal{O}((\log\left(\frac{1}{\epsilon}\right))^{d_0}/\sigma_t^4)$.

- Under Assumption 3.2, it holds

$$\int_{\mathbb{R}^{d_0}} \left\| \mathcal{T}_{\text{score}}(x, y, t)(\overline{h}, y|y) - \nabla \log p_t^h(\overline{h}|y) \right\|_2^2 \cdot p_t^h(\overline{h}|y) \mathrm{d}\overline{h} = \mathcal{O}\left( \frac{B_0^2}{\sigma_t^2} \cdot N^{-2\beta_0} \cdot (\log N)^{\beta_0 + 1} \right).$$

  Notably, for $\epsilon = \mathcal{O}(N^{-\beta_0})$, the approximation error has the upper bound $\widetilde{\mathcal{O}}(\epsilon^2/\sigma_t^2)$.

*Proof.* See Theorems D.1 and D.2 for the formal versions and Appendices H and I for proofs. □

**Remark A.2** (Comparing with Theorems 3.1 and 3.2). Recall $d_x \geq d_0$. The approximation error bounds are $\mathcal{O}((\log\left(\frac{1}{\epsilon}\right))^{d_x}/\sigma_t^4)$ in Theorem 3.1 and $\widetilde{\mathcal{O}}(\epsilon^2/\sigma_t^2)$ in Theorem 3.2. Theorem A.1 shows that the latent conditional DiT achieves better approximation and has the potential to bypass the challenges associated with the high dimensionality of initial data.

**Score and Distribution Estimation.** Based on Theorem A.1, we derive the score estimation bounds in Theorem D.3, and report the results for distribution estimation in next theorem.

**Theorem A.2** (Conditional Distribution Estimation). For $y \in [0,1]^{d_y}$, let $\widehat{P}_{t_0}(\cdot|y)$ denote *estimated* conditional distributions at $t_0$. Recall that $P_0(\cdot|y)$ is the conditional distribution of initial data $x_0$ given $y$ and $t \in [N^{-C_\sigma}, C_\alpha \log N]$. Assume $\text{KL}\left(P_0(\cdot|y) \mid N(0,I)\right) \leq c$ for some constant $c < \infty$.

- Assume $d_0 = \Omega(\frac{\log N}{\log \log N})$ and Assumption 3.1. Let $\widetilde{\nu}_1 := 68\beta_0 + 104C_\sigma$ and $\widetilde{\nu}_2 := 12d_0 + 12\beta_0 + 2$. it holds

$$\mathbb{E}_{\{x_i, y_i\}_{i=1}^n} \left[ \mathbb{E}_y \left[ \text{TV}\left( \widehat{P}_{t_0}(\cdot|y), P_0(\cdot|y) \right) \right] \right] = \mathcal{O}\left( n^{-\omega} (\log n)^{\frac{\widetilde{\nu}_2}{2} + \frac{3}{2}} \right),$$

  where $\omega = \min\left\{ \frac{C_\sigma}{2(\widetilde{\nu}_1 + C_\sigma + 3\beta_0)}, \frac{C_\alpha}{\widetilde{\nu}_1 + C_\sigma + 3\beta_0}, \frac{\beta_0}{2(\widetilde{\nu}_1 + C_\sigma + 3\beta_0)} \right\}$.

- Assume Assumption 3.2. Let $\widetilde{\nu}_3 := \frac{4}{\widetilde{d}}(12\beta_0 d_0 + 31\beta_0 \widetilde{d} + 6\beta_0) + \frac{12C_\alpha}{\widetilde{d}}(12d_0 + 25\widetilde{d} + 6) + 72C_\sigma$. For all $x \in \mathbb{R}^{d_x}$, it holds

$$\mathbb{E}_{\{x_i, y_i\}_{i=1}^n} \left[ \mathbb{E}_y \left[ \text{TV}\left( \widehat{P}_{t_0}(\cdot|y), P_0(\cdot|y) \right) \right] \right] = \mathcal{O}\left( n^{-\phi} (\log n)^{\max\{7, \frac{\beta_0 + 3}{2}, \frac{d_0 + 1}{2}\}} \right),$$

  where $\phi = \min\left\{ \frac{C_\sigma}{2(\widetilde{\nu}_3 + 6\beta_0)}, \frac{C_\alpha}{\widetilde{\nu}_3 + 6\beta_0}, \frac{\beta_0}{\widetilde{\nu}_3 + 6\beta_0} \right\}$.

*Proof.* Please see Appendix J.4 for a detailed proof. □

**Remark A.3** (Minimax Optimal Estimation). Following the same idea in Section 3.4, we show that the estimation error bound in Theorem A.2 is the optimal tight bound for the latent unconditional DiT. Specifically, by applying Lemma 3.1 and substituting $p(x|y)$ and $d_x$ by $p_t^h(\overline{h}|y)$ and $d_0$ respectively in Assumption 3.2, we follow the setting specified in Appendix J.5.

**Concluding Remarks.** We defer the discussion of our results and concluding remarks to Section 4. We extend our analysis to the setting of (Hu et al., 2024) and improve their results in Appendix E. Importantly, our bounds avoid the gigantic $2^{(1/\epsilon)^{2L}}$ term reported by Hu et al. (2024).

# B  NOTATION TABLE

We summarize our notations in the following table for easy reference.

Table 2: Mathematical Notations and Symbols

| Symbol | Description |
| --- | --- |
| $[I]$ | The index set $\{1, ..., I\}$, where $I \in \mathbb{N}^+$ |
| $a[i]$ | The $i$-th component of vector $a$ |
| $A_{ij}$ | The $(i,j)$-th entry of matrix $A$ |
| $\|x\|$ | Euclidean norm of vector $x$ |
| $\|x\|_1$ | 1-norm of vector $x$ |
| $\|x\|_2$ | 2-norm of vector $x$ |
| $\|x\|_\infty$ | Infinite norm of vector $x$ |
| $\|W\|_2$ | Spectral norm of matrix $W$ |
| $\|W\|_F$ | Frobenius norm of matrix $W$ |
| $\|W\|_{p,q}$ | $(p,q)$-norm of matrix $W$, where $p$-norm is over columns and $q$-norm is over rows |
| $\|f(x)\|_{L^2}$ | $L^2$-norm, where $f$ is a function |
| $\|f(x)\|_{L^2(P)}$ | $L^2(P)$-norm, where $f$ is a function and $P$ is a distribution |
| $\|f(\cdot)\|_{Lip}$ | Lipschitz-norm, where $f$ is a function |
| $d_p(f,g)$ | $p$-norm of the difference between functions $f$ and $g$ defined as $d_p(f,g) = \left( \int \|f(x) - g(x)\|^p \, dx \right)^{1/p}$ |
| $f_\sharp P$ | Pushforward measure, where $f$ is a function and $P$ is a distribution |
| $\mathrm{KL}(P,Q)$ | Kullback-Leibler (KL) divergence between distributions $P$ and $Q$ |
| $\mathrm{TV}(P,Q)$ | Total variation (TV) distance between distributions $P$ and $Q$ |
| $N(\mu, \sigma^2)$ | Normal distribution with mean $\mu$ and variance $\sigma^2$ |
| $a \lesssim b$ | There exist constants $C > 0$ such that $a \leq Cb$ |
| $n$ | Sample size |
| $x$ | Data point in original data space, $x \in \mathbb{R}^{d_x}$ |
| $y$ | Conditioning Label, $x \in \mathbb{R}^{d_y}$ |
| $h$ | Latent variable in low-dimensional subspace, $h \in \mathbb{R}^{d_0}$ |
| $\bar{h}$ | $\bar{h} = U^\top x$ |
| $p_h$ | The density function of $h$ |
| $U$ | The matrix with orthonormal columns to transform $h$ to $x$, where $U \in \mathbb{R}^{d \times d_0}$ |
| $B$ | Radius of Hölder ball for conditional density function $p(x\|y)$ |
| $B_0$ | Radius of Hölder ball for latent conditional density function $p(\bar{h}\|y)$ |
| $\beta$ | Hölder index for conditional density function $p(x\|y)$ |
| $\beta_0$ | Hölder index for latent conditional density function $p(\bar{h}\|y)$ |
| $D$ | Granularity in the construction of the transformer universal approximation |
| $N$ | Resolution of the discretization of the input domain |
| $\mathcal{R}$ | Score risk (expectation of squared $\ell^2$ difference between score estimator and ground truth) |
| $\mathcal{N}(\epsilon, \mathcal{F}, \|\cdot\|)$ | Covering number of collection $\mathcal{F}$ (see Definition J.5) |
| $T$ | Stopping time in the forward process of diffusion model |
| $t_0$ | Stopping time in the backward process of diffusion model |
| $\mu$ | Discretized step size in backward process |
| $p_t(\cdot)$ | The density function of $x$ at time $t$ |
| $p_t^h(\cdot)$ | The density function of $\bar{h}$ at time $t$ |
| $\psi$ | (Conditional) Gaussian density function |
| $\mathcal{T}^{h,s,r}$ | Transformer network function class (see Definition 2.2) |
| $f^{h,s,r}$ | Transformer block of $h$-head, $s$-hidden size, $r$-MLP dimension (see Definition 2.1) |
| $d$ | Input dimension of each token in the transformer network of DiT |
| $L$ | Token length in the transformer network of DiT |
| $\widetilde{d}$ | Latent data input dimension of each token in the transformer network of DiT |
| $\widetilde{L}$ | Latent data token length in the transformer network of DiT |
| $X$ | Sequence input of transformer network in DiT, where $X \in \mathbb{R}^{d \times L}$ |
| $H$ | Sequence latent data input of transformer network in DiT, where $X \in \mathbb{R}^{d \times L}$ |
| $E$ | Position encoding, where $E \in \mathbb{R}^{d \times L}$ |
| $R(\cdot)$ | Reshape layer in DiTs, $R(\cdot) : \mathbb{R}^{d_x} \to \mathbb{R}^{d \times L}$ |
| $\widetilde{R}(\cdot)$ | Reshape layer in latent DiTs, $\widetilde{R}(\cdot) : \mathbb{R}^{d_0} \to \mathbb{R}^{\widetilde{d} \times \widetilde{L}}$ |
| $R^{-1}(\cdot)$ | Reverse reshape layer in DiTs, $R^{-1}(\cdot) : \mathbb{R}^{d \times L} \to \mathbb{R}^{d_x}$ |
| $\widetilde{R}^{-1}(\cdot)$ | Reverse reshape layer in latent DiTs, $\widetilde{R}^{-1}(\cdot) : \mathbb{R}^{\widetilde{d} \times \widetilde{L}} \to \mathbb{R}^{d_0}$ |
| $W_U$ | The orthonormal matrix to approximate $U$, where $W_U \in \mathbb{R}^{d_x \times d_0}$ |

## C  RELATED WORKS, BROADER IMPACT AND LIMITATIONS

### C.1  RELATED WORKS

In the following, we discuss the recent success of the techniques used in our work. We first give the universality (universal approximation) of the transformer. Then, we discuss recent theoretical developments (approximation and estimation) in diffusion generative models.

**Universality of Transformers.**  The universality of transformers refers to their ability to approximate any sequence-to-sequence function with arbitrary precision. Yun et al. (2020) establish this by showing that transformers is capable of universally approximate sequence-to-sequence functions using deep stacks of feed-forward and self-attention layers. Additionally, Alberti et al. (2023) demonstrate universal approximation for architectures employing non-standard attention mechanisms. Recently, Kajitsuka and Sato (2024) show that even a single-layer transformer with self-attention suffices for universal approximation assuming all attention weights are rank-1. Moreover, Hu et al. (2024) leverage (Yun et al., 2020)'s universality results to analyze the approximation and estimation capabilities of DiT.

Our paper is motivated by and builds upon the works of Hu et al. (2024); Kajitsuka and Sato (2024); Yun et al. (2020). Specifically, we utilize and extend the transformer universality result from (Kajitsuka and Sato, 2024). We employ a relaxed contextual mapping property in (Kajitsuka and Sato, 2024) (see Appendix G.1). This generalization allows us to avoid the "double exponential" sample complexity bounds in previous DiT analyses (Hu et al., 2024, Remark 3.4) and establish transformer approximation in the simplest configuration — a single-layer, single-head attention model.

**Approximation and Estimation Theories of Diffusion Models.**  The theories of DiTs revolve around two main frontiers: score function approximation and statistical estimation (Chen et al., 2024a; Tang and Zhao, 2024). Score function approximation refers to the ability of the score network to approximate the score function. It leverages the universal approximation ability of the neural network in $L^p$ norms (Hayakawa and Suzuki, 2020), the approximation characterized as Taylor polynomial (Fu et al., 2024a) or B-Spline (Oko et al., 2023). Chen et al. (2023c) and Fu et al. (2024a) investigate score approximation under specific conditions, such as low-dimensional linear subspaces and Hölder smooth data assumptions, using ReLU-based models. Furthermore, Hu et al. (2024) presents the first characterization of score approximation in diffusion transformers (DiTs).

The statistical estimation includes score function and distribution estimation (Wu et al., 2024b; Dou et al., 2024a; Guo et al., 2024; Chen et al., 2023c). Under a $L_2$ accurate score estimation, several works provide the convergence bounds under either smoothness assumptions (Benton et al., 2024; Chen et al., 2022) or bounded second-order moment assumptions (Chen et al., 2023b; Lee et al., 2023). Chen et al. (2023c) provide the first complete estimation theory using ReLU networks without precise estimators. Oko et al. (2023) achieve nearly minimax optimal estimation rates for total variation and Wasserstein distances. Meanwhile, Dou et al. (2024b) define exact minimax optimality using kernel functions without characterizing the network architectures. In the realm of diffusion transformers, Hu et al. (2024) introduces the first complete estimation theory. Jiao et al. (2024a;b) demonstrate theoretical convergence for latent DiTs using ODE-based and Schrödinger bridge diffusion models.[3]

Our paper advances the foundational works of Fu et al. (2024b); Oko et al. (2023); Hu et al. (2024). We adopt the Hölder smooth data distribution assumption[4], a more practical approach than the bounded support assumption in (Oko et al., 2023). Unlike the simple ReLU networks in (Fu et al., 2024b), we provide a complete approximation and estimation analysis for conditional DiTs and establish their exact minimax optimality. Furthermore, while Hu et al. (2024) analyze DiTs, their estimation upper bounds are suboptimal. We refine this by avoiding the substantial double exponential term $2^{(1/\epsilon)^{2L}}$ reported by Hu et al. (2024, Remark 3.4) and present a provably tight, minimax optimal estimation. Recent works on minimax optimality of diffusion model such as (Tang et al., 2024; Fu

---

[3]Of independent interest, many works investigate the convergence rates of diffusion models under various score and data smoothness assumptions or with different samplers. Please see (Chen et al., 2025; Li et al., 2024a;b;c; Potaptchik et al., 2024; Wu et al., 2024c; Liang et al., 2024b;a; Gatmiry et al., 2024; Gu et al., 2024; Guo et al., 2023; Chen et al., 2024b; 2023b; 2022; Lee et al., 2023; 2022) and references therein.

[4]Recent work by Havrilla and Liao (2024) examines the generalization and approximation of transformers under Hölder smoothness and low-dimensional subspace assumptions.

et al., 2024b) demonstrate minimax-optimal under the total variation for $\mathrm{R}e\mathrm{LU}$ network. Additionally, Tang et al. (2024) extend the analysis to a more general setting by considering different smoothness levels for the data $x$ and condition $y$.

## C.2  BROADER IMPACT

This theoretical work aims to shed light on the foundations of generative diffusion models and is not expected to have negative social impacts.

## C.3  LIMITATIONS

Although our study provides a complete theoretical analysis of the conditional DiTs and establishes the minimax optimality of the unconditional DiT, we acknowledge three main limitations:

- The minimax optimality of conditional DiT remains not clear.

- We did not explore other architectures such as "adaptive layer norm" and "cross-attention" DiT. A potential direction is by establishing the universal approximation capacity of the transformer with cross-attention mechanisms.

- Although we achieve a better bound for the latent conditional DiT under the Lipschitz assumption than under the Hölder assumption, we do not show the minimax optimality under the Lipschitz assumption.

We leave these for future work.

Furthermore, there are limitations regarding the Hölder smooth data assumptions in Assumption 3.1 and Assumption 3.2. Our results in Section 3 and Appendix A depend on the Hölder smooth data assumptions. However, it is challenging to measure the smoothness of a given dataset (e.g., CIFAR10), because it requires knowledge of the dataset's exact distribution. Conversely, it is feasible to create a dataset with a predefined level of smoothness. To illustrate this, we provide two examples.

- Diffusion Models in Image Generation: When modeling conditional distributions of images given attributes (e.g., generating images based on class labels), these assumptions hold if the data distribution around these attributes is smooth and decays. In diffusion-based generative models, the data distribution often decays smoothly in high-dimensional space. The assumption that the density function decays exponentially reflects the natural behavior of image data, where pixels or features far from a central region or manifold are less likely. This is commonly observed in images with blank boundaries.

- Physical Systems with Gaussian-Like Decay: This applies to cases where the spatial distribution of a physical quantity, such as temperature, is smooth and governed by diffusion equations with exponential decay. In physics-based diffusion models, like those simulating the spread of particles or heat in a medium (e.g., stars in galaxies for astrophysics applications), the conditional density typically decays exponentially with distance from a central region.

# D    LATENT CONDITIONAL DiT WITH HÖLDER ASSUMPTION

In this section, we elaborate more on Appendix A — conditional DiTs under (i) low-dimensional linear latent (subspace) and (ii) Hölder data assumptions.

We extend the results on approximation and estimation of DiT from Section 3 by considering the latent conditional DiTs. Latent DiTs enables efficient data generation from latent space and therefore scales better in terms of spatial dimensionality (Rombach et al., 2022). Specifically, we assume the raw input $x \in \mathbb{R}^{d_x}$ has an intrinsic lower-dimensional representation in a $d_0$-dimensional subspace, where $d_0 \leq d_x$. This setting is common in both empirical (Peebles and Xie, 2023; Rombach et al., 2022) and theoretical studies (Hu et al., 2024; Chen et al., 2023c).

**Organization.** We present the statistical results under Hölder data smooth Assumptions 3.1 and 3.2 and state the results in Theorem D.1, Theorem D.2, Theorem D.3, and Theorem D.4, respectively. Appendix D.1 discusses score approximation. Appendix D.2 discusses score estimation. Appendix D.3 discusses distribution estimation. The proofs in this section primarily follow Appendices H and I.

Let $d_0$ denote the latent dimension. We summarize the key points of this section as follows:

**K1. Low-Dimensional Subspace Space Data Assumption.** We consider the setting that latent representation lives in a "Low-Dimensional Subspace" under Assumption A.1, following (Hu et al., 2024; Chen et al., 2023c).

> **Assumption D.1** (Low-Dimensional Linear Latent Space (Assumption A.1 Restated)). Data point $x = Uh$, where $U \in \mathbb{R}^{d_x \times d_0}$ is an unknown matrix with orthonormal columns. The latent variable $h \in \mathbb{R}^{d_0}$ follows a distribution $P_h$ with a density function $p_h$.

> For raw data $x \in \mathbb{R}^{d_x}$, we utilize linear encoder $W_U^\top \in \mathbb{R}^{d_0 \times d_x}$ and decoder $W_U \in \mathbb{R}^{d_x \times d_0}$ to convert the raw $x \in \mathbb{R}^{d_x}$ and latent $h \in \mathbb{R}^{d_0}$ data representations. Importantly, $x = Uh$ with $U \in \mathbb{R}^{d_x \times d_0}$ by Assumption A.1.

> For each input $x \in \mathbb{R}^{d_x}$ and corresponding label $y \in \mathbb{R}^{d_y}$, we use a transformer network to obtain a score estimator $s_W \in \mathbb{R}^{d_x}$. To utilize the transformer network as the score estimator, we introduce reshape layer to convert vector input $h \in \mathbb{R}^{d_0}$ to matrix (sequence) input $H \in \mathbb{R}^{\tilde{d} \times \tilde{L}}$. Specifically, the reshape layer in the network Figure 2 is defined as $\tilde{R}(\cdot) : \mathbb{R}^{d_0} \to \mathbb{R}^{\tilde{d} \times \tilde{L}}$ and its reverse $\tilde{R}^{-1}(\cdot) : \mathbb{R}^{\tilde{d} \times \tilde{L}} \to \mathbb{R}^{d_0}$, where $d_0 \leq d_x$, $\tilde{d} \leq d$, and $\tilde{L} \leq L$.

> We remark that the "low-dimensional data" assumption leads to tighter approximation rates than those of Sections 3.1 and 3.2 and estimation errors due to $d_0 \leq d_x$ (Theorems D.1 and D.2).

**K2. Hölder Smooth Assumption.** For approximation and estimation results for latent conditional DiTs (Theorems D.1 to D.4), we study the cases under both the generic and strong Hölder smoothness assumptions on latent representation $h \in \mathbb{R}^{d_0}$. Specifically, we assume the "latent" data is $\beta_0$-Hölder smooth with radius $B_0$ following Assumptions 3.1 and 3.2. We extend both approximation and estimation results from Section 3 to latent diffusion and establish the minimax optimality of latent conditional DiTs.

> **Assumption D.2** (Generic Hölder Smooth Data (Assumption 3.1 Restated)). The conditional density function $p_0^h(h_0|y)$ is defined on the domain $\mathbb{R}^{d_0} \times [0,1]^{d_y}$ and belongs to Hölder ball of radius $B_0 > 0$ for Hölder index $\beta_0 > 0$, denoted by $p_0^h(h_0|y) \in \mathcal{H}^{\beta_0}(\mathbb{R}^{d_0} \times [0,1]^{d_y}, B_0)$ (see Definition 3.1 for precise definition.) Also, for any $y \in [0,1]^{d_y}$, there exist positive constants $C_1, C_2$ such that $p_0^h(h_0|y) \leq C_1 \exp\left(-C_2 \|h_0\|_2^2 / 2\right)$.

> **Assumption D.3** (Stronger Hölder Smooth Data (Assumption 3.2 Restated)). Let function $f \in \mathcal{H}^{\beta_0}(\mathbb{R}^{d_0} \times [0,1]^{d_y}, B_0)$. Given a constant radius $B_0$, positive constants $C$ and $C_2$, we assume the conditional density function $p(h_0|y) = \exp\left(-C_2 \|h_0\|_2^2 / 2\right) \cdot f(h_0, y)$ and $f(h_0, y) \geq C$ for all $(h_0, y) \in \mathbb{R}^{d_0} \times [0,1]^{d_y}$.

**K3. Latent Score Network.** Under low-dimensional data assumption, we decompose the score function following (Hu et al., 2024; Chen et al., 2023c) (see Lemma D.1):

$$\nabla \log p_t(x|y) = U(\underbrace{\sigma_t^2 \nabla \log p_t^h(U^\top x|y) + U^\top x}_{:=q(U^\top x, y, t): \ \mathbb{R}^{d_0} \times [t_0, T] \ \rightarrow \ \mathbb{R}^{d_0}})/\sigma_t^2 - \underbrace{x/\sigma_t^2}_{\text{residual connection}} . \tag{D.1}$$

Based on this decomposition, we construct the model architecture in Figure 2. The network detail for approximate (D.1) are as follow: a transformer $g_\mathcal{T}(W_U^\top x, y, t) \in \mathcal{T}^{h,s,r}$ to approximate $q(U^\top x, y, t)$, a latent encoder $W_U^\top \in \mathbb{R}^{d_0 \times d_x}$ and decoder $W_U \in \mathbb{R}^{d_x \times d_0}$ to approximate $U^\top \in \mathbb{R}^{d_0 \times d_x}$ and $U \in \mathbb{R}^{d_x \times d_0}$, and a residual connection to approximate $-x/\sigma_t^2$.

We adopt the following transformer network class of one-layer single-head self-attention

$$\mathcal{T}_{\widetilde{R}}^{h,s,r} = \left\{ s_W(x, y, t) = \frac{1}{\sigma_t^2} W_U g_\mathcal{T}\left(W_U^\top x, y, t\right) - \underbrace{\frac{1}{\sigma_t^2} x}_{\text{residual connection}} \right\}, \tag{D.2}$$

where $g_\mathcal{T} \in \mathcal{T}^{h,s,r} = \{f_2^{\text{FF}} \circ f^{h,s,r} : \mathbb{R}^{\widetilde{d} \times \widetilde{L}} \rightarrow \mathbb{R}^{\widetilde{d} \times \widetilde{L}}\}$.

Let $h := W_U^\top x \in \mathbb{R}^{d_0}$ and $\bar{h} := U^\top x \in \mathbb{R}^{d_0}$ be the estimated and ground truth (according to Assumption A.1) latent representations, respectively. Here we construct a network $s_W(x, y, t)$ to approximate the score function in (D.1) (see Figure 2 for network illustration).

In Section 3, we give approximation results for score function $\nabla \log p_t(x|y)$ using conditional DiTs with a one-layer single-head self-attention transformer. We use the similar transformer architecture to approximate latent score function $\nabla \log p_t^h(\bar{h}|y)$. Specifically, the density function takes the form

$$p_t^h(\bar{h}|y) = \int \psi_t(\bar{h}|h) p_h(h|y) \mathrm{d}h,$$

where $\psi_t(\cdot|h)$ is Gaussian $N(\beta_t h, \sigma_t^2 I_{d_0})$ with $\beta_t = e^{-t/2}$, and $\sigma_t^2 = 1 - e^{-t}$.

Based on the latent network construction in (K3), we employ the same techniques presented in Section 3 for score function approximation and estimation. We restate for completeness. First, we decompose the conditional score function $\nabla \log p_t^h(\bar{h}|y)$ as following:

$$\nabla \log p_t^h\left(\bar{h}|y\right) = \frac{\nabla p_t^h\left(\bar{h}|y\right)}{p_t^h\left(\bar{h}|y\right)}. \tag{D.3}$$

By the definition of Gaussian kernel, we have

$$p_t^h\left(\bar{h}|y\right) = \int_{\mathbb{R}^{d_0}} (2\pi\sigma_t^2)^{-d_x/2} \underbrace{p_h(h|y)}_{\approx k_1\text{-order Taylor polynomial}} \exp\left(-\underbrace{\frac{\left\|\beta_t h - \bar{h}\right\|_2^2}{2\sigma_t^2}}_{\approx k_2\text{-order Taylor polynomial}}\right) \mathrm{d}h.$$

Similar to Section 3, our strategy is to expand above term-by-term with $k_1$- and $k_2$-order Taylor polynomials for fine-grained characterizations.

**Remark D.1.** Here in the latent density function, we have $(2\pi\sigma_t^2)^{-d_x/2}$ instead of $(2\pi\sigma_t^2)^{-d_0/2}$. However, the additional $(2\pi\sigma_t^2)^{-(d_x-d_0)/2}$ term does not affect the application of Section 3 into latent diffusion approximation.

Based on the low-dimensional data structure assumption, we have the following score decomposition terms: on-support score $s_+(U^\top x, y, t)$ and orthogonal score $s_-(x, y, t)$.

**Lemma D.1** (Score Decomposition, Lemma 1 of (Chen et al., 2023c)). Let data $x = Uh$ follow Assumption A.1. The decomposition of score function $\nabla \log p_t(x)$ is

$$\nabla \log p_t(x) = \underbrace{U \nabla \log p_t^h(\overline{h}|y)}_{s_+(\overline{h},y,t)} \underbrace{- (I_D - UU^\top) x / \sigma_t^2}_{s_-(x,t)}, \quad \overline{h} = U^\top x, \qquad \text{(D.4)}$$

where $p_t^h(\overline{h}|y) := \int \psi_t(\overline{h}|h) p_h(h|y) \, \mathrm{d}h$, $\psi_t(\cdot|h)$ is the Gaussian density function of $N(\beta_t h, \sigma_t^2 I_{d_0})$, $\beta_t = e^{-t/2}$ and $\sigma_t^2 = 1 - e^{-t}$.

Following the proof strategy of conditional DiTs in Appendices H and I with differences highlighted in (K1), (K2), and the latent network in (K3). To derive the approximation and estimation under generic and stronger Hölder assumptions results in Theorems 3.1 to 3.4 for data under low-dimensional data assumption, we just need to replace the input dimension $d$, $L$ to $\widetilde{d}$ and $\widetilde{L}$, and the input dimension $d_x$ with $d_0$, and consider the $\beta_0$-Hölder smoothness assumption on latent data.

To begin, we clarify the relation between initial data admits to $p(x|y) \in \mathcal{H}^\beta(\mathbb{R}^{d_x} \times [0,1]^{d_y}, B)$, and under linear transformed data Assumption A.1 admits to $p(\overline{h}|y) \in \mathcal{H}^{\beta_0}(\mathbb{R}^{d_0} \times [0,1]^{d_y}, B_0)$ where $\beta_0 = \beta$ and $B_0 \le \widetilde{C} B$ by Lemma D.2.

**Lemma D.2** (Transformation of Stronger Hölder Smooth Data Distribution under Linear Mapping). Let $f \in H^\beta(\mathbb{R}^{d_x} \times [0,1]^{d_y}, B)$ satisfy $f(x,y) \ge C > 0$ for all $(x,y) \in \mathbb{R}^{d_x} \times [0,1]^{d_y}$. Consider the conditional density function:

$$p(x|y) = f(x,y) \exp\left(-\frac{C_2}{2} \|x\|_2^2\right).$$

Suppose the data undergo the linear transformation $x = Uh$, where $U \in \mathbb{R}^{d_x \times d_0}$ has orthonormal columns ($U^\top U = I_{d_0}$) and $f_0(h|y) = f(Uh|y)$. The transformed density $p(h|y)$ becomes:

$$p(h|y) = f(Uh,y) \exp\left(-\frac{C_2}{2} \|h\|_2^2\right).$$

The following condition holds for Hölder smooth data that undergoes linear transformation: $f_0 \in H^\beta(\mathbb{R}^{d_x} \times [0,1]^{d_y}, B_0)$ with $B_0 \le \widetilde{C} B$, where $\widetilde{C} = \max\{C', C''\}$.

*Proof.* First, we compute the partial derivative of the transformed function $f_0(h|y) := f(Uh|y)$. From the definition of Hölder space Definition 3.1, and let $\alpha = (\alpha_h, \alpha_y)$ where $\alpha_h + \alpha_y \le k_1$. We compute the partial derivative up to the order of $k_1$ and show that it is bounded by some $C'$, that is

$$\partial_h^{\alpha_h} \partial_y^{\alpha_y} p(h|y) = \partial_h^{\alpha_h} \partial_y^{\alpha_y} \left[ f(Uh,y) \exp\left(-\frac{C_2}{2} \|h\|_2^2\right) \right]$$

$$= \sum_{\alpha \le \nu} \binom{\alpha}{\mu} \left(\partial_h^{\alpha_\mu} f(Uh,y)\right) \left(\partial_h^{(\alpha-\nu)} \exp\left(-\frac{C_2}{2} \|h\|_2^2\right)\right). \qquad \text{(By product rule)}$$

From the relation $\partial_h^{\alpha_h} f(Uh,y) = U^{\alpha_h} \partial_x^{\alpha_h} f(Uh,y)$ where $U^{\alpha_h}$ is the product of $U$ entries correspond to $\alpha_h$. Therefore, $\left\|\partial_h^{\alpha_h} \partial_y^{\alpha_y} f_0(h|y)\right\| \le C' B$ for some $C'$ depends on $U$ and $\alpha_h$. Since $f$ satisfied Hölder condition and the mapping $h \mapsto Uh$ is linear, for Hölder condition $|\alpha_h| + |\alpha_y| = k_1$ there exist $C''$ such that

$$\frac{\left|\partial_h^{\alpha_h} \partial_y^{\alpha_y} f_0(h|y) - \partial_h^{\alpha_h} \partial_y^{\alpha_y} f_0(h'|y')\right|}{\|(h,y) - (h',y')\|_\infty^\gamma} \le C'' B.$$

The bounded partial derivate up to order $k_1$ satisfied Hölder condition.

This completes the proof. $\qquad \square$

### D.1 SCORE APPROXIMATION

We present the approximation rate of latent score function under generic Hölder and stronger Hölder data assumption in Theorems D.1 and D.2, respectively.

**Theorem D.1** (Latent Conditional DiT Score Approximation, Formal Version of Theorem A.1). Assume Assumption 3.1 and $d_x = \Omega(\frac{\log N}{\log \log N})$. For any precision $0 < \epsilon < 1$ and smoothness $\beta_0 > 0$, let $\epsilon \le \mathcal{O}(N^{-\beta_0})$ for some $N \in \mathbb{N}$. Let $C_\alpha, C_\sigma > 0$ be some absolute constant. For any $y \in [0,1]^{d_y}$ and $t \in [N^{-C_\sigma}, C_\alpha \log N]$, there exists a $\mathcal{T}_{\text{score}}(x,y,t) \in \mathcal{T}_{\widetilde{R}}^{h,s,r}$ such that

$$\int_{\mathbb{R}^{d_0}} \left\| \mathcal{T}_{\text{score}}(\overline{h}, y, t) - \nabla \log p_t^h(\overline{h}|y) \right\|_2^2 \cdot p_t^h(\overline{h}|y) \mathrm{d}\overline{h} = \mathcal{O}\left( \frac{B_0^2}{\sigma_t^4} \cdot N^{-\beta_0} \cdot (\log N)^{d_0 + \frac{\beta_0}{2} + 1} \right).$$

*Proof Sketch.* The proof closely follows Theorem 3.1, with differences highlighted in (K1) and (K2). Specifically, we replace $d$, $L$ and $d_x$ with $\widetilde{d}$, $\widetilde{L}$ and $d_0$ respectively. Moreover, we consider the $\beta_0$-Hölder smoothness assumption on latent data detailed in (K2). This completes the proof.

Please see Appendix H for a detailed proof. □

**Theorem D.2** (Latent Conditional DiT Score Approximation under Stronger Hölder Assumption, Formal Version of Theorem A.1). Assume Assumption 3.2. For any precision $0 < \epsilon < 1$ and smoothness $\beta_0 > 0$, let $\epsilon \le \mathcal{O}(N^{-\beta_0})$ for some $N \in \mathbb{N}$. Let $C_\alpha, C_\sigma > 0$ be some absolute constant. For any $y \in [0,1]^{d_y}$ and $t \in [N^{-C_\sigma}, C_\alpha \log N]$, there exists a $\mathcal{T}_{\text{score}}(x,y,t) \in \mathcal{T}_{\widetilde{R}}^{h,s,r}$ such that

$$\int_{\mathbb{R}^{d_0}} \left\| \mathcal{T}_{\text{score}}(x,y,t)(\overline{h}, y, t) - \nabla \log p_t^h(\overline{h}|y) \right\|_2^2 \cdot p_t^h(\overline{h}|y) \mathrm{d}\overline{h} = \mathcal{O}\left( \frac{B_0^2}{\sigma_t^2} \cdot N^{-2\beta_0} \cdot (\log N)^{\beta_0 + 1} \right).$$

*Proof Sketch.* The proof closely follows Theorem I.1, with differences highlighted in (K1) and (K2). We replace the input dimension $d$, $L$ to $\widetilde{d}$ and $\widetilde{L}$, and the input dimension $d_x$ with $d_0$ in Theorem I.1 with the $\beta_0$-Hölder smoothness assumption on latent data detailed in (K2). This completes the proof.

Please see Appendix I for a detailed proof. □

**Remark D.2** (Score Approximation for Low-Dimensional Linear Latent Space). With the assumption of low-dimensional latent space Assumption A.1, Theorems D.1 and D.2 provide better approximation rates than Theorems 3.1 and 3.2 under Hölder smooth assumptions in Assumptions 3.1 and 3.2, respectively. Specifically, from Lemma D.2 we have $\beta_0 = \beta$ and $B_0 \lesssim B$. Therefore, Theorems D.1 and D.2 deliver better approximation error over Theorem 3.1, where $d_0 \le d_x$.

### D.2 SCORE ESTIMATION

In this section, we provide the extended results for Section 3.3 on score estimation with the estimator $\mathcal{T}_{\text{score}}$. We state the main results under Hölder data assumptions in Theorem D.3.

**Theorem D.3** (Conditional Score Estimation with Transformer). Consider $y \in [0,1]^{d_y}$ and $t \in [t_0, T]$ with $t_0 = N^{-C_\sigma}$ and $T = C_\alpha \log N$, where $C_\sigma, C_\alpha > 0$ are absolute constants such that $t_0 < 1$.

• Assume Assumption 3.1. Then, it holds

$$\mathbb{E}_{\{x_i, y_i\}_{i=1}^n} [\mathcal{R}(\widehat{s})] = \mathcal{O}\left( n^{-\frac{\beta_0}{\widetilde{\nu}_1 + C_\sigma + 3\beta_0}} \cdot (\log n)^{\widetilde{\nu}_2 + 2} \right).$$

• Assume Assumption 3.2. Then, it holds

$$\mathbb{E}_{\{x_i, y_i\}_{i=1}^n} [\mathcal{R}(\widehat{s})] = \mathcal{O}\left( n^{-\frac{2\beta_0}{\widetilde{\nu}_3 + 6\beta_0}} \cdot (\log n)^{\max\{13, \beta_0 + 2\}} \right).$$

*Proof Sketch.* The proof closely follows Theorem 3.3, with differences highlighted in (K1) and (K2). By replacing the input dimension $d$, $L$ to $\widetilde{d}$ and $\widetilde{L}$, and the input dimension $d_x$ with $d_0$ in Theorem 3.3, and under the the $\beta_0$-Hölder smoothness assumption on latent data detailed in (K2), the proof is complete. Please see Appendix J.2 for a detailed proof. □

**Remark D.3** (Comparing Score Estimation in Theorems 3.3 and D.3)**.** Invoking Lemma D.2 where $\beta_0 = \beta$ and $B_0 \lesssim B$ the sample complexity in Theorem D.3 improves Theorem 3.3.

### D.3 DISTRIBUTION ESTIMATION

In this section, we provide the extended results for Section 3.3 on distribution estimation with the estimator $\mathcal{T}_{\text{score}}$. We restate the main results under Hölder data assumptions in Theorem D.3.

**Theorem D.4** (Conditional Distribution Estimation, Theorem A.2 Restated)**.** For $y \in [0,1]^{d_y}$, let $\widehat{P}_{t_0}(\cdot|y)$ denote *estimated* conditional distributions at $t_0$. Recall that $P_0(\cdot|y)$ is the conditional distribution of initial data $x_0$ given $y$ and $t \in [N^{-C_\sigma}, C_\alpha \log N]$. Assume $\text{KL}\left(P_0(\cdot|y) \mid N(0,I)\right) \leq c$ for some constant $c < \infty$.

- Assume Assumption 3.1. Then, it holds

$$
\mathbb{E}_{\{x_i, y_i\}_{i=1}^n}\left[\mathbb{E}_y\left[\text{TV}\left(\widehat{P}_{t_0}(\cdot|y), P_0(\cdot|y)\right)\right]\right] = \mathcal{O}\left(n^{-\widetilde{\omega}}(\log n)^{\frac{\widetilde{\nu}_2}{2} + \frac{3}{2}}\right),
$$

where $\widetilde{\omega} = \min\left\{\frac{C_\sigma}{2(\widetilde{\nu}_1 + C_\sigma + 3\beta_0)}, \frac{C_\alpha}{\widetilde{\nu}_1 + C_\sigma + 3\beta_0}, \frac{\beta_0}{2(\widetilde{\nu}_1 + C_\sigma + 3\beta_0)}\right\}$.

- Assume Assumption 3.2. Then, it holds

$$
\mathbb{E}_{\{x_i, y_i\}_{i=1}^n}\left[\mathbb{E}_y\left[\text{TV}\left(\widehat{P}_{t_0}(\cdot|y), P_0(\cdot|y)\right)\right]\right] = \mathcal{O}\left(n^{-\widetilde{\phi}}(\log n)^{\frac{\widetilde{\nu}_2}{2} + \frac{3}{2}}\right),
$$

where $\widetilde{\phi} = \min\left\{\frac{C_\sigma}{2(\widetilde{\nu}_1 + 6\beta_0)}, \frac{C_\alpha}{\widetilde{\nu}_1 + 6\beta_0}, \frac{\beta_0}{\widetilde{\nu}_1 + 6\beta_0}\right\}$.

*Proof.* The proof closely follows Theorem 3.4, with differences highlighted in (K1) and (K2). By replacing the input dimension $d$, $L$ to $\widetilde{d}$ and $\widetilde{L}$, and the input dimension $d_x$ with $d_0$ in Theorem 3.4, and under the the $\beta_0$-Hölder smoothness assumption on latent data detailed in (K2), the proof is complete. Please see Appendix J.4 for a detailed proof. □

Next, we present the distribution estimation result for low-dimensional input data.

# E  LATENT CONDITIONAL DiT WITH LIPSCHITZ ASSUMPTION

In this section, we apply our techniques to the setting of (Hu et al., 2024) on DiT approximation and estimation theory — under (i) low-dimensional linear latent (subspace), (ii) Lipschitz score and (iii) sub-Gaussian data assumptions. Specifically, we extend their work by using the one-layer self-attention transformer universal approximation framework introduced in Appendix G.1.

Compared to (Hu et al., 2024), we consider classifier-free conditional DiTs, providing a holistic view of the theoretical guarantees under various assumptions. In particular, our sample complexity bounds avoid the gigantic double exponential term $2^{(1/\epsilon)^{2L}}$ reported in (Hu et al., 2024). We adopt the following three assumptions considered by Hu et al. (2024):

**(A1) Low-Dimensional Linear Latent Space Data Assumption.**

> **Assumption E.1** (Low-Dimensional Linear Latent Space, Assumption A.1 Restated). Data point $x = Uh$, where $U \in \mathbb{R}^{d_x \times d_0}$ is an unknown matrix with orthonormal columns. The latent variable $h \in \mathbb{R}^{d_0}$ follows a distribution $P_h$ with a density function $p_h$.

Under this data assumption, Chen et al. (2023a) show that the latent score function endows a neat decomposition into on-support $s_+$ and orthogonal $s_-$ terms (see Lemma D.1).

> **Lemma E.1** (Score Decomposition, Lemma 1 of (Chen et al., 2023c), Lemma D.1 Restated). Let data $x = Uh$ follow Assumption A.1. The decomposition of score function $\nabla \log p_t(x)$ is
>
> $$\nabla \log p_t(x) = \underbrace{U \nabla \log p_t^h(\overline{h}|y)}_{s_+(\overline{h},y,t)} \underbrace{- \left(I_D - UU^\top\right) x/\sigma_t^2}_{s_-(x,t)}, \ \overline{h} = U^\top x, \qquad \text{(E.1)}$$
>
> where $p_t^h\left(\overline{h}|y\right) := \int \psi_t(\overline{h}|h)p_h\left(h|y\right) \mathrm{d}h$, $\psi_t(\cdot|h)$ is the Gaussian density function of $N(\beta_t h, \sigma_t^2 I_{d_0})$, $\beta_t = e^{-t/2}$ and $\sigma_t^2 = 1 - e^{-t}$.

**(A2) Lipschitz Score Assumption.** We assume the on-support score function $s_+(\overline{h}, y, t)$ to be $L_{s_+}$-Lipschitz for any $\overline{h}$ and $y$.

> **Assumption E.2** ($L_{s_+}$-Lipschitz of $s_+(\overline{h}, y, t)$). The on-support score function $s_+(\overline{h}, y, t)$ is $L_{s_+}$-Lipschitz with respect to any $\overline{h} \in \mathbb{R}^{d_0}$ and $y \in \mathbb{R}^{d_y}$ for any $t \in [0, T]$. i.e., there exist a constant $L_{s_+}$, such that for any $\overline{h}$, $y$ and $\overline{h}'$, $y'$:
>
> $$\|s_+(\overline{h}, y, t) - s_+(\overline{h}', y', t)\|_2 \le L_{s_+}\|\overline{h} - \overline{h}'\|_2 + L_{s_+}\|y - y'\|_2.$$

**(A3) Light Tail Data Assumption.**

> **Assumption E.3** (Tail Behavior of $P_h$). The density function $p_h > 0$ is twice continuously differentiable. Moreover, there exist positive constants $A_0, A_1, A_2$ such that when $\|h\|_2 \ge A_0$, the density function $p_h\left(h|y\right) \le (2\pi)^{-d_0/2}A_1\exp(-A_2\|h\|_2^2/2)$.

We note that, the assumptions (A1) and (A3) are on data, and (A2) are on the score function. Notably, (A2) on the smoothness of score function is stronger than Hölder data smoothness assumptions considered in Section 3 and Appendix A.

**Organization.**  We study latent conditional DiTs under low-dimensional data Assumption E.1, Lipschitz smoothness Assumption E.2, and tail behavior of $P_h$ Assumption E.3 and states the results in Appendices E.1 to E.3, respectively. Appendix E.1 discusses score approximation. Appendix E.2 discusses score estimation. Appendix E.3 discusses distribution estimation. The proof in this section provided in Appendices E.4 to E.6. The proof strategy in this section follows (Hu et al., 2024).

Here we summarize the key settings of this section:

**S1. Lipschitz Smooth Assumption and Tail Behavior.** Following (Hu et al., 2024), we introduce two assumptions on Lipschitz smoothness for on-support score function $s_+$ and tail behavior

of $P_h$ in Assumptions E.2 and E.3, respectively. The on-support score function is defined as $s_+(U^\top x, y, t) = U\nabla \log p_t^h\left(U^\top x|y\right)$ (see Lemma D.1 for score decomposition).

**S2. Low-Dimensional Space.** We consider the setting of latent representation that is the data lives in a "Low-Dimensional Subspace" under Assumption A.1, following (Hu et al., 2024; Chen et al., 2023c). The raw data $x \in \mathbb{R}^{d_x}$ is supported by latent $h \in \mathbb{R}^{d_0}$ where $d_0 \leq d_x$.

**S3. Transformer Network.** We follow standard setting of "in-context" conditional DiTs by Peebles and Xie (2023) on latent representation. Recall Appendix A for the transformer network setting.

### E.1 SCORE APPROXIMATION

For completeness, we introduce the proofs from (Hu et al., 2024) for score approximation of the conditional latent diffusion model.

Instead of assuming Hölder smoothness of the initial conditional data distribution as in Appendix A, we use stricter assumptions on the latent density function. To be specific, we directly approximate the on-support latent score function, instead of approximating the denominator and nominator separately. From the score decomposition in (A.1), we define the on-support score function $s_+$ as following:

$$
\begin{aligned}
s_+(U^\top x, y, t) &= U \int \frac{\nabla_{\bar{h}}\psi_t(\bar{h}|h)p_h\left(h|y\right)}{\int \psi_t(\bar{h}|h')p_{h'}\left(h'|y\right)\mathrm{d}h'}\mathrm{d}h \\
&= U\nabla \log p_t^h\left(U^\top x|y\right).
\end{aligned}
\tag{E.2}
$$

Here we require two assumptions following the proof of (Hu et al., 2024) on tail behavior of density function and Lipschitz continuous for on-support score function. Assumption E.3 is the analogy of Assumption 3.1 for assuming the tail behavior of the density function. On the other hand, Assumption E.2 further assume the on-support score function $s_+$ to be $L_{s_+}$-Lipshitz. Note that this assumption is stricter than Assumption 3.1 since we make the Lipschitz assumption directly on the score function instead of on the latent density function.

**Theorem E.1** (Latent Score Approximation of Conditional DiT, modified from Theorem 3.1 in Hu et al. (2024)). *For any approximation error $\epsilon > 0$ and any data distribution $P_0$ under Assumptions A.1, E.2 and E.3, there exists a DiT score network $\mathcal{T}_{\mathrm{score}}(\bar{h}, y, t) \in \mathcal{T}_{\widetilde{R}}^{h,s,r}$ where $W = \{W_U, \mathcal{T}_{\mathrm{score}}\}$, such that for any $t \in [t_0, T]$, we have:*

$$
\|\mathcal{T}_{\mathrm{score}}(\cdot, t) - \nabla \log p_t(\cdot)\|_{L^2(P_t)} \leq \epsilon \cdot \sqrt{d_0 + d_y}/\sigma_t^2,
$$

*where $\sigma_t^2 = 1 - e^{-t}$ and the parameter bounds in the transformer network class satisfy*

$$
\|W_Q\|_2 = \|W_K\|_2 = \mathcal{O}\left(\widetilde{d} \cdot \epsilon^{-(\frac{1}{d}+2\widetilde{L})}(\log \widetilde{L})^{\frac{1}{2}}\right);
$$

$$
\|W_Q\|_{2,\infty} = \|W_K\|_{2,\infty} = \mathcal{O}\left(\widetilde{d}^{\frac{3}{2}} \cdot \epsilon^{-(\frac{1}{d}+2\widetilde{L})}(\log \widetilde{L})^{\frac{1}{2}}\right);
$$

$$
\|W_O\|_2 = \mathcal{O}\left(\widetilde{d}^{\frac{1}{2}}\epsilon^{\frac{1}{d}}\right); \|W_O\|_{2,\infty} = \mathcal{O}\left(\epsilon^{\frac{1}{d}}\right);
$$

$$
\|W_V\|_2 = \mathcal{O}(\widetilde{d}^{\frac{1}{2}}); \|W_V\|_{2,\infty} = \mathcal{O}(\widetilde{d});
$$

$$
\|W_1\|_2 = \mathcal{O}\left(\widetilde{d}\epsilon^{-\frac{1}{d}}\right), \|W_1\|_{2,\infty} = \mathcal{O}\left(\widetilde{d}^{\frac{1}{2}}\epsilon^{-\frac{1}{d}}\right);
$$

$$
\|W_2\|_2 = \mathcal{O}\left(\widetilde{d}\epsilon^{-\frac{1}{d}}\right); \|W_2\|_{2,\infty} = \mathcal{O}\left(\widetilde{d}^{\frac{1}{2}}\epsilon^{-\frac{1}{d}}\right);
$$

$$
\left\|E^\top\right\|_{2,\infty} = \mathcal{O}\left(\widetilde{d}^{\frac{1}{2}}\widetilde{L}^{\frac{3}{2}}\right).
$$

*Proof.* Please see Appendix E.4 for a detailed proof. □

## E.2 SCORE ESTIMATION

**Theorem E.2** (Score Estimation of Latent DiT). Under the Assumptions E.1 to E.3, we choose the score network $\mathcal{T}_{\text{score}}(x, y, t) \in \mathcal{T}_{\widetilde{R}}^{h,s,r}$ from Theorem E.1 using $\epsilon \in (0, 1)$ and $\widetilde{L} > 1$. With probability $1 - 1/\text{poly}(n)$, we have

$$\frac{1}{T - t_0} \int_{t_0}^{T} \|\mathcal{T}_{\text{score}}(\cdot, t) - \nabla \log p_t(\cdot)\|_{L^2(P_t)} \mathrm{d}t = \widetilde{\mathcal{O}}\left(\frac{1}{t_0^2} n^{\frac{-3}{2(1+3/\widetilde{d}+4\widetilde{L})}} \log^3 \widetilde{L} \log^3 n\right),$$

where $\widetilde{\mathcal{O}}$ hides the factor about $d_x, d_y, d_0, \widetilde{d}$ and $L_{s_+}$.

*Proof.* Please see Appendix E.5 for a detailed proof. □

## E.3 DISTRIBUTION ESTIMATION

In practice, DiTs generate data using discretized version with step size $\mu$. Let $\widehat{P}_{t_0}$ be the distribution generated by $\mathcal{T}_{\text{score}}(x, y, t)$ in Theorem E.2. Let $P_{t_0}^h$ and $p_{t_0}^h$ be the distribution and density function of on-support latent variable $\overline{h}$ at $t_0$. We have following results for distribution estimation.

**Theorem E.3** (Distribution Estimation of DiT, Modified from Theorem 3 of (Chen et al., 2023c)). Let $T = \mathcal{O}(\log n), t_0 = \mathcal{O}(\min\{c_0, 1/L_{s_+}\})$, where $c_0$ is the minimum eigenvalue of $\mathbb{E}_{P_h}[hh^\top]$. With the estimated DiT score network $\mathcal{T}_{\text{score}}(x, y, t)$ in Theorem E.2, we have the following with probability $1 - 1/\text{poly}(n)$.

(i) The accuracy to recover the subspace $U$ is

$$\|W_U W_U^\top - UU^\top\|_F^2 = \widetilde{\mathcal{O}}\left(\frac{1}{c_0} n^{\frac{-3}{2(1+3/\widetilde{d}+4\widetilde{L})}} \cdot \log^3 n\right). \tag{E.3}$$

(ii) $(W_B U)_\sharp^\top \widehat{P}_{t_0}$ denotes the pushforward distribution. With the conditions $\text{KL}(P_h \| N(0, I_{d_0})) < \infty$, and step size $\mu \leq \xi(n, t_0, L) \cdot t_0^2 / (d_0 \sqrt{\log d_0})$. There exists an orthogonal matrix $U \in \mathbb{R}^{d \times d}$ such that we have the following upper bound for the total variation distance

$$\text{TV}(P_{t_0}^h, (W_B U)_\sharp^\top \widehat{P}_{t_0}) = \widetilde{\mathcal{O}}\left(\frac{1}{t_0 \sqrt{c_0}} n^{\frac{-3}{4(1+3/\widetilde{d}+4\widetilde{L})}} \cdot \log^4 n\right), \tag{E.4}$$

where $\widetilde{\mathcal{O}}$ hides the factor about $d_x, d_0, d$, and $L_{s_+}$.

(iii) For the generated data distribution $\widehat{P}_{t_0}$, the orthogonal pushforward $(I - W_B W_B^\top)_\sharp \widehat{P}_{t_0}$ is $N(0, \Sigma)$, where $\Sigma \preceq at_0 I$ for a constant $a > 0$.

*Proof.* Please see Appendix E.6 for a detailed proof. □

**Remark E.1** (Compare with Existing Work). In (Chen et al., 2023c, Theorem 3), the upper bound for total variation distance with ReLU network is $\widetilde{\mathcal{O}}\left(\sqrt{1/(c_0 t_0)} n^{-1/(d+5)} \log^2 n\right)$. Therefore, for $n \gg 1$, Theorem E.3 gives tighter accuracy if $3d + 11 > 12/\widetilde{d} + 16\widetilde{L}$ where $\widetilde{d} \leq d$ and $\widetilde{L} \leq L$. On the other hand, under similar conditions for $d$ and $L$, Theorem E.3 suggest to achieve similar total variation distance we only require $\sqrt{t_0}$ early stopping time which is beneficial for empirical setting.

## E.4 PROOF OF SCORE APPROXIMATION (THEOREM E.1)

To begin with, we restate some auxiliary lemmas and their proofs here from (Chen et al., 2023c) for later convenience. Note that some of the proofs extend to the latent density function.

**Lemma E.2** (Modified from Lemma 16 in (Chen et al., 2023c)). Consider a probability density function $p_h(h|y) = \exp\left(-C\|h\|_2^2/2\right)$ for $h \in \mathbb{R}^{d_0}$ and constant $C > 0$. Let $r_h > 0$ be a fixed radius. Then it holds

$$\int_{\|h\|_2 > r_h} p_h(h|y)\,\mathrm{d}h \leq \frac{2d_0\pi^{d_0/2}}{C\Gamma(d_0/2+1)} r_h^{d_0-2} \exp\left(-Cr_h^2/2\right),$$

$$\int_{\|h\|_2 > r_h} \|h\|_2^2 p_h(h|y)\,\mathrm{d}h \leq \frac{2d_0\pi^{d_0/2}}{C\Gamma(d_0/2+1)} r_h^{d_0} \exp\left(-Cr_h^2/2\right).$$

**Lemma E.3** (Modified from Lemma 2 in (Chen et al., 2023c)). Suppose Assumption Assumption E.3 holds and $q$ is defined as:

$$q(\bar{h}, y, t) = \int \frac{h\psi_t(\bar{h}|h)\,p_h(h|y)}{\int \psi_t(\bar{h}|h)\,p_h(h|y)\,\mathrm{d}h}\mathrm{d}h, \quad \bar{h} = B^\top x.$$

Given $\epsilon > 0$, with $r_h = c\left(\sqrt{d_0\log(d_0/t_0) + \log(1/\epsilon)}\right)$ for an absolute constant $c$, it holds

$$\left\|q(\bar{h}, y, t)\,\mathbb{1}\{\|\bar{h}\|_2 \geq r_h\}\right\|_{L^2(P_t)} \leq \epsilon, \text{ for } t \in [t_0, T].$$

**Lemma E.4** (Modified from Theorem 1 in (Chen et al., 2023c)). We denote

$$\tau(r_h) = \sup_{t\in[t_0,T]} \sup_{\bar{h}\in[0,r_h]^{d_0}} \sup_{y\in[0,1]^{d_y}} \left\|\frac{\partial}{\partial t}q(\bar{h}, y, t)\right\|_2.$$

With $q(\bar{h}, y, t) = \int h\psi_t(\bar{h}|h)p_h(h|y)/(\int \psi_t(\bar{h}|h)p_h(h|y)\mathrm{d}h)\mathrm{d}h$ and $p_h$ satisfies Assumption E.3, we have a coarse upper bound for $\tau(r_h)$

$$\tau(r_h) = \mathcal{O}\left(\frac{1+\beta_t^2}{\beta_t}\left(L_{s_+} + \frac{1}{\sigma_t^2}\right)\sqrt{d_0}r_h\right) = \mathcal{O}\left(e^{T/2}L_{s_+}r_h\sqrt{d_0}\right).$$

*Proof of Lemma E.4.*

$$\frac{\partial}{\partial t}q(\bar{h}, y, t) = U\int \frac{h\frac{\partial}{\partial t}\psi_t(\bar{h}|h)p_h(h|y)}{\int \psi_t(\bar{h}|h)p_h(h|y)\mathrm{d}h}\mathrm{d}h - U\int \frac{h\psi_t(\bar{h}|h)p_h(h|y)\int \frac{\partial}{\partial t}\psi_t(\bar{h}|h)p_h(h|y)\mathrm{d}h}{\left(\int \psi_t(\bar{h}|h)p_h(h|y)\mathrm{d}h\right)^2}\mathrm{d}h$$

$$= U\int \frac{h\frac{\beta_t}{\sigma_t^2}\left(\|h\|_2^2 - (1+\beta_t^2)h^\top\bar{h} + \beta_t\|\bar{h}\|_2^2\right)\psi_t(\bar{h}|h)p_h(h|y)}{\int \psi_t(\bar{h}|h)p_h(h|y)\mathrm{d}h}\mathrm{d}h$$

$$- U\int \frac{h\psi_t(\bar{h}|h)p_h(h|y)\int \frac{\beta_t}{\sigma_t^2}\left(\|h\|_2^2 - (1+\beta_t^2)h^\top\bar{h} + \beta_t\|\bar{h}\|_2^2\right)\psi_t(\bar{h}|h)p_h(h|y)\mathrm{d}h}{\left(\int \psi_t(\bar{h}|h)p_h(h|y)\mathrm{d}h\right)^2}\mathrm{d}h$$

$$\overset{(i)}{=} \frac{\beta_t}{\sigma_t^2}U\left[\mathbb{E}_{P_h}\left[h\|h\|_2^2\right] - (1+\beta_t^2)\,\mathrm{Cov}\left[h|\bar{h}\right]\bar{h}\right],$$

where we plug in $\partial\psi_t(\bar{h}|h)/\partial t = \beta_t\left(\|h\|_2^2 - (1+\beta_t^2)h^\top\bar{h} + \beta_t\|\bar{h}\|_2^2\right)\psi_t(\bar{h}|h)/\sigma_t^2$ and collect terms in $(i)$. Since $P_h$ has a Gaussian tail, its third moment is bounded.

Then we bound $\left\|\mathrm{Cov}[h|\bar{h}]\right\|_{\mathrm{op}}$ by taking derivative of $s_+(\bar{h}, y, t)$ with respect to $\bar{h}$, here

$$s_+(\bar{h}, y, t) = U\frac{\beta_t}{\sigma_t^2}\int \frac{h \cdot \psi_t(\bar{h}|h)p_h(h|y)}{\int \psi_t(\bar{h}|h)p_h(h|y)\mathrm{d}h}\mathrm{d}h - U\frac{\bar{h}}{\sigma_t^2}.$$

Then we have

$$\frac{\partial}{\partial \overline{h}} s_+(\overline{h}, y, t) = \left(\frac{\beta_t}{\sigma_t^2}\right)^2 U \left[\int hh^\top \varphi(\overline{h}, y)\mathrm{d}h - \int h\varphi(\overline{h}, y)\mathrm{d}h \int h^\top \varphi(\overline{h}, y)\mathrm{d}h\right] - \frac{1}{\sigma_t^2} U$$

$$= \left(\frac{\beta_t}{\sigma_t^2}\right)^2 U \left[\mathrm{Cov}(h|\overline{h}) - \frac{1}{\sigma_t^2} I_{d_0}\right],$$

where

$$\varphi(\overline{h}, y) = \frac{\psi_t(\overline{h}|h)p_h(h|y)}{\int \psi_t(\overline{h}|h)p_h(h|y)\mathrm{d}h}.$$

Along with the $L_{s_+}$-Lipschitz property of $s_+$, we obtain

$$\left\|\mathrm{Cov}(h|\overline{h})\right\|_{\mathrm{op}} \leq \frac{\sigma_t^4}{\beta_t^2} \left(L_{s_+} + \frac{1}{\sigma_t^2}\right).$$

Therefore, we deduce

$$\tau(r_h) = \mathcal{O}\left(\frac{1 + \beta_t^2}{\beta_t} \left(L_{s_+} + \frac{1}{\sigma_t^2}\right) \sqrt{d_0} r_h\right) = \mathcal{O}\left(e^{T/2} L_{s_+} r_h \sqrt{d_0}\right),$$

as $P_h$ having sub-Gaussian tail implies $\mathbb{E}_{P_h}\left[h\|h\|_2^2\right]$ is bounded. $\qquad\square$

**Lemma E.5** (Modified from Lemma 10 in (Chen et al., 2023c)). *For any given $\epsilon > 0$, and $L$-Lipschitz function $g$ defined on $[0, 1]^{d_0} \times [0, 1]^{d_y}$, there exists a continuous function $\overline{f}$ constructed by trapezoid function that*

$$\left\|g - \overline{f}\right\|_\infty \leq \epsilon.$$

*Moreover, the Lipschitz continuity of $\overline{f}$ is bounded by*

$$\left|\overline{f}(x, y) - \overline{f}(x', y')\right| \leq 10d_0 L\|x - x'\|_2 + 10d_y L\|y - y'\|_2,$$

*for any $x, x' \in [0, 1]^{d_0}$ and $y, y' \in [0, 1]^{d_y}$*

*Proof of Lemma E.5.* This proof closely follows Lemma 10 in (Chen et al., 2023c). We divide the proof into two parts: First, we use a collection of Trapezoid function $\overline{f}$ to approximate the function $g$ defined on $[0, 1]^{d_0} \times [0, 1]^{d_y}$. Then we establish the Lipschitz continuity of the function $\overline{f}$ to facilitate the approximation with a transformer.

1. **Approximation by Trapezoid Function.** Given an integer $N > 0$, we choose $(N + 1)^{d_0}$ points in the hypercube $[0, 1]^{d_0}$ and $(N + 1)^{d_y}$ points in the hypercube $[0, 1]^{d_y}$. We denote the index of the hypercubes as $m = [m_1, m_2, \cdots, m_{d_0}]^\top \in \{0, \cdots, N\}$ and $n = [n_1, n_2, \cdots, n_{d_y}]^\top \in \{0, \cdots, N\}$. Next, we define a univariate trapezoid function (see Figure 3) as follow

$$\phi(a) = \begin{cases} 1, & |a| < 1 \\ 2 - |a|, & |a| \in [1, 2] \\ 0, & |a| > 2 \end{cases}. \tag{E.5}$$

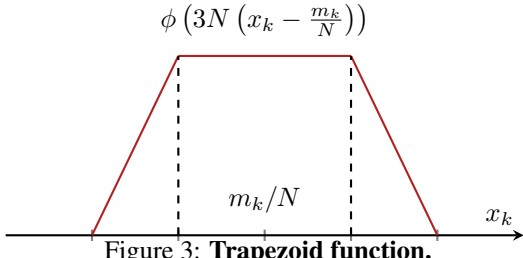

$$\phi\left(3N\left(x_k - \frac{m_k}{N}\right)\right)$$

$$m_k/N$$

$$x_k$$

Figure 3: **Trapezoid function.**

For any $x \in [0,1]^{d_0}$ and $y \in [0,1]^{d_y}$, we define a partition of unity based on a product of trapezoid functions indexed by $m$ and $n$,

$$\xi_{m,n}(x,y) = \mathbb{1}\left\{y \in \left(\frac{n-1}{N}, \frac{n}{N}\right]\right\} \prod_{k=1}^{d_0} \phi\left(3N\left(x_k - \frac{m}{N}\right)\right). \tag{E.6}$$

For example, the product of trapezoid function $\xi_{m,n}(x,y) \neq 0$ only if $y \in \left(\frac{n-1}{N}, \frac{n}{N}\right]$ and $x \in \left[\frac{m-2\cdot 1\cdot 3}{N}, \frac{m+2\cdot 1\cdot 3}{N}\right]$. For any target $L$-Lipschitz function $g$ with respect to $x$ and $y$, it is more convenient to write its Lipschitz continuity with respect to the $\ell_\infty$ norm, i.e.,

$$\begin{aligned}
|g(x,y) - g(x',y')| &\leq L\|x-x'\|_2 + L\|y-y'\|_2 \\
&\leq L\sqrt{d_0}\|x-x'\|_\infty + L\sqrt{d_y}\|y-y'\|_\infty.
\end{aligned} \tag{E.7}$$

We now define a collection of piecewise-constant functions as

$$P_{m,n}(x,y) = g(m,n) \quad \text{for} \quad m \in \{0,\ldots,N\}^{d_0} \text{ and } n \in \{0,\ldots,N\}^{d_y}.$$

We claim that $\bar{f}(x,y) = \sum_{m,n} \xi_{m,n}(x,y)P_{m,n}(x,y)$ is an approximation of $g$, with an approximation error evaluated as

$$\begin{aligned}
&\sup_{x\in[0,1]^{d_0}} \sup_{y\in[0,1]^{d_y}} \left|\bar{f}(x,y) - g(x,y)\right| \\
&= \sup_{x\in[0,1]^{d_0}} \sup_{y\in[0,1]^{d_y}} \left|\sum_{m,n} \xi_{m,n}(x,y)\left(P_{m,n}(x,y) - g(x,y)\right)\right| \\
&\leq \sup_{x\in[0,1]^{d_0}} \sup_{y\in[0,1]^{d_y}} \sum_{\substack{m:|x_k - m_k/N|\leq\frac{2}{3N} \\ n:|y_j - n_j/N|\in(-\frac{1}{2N},\frac{1}{2N}]}} |P_{m,n}(x,y) - g(x,y)| \\
&= \sup_{x\in[0,1]^{d_0}} \sup_{y\in[0,1]^{d_y}} \sum_{\substack{m:|x_k - m_k/N|\leq\frac{2}{3N} \\ n:|y_j - n_j/N|\in(-\frac{1}{2N},\frac{1}{2N}]}} |g(m,n) - g(x,y)| \\
&\leq L\sqrt{d_0}2^{d_0+1}\frac{1}{3N} + L\sqrt{d_y}1^{d_y}\frac{1}{2N} \qquad \text{(By Lipschitz continuity in (E.7))} \\
&= \frac{L}{N}\left(\frac{\sqrt{d_0}2^{d_0+1}}{3} + \frac{\sqrt{d_y}}{2}\right),
\end{aligned}$$

where the last inequality follows the Lipschitz continuity in (E.7) and using the fact that there are at most $2^{d_0}$ terms in the summation of $m$ and at most $1^{d_y}$ terms in the summation of $n$. By choosing $N = \lceil L\left(\sqrt{d_0}2^{d_0+1}/3 + \sqrt{d_y}/2\right)/\epsilon\rceil$, we have $\|g - \bar{f}\|_\infty \leq \epsilon$.

2. **Lipschitz Continuity.** Next we compute the Lipschitz of the function $\bar{f}$ with respect to $x$ and $y$. Suppose the approximation error $\epsilon > 0$ is small enough, then we have

$$
\begin{aligned}
&\left|\bar{f}(x, y) - \bar{f}(x', y')\right| \\
&\leq \left|\bar{f}(x, y) - g(x, y)\right| + \left|g(x, y) - g(x', y')\right| + \left|g(x', y') - \bar{f}(x', y')\right| \\
&\leq 2\epsilon + L\sqrt{d_0}\|x - x'\|_\infty + L\sqrt{d_y}\|y - y'\|_\infty \\
&\leq 10L\sqrt{d_0}\|x - x'\|_\infty + 10L\sqrt{d_y}\|y - y'\|_\infty \\
&\leq 10Ld_0\|x - x'\|_2 + 10Ld_y\|y - y'\|_2.
\end{aligned}
$$

This completes the proof. □

**Main Proof of Theorem E.1.** Now we are ready to state the main proof.

*Proof of Theorem E.1.* From low-dimensional data assumption, the score function $\log p_t(x|y)$ decomposes as the on-support and orthogonal component (see Lemma D.1). Recall the on-support score function is given by $\nabla \log p_t^h\left(\bar{h}|y\right) = U^\top s_+(\bar{h}, y, t)$ from (E.7). We use a latent score network to approximate the score function (see (K3)). Specifically, the latent score network includes a latent encoder and a latent decoder. The encoder approximates $U^\top \in \mathbb{R}^{d_0 \times d_x}$, and decoder approximates $U \in \mathbb{R}^{d_x \times d_0}$. At its core, we use the transformer $g_\mathcal{T}(W_U^\top x, y, t) \in \mathcal{T}^{h,s,r}$ to approximate $q(\bar{h}, y, t)$ as defined in (D.1). The expression for $q(\bar{h}, y, t)$ is given by:

$$
q(\bar{h}, y, t) = \sigma_t^2 \nabla \log p_t^h(U^\top x|y) + U^\top x = \sigma_t^2 U^\top(s_+(\bar{h}, y, t) + x/\sigma_t^2). \tag{E.8}
$$

We proceed as follows:

- **Step 1.** Approximate $q(\bar{h}, y, t)$ with a compact-supported continuous function $\bar{f}(\bar{h}, y, t)$.

- **Step 2.** Approximate $\bar{f}(\bar{h}, y, t)$ with a one-layer single-head transformer network.

**Step 1. Approximate $q(\bar{h}, y, t)$ with a Compact-Supported Continuous Function $\bar{f}(\bar{h}, y, t)$.** First, we partition $\mathbb{R}^{d_0}$ into a compact subset $H_1 := \{\bar{h} \mid \|\bar{h}\|_2 \leq r_h\}$ and its complement $H_2$, where the choice of $r_h$ comes from Lemma E.3. Next, we approximate $q(\bar{h}, y, t)$ on the two subsets by using the compact-supported continuous function $\bar{f}(\bar{h}, y, t)$. Finally, calculating the continuity of $\bar{f}$ gives an estimation error of $\sqrt{d_0 + d_y}\epsilon$ between $q(\bar{h}, y, t)$ and $\bar{f}(\bar{h}, y, t)$. We present the main proof as follows.

- **Approximation on $H_2 \times [0, 1] \times [t_0, T]$.** For any $\epsilon > 0$, by taking $r_h = c(\sqrt{d_0 \log(d_0/t_0) - \log \epsilon})$, we obtain from Lemma E.3 that

$$
\left\|q(\bar{h}, y, t)\mathbb{1}\{\|\bar{h}\|_2 \geq r_h\}\right\|_{L^2(P_t)} \leq \epsilon \quad \text{for} \quad t \in [t_0, T] \quad \text{and} \quad y \in [0, 1].
$$

So we set $\bar{f}(\bar{h}, y, t) = 0$ on $H_2 \times [0, 1] \times [t_0, T]$.

- **Approximation on $H_1 \times [0, 1] \times [t_0, T]$.** On $H_1 \times [0, 1] \times [t_0, T]$, we approximate

$$
q(\bar{h}, y, t) = [q_1(\bar{h}, y, t), q_2(\bar{h}, y, t), \cdots, q_{d_0}(\bar{h}, y, t)],
$$

by approximating each coordinate $q_k(\bar{h}, y, t)$ separately.

We firstly rescale the input by $h' = (\bar{h} + r_h \mathbb{1})/2r_h$ and $t' = t/T$, so that the transformed input space is $[0, 1]^{d_0} \times [0, 1]^{d_y} \times [t_0/T, 1]$. Here we do not need to rescale $y$, since it is already in $[0, 1]$ by definition. We implement such transformation by a single feed-forward layer.

By Assumption E.2, the on-support score $s_+(\bar{h}, y, t)$ is $L_{s_+}$-Lipschitz with respect to any $\bar{h} \in \mathbb{R}^{d_0}$ and $y \in \mathbb{R}^{d_y}$. This implies $q(\bar{h}, y, t)$ is $(1 + L_{s_+})$-Lipschitz in $\bar{h}$ and $y$. When taking the transformed inputs, $g(h', y, t') = q(2r_h h' - r_h \mathbb{1}, Tt')$ becomes $2r_h(1 + L_{s_+})$-Lipschitz in $h'$; each coordinate $g_k(h', y, t)$ is also $2r_h(1 + L_{s_+})$-Lipschitz in $h'$. Here we denote $L_* = 1 + L_{s_+}$.

Besides, $g(h', y, t')$ is $T\tau(r_h)$-Lipsichitz with respect to $t$, where

$$\tau(r_h) = \sup_{t \in [t_0, T]} \sup_{\bar{h} \in [0, r_h]^d} \sup_{y \in [0,1]^{d_y}} \left\| \frac{\partial}{\partial t} q(\bar{h}, y, t) \right\|_2.$$

We have a coarse upper bound for $\tau(r_h)$ in Lemma E.4. We restate it as follows:

$$\tau(r_h) = \mathcal{O}\left( \frac{1 + \beta_t^2}{\beta_t} \left( L_{s_+} + \frac{1}{\sigma_t^2} \right) \sqrt{d_0} r_h \right) = \mathcal{O}\left( e^{T/2} L_{s_+} r_h \sqrt{d_0} \right).$$

Since each $g_k(h', y, t)$ is Lipsichitz continuous, we apply Lemma E.5 to construct a collection of coordinate-wise functions, denoted as $\bar{f}_k(h', y, t)$. We concatenate $\bar{f}_k$'s together and construct $\bar{f} = [\bar{f}_1, \dots, \bar{f}_{d_0}]^\top$. According to the construction of trapezoid function in Lemma E.5, for any given $\epsilon$, we have the following relations:

$$\sup_{h', y, t' \in [0,1]_0^d \times [0,1]^{d_y} \times [t_0/T, 1]} \left\| \bar{f}(h', y, t') - g(h', y, t') \right\|_\infty \le \epsilon.$$

Considering the input rescaling (i.e., $\bar{h} \to h'$, $y \to y$ and $t \to t'$), we obtain:

– The constructed function is Lipschitz continuous in $\bar{h}$ and $y$, i.e., for any $\bar{h}_1, \bar{h}_2 \in H_1$, $y_1, y_2 \in [0,1]$ and $t \in [t_0, T]$, it holds

$$\left\| \bar{f}(\bar{h}_1, y_1, t) - \bar{f}(\bar{h}_2, y_2, t) \right\|_\infty \le 10 d_0 L_* \left\| \bar{h}_1 - \bar{h}_2 \right\|_2 + 10 d_y L_* \|y_1 - y_2\|_2. \tag{E.9}$$

– The function is also Lipschitz in $t$, i.e., for any $t_1, t_2 \in [t_0, T]$ and $\left\| \bar{h} \right\|_2 \le r_h$, it holds

$$\left\| \bar{f}(\bar{h}, y, t_1) - \bar{f}(\bar{h}, y, t_2) \right\|_\infty \le 10 \tau(r_h) \|t_1 - t_2\|_2.$$

To conclude, the construction of $\bar{f}(\bar{h}, y, t)$ uses a collection of trapezoid functions, as described in Lemma E.5. This ensures that $\bar{f}(\bar{h}, y, t) = 0$ for $\left\| \bar{h} \right\|_2 > r_h$, for all $t \in [t_0, T]$ and $y \in [0, 1]$. Consequently, the Lipschitz continuity of $\bar{f}(\bar{h}, y, t)$ with respect to $\bar{h}$ extends over the entire space $\mathbb{R}^{d_0}$.

• **Approximation Error Analysis under $L^2$ Norm.** We first decompose the $L^2$ approximation error of $\bar{f}$ into two terms ($\left\| \bar{h} \right\|_2 < r_h$ and $\left\| \bar{h} \right\|_2 < r_h$):

$$\left\| q(\bar{h}, y, t) - \bar{f}(\bar{h}, y, t) \right\|_{L^2(P_t^h)}$$
$$= \left\| (q(\bar{h}, y, t) - \bar{f}(\bar{h}, y, t)) \mathbb{1}\{ \left\| \bar{h} \right\|_2 < r_h \} \right\|_{L^2(P_t^h)} + \left\| q(\bar{h}, y, t) \mathbb{1}\{ \left\| \bar{h} \right\|_2 > r_h \} \right\|_{L^2(P_t^h)}.$$

By selecting $r_h = \mathcal{O}\left( \sqrt{d_0 \log(d_0/t_0) + \log(1/\epsilon)} \right)$ (see Lemma E.3), we bound the second term on the RHS of above expression as:

$$\left\| g(\bar{h}, y, t) \mathbb{1}\{ \left\| \bar{h} \right\|_2 > r_h \} \right\|_{L^2(P_t^h)} \le \epsilon.$$

For the first term, we bound

$$\left\|\big(q(\overline{h},y,t)-\overline{f}(\overline{h},y,t)\big)\mathbb{1}\big\{\big\|\overline{h}\big\|_2 < r_h\big\}\right\|_{L^2(P_t^h)}$$
$$\leq \sqrt{d_0+d_y}\sup_{h',y,t'\in[0,1]^{d_0}\times[0,1]^{d_y}\times[t_0/T,1]}\left\|\overline{f}(h',y,t')-g(h',y,t')\right\|_\infty$$
$$\leq \sqrt{d_0+d_y}\epsilon.$$

So we obtain

$$\left\|q(\overline{h},y,t)-\overline{f}(\overline{h},y,t)\right\|_{L^2(P_t^h)}\leq\Big(\sqrt{d_0+d_y}+1\Big)\epsilon.$$

Substituting $\epsilon$ with $\epsilon/2$ gives an approximation error for $\overline{f}(\overline{h},y,t)$ of $\sqrt{d_0+d_y}\epsilon$.

**Step 2. Approximate $\overline{f}(\overline{h},y,t)$ with One-Layer Self-Attention.** This step is based on the universal approximation of single-layer single-head transformers for compact-supported continuous function in Theorem G.2.

Recall the reshape layer $\widetilde{R}(\cdot)$ from Definition 2.3. We use $f(\cdot):=\widetilde{R}^{-1}\circ\widehat{g}_{\mathcal{T}}\circ\widetilde{R}(\cdot)$ to approximate $\overline{f}_t(\cdot):=\overline{f}(\cdot,t)$, where $\widehat{g}_{\mathcal{T}}(\cdot)\in\mathcal{T}^{h,s,r}=\{f_2^{(\mathrm{FF})}\circ f^{(\mathrm{SA})}\circ f_1^{(\mathrm{FF})}:\mathbb{R}^{\widetilde{d}\times\widetilde{L}}\to\mathbb{R}^{\widetilde{d}\times\widetilde{L}}\}$.

We first use $\widehat{f}_t(\cdot):=\widetilde{R}^{-1}\circ\widehat{g}_{\mathcal{T}}\circ\widetilde{R}(\cdot)$ to approximate the function $\overline{f}_t(\cdot)$ constructed at Step 1 and denote $H=R(\overline{h})$. Using Theorem G.2, we have:

$$\left\|\overline{f}_t(\overline{h},y)-\widehat{f}(\overline{h},y)\right\|_{L^2(P_t^h)}=\left(\int_{P_t^h}\left\|\overline{f}_t(\overline{h},y)-\widehat{f}(\overline{h},y)\right\|_2^2\mathrm{d}h\right)^{1/2} \tag{E.10}$$

$$=\left(\int_{P_t^h}\left\|\widetilde{R}\circ\overline{f}_t\circ\widetilde{R}^{-1}(H)-\widetilde{R}\circ\widehat{g}_{\mathcal{T}}\circ\widetilde{R}^{-1}(H)\right\|_F^2\mathrm{d}h\right)^{1/2}$$

$$=\left(\int_{P_t^h}\left\|\widetilde{R}\circ\overline{f}_t\circ\widetilde{R}^{-1}(H)-\widehat{g}_{\mathcal{T}}(H)\right\|_F^2\mathrm{d}h\right)^{1/2}$$

$$\leq\epsilon. \tag{E.11}$$

Along with Step 1, we obtain

$$\left\|q(\overline{h},y,t)-\widehat{f}(\overline{h},y)\right\|_{L^2(P_t^h)}\leq\left\|q(\overline{h},y,t)-\overline{f}(\overline{h},y,t)\right\|_{L^2(P_t^h)}+\left\|\overline{f}(\overline{h},y,t)-\widehat{g}_{\mathcal{T}}(\overline{h},y)\right\|_{L^2(P_t^h)}$$

$$\leq\Big(1+\sqrt{d_0+d_y}\Big)\epsilon.$$

The approximator $s_{\widehat{W}}$ for the score function $\nabla\log p_t(\overline{h}|y)$ is define in (D.2) where $s_{\widehat{W}}=(W_U\widehat{f}(U^\top x,y,t)-x)/\sigma_t^2$. The approximation error for such an approximator is

$$\left\|\nabla\log p_t(\cdot)-s_{\widehat{W}}(\cdot,t)\right\|_{L^2(P_t)}\leq\frac{1+\sqrt{d_0+d_y}}{\sigma_t^2}\epsilon,\quad\text{for all }t\in[t_0,T].$$

Finally, the parameter bounds in the transformer network class satisfy

$$\|W_Q\|_2=\|W_K\|_2=\mathcal{O}\Big(\widetilde{d}\cdot\epsilon^{-(\frac{1}{d}+2\widetilde{L})}(\log\widetilde{L})^{\frac{1}{2}}\Big);$$

$$\|W_Q\|_{2,\infty}=\|W_K\|_{2,\infty}=\mathcal{O}\Big(\widetilde{d}^{\frac{3}{2}}\cdot\epsilon^{-(\frac{1}{d}+2\widetilde{L})}(\log\widetilde{L})^{\frac{1}{2}}\Big);$$

$$\|W_O\|_2 = \mathcal{O}\left(\widetilde{d}^{\frac{1}{2}}\epsilon^{\frac{1}{d}}\right); \|W_O\|_{2,\infty} = \mathcal{O}\left(\epsilon^{\frac{1}{d}}\right);$$

$$\|W_V\|_2 = \mathcal{O}(\widetilde{d}^{\frac{1}{2}}); \|W_V\|_{2,\infty} = \mathcal{O}(\widetilde{d});$$

$$\|W_1\|_2 = \mathcal{O}\left(\widetilde{d}\epsilon^{-\frac{1}{d}}\right), \|W_1\|_{2,\infty} = \mathcal{O}\left(\widetilde{d}^{\frac{1}{2}}\epsilon^{-\frac{1}{d}}\right);$$

$$\|W_2\|_2 = \mathcal{O}\left(\widetilde{d}\epsilon^{-\frac{1}{d}}\right); \|W_2\|_{2,\infty} = \mathcal{O}\left(\widetilde{d}^{\frac{1}{2}}\epsilon^{-\frac{1}{d}}\right);$$

$$\left\|E^\top\right\|_{2,\infty} = \mathcal{O}\left(\widetilde{d}^{\frac{1}{2}}\widetilde{L}^{\frac{3}{2}}\right).$$

We refer to Appendix G.2 for the calculation of the hyperparameters configuration of this network. This completes the proof. □

### E.5 PROOF OF SCORE ESTIMATION (THEOREM E.2)

> **Lemma E.6** (Lemma 15 of (Chen et al., 2023c)). Let $\mathcal{G}$ be a bounded function class, i.e., there exists a constant $b$ such that any function $g \in \mathcal{G} : \mathbb{R}^{d_0} \mapsto [0, b]$. Let $z_1, z_2, \cdots, z_n \in \mathbb{R}^{d_0}$ be i.i.d. random variables. For any $\delta \in (0, 1), a \le 1$, and $c > 0$, we have
>
> $$P\left(\sup_{g \in \mathcal{G}} \frac{1}{n}\sum_{i=1}^{n} g(z_i) - (1+a)\mathbb{E}\left[g(z)\right] > \frac{(1+3/a)B}{3n}\log\frac{\mathcal{N}(c, \mathcal{G}, \|\cdot\|_\infty)}{\delta} + (2+a)c\right) \le \delta,$$
>
> $$P\left(\sup_{g \in \mathcal{G}} \mathbb{E}\left[g(z)\right] - \frac{1+a}{n}\sum_{i=1}^{n} g(z_i) > \frac{(1+6/a)B}{3n}\log\frac{\mathcal{N}(c, \mathcal{G}, \|\cdot\|_\infty)}{\delta} + (2+a)c\right) \le \delta.$$

**Main Proof of Theorem E.2.** Now we are ready to state the main proof.

*Proof of Theorem E.2.* Our proof is built on (Chen et al., 2023c, Appendix B.2).

Recall that the empirical score-matching loss is

$$\mathcal{L}(s_{\widehat{W}}) = \frac{1}{n}\sum_{i=1}^{n} \ell(x_i, y_i; s_{\widehat{W}}), \tag{E.12}$$

with the loss function $\ell$ for a data sample $(x, y)$ is defined as

$$\ell(x, y, s_{\widehat{W}}) = \int_{t_0}^{T} \frac{1}{T - t_0}\mathbb{E}_{(x_t|x_0=x,\tau)}\left[\|s(x_t, \tau y, t) - \nabla\log\phi_t(x_t|x_0)\|_2^2\right]\mathrm{d}t.$$

We organize the proof into the following three steps:

- **Step 1. Decomposing $\mathcal{L}\left(s_{\widehat{W}}\right)$:** We first decompose $\mathcal{L}$ into three terms $(A)$, $(B)$, and $(C)$.

- **Step 2. Bounding Each Term:** We then bound three terms separately using some helper from Lemma E.2 and Lemma E.6.

- **Step 3. Putting All Together:** Finally, we combine the above bounds and substitute the covering number of $\mathcal{S}\left(C_x\right)$ from Lemma J.3.

- **Step 1. Decomposing $\mathcal{L}\left(s_{\widehat{W}}\right)$:**

  Following (Chen et al., 2023c, Appendix B.2), for any $a \in (0, 1)$, we have:

  $$\mathcal{L}(s_{\widehat{W}})$$

$$\leq \underbrace{\mathcal{L}^{\text{trunc}}(s_{\widehat{W}}) - (1+a)\widehat{\mathcal{L}}^{\text{trunc}}(s_{\widehat{W}})}_{(A)} + \underbrace{\mathcal{L}(s_{\widehat{W}}) - \mathcal{L}^{\text{trunc}}(s_{\widehat{W}})}_{(B)} + (1+a)\underbrace{\inf_{s_W \in \mathcal{T}_{\widetilde{R}}^{h,s,r}} \widehat{\mathcal{L}}(s_W)}_{(C)}.$$

where

$$\mathcal{L}^{\text{trunc}}(s_{\widehat{W}}) := \mathbb{E}_{x \sim P_0}\big[\ell(x, \tau y, s_{\widehat{W}})\mathbb{1}\{\|x\|_2 \leq r_x\}\big], \quad r_x > B,$$

We denote

$$\eta := 4C_{\mathcal{T}}(C_{\mathcal{T}} + r_x)(r_x/d_x)^{d_x-2} \cdot \exp\big(-r_x^2/\sigma_t^2\big)/t_0(T - t_0),$$

$$r_x := \mathcal{O}\Big(\sqrt{d_0 \log d_0 + \log C_{\mathcal{T}} + \log(n/\bar{\delta})}\Big).$$

- **Step 2. Bounding Each Term:** We bound $(A)$, $(B)$, and $(C)$ term separately using some helper from Lemma E.2 and Lemma E.6.

  **Bounding term** $(A)$. For any $\bar{\delta} > 0$, following (Chen et al., 2023c, Appendix B.2) and applying Lemma E.6, we have the following for term $(A)$ with probability $1 - \bar{\delta}$,

  $$(A) = \mathcal{O}\left(\frac{(1+3/a)(C_{\mathcal{T}}^2 + r_x^2)}{nt_0(T - t_0)} \log \frac{\mathcal{N}\left(\frac{(T-t_0)(\epsilon_c - \eta)}{(C_{\mathcal{T}} + r_x)\log(T/t_0)}, \mathcal{T}^{h,s,r}, \|\cdot\|_2\right)}{\bar{\delta}} + (2+a)c\right),$$

  where $c \leq 0$ is a constant, and $\epsilon_c > 0$ is another constant to be determined later.

  By setting $\epsilon_c = \log(2/(nt_0(T - t_0)))$, then we have

  $$(A) = \mathcal{O}\left(\frac{(1+3/a)(C_{\mathcal{T}}^2 + r_x^2)}{nt_0(T - t_0)} \log \frac{\mathcal{N}\big((n(C_{\mathcal{T}} + r_x)t_0 \log(T/t_0))^{-1}, \mathcal{T}^{h,s,r}, \|\cdot\|_2\big)}{\bar{\delta}} + \frac{1}{n}\right), \tag{E.13}$$

  with probability $1 - \bar{\delta}$.

  **Bounding term** $(B)$. Following (Chen et al., 2023c, Appendix B.2) and applying Lemma E.2, we has the following bound for term $(B)$:

  $$(B) = \mathcal{O}\left(\frac{1}{t_0(T - t_0)}C_{\mathcal{T}}^2 r_x^{d_0} \frac{2^{-2/d_0+2}d_0}{\Gamma(d_0/2 + 1)} \exp\big(-C_2 r_x^2/2\big)\right). \tag{E.14}$$

  **Bounding term** $(C)$. In Theorem E.1, we approximate the score function with the network $\widehat{s}_W$ for any $\epsilon > 0$. We decompose the term $(C)$ into statistical error $(C_1)$ and approximation error $(C_2)$:

  $$(C) \leq \underbrace{\widehat{\mathcal{L}}(\widehat{s}_W) - (1+a)\mathcal{L}^{\text{trunc}}(\widehat{s}_W)}_{(C_1)} + (1+a)\underbrace{\mathcal{L}^{\text{trunc}}(\widehat{s}_W)}_{(C_2)}.$$

  Following (Chen et al., 2023c, Appendix B.2) and applying Lemma E.2 and Lemma E.6, we have the following bound for term $(C_1)$:

  $$(C_1) = \widehat{\mathcal{L}}^{\text{trunc}}(\widehat{s}_W) - (1+a)\mathcal{L}^{\text{trunc}}(\widehat{s}_W) = \mathcal{O}\left(\frac{(1+6/a)(C_{\mathcal{T}}^2 + r_x^2)}{nt_0(T - t_0)} \log \frac{1}{\bar{\delta}}\right),$$

  with probability $1 - \delta$.

Finally, for the term $(C_2)$ we use Theorem E.1 for score function approximation of $\mathcal{L}(\widehat{s}_W)$:

$$(C_2) = \mathcal{O}\left(\frac{d_0 + d_y}{t_0(T - t_0)}\epsilon^2\right) + (\text{const.}).$$

This give us the bound for term $(C) \leq (C_1) + (1 + a)(C_2)$ as

$$(C) \leq \mathcal{O}\left(\frac{(1 + 6/a)(C_\mathcal{T}^2 + r_x^2)}{nt_0(T - t_0)}\log\frac{1}{\bar{\delta}} + \frac{d_0 + d_y}{t_0(T - t_0)}\epsilon^2\right) + (\text{const.}). \qquad (\text{E.15})$$

- **Step 3. Putting All Together:** In the final steps, we combine three terms and substitute the covering number to get the score estimation bound for latent DiT.

  **Combining** $(A)$, $(B)$ **and** $(C)$. Following (Chen et al., 2023c, Appendix B.2), we set $a = \epsilon^2$ and get the overall bound:

$$\frac{1}{T - t_0}\int_{t_0}^{T}\left\|s_{\widehat{W}}(\cdot, t) - \nabla\log p_t(\cdot)\right\|_{L^2(P_t)}^2 \mathrm{d}t$$
$$= \mathcal{O}\left(\frac{(C_\mathcal{T}^2 + r_x^2)}{\epsilon^2 nt_0(T - t_0)}\log\frac{\mathcal{N}\left((n(C_\mathcal{T} + r_x)t_0\log(T/t_0))^{-1}, \mathcal{S}_{\mathcal{T}^{h,s,r}}, \|\cdot\|_2\right)}{\bar{\delta}} + \frac{1}{n} + \frac{d_0 + d_y}{t_0(T - t_0)}\epsilon^2\right),$$
$$(\text{E.16})$$

  with probability $1 - 3\bar{\delta}$.

  Before we move on to the covering number of $\mathcal{T}_{\widetilde{R}}^{h,s,r}$, we first compute the Lipschitz upper bound $L_\mathcal{T}$ and model output bound $C_\mathcal{T}$.

  **Lipschitz Upper Bound $L_\mathcal{T}$ and Model Output Bound $C_\mathcal{T}$.** We then compute the Lipschitz upper bound $L_\mathcal{T}$ for the transformer. We denote $\bar{f}_{t,R}(\cdot) = \widetilde{R}\circ\widehat{g}_t\circ\widetilde{R}^{-1}(\cdot)$ and $H = \left(\widetilde{R}(\bar{h}), y\right)$. We get the Lipschitz upper bound for $\widehat{f}_\mathcal{T} \in \mathcal{T}_{\widetilde{R}}^{h,s,r}$:

$$\left\|\widehat{f}_\mathcal{T}(H_1) - \widehat{f}_\mathcal{T}(H_2)\right\|_F \leq \left\|\widehat{f}_\mathcal{T}(H_1) - \bar{f}_{t,\widetilde{R}}(H_1)\right\|_F + \left\|\bar{f}_{t,\widetilde{R}}(H_1) - \bar{f}_{t,\widetilde{R}}(H_2)\right\|_F$$
$$+ \left\|\bar{f}_{t,\widetilde{R}}(H_2) - \widehat{f}_\mathcal{T}(H_2)\right\|_F$$
$$\leq 2\epsilon + \left\|\bar{f}_{t,\widetilde{R}}(H_1) - \bar{f}_{t,\widetilde{R}}(H_2)\right\|_F \qquad (\text{By (E.10)})$$
$$\leq 2\epsilon + 10(d_0 + d_y)L_{s_+}\|H_1 - H_2\|_F. \qquad (\text{By (E.9)})$$

  Then we get the upper bound of Lipschitzness of $\mathcal{T}_{\widetilde{R}}^{h,s,r}$:

$$L_\mathcal{T} = \mathcal{O}\left((d_0 + d_y)L_{s_+}\right). \qquad (\text{E.17})$$

  Next, we compute the model output bound for $\mathcal{T}_{\widetilde{R}}^{h,s,r}$. For the output of the constructed transformer $\widehat{f}_\mathcal{T} \in \mathcal{T}^{h,s,r}$, according to (G.17), the output of the network is lower bounded by $\mathcal{O}(1)$. Thus with the Lipschitz upper bound $L_\mathcal{T} = \mathcal{O}((d_0 + d_y)L_{s_+})$, we have $\|\widehat{f}_\mathcal{T}(H)\|_F = \mathcal{O}((d_0 + d_y)L_{s_+}r_h)$, where $\|H\|_F \leq r_h$. With $r_h = c(\sqrt{d_0\log(d_0/t_0) + \log(1/\epsilon)})$, we obtain

$$C_\mathcal{T} = \mathcal{O}\left((d_0 + d_y)L_{s_+} \cdot \sqrt{d_0\log(d_0/t_0) + \log(1/\epsilon)}\right). \qquad (\text{E.18})$$

  **Covering Number of $\mathcal{T}_{\widetilde{R}}^{h,s,r}$.** The next step is to calculate the covering number of $\mathcal{T}_{\widetilde{R}}^{h,s,r}$. In particular, $\mathcal{T}_{\widetilde{R}}^{h,s,r}$ consists of two components: (i) Matrix $W_U$ with orthonormal columns; (ii)

Network function $g_{\mathcal{T}}$. Suppose we have $W_{U1}, W_{U2}$ and $g_1, g_2$ such that $\|W_{U1} - W_{U2}\|_F \leq \delta_1$ and $\sup_{\|x\|_2 \leq 3r_x + \sqrt{d_x \log d_x}, y \in [0,1], t \in [t_0, T]} \|g_1(x, y, t) - g_2(x, y, t)\|_2 \leq \delta_2$, where $g_1 = \widetilde{R}^{-1} \circ g_{\mathcal{T}1} \circ \widetilde{R}$ and $g_2 = \widetilde{R}^{-1} \circ g_{\mathcal{T}2} \circ \widetilde{R}$. Then we evaluate

$$\sup_{\|x\|_2 \leq 3r_x + \sqrt{d_x \log d_x}, y \in [0,1], t \in [t_0, T]} \|s_{W_{U1}, g_{\mathcal{T}1}}(x, y, t) - s_{W_{U2}, g_{\mathcal{T}2}}(x, y, t)\|_2$$

$$= \frac{1}{\sigma_t^2} \sup_{\|x\|_2 \leq 3r_x + \sqrt{d_x \log d_x}, y \in [0,1], t \in [t_0, T]} \left\| W_{U1} g_1(W_{U1}^\top x, y, t) - W_{U2} g_2(W_{U2}^\top x, y, t) \right\|_2$$

$$\leq \frac{1}{\sigma_t^2} \sup_{\|x\|_2 \leq 3r_x + \sqrt{d_x \log d_x}, y \in [0,1], t \in [t_0, T]} \left( \underbrace{\left\| W_{U1} g_1(W_{U1}^\top x, y, t) - W_{U1} g_1(W_{U2}^\top x, y, t) \right\|_2}_{\text{1st term}} \right.$$

$$\left. + \underbrace{\left\| W_{U1} g_1(W_{U2}^\top x, y, t) - W_{U1} g_2(W_{U2}^\top x, y, t) \right\|_2}_{\text{2nd term}} + \underbrace{\left\| W_{U1} g_2(W_{U2}^\top x, y, t) - W_{U2} g_2(W_{U2}^\top x, y, t) \right\|_2}_{\text{3rd term}} \right)$$

$$\leq \frac{1}{\sigma_t^2} \left( \underbrace{L_{\mathcal{T}} \delta_1 \sqrt{d_0}(3r_x + \sqrt{d_x \log d_x})}_{\text{1st term}} + \underbrace{\delta_2}_{\text{2nd term}} + \underbrace{\delta_1}_{\text{3rd term}} \right), \tag{E.19}$$

where $L_{\mathcal{T}}$ upper bounds the Lipschitz constant of $g_{\mathcal{T}}$ (see (E.17)).

For the set $\{W_B \in \mathbb{R}^{d_x \times d_0} : \|W_B\|_2 \leq 1\}$, its $\delta_1$-covering number is $\left(1 + 2\sqrt{d_0}/\delta_1\right)^{d_x d_0}$ (Chen et al., 2023c, Lemma 8). The $\delta_2$-covering number of $f$ needs further discussion as there is a reshaping process in our network. For the input reshaped from $\overline{h} \in \mathbb{R}^{d_0}$ to $H \in \mathbb{R}^{\widetilde{d} \times \widetilde{L}}$, we have

$$\left\| \overline{h} \right\|_2 \leq r_x \iff \|H\|_F \leq r_x,$$

Thus we have

$$\sup_{\|\overline{h}\|_2 \leq 3r_x + \sqrt{D \log D}, y \in [0,1], t \in [t_0, T]} \left\| g_1(\overline{h}, y, t) - g_2(\overline{h}, y, t) \right\|_2 \leq \delta_2,$$

$$\iff \sup_{\|H\|_F \leq 3r_x + \sqrt{D \log D}, y \in [0,1], t \in [t_0, T]} \|g_{\mathcal{T}1}(H) - g_{\mathcal{T}2}(H)\|_2 \leq \delta_2.$$

Next we follow the covering number property for sequence-to-sequence transformer $\mathcal{T}_{\widetilde{R}}^{h,s,r}$, i.e., Lemma J.2 and get the following $\delta_2$-covering number

$$\log \mathcal{N}\left(\epsilon_c, \mathcal{T}_{\widetilde{R}}^{h,s,r}, \|\cdot\|_2\right) \tag{E.20}$$

$$\leq \frac{\log(nL)}{\epsilon_c^2} \cdot \alpha^2 \left( \underbrace{((C_F)^2 C_{OV}^{2,\infty})^{\frac{2}{3}}}_{\text{1st term}} + \underbrace{(d + d_y)^{\frac{2}{3}} (C_F^{2,\infty})^{\frac{4}{3}}}_{\text{2nd term}} + \underbrace{(d + d_y)^{\frac{2}{3}} (2(C_F)^2 C_{OV} C_{KQ}^{2,\infty})^{\frac{2}{3}}}_{\text{3rd term}} \right)^3, \tag{E.21}$$

where

$$\alpha := \prod_{j<i} (C_F)^2 C_{OV}(1 + 4C_{KQ})(C_X + C_E).$$

Recall that from the network configuration in Theorem E.1, we have the following bound:

$$\|W_Q\|_2 = \|W_K\|_2 = \mathcal{O}\Big(\widetilde{d}\cdot\epsilon^{-(\frac{1}{d}+2\widetilde{L})}(\log\widetilde{L})^{\frac{1}{2}}\Big);$$

$$\|W_Q\|_{2,\infty} = \|W_K\|_{2,\infty} = \mathcal{O}\Big(\widetilde{d}^{\frac{3}{2}}\cdot\epsilon^{-(\frac{1}{d}+2\widetilde{L})}(\log\widetilde{L})^{\frac{1}{2}}\Big);$$

$$\|W_O\|_2 = \mathcal{O}\left(\widetilde{d}^{\frac{1}{2}}\epsilon^{\frac{1}{d}}\right); \|W_O\|_{2,\infty} = \mathcal{O}\left(\epsilon^{\frac{1}{d}}\right);$$

$$\|W_V\|_2 = \mathcal{O}(\widetilde{d}^{\frac{1}{2}}); \|W_V\|_{2,\infty} = \mathcal{O}(\widetilde{d});$$

$$\|W_1\|_2 = \mathcal{O}\left(\widetilde{d}\epsilon^{-\frac{1}{d}}\right), \|W_1\|_{2,\infty} = \mathcal{O}\left(\widetilde{d}^{\frac{1}{2}}\epsilon^{-\frac{1}{d}}\right);$$

$$\|W_2\|_2 = \mathcal{O}\left(\widetilde{d}\epsilon^{-\frac{1}{d}}\right); \|W_2\|_{2,\infty} = \mathcal{O}\left(\widetilde{d}^{\frac{1}{2}}\epsilon^{-\frac{1}{d}}\right);$$

$$\left\|E^{\top}\right\|_{2,\infty} = \mathcal{O}\left(\widetilde{d}^{\frac{1}{2}}\widetilde{L}^{\frac{3}{2}}\right).$$

Note that $W_{K,Q} = W_Q W_K^{\top}$ and $W_{O,V} = W_O W_V^{\top}$. Combining every component and substitute into (E.20), we have three respective terms bounded as

$$1^{\text{st}}\text{ term} = \mathcal{O}\Big(\widetilde{d}^2\epsilon^{-2/(3\widetilde{d})}\Big),$$

$$2^{\text{nd}}\text{ term} = \mathcal{O}\Big((d_0+d_y)^{2/3}\widetilde{d}^{2/3}\epsilon^{-4/(3\widetilde{d})}\Big),$$

$$3^{\text{rd}}\text{ term} = \mathcal{O}\Big((d_0+d_y)^{2/3}\cdot\Big(\log\widetilde{L}\Big)^{2/3}\cdot\widetilde{d}^4\cdot\epsilon^{(-2/3)(3/\widetilde{d}+4\widetilde{L})}\Big).$$

Apparently the $3^{\text{rd}}$ term dominates the other two. For the $\alpha^2$ term, we write

$$\alpha^2 = \mathcal{O}\Big(\widetilde{d}^{10}\epsilon^{-2(3/\widetilde{d}+4\widetilde{L})}\Big(\log\widetilde{L}\Big)C_x'\Big),$$

where $C_x' = \big(C_x + (d_0+d_y)^{3/2}\big)^2$.

Combining the above bound we get the log-covering number of $\mathcal{T}_2$ as

$$\log\mathcal{N}\left(\epsilon_c, \mathcal{T}_{\widetilde{R}}^{h,s,r}, \|\cdot\|_2\right) \lesssim \mathcal{O}\left(\frac{\log(n\widetilde{L})\log^3(\widetilde{L})}{\epsilon_c^2}\widetilde{d}^{22}(d_0+d_y)^2\epsilon^{-4(3/\widetilde{d}+4\widetilde{L})}C_x^2\right). \quad\text{(E.22)}$$

Substituting the log-covering number of $\mathcal{T}_{\widetilde{R}}^{h,s,r}$ into (E.16), we have

$$\frac{1}{T-t_0}\int_{t_0}^{T}\left\|s_{\widehat{W}}(\cdot,t) - \nabla\log p_t(\cdot)\right\|_{L^2(P_t)}^2 \mathrm{d}t$$

$$= \mathcal{O}\left(\frac{\big(C_{\mathcal{T}}^2 + \log\big(\frac{n}{\delta}\big)\big)}{n\epsilon^2 t_0(T-t_0)}\left(\frac{\log(n\widetilde{L})\log^3(\widetilde{L})}{(T-t_0)n^2}\widetilde{d}^{22}(d_0+d_y)^2\epsilon^{-4(\frac{3}{d}+4\widetilde{L})}C_x^2\right) + \frac{1}{n} + \frac{d_0+d_y}{t_0(T-t_0)}\epsilon^2\right)$$

$$\text{(By (E.16))}$$

$$= \mathcal{O}\left(\frac{(\widetilde{d}+d_0)^2 L_{s_+}^2(d_0\log\big(\frac{d_0}{t_0}\big) + \log\big(\frac{1}{\epsilon}\big)) + \log\big(\frac{n}{\delta}\big)}{n\epsilon^2 t_0(T-t_0)}\left(\frac{\log(n\widetilde{L})\log^3(\widetilde{L})}{(T-t_0)n^2}\widetilde{d}^{22}(\widetilde{d}+d_y)^2\epsilon^{-4(\frac{3}{d}+4\widetilde{L})}C_x^2\right)\right.$$

$$\left. + \frac{d_0+d_y}{t_0(T-t_0)}\epsilon^2\right). \qquad\qquad\text{(By (E.17) and (E.18))}$$

**Balancing Error Terms.** To balance the error term, we set $\epsilon = n^{-3/4\left(1+3/\widetilde{d}+4\widetilde{L}\right)}$. Also setting $\bar{\delta} = 1/3n$ then we have

$$\frac{1}{T - t_0} \int_{t_0}^{T} \left\| s_{\widehat{W}}(\cdot, t) - \nabla \log p_t(\cdot) \right\|_{L^2(P_t)}^2 \mathrm{d}t = \mathcal{O}\left( \frac{\widetilde{d}^{22}(\widetilde{d} + d_0)^2(\widetilde{d} + d_y)^2}{t_0^2} n^{\frac{-3}{2(1+3/\widetilde{d}+4\widetilde{L})}} \log^3 \widetilde{L} \log^3 n \right)$$

(E.23)

with probability of $1 - \frac{1}{n}$.

This completes the proof. □

### E.6 PROOF OF DISTRIBUTION ESTIMATION (THEOREM E.3)

Our proof is built on Chen et al. (2023c, Appendix C). The main difference between our work and Chen et al. (2023c) is our score estimation error from Theorem E.2. This is based on our universal approximation of transformers.

Consequently, only the subspace error and the total variation distance differ from Chen et al. (2023c, Theorem 3).

**Proof Sketch of (i).** We show that if the orthogonal score increases significantly, the mismatch between the column span of $U$ and $W_U$ will be greatly amplified. Therefore, an accurate score network estimator forces $U$ and $W_U$ to align with each other.

**Proof Sketch of (ii).** We conduct the proof via 2 steps:

- **Step 1: Total Variation Distance Bound.** We obtain the discrete result from the continuous-time generated distribution $\widehat{P}_{t_0}$ by adding discretization error (Chen et al., 2023c, Lemma 4). It suffices to bound the divergence between the following two stochastic processes:

  - For the ground-truth backward process, consider $h_t^{\leftarrow} = B^\top y_t$ and the following SDE:

$$\mathrm{d}h_t^{\leftarrow} = \left[ \frac{1}{2} h_t^{\leftarrow} + \nabla \log p^h T - t(h_t^{\leftarrow}) \right] \mathrm{d}t + \mathrm{d}\overline{U}_t^h.$$

    Denote the marginal distribution of the ground-truth process as $P_{t_0}^h$.

  - For the learned process, consider $\widetilde{h}_t^{\leftarrow, r}$ and the following SDE:

$$\mathrm{d}\widetilde{h}_t^{\leftarrow, r} = \left[ \frac{1}{2} \widetilde{h}_t^{\leftarrow, r} + \widetilde{s}_{f,M}^h(\widetilde{h}_t^{\leftarrow, r}, T - t) \right] \mathrm{d}t + \mathrm{d}\overline{U}_t^h,$$

    where $\widetilde{s}_{f,M}^h(z, t) := [M^\top f(Mz, t) - z]/\sigma_t^2$ and $M$ is an orthogonal matrix. Following the notation in (Chen et al., 2023c), we use $(W_U M)_\sharp^\top \widehat{P}_{t_0}$ to denote the marginal distribution of $\widehat{P}_{t_0}$. We first calculate the latent score matching error, i.e., the error between $\nabla \log p_t^h(h, y)$ and $\widetilde{s}_{M,f}^h(h, y, t)$. Then, we adopt Girsanov's Theorem (Chen et al., 2022) and bound the difference in the KL divergence of the above two processes to derive the score-matching error bound.

**Proof Sketch of (iii).** We derive item (iii) by solving the orthogonal backward process of the diffusion model.

**Definition E.1.** For later convenience, we define $\xi(n, t_0, \widetilde{d}, \widetilde{L}) := \frac{1}{t_0^2} n^{\frac{-3}{2(1+3/\widetilde{d}+4\widetilde{L})}} \log^3 n$.

Here we include a few auxiliary lemmas from Chen et al. (2023c) without proofs. Recall the definition of Lipschitz norm: for a given function $f$, $\|f(\cdot)\|_{Lip} = \sup_{x \neq y}(\|f(x) - f(y)\|_2/\|x - y\|_2)$.

**Lemma E.7** (Lemma 3 of Chen et al. (2023c)). Assume that the following holds

$$\mathbb{E}_{h \sim P_h} \|\nabla \log p_h(h|y)\|_2^2 \le C_{sh}, \quad \lambda_{\min} \mathbb{E}_{h \sim P_h}[hh^\top] \ge c_0, \quad \mathbb{E}_{h \sim P_h} \|h\|_2^2 \le C_h,$$

where $\lambda_{\min}$ denotes the smallest eigenvalue. We denote

$$\overline{\mathbb{E}}[\phi(\cdot, t)] = \int_{t_0}^T \frac{1}{\sigma_t^4} \mathbb{E}_{x \sim P_t}[\phi(\cdot, t)] dt.$$

We set $t_0 \le \min\{2\log(d_0/C_{sh}), 1, 2\log(c_0), c_0\}$ and $T \ge \max\{2\log(C_h/d_0), 1\}$. Suppose we have

$$\overline{\mathbb{E}} \|W_B f(W_B^\top x, y, t) - Uq(B^\top x, y, t)\|_2^2 \le \epsilon.$$

Then we have

$$\|W_U W_U^\top - UU^\top\|_{\mathrm{F}}^2 = \mathcal{O}(\epsilon t_0/c_0),$$

and there exists an orthogonal matrix $M \in \mathbb{R}^{d_0 \times d_0}$, such that:

$$\overline{\mathbb{E}} \|M^\top f(Mh, y, t) - q(h, y, t)\|_2^2$$
$$= \epsilon \cdot \mathcal{O}\left(1 + \frac{t_0}{c_0}\left[(T - \log t_0)d_0 \cdot \max_t \|f(\cdot, t)\|_{\mathrm{Lip}}^2 + C_s h\right] + \frac{\max_t \|f(\cdot, t)\|_{\mathrm{Lip}}^2 \cdot C_h}{c_0}\right).$$

**Lemma E.8** (Lemma 4 of Chen et al. (2023c)). Assume that $P_h$ is sub-Gaussian, $f(h, y, t)$ and $\nabla \log p_t^h(h|y)$ are Lipschitz in both $h$, $y$ and $t$. Assume we have the latent score matching error-bound

$$\int_{t_0}^T \mathbb{E}_{h \sim P_t^h} \|\widetilde{s}_{M,f}^h(h_t, y, t) - \nabla \log p_t^h(h_t|y)\|_2^2 \, dt \le \epsilon_{\mathrm{latent}}(T - t_0).$$

Then we have the following latent distribution estimation error for the undiscretized backward SDE

$$\mathrm{TV}\left(P_{t_0}^h, \widehat{P}_{t_0}^h\right) \lesssim \sqrt{\epsilon_{\mathrm{latent}}(T - t_0)} + \sqrt{\mathrm{KL}\left(P_h \| N\left(0, I_{d_0}\right)\right)} \cdot \exp(-T).$$

Furthermore, we have the following latent distribution estimation error for the discretized backward SDE

$$\mathrm{TV}\left(P_{t_0}^h, \widehat{P}_{t_0}^{h,\mathrm{dis}}\right) \lesssim \sqrt{\epsilon_{\mathrm{latent}}(T - t_0)} + \sqrt{\mathrm{KL}\left(P_h \| N\left(0, I_{d_0}\right)\right)} \cdot \exp(-T) + \sqrt{\epsilon_{\mathrm{dis}}(T - t_0)},$$

where

$$\epsilon_{\mathrm{dis}} = \left(\frac{\max_h \|f(h, y, \cdot)\|_{\mathrm{Lip}}}{\sigma(t_0)} + \frac{\max_{h,t} \|f(h, y, t)\|_2}{t_0^2}\right)^2 \eta^2$$
$$+ \left(\frac{\max_t \|f(\cdot, y, t)\|_{\mathrm{Lip}}}{\sigma(t_0)}\right)^2 \eta^2 \max\left\{\mathbb{E}\|h_0\|^2, d_0\right\} + \eta d_0,$$

and $\eta$ is the step size in the backward process.

**Lemma E.9** (Lemma 6 of Chen et al. (2023c)). Consider the following discretized SDE with step size $\mu$ satisfying $T - t_0 = K_T\mu$

$$\mathrm{d}y_t = \left[\frac{1}{2} - \frac{1}{\sigma(T - k\mu)}\right] y_{k\mu}\mathrm{d}t + \mathrm{d}U_t, \text{ for } t \in [k\mu, (k+1)\mu),$$

where $Y_0 \sim N(0, I)$. Then when $T > 1$ and $t_0 + \mu \le 1$, we have $Y_{T-t_0} \sim N\left(0, \sigma^2 I\right)$ with $\sigma^2 \le e\left(t_0 + \mu\right)$.

**Lemma E.10** (Lemma 10 in Chen et al. (2023c)). Assume that $\nabla \log p_h(h|y)$ is $L_h$-Lipschitz. Then we have $\mathbb{E}_{h \sim P_h} \|\nabla \log p_h(h|y)\|_2^2 \le d_0 L_h$.

**Main Proof of Theorem E.3.**  Now we are ready to state the main proof.

*Proof of Theorem E.3.* Recall that in (E.23), we have

$$\xi(n, t_0, \widetilde{d}, \widetilde{L}) := \frac{1}{t_0^2} n^{\frac{-3}{2(1+3/\widetilde{d}+4\widetilde{L})}} \log^3 L \log^3 n.$$

- **Proof of (i).** With Lemma E.7, we replace $\epsilon$ to be $\epsilon(T - t_0)^2$ and we set $C_{sh} = L_h d_0$ by Lemma E.10, we have

$$\left\|W_U W_U^\top - UU^\top\right\|_F^2 = \mathcal{O}\left(\frac{t_0^2 \xi(n, t_0, \widetilde{d}, \widetilde{L})}{c_0}\right).$$

We substitute the score estimation error in Theorem E.2 and $T = \mathcal{O}(\log n)$ into the bound above, we deduce

$$\left\|W_U W_U^\top - UU^\top\right\|_F^2 = \widetilde{\mathcal{O}}\left(\frac{1}{c_0} n^{\frac{-3}{2(1+3/\widetilde{d}+4\widetilde{L})}} \cdot \log^3 n\right).$$

We note that $\log n$ is great enough to make $T$ satisfies $T \ge \max\{\log(C_h/d_0 + 1), 1\}$ where $C_h \ge \mathbb{E}_{h \sim P_h}\|h\|_2^2$.

- **Proof of (ii).** Lemma E.7 and Lemma E.10 imply that

$$\bar{\mathbb{E}}\left\|M^\top f(Mh, y, t) - q(h, y, t)\right\|_2^2 = \mathcal{O}(\epsilon_{\text{latent}}(T - t_0)),$$

where

$$\epsilon_{\text{latent}} = \epsilon \cdot \mathcal{O}\left(\frac{t_0}{c_0}\left[(T - \log t_0)d_0 \cdot L_{s_+}^2 + d_0 L_h\right] + \frac{L_{s_+}^2 \cdot C_h}{c_0}\right).$$

Through the algebra calculation, we get

$$\bar{\mathbb{E}}\left\|M^\top f(Mh, y, t) - q(h, y, t)\right\|_2^2 = \int_{t_0}^T \mathbb{E}_{h \sim P_t^h}\left\|\frac{U^\top f(Uh, y, t) - h}{\sigma_t^2} - \nabla \log p_t^h(h|y)\right\|_2^2 \mathrm{d}t$$
$$\le \epsilon_{\text{latent}}(T - t_0).$$

With $\epsilon_{\text{latent}}$ and Lemma E.8, we obtain

$$\mathrm{TV}(P_{t_0}^h, (W_U M)_\sharp^\top \widehat{P}_{t_0}^{\text{dis}})$$

$$\lesssim \sqrt{\epsilon_{\text{latent}}\left(T-t_0\right)} + \sqrt{\text{KL}\left(P_h \| N\left(0, I_{d_0}\right)\right)}\exp(-T) + \sqrt{\epsilon_{\text{dis}}\left(T-t_0\right)}$$
$$= \widetilde{\mathcal{O}}\left(\frac{1}{t_0\sqrt{c_0}}n^{\frac{-3}{2(1+3/\tilde{d}+4\tilde{L})}} \cdot \log^3 n + \frac{1}{n} + \mu\frac{\sqrt{d_0^2\log d_0}}{t_0^2} + \sqrt{\mu}\sqrt{d_0}\right).$$

As we choose time step $\mu = \mathcal{O}\left(t_0^2/d_0\sqrt{\log d_0}n^{\frac{-3}{4(1+3/\tilde{d}+4\tilde{L})}}\right)$, we obtain

$$\text{TV}(P_{t_0}^h, (W_U M)_\sharp^\top \widehat{P}_{t_0}^{\text{dis}}) = \widetilde{\mathcal{O}}\left(\frac{1}{t_0\sqrt{c_0}}n^{\frac{-3}{2(1+3/\tilde{d}+4\tilde{L})}} \cdot \log^3 n\right).$$

By definition, $\widehat{P}_{t_0}^{h,\text{dis}} = (UW_B)_\sharp^\top \widehat{P}_{t_0}^{\text{dis}}$. This completes the proof of the total variation distance.

- **Proof of (iii).** We apply Lemma E.9 due to our score decomposition. With the marginal distribution at time $T - t_0$ and observing $\mu \ll t_0$, we obtain the last property.

This completes the proof. $\qquad\square$

# F SUPPLEMENTARY THEORETICAL BACKGROUND

In this section, we provide an overview of the conditional diffusion model and classifier guidance in Appendix F.1 and classifier-free guidance in Appendix F.2.

## F.1 CONDITIONAL DIFFUSION PROCESS

Conditional diffusion models use the conditional information (guidance) $y$ to generate samples from conditional data distribution $P(\cdot|y = \text{guidance})$. Depending on the model's objective, the guidance is either a label for generating categorical images, a text prompt for generating images from input sentences, or an image region for tasks like image editing and restoration. Throughout this paper, we coin diffusion models with label guidance $y$ as conditional diffusion models (CDMs). Practically, implement a conditional diffusion model characterized as classifier and classifier-free guidance. The classifier guidance diffusion model combines the unconditional score function with the gradient of an external classifier trained on corrupted data. On the other hand, classifier-free guidance integrates the conditional and unconditional score function by randomly ignoring $y$ with mask signal (see (F.6)). In this paper, we focus on the latter approach.

Specifically, we consider data $x \in \mathbb{R}^{d_x}$ and label $y \in \mathbb{R}^{d_y}$ with initial conditional distribution $P(x|y)$. The diffusion process (forward Ornstein–Uhlenbeck process) is characterized by:

$$\mathrm{d}X_t = -\frac{1}{2}X_t\mathrm{d}t + \mathrm{d}W_t \quad \text{with} \quad X_0 \sim P(x|y), \tag{F.1}$$

where $W_t$ is a Wiener process. The distribution at any finite time $t$ is denoted by $P_t(x|y)$, and $X_\infty$ follows standard Gaussian distribution. Up to a sufficiently large terminating time $T$, we generate samples by the reverse process:

$$\mathrm{d}X_t^{\leftarrow} = \left[\frac{1}{2}X_t^{\leftarrow} + \nabla \log p_{T-t}(X_t^{\leftarrow}|y)\right]\mathrm{d}t + \mathrm{d}\overline{W}_t \quad \text{with} \quad X_0^{\leftarrow} \sim P_T(x|y), \tag{F.2}$$

where the term $\nabla \log p_{T-t}(X_t^{\leftarrow}|y)$ represents the conditional score function. We have $X_t|X_0 \sim N(\alpha_t X_0, \sigma_t^2 I)$ with $\alpha_t = e^{-t/2}$ and $\sigma_t^2 = 1 - e^{-t}$.

We use a score network $\widehat{s}$ to estimate the conditional score function $\nabla \log p_t(x|y)$, and the quadratic loss of the conditional diffusion model is given by

$$\widehat{s} := \underset{s \in \mathcal{T}_R^{h,s,r}}{\mathrm{argmin}} \, \mathbb{E}_t \left[ \mathbb{E}_{(x_0,y)} \left[ \mathbb{E}_{(x' \sim x'|x_0)} \left[ \|s(x', y, t) - \nabla_{x'} \log p_t(x'|x_0)\|_2^2 \right] \right] \right], \tag{F.3}$$

where $t \sim \mathrm{Unif}(t_0, T)$.

With the estimate score network $\widehat{s}$ in (F.3), we generates the conditional sample in the backward process as follows:

$$\mathrm{d}\widetilde{X}_t^{\leftarrow} = \left[\frac{1}{2}\widetilde{X}_t^{\leftarrow} + \widehat{s}\left(\widetilde{X}_t^{\leftarrow}, y, T - t\right)\right]\mathrm{d}t + \mathrm{d}\overline{W}_t \quad \text{with} \quad \widetilde{X}_0^{\leftarrow} \sim N(0, I_d). \tag{F.4}$$

Classifier guidance (Song et al., 2021; Dhariwal and Nichol, 2021) and classifier-free guidance (Ho and Salimans, 2022) are piratical implementations for conditional score estimation. For classifier guidance (Song et al., 2021; Dhariwal and Nichol, 2021), it use the gradient of the classifier to improve the conditional sample quality of the diffusion model. According to Bayes rule, the conditional score function has the relation:

$$\nabla_x \log p_t(x_t|y) = \underbrace{\nabla \log p_t(x_t)}_{\text{Approximate by } \widehat{s}} + \underbrace{\nabla_x \log p_t(y|x_t)}_{\text{Guidance from classifier}}. \tag{F.5}$$

It uses the neural network to approximate the unconditional score function $\nabla \log \widehat{p}_t(x_t)$ along with external classifier to approximate $\widehat{p}_t(y|x_t)$ and compute the gradient of the classifier logits as the guidance $\nabla \log \widehat{p}_t(y|x_t)$.

## F.2 CLASSIFIER-FREE GUIDANCE

Classifier-free guidance (Ho and Salimans, 2022) provides a widely used approach for training condition diffusion models. It not only simplifies the training pipeline but also improves performance and removes the need for an external classifier. Classifier-free guidance diffusion model approximates both conditional and unconditional score functions by neural networks $s_W$, where $W$ is the network parameters.

Our primary goal is to establish the theoretical guarantee for selecting conditional score estimator $\widehat{s}(x, y, t)$ chosen from the transformer architecture class and bound the error for such estimation. Based on previous work by Dhariwal and Nichol (2021); Fu et al. (2024b); Sohl-Dickstein et al. (2015); Ho and Salimans (2022), we adopt the unified setting for the conditional diffusion model. First we define the mask signal as $\tau := \{\emptyset, \text{id}\}$, where $\emptyset$ denotes the the absence of guidance $y$ and id denotes otherwise. Unites the learning of conditional and unconditional scores by randomly ignoring the guidance $y$. Therefore we write the function class of the score estimator as

$$s(x, y, t) = \begin{cases} s_1(x, y, t), & \text{if} \quad y \in \mathbb{R}^{d_y} \\ s_2(x, t), & \text{if} \quad y = \emptyset. \end{cases} \tag{F.6}$$

Both $s_1(x, y, t)$ and $s_2(x, t)$ belong to the transformer function class with slight adaption. Following Fu et al. (2024b), we consider $P(\tau = \text{id}) = P(\tau = \emptyset) = \frac{1}{2}$ without loss of generality, and we have the following objective function for score matching:

$$\widehat{s} := \underset{s_W \in \mathcal{T}_R^{h,s,r}}{\arg\min} \; \mathbb{E}_t \left[ \mathbb{E}_{(x_0,y)} \left[ \mathbb{E}_{(\tau, x' \sim x'|x_0)} \left[ \| s_W(x', \tau y, t) - \nabla_{x'} \log p_t(x'|x_0) \|_2^2 \right] \right] \right].$$

In practice, the loss function is given by

$$\ell(x_0, y; s_W) = \int_{T_0}^{T} \frac{1}{T - T_0} \mathbb{E}_{\tau, x_t|x_0 \sim N(\alpha_t x_0, \sigma_t^2 I_{d_x})} \left[ \| s_W(x_t, \tau y, t) - \nabla_{x_t} \log p_t(x_t|x_0) \|_2^2 \right] \mathrm{d}t, \tag{F.7}$$

where $T_0$ is a small value for stabilize training (Vahdat et al., 2021). To train $s_W$ we select $n$ i.i.d. training samples $\{x_{0,i}, y_i\}_{i=1}^n$, where $x_{0,i} \sim P_0(\cdot|y_i)$. We utilize the following empirical loss:

$$\widehat{\mathcal{L}}(s_W) = \frac{1}{n} \sum_{i=1}^{n} \ell(x_{0,i}, y_i; s_W). \tag{F.8}$$

With the estimate score function $s_W(x, y, t)$ from minimizing the empirical loss in (F.8), we use $s_W(x, y, t)$ to generate new samples. In the classifier-free guidance setting, we generate a new conditional sample by replacing the approximation $s_W$ in (F.4) with $\widetilde{s}_W$, defined as:

$$\widetilde{s}_W(x, y, t) = (1 + \eta) \cdot s_W(x, y, t) - \eta \cdot s_W(x, \emptyset, t), \tag{F.9}$$

where the strength of guidance $\eta > 0$. The proper choice of $\eta$ is crucial for balancing trade-offs between conditional guidance and unconditional ones. The choice directly impacts the performance of the generation process. Wu et al. (2024b) theoretically study the effect of guidance $\eta$ on Gaussian mixture model. They demonstrate that strong guidance improves classification confidence but reduces sample diversity. For more detailed related work, refer to Appendix C.1.

## G  UNIVERSAL APPROXIMATION OF TRANSFORMERS

In this section, we discuss the universal approximation theory of transformers.

In Appendix G.1, we present the universal approximation results of transformers for score approximation in Section 3. We emphasize that most of the material in Appendix G.1 is not original and is drawn from (Hu et al., 2025; Kajitsuka and Sato, 2024; Yun et al., 2020).

In Appendix G.2, we compute the parameter norm bounds of the transformers used for score approximation. These bounds are crucial for calculating the covering number of the transformers and are essential for score and distribution estimation in Section 3.3.

### G.1  TRANSFORMERS AS UNIVERSAL APPROXIMATORS

The key idea for demonstrating the transformers' ability to capture the entire sequence lies in the concept of contextual mapping (Hu et al., 2025; Kajitsuka and Sato, 2024; Yun et al., 2020). We first restate the background of a $(\gamma, \delta)$-contextual mapping in Definition G.3, using the definition of vocabulary (Definition G.1) and token separation (Definition G.2) in the input sequences.

**Background: Contextual Mapping.**  Let $Z, Y \in \mathbb{R}^{d \times L}$ represent input embeddings and output label sequences, respectively, where $Z_{:,k} \in \mathbb{R}^d$ denotes the $k$-th token (column) of each $Z$ sequence. The vocabulary corresponding to the $i$-th sequence at the $k$-th index is defined in Definition G.1.

**Definition G.1** (Vocabulary).  We define the $i$-th vocabulary set for $i \in [N]$ by $\mathcal{V}^{(i)} = \bigcup_{k \in [L]} Z_{:,k}^{(i)} \subset \mathbb{R}^d$, and the whole vocabulary set $\mathcal{V}$ is defined by $\mathcal{V} = \bigcup_{i \in [N]} \mathcal{V}^{(i)} \subset \mathbb{R}^d$.

In line with prior works (Hu et al., 2025; Kajitsuka and Sato, 2024; Kim et al., 2022; Yun et al., 2020), we assume the embeddings separateness to be $(\gamma_{\min}, \gamma_{\max}, \delta)$-separated, as defined in Definition G.2.

**Definition G.2** (Tokenwise Separateness).  Let $Z^{(1)}, \ldots, Z^{(N)} \in \mathbb{R}^{d \times L}$ be embeddings. Then, $Z^{(1)}, \ldots, Z^{(N)}$ are called tokenwise $(\gamma_{\min}, \gamma_{\max}, \delta)$-separated if the following three conditions hold.

(i)  For any $i \in [N]$ and $k \in [n]$, $\|Z_{:,k}^{(i)}\| > \gamma_{\min}$ holds.

(ii)  For any $i \in [N]$ and $k \in [n]$, $\|Z_{:,k}^{(i)}\| < \gamma_{\max}$ holds.

(iii)  For any $i, j \in [N]$ and $k, l \in [n]$ if $Z_{:,k}^{(i)} \neq Z_{:,l}^{(j)}$, then $\|Z_{:,k}^{(i)} - Z_{:,l}^{(j)}\| > \delta$ holds.

Note that when only conditions (ii) and (iii) hold, we denote this as $(\gamma, \delta)$-separateness. Moreover, if only condition (iii) holds, we denote it as $(\delta)$-separateness.

Next, we define a $(\gamma, \delta)$-contextual mapping, building from conditions (ii) and (iii) in the definition of Definition G.2. The contextual mapping extends the concept of token separateness to captures the relationships between tokens across different input sequences effectively. This allows transformers' to utilize self-attention for full context representation.

**Definition G.3** (Contextual Mapping).  Let $Z^{(1)}, \ldots, Z^{(N)} \in \mathbb{R}^{d \times L}$ be embeddings. Then, a map $q : \mathbb{R}^{d \times L} \to \mathbb{R}^{d \times L}$ is called an $(\gamma, \delta)$-contextual mapping if the following two conditions hold:

1.  For any $i \in [N]$ and $k \in [L]$, $\|q(Z^{(i)})_{:,k}\| < \gamma$ holds.

2.  For any $i, j \in [N]$ and $k, l \in [L]$ such that $\mathcal{V}^{(i)} \neq \mathcal{V}^{(j)}$ or $Z_{:,k}^{(i)} \neq Z_{:,l}^{(j)}$, $\|q(Z^{(i)})_{:,k} - q(Z^{(j)})_{:,l}\| > \delta$ holds.

Note that $q\left(Z^{(i)}\right)$ for $i \in [N]$ is called a *context ID* of $Z^{(i)}$.

**Helper Lemmas.**  For completeness, we restate existing lemmas before presenting the proof of one-layer single-head attention mechanism as the contextual mapping in Theorem G.1.

**Lemma G.1** (Boltz Preserves Distance, Lemma 1 of (Kajitsuka and Sato, 2024)). Given $(\gamma, \delta)$-tokenwise separated vectors $z^{(1)}, \ldots, z^{(N)} \in \mathbb{R}^n$ with no duplicate entries in each vector:

$$z_s^{(i)} \neq z_t^{(i)},$$

where $i \in [N]$ and $s, t \in [L], s \neq t$. Also, let

$$\delta \geq 4 \ln n.$$

Then, the outputs of the Boltzmann operator has the following properties:

$$\left| \text{Boltz}\left(z^{(i)}\right) \right| \leq \gamma, \tag{G.1}$$

$$\left| \text{Boltz}\left(z^{(i)}\right) - \text{Boltz}\left(z^{(j)}\right) \right| > \delta' = \ln^2(n) \cdot e^{-2\gamma} \tag{G.2}$$

for all $i, j \in [N], i \neq j$.

**Lemma G.2** (Lemma 13 of (Park et al., 2021)). For any finite subset $\mathcal{X} \subset \mathbb{R}^d$, there exists at least one unit vector $u \in \mathbb{R}^d$ such that

$$\frac{1}{|\mathcal{X}|^2} \sqrt{\frac{8}{\pi d}} \|x - x'\| \leq \left| u^\top \left(x - x'\right) \right| \leq \|x - x'\|$$

for any $x, x' \in \mathcal{X}$.

Lemma G.2 provides the existence of a unit vector $u \in \mathbb{R}^d$ that bounds the inner product of the difference between points in a finite subset $\mathcal{X} \subset \mathbb{R}^d$.

We are now ready to restate the construction of rank-$\rho$ weight matrices in a self-attention layer following (Hu et al., 2025) in Lemma G.3.

**Lemma G.3** (Construction of Weight Matrices, Lemma D.2 of (Hu et al., 2025)). Given a dataset with a $(\gamma_{\min}, \gamma_{\max}, \epsilon)$-separated finite vocabulary $\mathcal{V} \subset \mathbb{R}^d$. There exists rank-$\rho$ weight matrices $W_K, W_Q \in \mathbb{R}^{s \times d}$ such that

$$\left| (W_K v_a)^\top (W_Q v_c) - (W_K v_b)^\top (W_Q v_c) \right| > \delta,$$

for any $\delta > 0$, any $\min(d, s) \geq \rho \geq 1$ and any $v_a, v_b, v_c \in \mathcal{V}$ with $v_a \neq v_b$. In addition, the matrices are constructed as

$$W_K = \sum_{i=1}^{\rho} p_i q_i^\top \in \mathbb{R}^{s \times d}, \quad W_Q = \sum_{j=1}^{\rho} p_j' q_j'^\top \in \mathbb{R}^{s \times d},$$

where for at least one $i$, $q_i, q_i' \in \mathbb{R}^d$ are unit vectors that satisfy Lemma G.2, and $p_i, p_i' \in \mathbb{R}^s$ satisfies

$$\left| p_i^\top p_i' \right| = 5 \left(|\mathcal{V}| + 1\right)^4 d \frac{\delta}{\epsilon \gamma_{\min}}.$$

*Proof of Lemma G.3.* For completeness, we restate the key point from the proof in (Hu et al., 2025).

First, applying Lemma G.2 to $\mathcal{V} \cup \{0\}$, there exists at least one unit vector $q \in \mathbb{R}^d$ such that for any $v_a, v_b \in \mathcal{V} \cup \{0\}$ and $v_a \neq v_b$ the following holds:

$$\frac{1}{\left(|\mathcal{V}| + 1\right)^2 d^{0.5}} \|v_a - v_b\| \leq \left| q^\top \left(v_a - v_b\right) \right| \leq \|v_a - v_b\|.$$

Following (Hu et al., 2025), let $v_b = 0$ and all unit vector $q \in \{q \in \mathbb{R}^n : \|q\| = 1\}$, and select some arbitrary vector pairs $p_i, p_i' \in \mathbb{R}^s$ that satisfy the constraint:

$$\left|p_i^\top p_i'\right| = (|\mathcal{V}| + 1)^4 \, d \frac{\delta}{\epsilon \gamma_{\min}}. \tag{G.3}$$

By constructing the weight matrices as follows:

$$W_K = \sum_{i=1}^{\rho} p_i q_i^\top \in \mathbb{R}^{s \times d}, \quad W_Q = \sum_{j=1}^{\rho} p_j' q_j'^\top \in \mathbb{R}^{s \times d},$$

where for at least one $i$, $p_i, p_i'$ satisfies (G.3) and $q_i, q_j' \in \mathbb{Q}$, we are able to demonstrate that:

$$\left|(W_K v_a)^\top (W_Q v_c) - (W_K v_b)^\top (W_Q v_c)\right| > \delta.$$

This completes the proof.

$\square$

**Any-Rank Attention is Contextual Mapping.** Next, we present the generalized result where self-attention mechanisms of any rank serve as contextual mappings, extending the low-rank analysis in (Kajitsuka and Sato, 2024) to *any-rank* attention weights, as shown in (Hu et al., 2025).

**Theorem G.1** (Any-Rank Attention is $(\gamma, \delta)$-Contextual Mapping, Lemma 2.2 of (Hu et al., 2025))**.** Given embeddings $Z^{(1)}, \ldots, Z^{(N)} \in \mathbb{R}^{d \times L}$ which are $(\gamma_{\min}, \gamma_{\max}, \epsilon)$-tokenwise separated and vocabulary set $\mathcal{V} = \bigcup_{i \in [N]} \mathcal{V}^{(i)} \subset \mathbb{R}^d$. Also, let $Z^{(1)}, \ldots, Z^{(N)} \in \mathbb{R}^{d \times L}$ be embedding sequences with no duplicate word token in each sequence, that is, $Z_{:,k}^{(i)} \neq Z_{:,l}^{(i)}$, for any $i \in [N]$ and $k, l \in [L]$. Then, there exists a 1-layer single head attention with weight matrices $W_O \in \mathbb{R}^{d \times s}$ and $W_V, W_K, W_Q \in \mathbb{R}^{s \times d}$, that is a $(\gamma, \delta)$-contextual mapping for the embeddings $Z^{(1)}, \ldots, Z^{(N)}$ with

$$\gamma = \gamma_{\max} + \epsilon/4, \quad \delta = \exp\left(-5\epsilon^{-1}|\mathcal{V}|^4 d\kappa\gamma_{\max} \log L\right),$$

where $\kappa = \gamma_{\max}/\gamma_{\min}$.

Theorem G.1 shows that any-rank self-attention is able to distinguish two identical tokens in distinct contexts. Specifically, this holds for embeddings $Z_{:,k}^{(i)} = Z_{:,l}^{(j)}$ when the vocabulary sets $\mathcal{V}^{(i)} \neq \mathcal{V}^{(j)}$.

Since the proof of Theorem G.1 is crucial for subsequent analysis, we restate it for later convenience.

*Proof Sketch.* Hu et al. (2025) generalize Theorem 2 of (Kajitsuka and Sato, 2024), where all weight matrices have to be rank-1. This is achieved by constructing the weight matrices as an outer product sum $\sum_i^{\rho} u_i v_i^\top$, where $u_i \in \mathbb{R}^s, v_i \in \mathbb{R}^d$. The proof in (Hu et al., 2025) is divided into two parts:

- **Construction of Softmax Self-Attention**: Different input tokens are mapped to unique contextual embeddings by configuring the weight matrices according to Lemma G.3.

- **Handling Identical Tokens in Different Contexts**: Use the tokenwise separateness guaranteed by Lemma G.3 to handle identical tokens appearing in different contexts. Additionally, Lemma G.1 shows Boltz preserves separateness properties.

With these, we prove that self-attention distinguish input embeddings $Z_{:,k}^{(i)} = Z_{:,l}^{(j)}$ when the vocabulary sets $\mathcal{V}^{(i)} \neq \mathcal{V}^{(j)}$.

$\square$

*Proof of Theorem G.1.* For completeness, we restate the key point from the proof in (Hu et al., 2025).

The proof consists of two parts: First, we show that the attention layer maps different tokens to unique IDs. Second, we show that self-attention distinguishes duplicate input tokens when they appear in different contexts.

For the first part, by utilizing Lemma G.3 and set the weight matrices as follows:

- **Weight Matrices $W_K$ and $W_Q$:**

$$W_K = \sum_{i=1}^{\rho} p_i q_i^\top \in \mathbb{R}^{s \times d},$$

$$W_Q = \sum_{j=1}^{\rho} p_j' q_j'^\top \in \mathbb{R}^{s \times d},$$

where $p_i, p_j' \in \mathbb{R}^s$ and $q_i, q_j' \in \mathbb{R}^d$. In addition, let $\delta = 4 \ln n$ and $p_1, p_1' \in \mathbb{R}^s$ be an arbitrary vector pair that satisfies

$$\left| p_1^\top p_1' \right| = (|\mathcal{V}| + 1)^4 \, d \frac{\delta}{\epsilon \gamma_{\min}}. \tag{G.4}$$

- **Weight Matrices $W_V$ and $W_O$:** In addition, for the other two weight matrices $W_O \in \mathbb{R}^{d \times s}$ and $W_V \in \mathbb{R}^{s \times d}$, we set

$$W_V = \sum_{i=1}^{\rho} p_i'' q_i''^\top \in \mathbb{R}^{s \times d}, \tag{G.5}$$

where $q'' \in \mathbb{R}^d$, $q_1'' = q_1$ and $p_i'' \in \mathbb{R}^s$ is some nonzero vector that satisfies

$$\|W_O p_i''\| = \frac{\epsilon}{4 \rho \gamma_{\max}}. \tag{G.6}$$

This can be accomplished, e.g., $W_O = \sum_{i=1}^{\rho} p_i''' p_i''^\top$ for any vector $p_i'''$ which satisfies $\|p_i'''\| = \epsilon/(4\rho^2 \gamma_{\max} \|p_i''\|^2)$ for any $i \in [\rho]$.

- **Mapping Condition:** With above weights construction, for $i \in [N]$ and $k \in [L]$, we have

$$\left\| W_O \left( W_V Z^{(i)} \right) \mathrm{Softmax} \left[ \left( W_K Z^{(i)} \right)^\top \left( W_Q Z_{:,k}^{(i)} \right) \right] \right\| < \frac{\epsilon}{4}. \tag{G.7}$$

For the second part, we prove that with the weight matrices $W_O, W_V, W_K, W_Q$ configured above, the attention layer distinguishes duplicate input tokens with different context, $Z_{:,k}^{(i)} = Z_{:,l}^{(j)}$ with different vocabulary sets $\mathcal{V}^{(i)} \neq \mathcal{V}^{(j)}$.

We define $a^{(i)}, a^{(j)}$ as

$$a^{(i)} = \left( W_K Z^{(i)} \right)^\top \left( W_Q Z_{:,k}^{(i)} \right) \in \mathbb{R}^n, \quad a^{(j)} = \left( W_K Z^{(j)} \right)^\top \left( W_Q Z_{:,l}^{(j)} \right) \in \mathbb{R}^n,$$

where $a^{(i)}$ and $a^{(j)}$ are tokenwise $(\gamma, \delta)$-separated. Specifically, the following inequality holds

$$|a_{k'}^{(i)}| \leq (|\mathcal{V}| + 1)^4 \, d \frac{\delta}{\epsilon \gamma_{\min}} \gamma_{\max}^2.$$

Since $\mathcal{V}^{(i)} \neq \mathcal{V}^{(j)}$ and there is no duplicate token in $Z^{(i)}$ and $Z^{(j)}$ respectively, we use Lemma G.1 and obtain

$$
\begin{aligned}
&\left| \mathrm{Boltz}\left(a^{(i)}\right) - \mathrm{Boltz}\left(a^{(j)}\right) \right| \\
&= \left| \left(a^{(i)}\right)^{\top} \mathrm{Softmax}\left[a^{(i)}\right] - \left(a^{(j)}\right)^{\top} \mathrm{Softmax}\left[a^{(j)}\right] \right| \\
&> \delta' \\
&= (\ln n)^2 e^{-2\gamma}.
\end{aligned}
\tag{G.8}
$$

Additionally, using Lemma G.3 and (G.4), and assuming $Z^{(i)}_{:,k} = Z^{(j)}_{:,l}$, we have

$$
\left| \left(a^{(i)}\right)^{\top} \mathrm{Softmax}\left[a^{(i)}\right] - \left(a^{(j)}\right)^{\top} \mathrm{Softmax}\left[a^{(j)}\right] \right|
\tag{G.9}
$$
$$
\leq \sum_{i=1}^{\rho} \gamma_{\max} \cdot (|\mathcal{V}|+1)^4 \frac{\pi d}{8} \frac{\delta}{\epsilon \gamma_{\min}} \cdot \left| \left(q_i^{\top} Z^{(i)}\right) \mathrm{Softmax}\left[a^{(i)}\right] - \left(q_i^{\top} Z^{(j)}\right) \mathrm{Softmax}\left[a^{(j)}\right] \right|.
$$

By combining (G.8) and (G.9), we have

$$
\sum_{i=1}^{\rho} \left| \left(q_i^{\top} Z^{(i)}\right) \mathrm{Softmax}\left[a^{(i)}\right] - \left(q_i^{\top} Z^{(j)}\right) \mathrm{Softmax}\left[a^{(j)}\right] \right| > \frac{\delta'}{(|\mathcal{V}|+1)^4} \frac{\epsilon \gamma_{\min}}{d \delta \gamma_{\max}}.
\tag{G.10}
$$

Finally, using (G.6) and (G.10). we derive the lower bound of the difference between the self-attention outputs of $Z^{(i)}, Z^{(j)}$ as follows:

$$
\left\| f_S^{(\mathrm{SA})}\left(Z^{(i)}\right)_{:,k} - f_S^{(\mathrm{SA})}\left(Z^{(j)}\right)_{:,l} \right\| > \frac{\epsilon}{4\gamma_{\max}} \frac{\delta'}{(|\mathcal{V}|+1)^4} \frac{\epsilon \gamma_{\min}}{d \delta \gamma_{\max}},
\tag{G.11}
$$

where $\delta = 4 \ln L$ and $\delta' = \ln^2(L) e^{-2\gamma}$ with $\gamma = (|\mathcal{V}|+1)^4 d \delta \gamma_{\max}^2 / (\epsilon \gamma_{\min})$.

This completes the proof. $\qquad\square$

With Theorem G.1 establishing that any-rank attention is a contextual mapping, we restate the universal approximation result for transformers with a single self-attention layer from (Hu et al., 2025; Kajitsuka and Sato, 2024).

> **Theorem G.2** (Transformers with 1-Layer Self-Attention are Universal Approximators, Modified from Proposition 1 of (Kajitsuka and Sato, 2024)). *Let $0 \leq p < \infty$ and $f^{(\mathrm{FF})}, f^{(\mathrm{SA})}$ be feedforward neural network layers and a single-head self-attention layer. Then, for any permutation equivariant, continuous function $f$ with compact support and $\epsilon > 0$, there exists $f' \in \mathcal{T}_R^{h,s,r}$ such that $d_p(f, f') < \epsilon$, where $d_p := (\int \|f(Z) - g(Z)\|_p^p \mathrm{d}Z)^{1/p}$, and $\|\cdot\|_p$ is the element-wise $\ell_p$-norm.*

*Proof of Theorem G.2.* We restate the proof from (Kajitsuka and Sato, 2024) for completeness.

The proof consists of the following steps:

1. Approximate by Step Function: Given a permutation equivariant continuous function $f$ on a compact set, there exists a Transformer $f' \in \mathcal{T}_R^{h,s,r}$ with one self-attention layer to approximate $f$ by step function with arbitrary precision in terms of $p$-norm.

2. Quantization via $f_1^{\mathrm{FF}}$: The first feed-forward network $f_1^{\mathrm{FF}}$ quantize the input domain, reducing the problem to memorization of finite samples.

3. Contextual Mapping $f^{(\text{SA})}$ and Memorization $f_2^{\text{FF}}$: According to Theorem G.1, we construct any-rank attention $f^{(\text{SA})}$ to be contextual mapping. Then use the second feed-forward $f_2^{\text{FF}}$ to memorize the *context ID* with its corresponding label.

The details for the three steps are below.

1. Since $f$ is a continuous function on a compact set, $f$ has maximum and minimum values on the domain. By scaling with $f_1^{\text{FF}}$ and $f_2^{\text{FF}}$, $f$ is assumed to be normalized without loss of generality: That is for any $Z \in \mathbb{R}^{d \times L} \setminus [0, 1]^{d \times L}$, we have $f(Z) = 0$. For any $X \in [-1, 1]^{d \times L}$, the function $f(X)$ satisfies $-1 \leq f(X) \leq 1$.

   Let $D \in \mathbb{N}$ be the granularity of a grid

   $$\mathbb{G}_D = \{1/D, 2/D, \dots, 1\}^{d \times L} \subset \mathbb{R}^{d \times L}$$

   such that a piece-wise constant approximation

   $$\overline{f}(X) = \sum_{L \in \mathbb{G}_D} f(L) \, 1_{Z \in L + [-1/D, 0)^{d \times L}}$$

   satisfies

   $$d_p(f, \overline{f}) < \epsilon/3. \tag{G.12}$$

   Such a $D$ always exists because of uniform continuity of $f$.

2. We use $f_1^{\text{FF}}$ to quantize the input domain into $\mathbb{G}_D$.

   We first define the following two terms for first feed-forward neural network to approximate.

   - The quantize term $(\text{quant}_D^{d \times L} : \mathbb{R}^{d \times L} \to \mathbb{R}^{d \times L})$: Quantize $[0, 1]$ into $\{1/D, \dots, 1\}$, while it projects $\mathbb{R} \setminus [0, 1]$ to 0 by shifting and stacking step function.

   $$\sum_{t=0}^{D-1} \frac{\text{ReLU}\left[x/\delta - t/\delta D\right] - \text{ReLU}\left[x/\delta - 1 - t/\delta D\right]}{D}$$

   $$\approx \text{quant}_D(x) = \begin{cases} 0 & x < 0 \\ 1/D & 0 \leq x < 1/D \\ \vdots & \vdots \\ 1 & 1 - 1/D \leq x \end{cases}. \tag{G.13}$$

   - The penalty term $(\text{penalty})$: Identify whether an input sequence is in $[0, 1]^{d \times L}$. This is defined by

   $$\text{ReLU}\left[(x-1)/\delta\right] - \text{ReLU}\left[(x-1)/\delta - 1\right] - \text{ReLU}\left[-x/\delta\right] - \text{ReLU}\left[-x/\delta - 1\right]$$

   $$\approx \text{penalty}(x) = \begin{cases} -1 & x \leq 0 \\ 0 & 0 < x \leq 1 \\ -1 & 1 < x \end{cases}. \tag{G.14}$$

   Combining these components together, the first feed-forward neural network layer $f_1^{\text{FF}}$ approximates the following function:

   $$\overline{f}_1^{(\text{FF})}(X) = \text{quant}_D^{d \times L}(X) + \sum_{t=1}^{d} \sum_{k=1}^{L} \text{penalty}(X_{t,k}) \tag{G.15}$$

Note that this function quantizes inputs in $[0,1]^{d \times L}$ with granularity $D$, while every element of the output is non-positive for inputs outside $[0,1]^{d \times L}$. In particular, the norm of the output is upper-bounded by

$$\max_{X \in \mathbb{R}^{d \times L}} \left\| f_1^{\mathrm{FF}}(X)_{:,k} \right\| = \underbrace{dL}_{\text{Total number of elements in X}} \times \underbrace{\sqrt{d}}_{\text{Maximum Euclidean norm in } d\text{-dimensional space}} \tag{G.16}$$

for any $k \in [L]$.

3. Let $\widetilde{\mathbb{G}}_D \subset \mathbb{G}_D$ be a sub-grid

$$\widetilde{\mathbb{G}}_D = \left\{ G \in \mathbb{G}_D \mid \forall k, l \in [L],\ G_{:,k} \neq G_{:,l} \right\},$$

and consider memorization of $\widetilde{\mathbb{G}}_D$ with its labels given by $f(G)$ for each $G \in \widetilde{\mathbb{G}}_D$. Using our modified any-rank attention is contextual mapping in Theorem G.1 allows us to construct a self-attention $f^{(\mathrm{SA})}$ to be a contextual mapping for such input sequences, because $\widetilde{\mathbb{G}}_D$ can be regarded as tokenwise $(1/D, \sqrt{d}, 1/D)$-separated input sequences. By taking sufficiently large granularity $D$ of $\mathbb{G}_D$, the number of cells with duplicate tokens, that is, $|\mathbb{G}_D \setminus \widetilde{\mathbb{G}}_D|$ is negligible.

From the way the self-attention $f^{(\mathrm{SA})}$ is constructed, we have

$$\left\| f^{(\mathrm{SA})}(X)_{:,k} - X_{:,k} \right\| < \frac{1}{4\sqrt{d}D} \max_{k' \in [L]} \|X_{:,k'}\|$$

for any $k \in [L]$ and $X \in \mathbb{R}^{d \times L}$.

If we take large enough $D$, every element of the output for $X \in \mathbb{R}^{d \times L} \setminus [0,1]^{d \times L}$ is upper-bounded by

$$f^{(\mathrm{SA})} \circ f_1^{\mathrm{FF}}(X)_{t,k} < \frac{1}{4D} \quad (\forall t \in [d],\ k \in [L]),$$

while the output for $X \in [0,1]^{d \times L}$ is lower-bounded by

$$f^{(\mathrm{SA})} \circ f_1^{\mathrm{FF}}(X)_{t,k} > \frac{3}{4D} \quad (\forall t \in [d],\ k \in [L]).$$

Finally, we construct $\mathrm{bump}$ function of scale $R > 0$ to map each input sequence $L \in \widetilde{\mathbb{G}}_D$ to its labels $f(L)$ and for input sequence outside the range $X \in (-\infty, 1/4D)^{d \times L}$ to 0 using the second feed-forward $f_2^{\mathrm{FF}}$. Precisely, $\mathrm{bump}$ function of scale $R > 0$ is given by

$$\begin{aligned}
&\mathrm{bump}_R(x) \\
&= \frac{f(L)}{dL} \sum_{t=1}^{d} \sum_{k=1}^{L} (\mathrm{ReLU}\left[R(X_{t,k} - G_{t,k}) - 1\right] - \mathrm{ReLU}\left[R(Z_{t,k} - G_{t,k})\right] \\
&\qquad\qquad + \mathrm{ReLU}\left[R(Z_{t,k} - G_{t,k}) + 1\right]) + \mathrm{ReLU}[R(G_{t,k} - Z_{t,k})]
\end{aligned} \tag{G.17}$$

for each input sequence $G \in \widetilde{\mathbb{G}}_D$ and add up these functions to implement $f_2^{\mathrm{FF}}$.

In addition, the value of $f_2^{(\mathrm{FF})}$ is always bounded: $0 \leq f_2^{(\mathrm{FF})} \leq 1$. Thus, by taking sufficiently small $\delta > 0$ to quantize the step function, we have

$$d_p \left( f_2^{(\mathrm{FF})} \circ f^{(\mathrm{SA})} \circ f_1^{(\mathrm{FF})}, f_2^{(\mathrm{FF})} \circ f^{(\mathrm{SA})} \circ \overline{f}_1^{(\mathrm{FF})} \right) < \frac{\epsilon}{3}. \tag{G.18}$$

Taking large enough $D$ to make duplicate tokens negligible, we have

$$d_p \left( f_2^{\text{(FF)}} \circ f^{\text{(SA)}} \circ \overline{f}_1^{\text{(FF)}}, \overline{f} \right) < \frac{\epsilon}{3}. \tag{G.19}$$

Combining estimation of step function (G.12), estimation of quantization (G.18) and estimatation of duplicate tokens (G.19) together, we get the approximation error of the any-rank Transformer as

$$d_p \left( f_2^{\text{(FF)}} \circ f^{\text{(SA)}} \circ \overline{f}_1^{\text{(FF)}}, f \right) < \epsilon. \tag{G.20}$$

This completes the proof. □

### G.2 PARAMETER NORM BOUNDS FOR TRANSFORMER APPROXIMATION

In the analysis of the approximation ability of transformers in (Kajitsuka and Sato, 2024), universal approximation is ensured by using a sufficiently large granularity $D$, a sufficiently small $\delta$ in $f_1^{\text{(FF)}}$, and an appropriate scaling factor $R$ in $f_2^{\text{(FF)}}$. Here, we provide a detailed discussion on parameter bounds for matrices in $\mathcal{T}_R^{h,r,s}$, focusing on the choice of granularity and scaling factor.

**Lemma G.4** (Order of Granularity and Scaling Factor). Recall universal approximation of transformer, Theorem G.2. Suppose the target function $f$ is defined on domain $\Omega := [-I, I]^{d \times L}$ for some $I \in \mathbb{N}$. Further, suppose the target function is Lipschitz continuous on $\Omega$ with respect to the element-wise $\ell_p$-norm. The order for the granularity and the scaling factor follows $D = \mathcal{O}(\epsilon^{-1})$ and $R = \mathcal{O}(ID)$, and the parameter $\delta$ for the first feed-forward layer in (G.13) follows $\delta = o(D^{-1})$.

*Proof.* We investigate the more precise choice of $D$, $R$, and $\delta$ respectively. We begin by deriving results for the domain $[0, 1]^{d \times L}$ and then extend them to $[-I, I]^{d \times L}$.

- **Bound on Granularity** $D$. Note that there are $\mathcal{O}(D^{-d}|\mathbb{G}_D|)$ omitted duplicated input. Clearly, $\left| \mathbb{G}_D \setminus \widetilde{\mathbb{G}}_D \right|$ becomes negligible by taking sufficiently large granularity. However, we aim to evaluate the corresponding order of $D$. By the extreme value theorem, the continuous function $f$ on $[0, 1]^{d \times L}$ is bounded by some constant, denoted by $C_B > 0$. Moreover, the total number of omitted points are $\mathcal{O}(D^{d(L-1)})$ and the probability for selecting each point in $\mathbb{G}_D$ is $1/D^{dL}$.

  Therefore, the corresponding error is bounded by $\mathcal{O}(D^{-d/p})$. Since we require error to be bounded $\epsilon/3$, setting $D = \mathcal{O}(\epsilon^{-p/d})$ for some constant $p > 0$ guarantees the result. Specifically, let $f(\cdot) : [0, 1]^{d \times L} \rightarrow [0, 1]^{d \times L}$ be the target function and $\overline{f}(\cdot)$ be the piece-wise constant approximation of granularity $D$. The $p$-norm difference between $f(\cdot)$ and $\overline{f}(\cdot)$. (G.12) gives

$$\begin{aligned}
d_p(f, \overline{f}) &= (\int \|f(x) - \overline{f}(x)\|^p \mathrm{d}x)^{1/p} \\
&= (\mathcal{O}(D^{dL-d}) \cdot B^p (1/D)^{dL})^{1/p} \\
&= \mathcal{O}(D^{(dL-d)/p}) \cdot \mathcal{O}(D^{-dL/p}) \\
&= \mathcal{O}(D^{-d/p}).
\end{aligned}$$

  Here, $\mathcal{O}(D^{-d/p}) = \epsilon$ implies $D = \mathcal{O}(\epsilon^{-p/d})$ for some constant $p > 0$. For simplicity, we use $D = \mathcal{O}(\epsilon^{-1/d})$ in our analysis without loss of generality. Furthermore, for Lipschitz continuous target function $f$, there exist a grid $\mathbb{G}_D$ on domain $[-I, I]^{d \times L}$ such that

$$d_p(f(Z), g_1(Z)) < L_f \|Z - Z'\|_2 \le L_f \|Z - Z'\|_F \le \sqrt{dL} L_f / D,$$

where $Z' \in \mathbb{G}_D$ and $L_f$ is the Lipchitz constant with respect to the matrix 2-norm. Therefore, it suffices to take $\epsilon = \sqrt{dL}L_f/D$. Altogether, we take $D = O(\epsilon^{-1})$ such that Theorem G.2 holds.

- **Bound on Parameter $\delta$ in $f_1^{(\mathbf{FF})}$.** We ensure the error within region $(i/D, i/D + \delta)$ does not affect the desired interval $(i/D, (i+1)/D)$ for $i \in [D]$. Thus, we take $\delta = o(1/D)$.

- **Bound on Scaling Factor in $f_2^{(\mathbf{FF})}$.** We first ensure that $R > 0$ is large enough such that it maps input $Z \in (-\infty, \frac{1}{4D})^{d \times L}$ to zero. Since $Z_{t,k} - L_{t,k} \leq -\frac{3}{4D}$, by taking $R = \mathcal{O}(D)$, we obtain (G.17) with all $\mathrm{ReLU}(\cdot)$ output zero. Then, we ensure that $R > 0$ is large enough such that it maps $L \in \widetilde{\mathbb{G}} \subset (\frac{3}{4D}, \infty)^{d \times L}$ to the corresponding label $f(L)$. From (G.17), we achieve this by:

$$\sum_{t=1}^{d} \sum_{k=1}^{L} \mathrm{ReLU}\left[RS - 1\right] - \mathrm{ReLU}\left[RS\right] + \mathrm{ReLU}\left[RS + 1\right] \mathrm{ReLU}[-RS] = dL,$$

where $S \coloneqq Z_{t,k} - L_{t,k} = \mathcal{O}(D^{-1})$. For any $S \in \mathbb{R}$, it suffices to take $R = \mathcal{O}(D)$ such that $|RS| \leq 1$ for any uniform continuous target function on $[0,1]^{d \times L}$. To extend this to $[-I, I]^{d \times L}$ with the identical approximation error, we scale the granularity $D \to 2ID$. In sum, we have $R = O(ID)$ in order to achieve $\epsilon$ precision on Lipschitz continuous target function on $[-I, I]^{d \times L}$.

This completes the proof. $\qquad\qquad\qquad\qquad\qquad\qquad\qquad\qquad\qquad\qquad\qquad\qquad\square$

Building upon Lemma G.4, we extend the results to derive explicit parameter bounds for matrices regarding the transformer-based universal approximation framework. That is, we ensure a more precise quantification of parameter constraints across the architecture.

**Lemma G.5** (Transformer Matrices Bounds). Consider an input sequence $Z \in [0,1]^{d \times L}$. Let $f(Z) : [0,1]^{d \times L} \to \mathbb{R}^{d \times L}$ be any permutation equivariant and Lipschitz continuous sequence-to-sequence function. Then, given a transformer network $f' \in \mathcal{T}_R^{r,h,s}$ (Definition 2.4) that approximates $f$ within $\epsilon$ precision, i.e., $d_p(f, f') < \epsilon$, the following parameter bounds hold for $d \geq 1$ and $L \geq 2$

$$C_{KQ}, C_{KQ}^{2,\infty} = O(I^{4d+2}\epsilon^{-4d-2}); \quad C_{OV}, C_{OV}^{2,\infty} = O(\epsilon);$$
$$C_F, C_F^{2,\infty} = O\left(I\epsilon^{-1} \cdot \max \|f(Z)\|_F\right); \quad C_E = O(1),$$

where $O(\cdot)$ hides polynomial and logarithmic factors depending on $d$ and $L$.

*Proof.* We denote the separatedness of the input tokens by $(\gamma_{\min}, \gamma_{\max}, \epsilon_s)$ and the separatedness of the output tokens by $(\gamma, \delta_s)$. We denote parameter taken in $f_1^{\mathrm{FF}}$ corresponding to granularity by $\delta_{f_1}$.

- **Bounds for $W_Q$ and $W_K$ in $f^{(\mathbf{SA})}$.**

  From the universal approximation theorem of transformer Theorem G.2, with $p_i, p_i' \in \mathbb{R}^s$ and $q_i, q_i'$, being any unit vectors in $\mathbb{R}^d$, we construct rank $\rho$ matrix $W_Q$ and $W_K$ as

  $$W_K = \sum_{i=1}^{\rho} p_i q_i^\top \in \mathbb{R}^{s \times d},$$

  $$W_Q = \sum_{i=1}^{\rho} p_i' q_i'^\top \in \mathbb{R}^{s \times d},$$

  with the identity $p_i^\top p_i' = (|\mathcal{V}| + 1)^4 d\delta_s/(\epsilon_s \gamma_{\min})$. With this, we have the bound for $p_i, p_i'$:

  $$\|p_i\| = \mathcal{O}\left(|\mathcal{V}|^2 \sqrt{d\frac{\delta_s}{\epsilon_s \gamma_{\min}}}\right), \qquad \|p_i'\| = \mathcal{O}\left(|\mathcal{V}|^2 \sqrt{d\frac{\delta_s}{\epsilon_s \gamma_{\min}}}\right). \qquad (\mathrm{G.21})$$

Summing over the set of $p_i^\top p_i'$ for $i = [\rho]$, we obtain the bound for rank $\rho$ matrix $W_Q$ and $W_K$

$$\|W_Q\|_2 = \sup_{\|x\|_2=1} \|W_Q x\|_2 \leq C_Q = \mathcal{O}\left(\sqrt{\rho}|\mathcal{V}|^2 \sqrt{d \frac{\delta_c}{\epsilon_c \gamma_{\min}}}\right),$$

$$\|W_Q\|_{2,\infty} = \max_{1 \leq i \leq d} \|(W_Q)_{(i,:)}\|_2 \leq C_Q^{2,\infty} = \mathcal{O}\left(\rho|\mathcal{V}|^2 \sqrt{d \frac{\delta_s}{\epsilon_s \gamma_{\min}}}\right),$$

$$\|W_K\|_2 = \sup_{\|x\|_2=1} \|W_K x\|_2 \leq C_K = \mathcal{O}\left(\sqrt{\rho}|\mathcal{V}|^2 \sqrt{d \frac{\delta_s}{\epsilon_s \gamma_{\min}}}\right),$$

$$\|W_K\|_{2,\infty} = \max_{1 \leq i \leq d} \|(W_K)_{(i,:)}\|_2 \leq C_K^{2,\infty} = \mathcal{O}\left(\rho|\mathcal{V}|^2 \sqrt{d \frac{\delta_s}{\epsilon_s \gamma_{\min}}}\right),$$

where $\rho \leq s$ and the head size $s \leq d$. After the first step quantization, we obtain vocabulary bounds $|\mathcal{V}| = \mathcal{O}(D^d)$ and output sequences with $(1/D, \sqrt{d}, 1/D)$ tokenwise separatedness, and we take $\delta_s = 4 \log L$ in Theorem G.2 to ensure that $f^{(\text{SA})}$ is a contextual mapping.

Furthermore, to extend this to domain $[-I, I]^{d \times L}$, we rescale the granularity to $D \to O(DI)$. This gives $D = \mathcal{O}(\epsilon^{-1})$. Lastly, recall $W_{KQ} := W_K^\top W_Q$, the bounds on $W_{KQ}$ follows

$$\|W_{KQ}\|_2 \leq C_{KQ} = O(\epsilon^{-4d-2} \cdot I^{4d+2}); \quad \|W_{KQ}\|_{2,\infty} \leq C_{KQ}^{2,\infty} = O(\epsilon^{-4d-2} \cdot I^{4d+2}).$$

- **Bounds for $W_O$ and $W_V$ in $f^{(\text{SA})}$.**

  Following the construction of $W_Q$ and $W_K$ in Theorem G.2, we have

  $$W_V = \sum_{i=1}^\rho p_i'' q_i''^\top \in \mathbb{R}^{s \times d}, \quad W_O = \sum_{i=1}^\rho p_i''' p_i''^\top \in \mathbb{R}^{d \times s},$$

  with the identity $\|p_i'''\| \lesssim \epsilon_s / (4\rho \gamma_{\max} \|p_i''\|)$ from (G.6), and $p_i'' \in \mathbb{R}^s$ is any nonzero vector.

  Along with the $(\gamma_{\min} = 1/D, \gamma_{\max} = \sqrt{d}, \epsilon_s = 1/D)$ separateness and taking $D = \mathcal{O}(\epsilon^{-1})$, we have the following bounds for $W_V$ and $W_O$:

  $$\|W_V\|_2 = \sup_{\|x\|_2=1} \|W_V x\|_2 \leq C_V = \mathcal{O}(\sqrt{\rho}) = \mathcal{O}\left(\sqrt{d}\right),$$

  $$\|W_V\|_{2,\infty} = \max_{1 \leq i \leq d} \|(W_V)_{(i,:)}\|_2 \leq C_V^{2,\infty} = \mathcal{O}(\rho) = \mathcal{O}(d),$$

  $$\|W_O\|_2 = \sup_{\|x\|_2=1} \|W_O x\|_2 \leq C_O = \mathcal{O}\left(\sqrt{\rho} \cdot \rho^{-1} \cdot \gamma_{\max}^{-1} \cdot \epsilon_s\right) = \mathcal{O}\left(d^{-1}\epsilon\right)$$

  $$\|W_O\|_{2,\infty} = \max_{1 \leq i \leq s} \|(W_O)_{(i,:)}\|_2 \leq C_O^{2,\infty} = \mathcal{O}\left(\rho \cdot \rho^{-1} \cdot \gamma_{\max}^{-1} \cdot \epsilon_s\right) = \mathcal{O}\left(d^{-1/2}\epsilon\right),$$

  where we use $\rho \leq d$. Extending bounds to the case where $f$ is defined on $[-I, I]^{d \times L}$, we have

  $$\|W_{OV}\|_2 = \|W_O W_V\|_2 \leq C_{OV} = O(\epsilon); \quad \|W_{OV}\|_{2,\infty} = \|W_O W_V\|_{2,\infty} \leq C_{OV}^{2,\infty} = O(\epsilon)$$

- **Bounds for $W_1$ in $f_1^{\text{FF}}$.**

  In order to approximate the quantization in Theorem G.2, we set up $f_1^{\text{FF}}$ as in (G.13) where every entry of $W_1$ in the layer is bounded by $\mathcal{O}(t/\delta D)$ and $t = \mathcal{O}(D)$. Therefore, we have

  $$\|W_1\|_{2,\infty} \leq C_{F_1}^{2,\infty} = \mathcal{O}\left(\frac{\sqrt{d}}{\delta}\right), \quad \|W_1\|_2 \leq \|W_1\|_F \leq C_{F_1} = \mathcal{O}\left(\frac{d}{\delta}\right).$$

Furthermore, by Lemma G.4, we have $\delta = \nu D^{-1}$ for some $\nu \in (0, 1)$. Then,

$$\|W_1\|_{2,\infty} \leq C_{F_1}^{2,\infty} = \mathcal{O}\left(D\right) = \mathcal{O}(\epsilon^{-1}), \quad \|W_1\|_2 \leq \|W_1\|_F \leq C_{F_1} = \mathcal{O}\left(D\right) = \mathcal{O}(\epsilon^{-1}).$$

- **Bounds on $W_2$ in $f^{\mathbf{FF}}$.**

  The bounds for $\|W_2\|_2, \|W_2\|_{2,\infty}$ in (G.17) follow the same argument as for $W_1$, with the replacement of the largest element with the scaling factor $R$. So we have

$$\|W_2\|_{2,\infty} \leq C_{F_2}^{2,\infty} = \mathcal{O}\left(\sqrt{d}R\right), \quad \|W_2\|_2 \leq C_{F_2} = \mathcal{O}\left(dR\right).$$

  Furthermore, by Lemma G.4, we have $R = \mathcal{O}(D) = \mathcal{O}(\epsilon^{-1})$. Then,

$$\|W_2\|_{2,\infty} \leq C_{F_2}^{2,\infty} = \mathcal{O}\left(D\right) = \mathcal{O}\left(\frac{\max\|f(Z)\|_F}{\epsilon}\right), \quad \|W_2\|_2 \leq C_{F_2} = \mathcal{O}\left(D\right) = \mathcal{O}\left(\frac{\max\|f(Z)\|_F}{\epsilon}\right).$$

- **Bounds on Positional Encoding Matrix $E$.**

  For $\left\|E^\top\right\|_2, \left\|E^\top\right\|_{2,\infty}$, it suffices to set the positional encoding:

$$E = \begin{pmatrix} 2\gamma_{\max} & 4\gamma_{\max} & \cdots & 2L\gamma_{\max} \\ \vdots & \vdots & \ddots & \vdots \\ 2\gamma_{\max} & 4\gamma_{\max} & \cdots & 2L\gamma_{\max} \end{pmatrix}.$$

  Since the $\ell_2$ norm over every row is identical, we have

$$\left\|E^\top\right\|_{2,\infty} = \left(\sum_{i=1}^{L}(2i\gamma_{\max})^2\right)^{\frac{1}{2}} = \left(4\gamma_{\max}^2 \frac{L(L+1)(2L+1)}{6}\right)^2 = \mathcal{O}\left(\gamma_{\max}L^{\frac{3}{2}}\right).$$

  Recall that we have the relation $\gamma_{\max} = \sqrt{d}$ in the self-attention layer. Therefore,

$$\|E^\top\|_{2,\infty} \leq C_E = \mathcal{O}(d^{1/2}L^{3/2}). \tag{G.22}$$

This completes the proof. $\qquad\square$

# H  PROOF OF THEOREM 3.1

Our proof leverages the local smoothness in Hölder spaces alongside the universal approximation property of transformers. Results presented in Appendix G handle uniform continuous functions and do not consider the higher-order regularity of the target function. However, bounding the weight matrices in transformer network function class becomes infeasible without information on the target function regularity. Therefore, we construct a function approximator for the score function that captures Hölder smoothness. Then, we approximate the constructed function using transformers.

- **Step 1: Function Approximation for the Score Function.** Recall that (i) $p_t(x|y)$ follows the integral form shown in (3.1) (ii) the score function has the expression $\nabla \log p_t(x|y) = p_t(x|y)/\nabla p_t(x)$. We perform a $k_1$-th order and a $k_2$-th order Taylor expansion for $p_0(x|y)$ and $\exp(\cdot)$ to construct two function approximators $f_1(x, y, t)$ and $f_2(x, y, t)$ for $p_t(x|y)$ and $\nabla p_t(x|y)$, respectively. These functions inherit the Hölder smoothness property of the density function class (Definition 3.1).

- **Step 2: Universal Approximation of Transformers on a Bounded Domain.** We utilize the universal approximation of transformers to approximate the constructed function on a bounded domain with arbitrary precision. Notably, the constructed functions possess properties stronger than uniform continuity, allowing us to bound the weight matrices in the transformers. We reiterate that these bounds are essential for the later analysis of the transformer's estimation capacity.

- **Step 3: Approximation on the Full Domain with Sub-Gaussianity.** We extend the approximation results to full space $\mathbb{R}^{d_x}$ by leveraging the sub-Gaussian property of the target density function. Specifically, we apply the Gaussian tail bounds to control the error outside the bounded domain.

**Organization.**  Appendix H.1 provides auxiliary lemmas. Appendix H.2 presents the approximation of score function on a bounded domain. Appendix H.3 includes the main proof of Theorem 3.1.

## H.1  AUXILIARY LEMMAS

In this section, we introduce auxiliary lemmas for the score approximation. Specifically, Lemma H.1 establishes an upper bound on the score function in the $\ell_\infty$-distance. Further, Lemma H.2 presents an integral domain clipping technique that enables the score approximation on a bounded domain.

**Bounds on the Score Functions.** We present a theoretical upper bound on the score function. We remark that we use this bound to determine the transformer model output bound $C_{\mathcal{T}}$ as well.

**Lemma H.1** (Upper Bounds on the Score Function in $\ell_\infty$ Distance, Lemma A.10 of (Fu et al., 2024b)). Assume Assumption 3.1. Then, there exists a positive constant $K$ such that

$$\|\nabla \log p_t(x|y)\|_\infty \leq \frac{K(\|x\| + 1)}{\sigma_t^2}.$$

Before presenting the clipping technique, we first introduce the multi-index notation for clarity.

**Definition H.1** (Multi-Index). Let $x \in \mathbb{R}^{d_x}$. We say $\alpha = (\alpha[1], \ldots, \alpha[d_x])$ is a $d_x$-dimensional multi-index if every $\alpha[i]$ is a non-negative integer and satisfies (i) factorial operation: $\alpha! := \alpha[1]! \cdots \alpha[d_x]!$ (ii) derivative operation: $\partial^\alpha := \partial^{\alpha[1]} \cdots \partial^{\alpha[d_x]}$ (iii) power operation: $x^\alpha := x[1]^{\alpha[1]} \cdots x[d_x]^{\alpha[d_x]}$.

Furthermore, when contexts are clear, we use $p_0(x|y)$, $p(x_0|y)$ and $p_0(x_0|y)$ to denote the true (target) density function throughout this section interchangeably.

**Clipping Integral Domain.**  Let $\epsilon \in (0, 1)$ be a precision parameter and $B_x$ be a bounded domain dependent on $\epsilon$ and $x \in \mathbb{R}^{d_x}$. Considering the integral form of $p_t(x|y)$:

$$p_t(x_t|y) = \int_{\mathbb{R}^{d_x}} \frac{\mathrm{d}x_0}{\sigma_t^{d_x}(2\pi)^{d_x/2}} \cdot p_0(x_0|y) \cdot \exp\left(-\frac{\|\alpha_t x_0 - x_t\|^2}{2\sigma_t^2}\right),$$

the next lemma shows that the integral outside of the $B_x$ is bounded by $\epsilon$.

**Lemma H.2** (Approximating Clipped Multi-Index Gaussian Integral, Lemma A.8 of (Fu et al., 2024b)). *Assume Assumption 3.1. Let $\kappa \in \mathbb{Z}_+^{d_x}$ be an integer vector with $\|\kappa\|_1 \leq n$. Then, there exists a constant $C(n, d_x) \geq 1$, such that for any $x \in \mathbb{R}^{d_x}$ and $0 < \epsilon \leq 1/e$, it holds*

$$\int_{\mathbb{R}^{d_x} \setminus B_x} \left| \left( \frac{\alpha_t x_0 - x}{\sigma_t} \right)^\kappa \right| \cdot p(x_0|y) \cdot \frac{1}{\sigma_t^d (2\pi)^{d/2}} \exp\left( -\frac{\|\alpha_t x_0 - x\|^2}{2\sigma_t^2} \right) \mathrm{d}x_0 \leq \epsilon. \tag{H.1}$$

*where*

$$B_x := \left[ \frac{x - \sigma_t C(n, d_x)\sqrt{\log(1/\epsilon)}}{\alpha_t}, \frac{x + \sigma_t C(n, d_x)\sqrt{\log(1/\epsilon)}}{\alpha_t} \right]$$

$$\bigcap \left[ -C(n, d_x)\sqrt{\log(1/\epsilon)}, C(n, d_x)\sqrt{\log(1/\epsilon)} \right]^{d_x}.$$

In Appendix H.2, we approximate the score function on $B_x$ and align the approximation error with the error $\epsilon$ introduced by the integral domain clipping. We specify on the approximation error in Lemma H.3 and Lemma H.4. For now, with hindsight, we set $\epsilon = N^{-\beta}$ for some $N \in \mathbb{N}$. This gives:

$$B_{x,N} := \underbrace{\left[ \frac{x - \sigma_t C(0, d_x)\sqrt{\beta \log N}}{\alpha_t}, \frac{x + \sigma_t C(0, d_x)\sqrt{\beta \log N}}{\alpha_t} \right]}_{\text{(I)}}$$

$$\bigcap \underbrace{\left[ -C(0, d_x)\sqrt{\beta \log N}, C(0, d_x)\sqrt{\beta \log N} \right]^{d_x}}_{\text{(II)}}, \tag{H.2}$$

and the clipping error follows Lemma H.2:

$$p_t(x|y) = \int_{\mathbb{R}^{d_x} \setminus B_{x,N}} p(x_0|y) \cdot \frac{1}{\sigma_t^d (2\pi)^{d/2}} \exp\left( -\frac{\|\alpha_t x_0 - x\|^2}{2\sigma_t^2} \right) \mathrm{d}x_0 \leq \epsilon = N^{-\beta}.$$

Next, we present the the score approximation on a bounded domain.

## H.2 SCORE APPROXIMATION ON A BOUNDED DOMAIN

In this section, we approximate components of the score function and incorporate the Hölder smoothness index (Definition 3.1) $\beta > 0$. Specifically, Lemma H.3 approximates $p_t(x|y)$, and Lemma H.4 approximates $\nabla p_t(x|y)$. Then, we approximate the constructed function on a bounded domain by leveraging the universal approximation of transformers (Appendix G.1).

**Step 1: Function Approximator for the Score Function.** Recall that the score function has the form $\nabla \log p_t(x|y) = \nabla p_t(x|y)/p_t(x|y)$. First, we write $p_t(x|y)$ into the product of $p(x_0|y)$ and $\exp(\cdot)$:

$$p_t(x|y) = \int_{\mathbb{R}^{d_x}} p(x_0|y) p_t(x|x_0) \mathrm{d}x_0 = \int_{\mathbb{R}^{d_x}} \frac{1}{\sigma_t^{d_x} (2\pi)^{d_x/2}} p(x_0|y) \exp\left( -\frac{\|\alpha_t x_0 - x\|^2}{2\sigma_t^2} \right) \mathrm{d}x_0.$$

Let $B_{x,N}$ be the bounded domain defined in Lemma H.2. Then, we approximate $p(x_0|y)$ and $\exp\left( -\frac{\|\alpha_t x_0 - x\|^2}{2\sigma_t^2} \right)$ with $k_1$-order Taylor polynomial and $k_2$-order Taylor polynomial within $B_{x,N}$ respectively. Altogether, we approximate $p_t(x|y)$ with the following *diffused local polynomial*:

$$f_1(x, y, t) = \sum_{v \in [N]^{d_x}, w \in [N]^{d_y}} \sum_{\|n_x\|_1 + \|n_y\|_1 \leq k_1} \frac{R_B^{\|n_x\|}}{n_x! n_y!} \frac{\partial^{n_x + n_y} p}{\partial x^{n_x} \partial y^{n_y}} \Bigg|_{x = R_B(\frac{v}{N} - \frac{1}{2}), y = \frac{w}{N}} \Phi_{n_x, n_y, v, w}(x, y, t),$$

$$\tag{H.3}$$

where

- $g(x[i], n_x[i], v[i], u) := \frac{1}{\sigma_t \sqrt{2\pi}} \int_{B_{v_i,n_i,x_i}} \left( \frac{x_0}{R} + \frac{1}{2} - \frac{v[i]}{N} \right)^{n_x[i]} \frac{1}{u!} \left( -\frac{\|x[i] - \sigma_t x_0^2\|}{2\sigma_t^2} \right)^u dx_0,$

- $B_{v_i,n_i,x_i} := \left[ \frac{x[i] - \sigma_t C(0,d_x)\sqrt{\beta \log N}}{\alpha_t}, \frac{x[i] + \sigma_t C(0,d_x)\sqrt{\beta \log N}}{\alpha_t} \right] \bigcap \left[ (\frac{v_i-1}{N} - \frac{1}{2})R_B, (\frac{v_i}{N} - \frac{1}{2})R_B \right],$

- $\Phi_{n_x,n_y,v,w}(x,y,t) := \left( y - \frac{w}{N} \right)^{n_y} \prod_{j=1}^{d_y} \phi \left( 3N(y[j] - \frac{w}{N}) \right) \prod_{i=1}^{d_x} \sum_{u < k_2} g(x[i], n_x[i], v[i], u).$

**Remark H.1** (Diffused Local Polynomial). The expression of the diffused local polynomial given (H.3) arises from the Taylor expansion applied on each grid point within $[0,1]^{d_x+d_y}$, where we use $v \in [N]^{d_x}$ and $w \in [N]^{d_y}$ to denote the specific grid point undergoing the Taylor approximation. Furthermore, Assumption 3.1 imposed on $p(x_0|y)$ allows us to leverage the Hölder smooth property to establish an upper bound on the error arising from the remainder term in the Taylor expansion.

The next lemma specifies the approximation error for $p_t(x|y)$ using $f(x,y,t)$.

**Lemma H.3** (Approximation of $p_t(x|y)$ by $f_1(x,y,t)$, Lemma A.4 of (Fu et al., 2024b)). Assume Assumption 3.1. For any $x \in \mathbb{R}^{d_x}, y \in [0,1]^{d_y}, t > 0$, and a sufficiently large $N > 0$, there exists a diffused local polynomial $f_1(x,y,t)$ with at most $N^{d_x+d_y}(d_x + d_y)^{k_1}$ monomials such that

$$|f_1(x,y,t) - p_t(x|y)| \lesssim BN^{-\beta} \log^{\frac{d_x+k_1}{2}} N.$$

To avoid the score from exploding, we need to set a threshold for $f_1(x,y,t)$:

**Definition H.2** (Truncated Approximator of $p_t(x|y)$). Let $\epsilon_{\text{low}}$ be a positive real number. Let $f_1(x,y,t)$ be the diffused local polynomial defined in (H.3). Then, we define:

$$f_{1,\text{clip}}(x,y,t) := \max\{\epsilon_{\text{low}}, f_1(x,y,t)\}.$$

Similarly, we have the approximation for $\nabla p_t(x|y)$ based on the diffused local polynomial:

**Lemma H.4** (Approximation of $\nabla p_t(x|y)$ by $f_2(x,y,t)$, Lemma A.6 of (Fu et al., 2024b)). Assume Assumption 3.1. Let $f_2 := (f_2[1], \ldots, f_2[d_x])^\top \in \mathbb{R}^{d_x}$ where $f_2[i]$ is a diffused local polynomial for all $i \in [d_x]$. Then, for any $x \in \mathbb{R}^{d_x}, y \in [0,1]^{d_y}, t > 0$, and a sufficiently large $N > 0$, it holds

$$|f_2(x,y,t)[i] - \sigma_t \nabla p_t(x|y)[i]| \lesssim BN^{-\beta} \log^{\frac{d_x+k_1+1}{2}} N,$$

where each $f_2[i]$ contains at most $N^{d_x+d_y}(d_x + d_y)^{k_1}$ monomials.

To this end, we complete the approximation of $p_t$ and $\nabla p_t$ with diffused local polynomial $f_1$ and $f_2$.

**Step 2. Score Approximation with Transformers on a Bounded Domain.** We use transformers to approximate a function approximator for the score constructed with $f_1(x,y,t)$ and $f_2(x,y,t)$.

**Lemma H.5** (Approximate the Score Approximator with Transformers). Assume Assumption 3.1. Let $C_x(d_x, \beta, C_1, C_2)$ be a positive constant. Let $f_3(x,y,t) := f_2/(\sigma_t \cdot f_{1,\text{clip}})$ be the target function. Then, for any $\epsilon \in (0,1)$, any $x \in [-C_x\sqrt{\log N}, C_x\sqrt{\log N}]^{d_x}$, any $y \in [0,1]^{d_y}$ and any $t \in [N^{-C_\sigma}, C_\alpha \log N]$, there exists a transformer $g(x,y,t) \in \mathcal{T}_R^{h,s,r}$ such that

$$\int_{\|x\| \leq C_x\sqrt{\log N}} \|g(x,y,t) - f_3(x,y,t)\|_2^2 dx \leq \epsilon^2.$$

Furthermore, the parameter bounds in transformer network function class satisfy:

$$C_{KQ}, C_{KQ}^{2,\infty} = O((\log N)^{4d+2}\epsilon^{-4d-2}); C_{OV}, C_{OV}^{2,\infty} = O(\epsilon);$$
$$C_F, C_F^{2,\infty} = O(\log N\epsilon^{-1} \cdot \max\|f_3\|_F); \quad C_E = O(1),$$

where $O(\cdot)$ hides all polynomial factors depending on $d_x, d, L, \beta, C_1, C_2$.

*Proof.* Since the diffused local polynomials integrates over some polynomial functions and $\sigma_t$ is a smooth function, the target $f_3(x, y, t)$ is Lipschitz continuous. Therefore, for any $\epsilon \in (0, 1)$,

$$d_2(g, f) = \left( \int \|g(x, y, t) - f_3(x, y, t)\|_2^2 \mathrm{d}x \right)^{1/2} \le \epsilon. \qquad \text{(By Theorem G.2)}$$

The parameter bounds in the transformer network class follow Lemma G.5.

This completes the proof. $\qquad\qquad\qquad\qquad\qquad\qquad\qquad\qquad\qquad\qquad\qquad\qquad\quad\square$

Next lemma incorporates previous approximation results into an unified transformer architecture.

**Lemma H.6** (Approximation Score Function with Transformer on Supported Domain). Assume Assumption 3.1. Consider $t \in [N^{-C_\sigma}, C_\alpha \log N]$, for constant $C_\sigma, C_\alpha$, and $(x, y) \in [-C_x\sqrt{\log N}, C_x\sqrt{\log N}]^{d_x} \times [0, 1]^{d_y}$, where $N \in \mathbb{N}$ and $C_x$ depends on $d, \beta, B, C_1, C_2$. Then, there exist a transformer network $\mathcal{T}_{\text{score}}(x, y, t) \in \mathcal{T}_R^{h,s,r}$ such that

$$\int \left( p_t(x|y) \right)^2 \|\nabla \log p_t(x|y) - \mathcal{T}_{\text{score}}(x, y, t)\|_2^2 \lesssim \frac{B^2}{\sigma_t^4} N^{-2\beta} (\log N)^{\frac{3d_x}{2} + k_1 + 1}.$$

The parameter bounds in the transformer network class satisfy

$$C_{KQ}, C_{KQ}^{2,\infty} = O\left( N^{4\beta d + 2\beta} (\log N)^{4d+2} \right); C_{OV}, C_{OV}^{2,\infty} = O\left( N^{-\beta} \right);$$

$$C_F, C_F^{2,\infty} = O\left( N^\beta (\log N)^{\frac{d_x+\beta+3}{2}} \right); \ C_E = O(1); \ C_\mathcal{T} = O(\sqrt{\log N}/\sigma_t^2),$$

where $O(\cdot)$ hides all polynomial factors depending on $d_x, d, L, \beta, C_1, C_2$.

*Proof.* Recall Lemma H.1, Lemma H.3, Definition H.2 and Lemma H.4. We define:

$$f_3(x, y, t) = \min \left( \frac{f_2}{\sigma_t f_{1,\text{clip}}}, \frac{K}{\sigma_t^2} (C_x \sqrt{d_x \log N} + 1) \right),$$

where we set $f_{1,\text{clip}} = \{f_1, \epsilon_{\text{low}}\}$ to prevent the score explosion. We specify the coice of $\epsilon_{\text{low}}$ later. We proceed with the following two steps:

- **Step A. Approximate Score Function with $f_3$.** For any $i \in [d_x]$, it holds

$$|(\nabla \log p_t)[i] - f_3[i]|$$
$$\le \left| (\nabla \log p_t)[i] - \frac{f_2[i]}{\sigma_t f_{1,\text{clip}}} \right|$$
$$\le \left| \frac{(\nabla p_t)[i]}{p_t} - \frac{(\nabla p_t)[i]]}{f_{1,\text{clip}}} \right| + \left| \frac{(\nabla p_t)[i]}{f_{1,\text{clip}}} - \frac{f_2[i]}{\sigma_t f_{1,\text{clip}}} \right|.$$

From Lemma H.1, the bound on the score implies $(\nabla p_t)[i] \le K(\sqrt{d_x \log N} + 1)p_t/\sigma_t^2$. Therefore,

$$|(\nabla \log p_t)[i] - f_3[i]|$$
$$\le \frac{K}{\sigma_t^2} (\sqrt{d \log N} + 1)p_t \left| \frac{1}{p_t} - \frac{1}{f_{1,\text{clip}}} \right| + \frac{1}{f_{1,\text{clip}}} \left| \frac{(\nabla \sigma_t p_t)[1] - f_2[1]}{\sigma_t} \right|,$$
$$\lesssim \frac{1}{f_{1,\text{clip}}} \left( \frac{1}{\sigma_t^2} \sqrt{\log N} |p_t - f_{1,\text{clip}}| + \frac{(\nabla \sigma_t p_t)[1] - f_2[1]}{\sigma_t} \right). \qquad \text{(By dropping Constant Terms)}$$

From Lemma H.3, we have

$$|f_1 - p_t| \leq BN^{-\beta} \log^{\frac{d_x + k_1}{2}} N.$$

We set $\epsilon_{\text{low}} = C_3 N^{-\beta} \log^{\frac{d_x + k_1}{2}} N \leq p_t$ such that $f_1 \geq p_t / 2$ by the choice of constant $C_3$. Then,

$$
\begin{aligned}
&|(\nabla \log p_t)[1] - f_3[1]| \\
&\lesssim \frac{1}{p_t} \Big( \frac{1}{\sigma_t^2} \sqrt{\log N} |p_t - f_{1,\text{clip}}| + \frac{(\nabla \sigma_t p_t)[1] - f_2[1]}{\sigma_t} \Big), && \text{(By the choice of } \epsilon_{\text{low}}) \\
&\lesssim \frac{B}{\sigma_t^2 p_t} N^{-\beta} (\log N)^{\frac{d_x + k_1 + 1}{2}}. && \text{(By Lemma H.3 and Lemma H.4)}
\end{aligned}
$$

By the symmetry of each coordinate, it holds

$$\|\nabla \log p_t - f_3\|_\infty \leq \|\nabla \log p_t - f_3\|_2 \lesssim \frac{B}{\sigma_t^2 p_t} N^{-\beta} (\log N)^{\frac{d_x + k_1 + 1}{2}}. \tag{H.4}$$

- **Step B: Approximate $f_3$ with Transformer $\mathcal{T}_{\text{score}}$.** We use transformers to approximate $f_3$ to an accuracy of order $N^{-\beta}$ such that it aligns with the error order in (H.4). By Lemma H.5, we have

$$\int \|\mathcal{T}_{\text{score}}(x, y, t) - f_3(x, y, t)\|_2^2 \mathrm{d}x \leq \epsilon^2.$$

By setting $\epsilon = N^{-\beta}$, it holds

$$
\begin{aligned}
&\int p_t^2 \|\mathcal{T}_{\text{score}}(x, y, t) - \nabla \log p_t(x|y)\|_2^2 \mathrm{d}x \\
&\leq \int p_t^2 \|\mathcal{T}_{\text{score}}(x, y, t) - f_3(x, y, t)\|_2^2 \mathrm{d}x + \int p_t^2 \|f_3(x, y, t) - \nabla \log p_t(x|y)\|_2^2 \mathrm{d}x \\
&\hspace{8cm} \text{(By triangle inequality)} \\
&\leq \int \|\mathcal{T}_{\text{score}}(x, y, t) - f_3(x, y, t)\|_2^2 \mathrm{d}x + \int p_t^2 \|f_3(x, y, t) - \nabla \log p_t(x|y)\|_2^2 \mathrm{d}x \\
&\hspace{8cm} \text{(By } p_t(x|y) \in [0, 1]) \\
&\lesssim \int \|\mathcal{T}_{\text{score}}(x, y, t) - f_3(x, y, t)\|_2^2 \mathrm{d}x + \frac{B^2}{\sigma_t^4} N^{-2\beta} (\log N)^{d_x + k_1 + 1} \int \mathrm{d}x && \text{(By (H.4))} \\
&\lesssim \int \|\mathcal{T}_{\text{score}}(x, y, t) - f_3(x, y, t)\|_2^2 \mathrm{d}x + \frac{B^2}{\sigma_t^4} N^{-2\beta} (\log N)^{\frac{3d_x}{2} + k_1 + 1} && \text{(By } \|x\|_\infty \leq C_x \sqrt{\log N}) \\
&\leq \frac{B^2}{\sigma_t^4} N^{-2\beta} (\log N)^{\frac{3d_x}{2} + k_1 + 1}. && \text{(By Lemma H.5)}
\end{aligned}
$$

Next, by Lemma H.1 and $x \in [-C_x \sqrt{\log N}, C_x \sqrt{\log N}]^{d_x}$, we have

$$\|\nabla \log p_t\|_\infty = O(\sqrt{\log N}),$$

Therefore we have $C_{\mathcal{T}} = O(\sqrt{\log N})$, and by (H.4) we have

$$|\nabla \log p_t[i] - f_3[i]| = O(N^{-\beta} (\log N)^{\frac{d_x + k_1 + 1}{2}}).$$

This implies

$$\|f_3\|_2 = O(\sqrt{\log N} + N^{-\beta} (\log N)^{\frac{d_x + k_1 + 1}{2}}).$$

We take a looser bound on $f_3$ such that it holds for all $d_x$:

$$\|f_3\|_2 \le d_x \|f_3\|_\infty = O((\log N)^{\frac{d_x+\beta+1}{2}}),$$

where we use $k_1 \le \beta$.

Then, the parameter bounds follow Lemma H.5 with $\epsilon = N^{-\beta}$. Therefore, we have

$$C_{KQ}, C_{KQ}^{2,\infty} = O(N^{4\beta d+2\beta}(\log N)^{4d+2}); C_{OV}, C_{OV}^{2,\infty} = O(N^{-\beta});$$
$$C_F, C_F^{2,\infty} = O(N^\beta (\log N)^{\frac{d_x+\beta+3}{2}}); \ C_E = O(1); \ C_\mathcal{T} = O(\sqrt{\log N}/\sigma_t^2).$$

This completes the proof. $\qquad\qquad\square$

## H.3    Main Proof of Theorem 3.1

In Lemma H.6, we establish the score approximation with transformer that incorporates every essential components and encodes the Hölder smoothness in the final result.

However, it is only valid within the input domain $[C_x\sqrt{\log N}, C_x\sqrt{\log N}]^{d_x} \times [0,1]^{d_y}$, and we also excludes region $p_t < \epsilon_{\text{low}}$ where the problem of score explosion remains unaddressed.

To combat this, we introduce two additional lemmas.

**Lemma H.7** (Truncate $x$ for Score Function, Lemma A.1 of (Fu et al., 2024b)). Assume Assumption 3.1. For any $R_1 > 1$, $y, t > 0$ we have

$$\int_{\|x\|_\infty \ge R_1} p_t(x|y)dx \le R_1 \exp(-C_2' R_1^2),$$

$$\int_{\|x\|_\infty \ge R_1} \|\nabla \log p_t(x|y)\|_2^2 p_t(x|y)dx \le \frac{R_1^3}{\sigma_t^4}\exp(-C_2' R_1^2),$$

where $C_2' = C_2/(2\max(C_2, 1))$.

**Remark H.2.** Because we only impose assumption on the light tail property of the conditional distribution in Assumption 3.1, the unboundedness of $x$ necessitates a truncation for integrals regarding $x$, or else the result would diverge.

Furthermore, we address the explosion of score function with the second lemma.

**Lemma H.8** (Lemma A.2 of (Fu et al., 2024b)). Assume Assumption 3.1. For any $R_2, y, \epsilon_{\text{low}} > 0$ we have

$$\int_{\|x\|_\infty \le R_2} \mathbb{1}\{|p_t(x|y)| < \epsilon_{\text{low}}\} \cdot p_t(x|y)dx \le R_2^{d_x}\epsilon_{\text{low}},$$

$$\int_{\|x\|_\infty \le R_2} \mathbb{1}\{|p_t(x|y)| < \epsilon_{\text{low}}\} \cdot \|\nabla \log p_t(x|y)\|_2^2 p_t(x|y)dx \le \frac{1}{\sigma_t^4}R_2^{d_x+2}\epsilon_{\text{low}}.$$

**Remark H.3.** Recall that the score function has the form $\nabla \log p_t(x|y) = \nabla p_t(x|y)/p_t(x|y)$. It is essential to set a threshold for $p_t(x|y)$ prevents the explosion of the score function.

We begin the proof of Theorem 3.1.

*Proof Sketch of Theorem 3.1.* In the following proof, we give error bound for the three terms:

- **(A.1): The approximation for $\|x\|_\infty > R_1$.**

  This step controls the error from truncation of $\mathbb{R}^{d_x}$ with radius $R_1$ in $\ell_2$ distance. We approximate the error with Lemma H.7

- **(A.2): The approximation for $\mathbb{1}\{p_t(x|y) < \epsilon_{\text{low}}\}$ and $\{\|x\|_\infty \le R_1\}$.**

  This step controls the error from setting a threshold to prevent score explosion within the bounded domain $\|x\|_\infty \le R_1$. We approximate the error with Lemma H.8.

- **(A.3) The approximation for $\mathbb{1}\{p_t(x|y) \ge \epsilon_{\text{low}}\}$ and $\{\|x\|_\infty \le R_1\}$.**

  With previous two steps ensuring the bounded domain and preventing the divergence of score function, we approximate with Lemma H.6.

$\square$

*Proof of Theorem 3.1.* First, we set $R_1 = R_2 = \sqrt{2\beta \log N / C_2'}$ in Lemma H.7 and Lemma H.8. Next, we expand the target into three parts $(A_1)$, $(A_2)$, and $(A_3)$:

$$\int_{\mathbb{R}^{d_x}} \|s(x, y, t) - \nabla \log p_t(x|y)\|_2^2 \cdot p_t(x|y) \mathrm{d}x$$

$$= \underbrace{\int_{\|x\|_\infty > \sqrt{\frac{2\beta}{C_2'} \log N}} \|s(x, y, t) - \nabla \log p_t(x|y)\|_2^2 \cdot p_t(x|y) \mathrm{d}x,}_{(A_1)}$$

$$+ \underbrace{\int_{\|x\|_\infty \le \sqrt{\frac{2\beta}{C_2'} \log N}} \mathbb{1}\{|p_t(x|y)| < \epsilon_{\text{low}}\} \|s(x, y, t) - \nabla \log p_t(x|y)\|_2^2 \cdot p_t(x|y) \mathrm{d}x}_{(A_2)}$$

$$+ \underbrace{\int_{\|x\|_\infty \le \sqrt{\frac{2\beta}{C_2'} \log N}} \mathbb{1}\{|p_t(x|y)| \ge \epsilon_{\text{low}}\} \|s(x, y, t) - \nabla \log p_t(x|y)\|_2^2 \cdot p_t(x|y) \mathrm{d}x}_{(A_3)}.$$

We derive the bound for $(A_1), (A_2), (A_3)$ and combine these results.

- **Bounding $(A_1)$.** We apply Lemma H.7. Note that we have $\|s(x, y, t)\|_\infty \lesssim \sqrt{\log N} / \sigma_t^2$ from the construction of the score estimator in Lemma H.6.

$$\int_{\|x\|_\infty > \sqrt{\frac{2\beta}{C_2'} \log N}} \|s(x, y, t) - \nabla \log p_t(x|y)\|_2^2 \cdot p_t(x|y) \mathrm{d}x \qquad \text{(By expanding the } \ell_2 \text{ norm)}$$

$$\le 2 \int_{\|x\|_\infty > \sqrt{\frac{2\beta}{C_2'} \log N}} \|s(x, y, t)\|_2^2 \cdot p_t(x|y) \mathrm{d}x + 2 \int_{\|x\|_\infty > \sqrt{\frac{2\beta}{C_2'} \log N}} \|\nabla \log p_t(x|y)\|_2^2 \cdot p_t(x|y) \mathrm{d}x$$

$$\text{(By } \|\cdot\|_2^2 \le d_x \|\cdot\|_\infty^2)$$

$$\le 2d_x \int_{\|x\|_\infty > \sqrt{\frac{2\beta}{C_2'} \log N}} \|s(x, y, t)\|_\infty^2 \cdot p_t(x|y) \mathrm{d}x + 2 \int_{\|x\|_\infty > \sqrt{\frac{2\beta}{C_2'} \log N}} \|\nabla \log p_t(x|y)\|_2^2 \cdot p_t(x|y) \mathrm{d}x$$

$$\text{(By the } \ell_\infty \text{ bound on the score function)}$$

$$\lesssim 2d_x \left(\frac{\sqrt{\log N}}{\sigma_t^2}\right)^2 \int_{\|x\|_\infty > \sqrt{\frac{2\beta}{C_2'} \log N}} p_t(x|y) \mathrm{d}x + 2 \int_{\|x\|_\infty > \sqrt{\frac{2\beta}{C_2'} \log N}} \|\nabla \log p_t(x|y)\|_2^2 \cdot p_t(x|y) \mathrm{d}x$$

$$\text{(By Lemma H.7 and dropping constant)}$$

$$\lesssim 2d_x \cdot \frac{\log N}{\sigma_t^4} \cdot \sqrt{\frac{2\beta}{C_2'} \log N} \cdot N^{-2\beta} + \frac{2}{\sigma_t^4} \left(\frac{2\beta}{C_2'} \log N\right)^{\frac{3}{2}} N^{-2\beta}$$

$$\text{(By dropping constant and lower order term)}$$

$$\lesssim \frac{1}{\sigma_t^4} N^{-2\beta} (\log N)^{\frac{3}{2}}.$$

- **Bounding** ($A_2$). We apply Lemma H.8. Recall that we set $\epsilon_{\text{low}} = C_3 N^{-\beta} (\log N)^{\frac{d_x+k_1}{2}}$ (Lemma H.6).

$$\int_{\|x\|_\infty \le \sqrt{\frac{2\beta}{C_2'} \log N}} \mathbb{1}\{|p_t(x|y)| < \epsilon_{\text{low}}\} \|s(x,y,t) - \nabla \log p_t(x|y)\|_2^2 \cdot p_t(x|y) \mathrm{d}x$$

(By expanding the $\ell_2$ norm)

$$\le \int_{\|x\|_\infty \le \sqrt{\frac{2\beta}{C_2'} \log N}} 2\mathbb{1}\{|p_t(x|y)| < \epsilon_{\text{low}}\} \left( \|s(x,y,t)\|_2^2 + \|\nabla \log p_t(x|y)\|_2^2 \right) \cdot p_t(x|y) \mathrm{d}x$$

(By $\|\cdot\|_2^2 \le d_x \|\cdot\|_\infty^2$)

$$\le \int_{\|x\|_\infty \le \sqrt{\frac{2\beta}{C_2'} \log N}} \mathbb{1}\{|p_t(x|y)| < \epsilon_{\text{low}}\} \left( d_x \|s(x,y,t)\|_\infty^2 + \|\nabla \log p_t(x|y)\|_2^2 \right) \cdot p_t(x|y) \mathrm{d}x$$

(By the $\ell_\infty$ bound on the score function)

$$\lesssim \int_{\|x\|_\infty \le \sqrt{\frac{2\beta}{C_2'} \log N}} \mathbb{1}\{|p_t(x|y)| < \epsilon_{\text{low}}\} \left( d_x \left( \frac{\sqrt{\log N}}{\sigma_t^2} \right)^2 + \|\nabla \log p_t(x|y)\|_2^2 \right) \cdot p_t(x|y) \mathrm{d}x$$

(By Lemma H.8 and dropping constant)

$$\lesssim d_x \left( \frac{\sqrt{\log N}}{\sigma_t^2} \right)^2 \left( \frac{2\beta}{C_2'} \log N \right)^{\frac{d_x}{2}} \epsilon_{\text{low}} + \left( \frac{2\beta}{C_2'} \log N \right)^{\frac{d_x+2}{2}} \frac{\epsilon_{\text{low}}}{\sigma_t^4}$$

(By dropping constant and lower order term)

$$\lesssim \frac{1}{\sigma_t^4} (\log N)^{\frac{d_x+2}{2}} \epsilon_{\text{low}}.$$

- **Bounding** ($A_3$). We apply Lemma H.6.

$$\int_{\|x\|_\infty \le \sqrt{\frac{2\beta}{C_2'} \log N}} \mathbb{1}\{|p_t(x|y)| \ge \epsilon_{\text{low}}\} \|s(x,y,t) - \nabla \log p_t(x|y)\|_2^2 \cdot p_t(x|y) \mathrm{d}x$$

$$= \int_{\|x\|_\infty \le \sqrt{\frac{2\beta}{C_2'} \log N}} \frac{\mathbb{1}\{|p_t(x|y)| \ge \epsilon_{\text{low}}\}}{p_t(x|y)} d_x \|s(x,y,t) - \nabla \log p_t(x|y)\|_2^2 \cdot p_t^2(x|y) \mathrm{d}x$$

(Multiply with $p_t/p_t$)

$$\le \int_{\|x\|_\infty \le \sqrt{\frac{2\beta}{C_2'} \log N}} \frac{\mathbb{1}\{|p_t(x|y)| \ge \epsilon_{\text{low}}\}}{\epsilon_{\text{low}}} d_x \|s(x,y,t) - \nabla \log p_t(x|y)\|_2^2 \cdot p_t^2(x|y) \mathrm{d}x$$

(By $1/p_t < 1/\epsilon_{\text{low}}$)

$$\lesssim \frac{B^2}{\sigma_t^4} N^{-2\beta} (\log N)^{\frac{3d_x}{2}+k_1+1} \cdot \frac{1}{\epsilon_{\text{low}}}.$$

(Lemma H.6)

- **Combining Three Upper-Bounds.**

  Combining ($A_1$), ($A_2$) and ($A_3$), we have

$$\int_{\mathbb{R}^d} \|s(x,y,t) - \nabla \log p_t(x|y)\|_2^2 p_t(x|y) \mathrm{d}x$$

$$\lesssim \underbrace{\frac{N^{-2\beta}(\log N)^{\frac{3}{2}}}{\sigma_t^4}}_{(A_1)} + \underbrace{\frac{\epsilon_{\text{low}}(\log N)^{\frac{d_x+2}{2}}}{\sigma_t^4}}_{(A_2)} + \underbrace{\frac{B^2}{\sigma_t^4 \epsilon_{\text{low}}} N^{-2\beta}(\log N)^{\frac{3d_x}{2}+k_1+1}}_{(A_3)}.$$

By replacing $\epsilon_{\text{low}}$ with $C_3 N^{-\beta}(\log N)^{d_x+k_1/2}$ and using the relation $k_1 \leq \beta$,[5] we obtain

$$\int_{\mathbb{R}^d} \|s(x,y,t) - \nabla \log p_t(x|y)\|_2^2 p_t(x|y)\mathrm{d}x = \mathcal{O}\left(\frac{B^2}{\sigma_t^4} N^{-\beta}(\log N)^{d_x+\frac{\beta}{2}+1}\right).$$

Last, the transformer parameter norm bounds follow Lemma H.6.

This completes the proof. $\qquad\square$

---

[5]Recall the definition of the Hölder smoothness, Definition 3.1.

# I  PROOF OF THEOREM 3.2

We provide the formal version of Theorem 3.2 at the end of Appendix I.2.

**Organization.**  Appendix I.1 includes auxiliary lemmas for supporting our proof. Appendix I.2 includes the formal version and main proof of Theorem 3.2.

## I.1  AUXILIARY LEMMAS

We utilize the condition assumed in Assumption 3.2 to achieve the decomposition.

**Lemma I.1** (Lemma B.1 of Fu et al. (2024b))**.** Assume Assumption 3.2. The conditional distribution at time $t$ has the following expression:

$$p_t(x|y) = \frac{1}{(\alpha_t^2 + C_2\sigma_t^2)^{d_x/2}} \exp\left(-\frac{C_2\|x\|_2^2}{2(\alpha_t^2 + C_2\sigma_t^2)}\right) h(x, y, t).$$

Moreover, the score function has the following expression:

$$\nabla \log p_t(x|y) = \frac{-C_2 x}{\alpha_t^2 + C_2\sigma_t^2} + \frac{\nabla h(x, y, t)}{h(x, y, t)},$$

where $h(x, y, t) = \int \frac{f(x_0, y)}{\widehat{\sigma}_t^d (2\pi)^{d/2}} \exp\left(-\frac{\|x_0 - \widehat{\alpha}_t x\|^2}{2\widehat{\sigma}_t^2}\right) dx_0$, $\widehat{\sigma}_t = \frac{\sigma_t}{(\alpha_t^2 + C_2\sigma_t^2)^{1/2}}$, and $\widehat{\alpha}_t = \frac{\alpha_t}{\alpha_t^2 + C_2\sigma_t^2}$.

*Proof.* Let $z := x_0$ denote the initial data distribution.

By Assumption 3.2, we have $p(z|y) = \exp\left(-C_2\|z\|_2^2/2\right) \cdot f(z, y)$. Therefore,

$$p_t(x|y) = \int \frac{1}{\sigma_t^d (2\pi)^{d/2}} p(z|y) \exp\left(-\frac{\|x - \alpha_t z\|^2}{2\sigma_t^2}\right) dz,$$

$$= \frac{1}{\sigma_t^d (2\pi)^{d/2}} \int \exp\left(-\frac{C_2\|z\|_2^2}{2}\right) f(z, y) \exp\left(-\frac{\|x - \alpha_t z\|^2}{2\sigma_t^2}\right) dz. \qquad (\text{I.1})$$

We rearrange the two exponential terms in (I.1) into

$$\exp\left(-\frac{C_2\|z\|_2^2}{2}\right) \exp\left(-\frac{\|x - \alpha_t z\|^2}{2\sigma_t^2}\right) = \exp\left(-\frac{1}{2\sigma_t^2} \sum_{i=1}^{d} (x[i]^2 - 2\alpha_t x[i]z[i] + \alpha_t^2 z[i]^2 + C_2\sigma_t^2 z[i]^2)\right).$$

To simplify the summation in the exponents, we first rewrite the term for a single coordinate.

Without loss of generality, we derive the first coordinate of the fucntion:

$$\exp\left(-\frac{1}{2\sigma_t^2}(x[1]^2 - 2\alpha_t x[1]z[1] + \alpha_t^2 z[1]^2 + C_2\sigma_t^2 z[1]^2)\right),$$

$$= \exp\left(-\frac{1}{2\sigma_t^2}(\alpha_t^2 + C_2\sigma_t^2)\left(z[1]^2 - \frac{2\alpha_t}{\alpha_t^2 + C_2\sigma_t^2}x[1]z[1] + \frac{x[1]^2}{\alpha_t^2 + C_2\sigma_t^2}\right)\right),$$

$$= \exp\left(-\frac{1}{2\sigma_t^2}(\alpha_t^2 + C_2\sigma_t^2)\left(z[1] - \frac{\alpha_t x[1]}{\alpha_t^2 + C_2\sigma_t^2}\right)^2 - \frac{1}{2\sigma_t^2}\left(\frac{-\alpha_t^2}{\alpha_t^2 + C_2\sigma_t^2} + 1\right)x[1]^2\right),$$

$$= \exp\left(-\frac{1}{2\sigma_t^2}(\alpha_t^2 + C_2\sigma_t^2)\left(z[1] - \frac{\alpha_t x[1]}{\alpha_t^2 + C_2\sigma_t^2}\right)^2\right) \exp\left(-\frac{C_2 x[1]^2}{2(\alpha_t^2 + C_2\sigma_t^2)}\right).$$

The other $d_x - 1$ coordinates abide by the same derivation.

Next, we rewrite the the exponential with product of all coordinates by

$$
\exp\left(-\frac{C_2\|z\|_2^2}{2}\right)\exp\left(-\frac{\|x-\alpha_t z\|^2}{2\sigma_t^2}\right),
$$

$$
= \exp\left(-\frac{1}{2\sigma_t^2}(\alpha_t^2+C_2\sigma_t^2)\left\|z-\frac{\alpha_t x}{\alpha_t^2+C_2\sigma_t^2}\right\|^2\right)\exp\left(-\frac{C_2}{2(\alpha_t^2+C_2\sigma_t^2)}\|x\|_2^2\right). \quad \text{(I.2)}
$$

Define $\widehat{\alpha}_t := \alpha_t/\alpha_t^2 + C_2\sigma_t^2$ and $\widehat{\sigma}_t^2 = \sigma_t^2/\alpha_t^2 + C_2\sigma_t^2$. Next, we plug (I.2) into (I.1) and write

$$
p_t(x|y)
$$

$$
= \frac{1}{\sigma_t^d(2\pi)^{d/2}}\exp\left(-\frac{C_2\|x\|_2^2}{2(\alpha_t^2+C_2\sigma_t^2)}\right)\int f(z,y)\exp\left(-\frac{1}{2\sigma_t^2}(\alpha_t^2+C_2\sigma_t^2)\left\|z-\frac{\alpha_t x}{\alpha_t^2+C_2\sigma_t^2}\right\|^2\right)\mathrm{d}z,
$$

$$
= \frac{1}{\sigma_t^d(2\pi)^{d/2}}\exp\left(-\frac{C_2\|x\|_2^2}{2(\alpha_t^2+C_2\sigma_t^2)}\right)\int f(z,y)\exp\left(-\frac{\|z-\widehat{\alpha}_t x\|^2}{2\widehat{\sigma}_t^2}\right)\mathrm{d}z. \quad \text{(I.3)}
$$

Last, we define $h(x,y,t) = \int \frac{1}{\widehat{\sigma}_t^d(2\pi)^{d/2}}f(z,y)\exp\left(-\frac{\|z-\widehat{\alpha}_t x\|^2}{2\widehat{\sigma}_t^2}\right)\mathrm{d}z$ and plug it back to (I.3).

This completes the proof. □

Next, we provide lemma that provides bound on $h(x,y,t)$ and $\nabla h(x,y,t)$ in Lemma I.1

**Lemma I.2** (Lemma B.8 of (Fu et al., 2024b)). Under Assumption 3.2, we have the following bounds for $h(x,y,t)$ and $\frac{\widehat{\sigma}_t}{\widehat{\alpha}_t}\nabla h(x,y,t)$

$$
C_1 \le h(x,y,t) \le B, \quad \left\|\frac{\widehat{\sigma}_t}{\widehat{\alpha}_t}\nabla h(x,y,t)\right\|_\infty \le \sqrt{\frac{2}{\pi}}B,
$$

where $C_1$ and $B$ are the hyperparameters of $\mathcal{H}^\beta(\mathbb{R}^{d_x}\times[0,1]^{d_y},B)$ in Assumption 3.2.

**Remark I.1** (Bound on $h$ and $\nabla h$). We reiterate that Lemma I.2 drives the key distinction between the analyses in Theorem 3.1 and Theorem 3.2. Specifically, in Appendix H.3, the decomposed term containing the threshold $\epsilon_{\text{low}}$ results in lower approximation rate, while bounds on $h$ and $\nabla h$ eliminate the need of the threshold with $h$'s lower bound $C_1$, rendering faster approximation rate.

This step parallels Lemma H.2; however, the discretization differs due to the structure of $h$.

**Lemma I.3** (Clipping Integral, Lemma B.10 of Fu et al. (2024b)). Assume Assumption 3.2. Consider any integer vector $\kappa \in \mathbb{Z}_+^{d_x}$ with $\|\kappa\|_1 \le n$. There exists a constant $C(n,d_x)$, such that for any $x \in \mathbb{R}^{d_x}$ and $0 < \epsilon \le 0.99$, it holds

$$
\int_{\mathbb{R}^{d_x}\setminus B_x}\left|\left(\frac{\widehat{\alpha}_t x_0-x}{\widehat{\sigma}_t}\right)^\kappa\right|\cdot p(x_0|y)\cdot\frac{1}{\widehat{\sigma}_t^d(2\pi)^{d/2}}\exp\left(-\frac{\|\widehat{\alpha}_t x_0-x\|^2}{2\widehat{\sigma}_t^2}\right)\mathrm{d}x_0 \le \epsilon, \quad \text{(I.4)}
$$

where $\left(\frac{\widehat{\alpha}_t x_0-x}{\widehat{\sigma}_t}\right)^\kappa := \left(\left(\frac{\widehat{\alpha}_t x_0[1]_1-x[1]}{\widehat{\sigma}_t}\right)^{\kappa[1]},\left(\frac{\widehat{\alpha}_t x_0[2]-x[2]}{\widehat{\sigma}_t}\right)^{\kappa[2]},\dots,\left(\frac{\widehat{\alpha}_t x_0[d_x]-x[d_x]}{\widehat{\sigma}_t}\right)^{\kappa[d_x]}\right)$ and

$$
B_x := \left[\widehat{\alpha}_t x-C(n,d)\widehat{\sigma}_t\sqrt{\log\epsilon^{-1}},\widehat{\alpha}_t x+C(n,d)\widehat{\sigma}_t\sqrt{\log\epsilon^{-1}}\right]^{d_x}.
$$

**Step 2: Approximate $h$ and $\nabla h$ with Polynomials.** Similar to the construction of the diffused local polynomials, the following two lemmas render the first step approximation for $h(x,y,t)$ and $\nabla h(x,y,t)$ that captures the local smoothness.

**Lemma I.4** (Approximation with Diffused Local Polynomials, Lemma B.4 of (Fu et al., 2024b)). Assume Assumption 3.2. For sufficiently larger $N > 0$ and constant $C_2$, there exists a diffused local polynomial $f_1(x, y, t)$ with at most $N^{d+d_y}(d + d_y)^{k_1}$ monomials such that

$$|f_1(x, y, t) - h(x, y, t)| \lesssim BN^{-\beta} \log^{\frac{k_1}{2}} N,$$

for any $x \in [-C_x\sqrt{\log N}, C_x\sqrt{\log N}]^{d_x}, y \in [0, 1]^{d_y}$ and $t > 0$.

**Lemma I.5** (Counterpart of Lemma I.4, Lemma B.6 of (Fu et al., 2024b)). Assume Assumption 3.2. For sufficiently larger $N > 0$ and constant $C_2$, there exists a diffused local polynomial $f_2(x, y, t) \in \mathcal{T}_R^{h,s,r}$ with at most $N^{d_x+d_y}(d_x + d_y)^{k_1}$ monomials $f_2[i](x, y, t)$ such that

$$\left| f_2[i](x, y, t) - \left( \frac{\widehat{\sigma}_t}{\widehat{\alpha}_t} \nabla h(x, y, t) \right)[i] \right| \lesssim BN^{-\beta} \log^{\frac{k_1+1}{2}} N,$$

for any $x \in \mathbb{R}^{d_x}, y \in [0, 1]^{d_y}$ and $t > 0$.

**Approximation of the Score Approximator with Transformers**  We apply the universal approximation of transformers to approxiamte a score approximator constructed with $f_1$ and $f_2$.

**Lemma I.6** (Approximate the Score Approximator with Transformers). Assume Assumption 3.1. Let $C_x(d_x, \beta, C_1, C_2)$ be a positive constant. Let $f_3(x, y, t)$ be the target function:

$$f_3 := \frac{\widehat{\alpha}_t}{\widehat{\sigma}_t} \cdot \frac{f_2(x, y, t)}{f_1(x, y, t)} - \frac{C_2 x}{\alpha_t^2 + C_2\sigma_t^2}.$$

Then, for any $\epsilon \in (0, 1)$, any $x \in [-C_x\sqrt{\log N}, C_x\sqrt{\log N}]^{d_x}$, any $y \in [0, 1]^{d_y}$ and any $t \in [N^{-C_\sigma}, C_\alpha \log N]$, there exists a transformer $g(x, y, t) \in \mathcal{T}_R^{h,s,r}$ such that

$$\int_{\|x\| \leq C_x\sqrt{\log N}} \|g(x, y, t) - f_3(x, y, t)\|_2^2 \leq \epsilon^2.$$

Furthermore, the parameter bounds in transformer network function class follow Lemma H.5.

We introduce the counterpart of Lemma H.6. It is the core of the proof for Theorem I.1.

**Lemma I.7** (Score Approximation with Transformer). Assume Assumption 3.2. For sufficiently large integer $N$, there exists a mapping from transformer $\mathcal{T}_{\text{score}} \in \mathcal{T}_R^{h,s,r}$ such that

$$\int \left\| \mathcal{T}_{\text{score}} - \nabla \log h(x, y, t) + \frac{C_2 x}{\alpha_t^2 + C_2\sigma_t^2} \right\|_2^2 dx \lesssim \frac{B^2}{\sigma_t^2} N^{-2\beta}(\log N)^{k_1+1},$$

for any $x \in [-C_x\sqrt{\log N}, C_x\sqrt{\log N}]^{d_x}, y \in [0, 1]^{d_y}$ and $t \in [N^{-C_\sigma}, C_\alpha \log N]$. Furthermore, the parameter bounds in transformer network function class follow Lemma H.6.

*Proof.* Our proof follows the proof structure of (Fu et al., 2024b, Proposition B.3).

We establish the the first-step approximator $f_3$ with the form:

$$f_3(x, y, t) := \frac{\widehat{\alpha}_t}{\widehat{\sigma}_t} \cdot \frac{f_2(x, y, t)}{f_1(x, y, t)} - \frac{C_2 x}{\alpha_t^2 + C_2\sigma_t^2}.$$

We derive the error bound on the approximation of the first term containing Taylor polynomials in $f_3$. We incorporate second term containing the linear function in $x$ into the the transformer architecture.

We proceed with the following two steps.

- **Step A. Approximate Scroe Function with $f_3$.**

We first construct $f_1(x, y, t)$ and $f_2(x, y, t)$ from Lemma I.4 and Lemma I.5 to approximate $h(x, y, t)$ and $\nabla h(x, y, t)$ respectively.

From Lemma I.2, we have $C_1 \leq h \leq B$ and $\left\|\frac{\widehat{\sigma}_t \nabla h}{\widehat{\alpha}_t}\right\|_\infty \leq \sqrt{\frac{2}{\pi}}B$. Next, by Lemma I.4 and Lemma I.5, we select a sufficiently large $N$ such that $\frac{C_1}{2} \leq f_1 \leq 2B$ and $f_2 \leq B$.

Without loss of generality, we begin by bounding the first coordinate of $\nabla h$, denoted as $\nabla h[1]$:

$$
\begin{aligned}
&\left|\frac{\nabla h[1]}{h} - \frac{\widehat{\alpha}_t}{\widehat{\sigma}_t}\frac{f_2[1]}{f_1}\right| \\
&\leq \left|\frac{\nabla h[1]}{h} - \frac{\nabla h[1]]}{f_1}\right| + \left|\frac{\nabla h[1]}{f_1} - \frac{\widehat{\alpha}_t}{\widehat{\sigma}_t}\frac{f_2[1]]}{f_1}\right|, \\
&\leq \left|\frac{\nabla h[1]]}{h \cdot f_1}\right||f_1 - h| + \frac{\widehat{\alpha}_t}{\widehat{\sigma}_t}\left|\frac{1}{f_1}\right|\left|f_2 - \frac{\widehat{\sigma}_t}{\widehat{\alpha}_t}\nabla h[1]]\right|, \\
&\lesssim \frac{\widehat{\alpha}_t}{\widehat{\sigma}_t}\left(|f_1 - h| + \left|f_2 - \frac{\widehat{\sigma}_t}{\widehat{\alpha}_t}\nabla h[1]\right|\right), &&\text{(By bounds on } h, \nabla h, f_1, f_2) \\
&\lesssim \frac{\widehat{\alpha}_t}{\widehat{\sigma}_t}\left(BN^{-\beta}(\log N^{\frac{k_1}{2}} + BN^{-\beta}(\log N^{\frac{k_1+1}{2}}))\right), &&\text{(By Lemma I.4 and Lemma I.5)} \\
&\lesssim \frac{1}{\sigma_t}\left(BN^{-\beta}(\log N^{\frac{k_1+1}{2}})\right).
\end{aligned}
$$

Note that in the last line, we utilize

$$
\frac{\widehat{\alpha}_t}{\widehat{\sigma}_t} = \frac{\alpha_t}{\sigma_t}\frac{1}{\sqrt{\alpha_t^2 + C_2\sigma_t^2}} = \frac{1}{\sigma_t}\frac{1}{\sqrt{1 + C_2\left(\sigma_t/\alpha_t\right)^2}} = \frac{1}{\sigma_t}\frac{1}{\sqrt{1 + C_2\frac{\sigma_t^2}{1-\sigma_t^2}}} = \mathcal{O}(\sigma_t^{-1}).
$$

By the symmetry of each coordinate in $\nabla h$, we obtain the $\ell_\infty$ bounds:

$$
\left\|\frac{\nabla h_{(x,y,t)}}{h(x,y,t)} - \frac{\widehat{\alpha}_t}{\widehat{\sigma}_t}\frac{f_2(x,y,t)}{f_1(x,y,t)}\right\|_\infty \lesssim \frac{B}{\sigma_t}N^{-\beta}(\log N)^{\frac{k_1+1}{2}}. \tag{I.5}
$$

- **Step B. Approximate $f_3$ with Transformer $\mathcal{T}_{\text{score}}$.** We use transformers to approximate $f_3$ to an accuracy of order $N^{-\beta}$ such that it aligns with the error order in (H.4). By Lemma I.6, we have

$$
\int \|\mathcal{T}_{\text{score}}(x, y, t) - f_3(x, y, t)\|_2^2 \mathrm{d}x \leq \epsilon^2.
$$

By setting $\epsilon = N^{-\beta}$, it holds

$$
\begin{aligned}
&\int p_t^2\|\mathcal{T}_{\text{score}}(x, y, t) - \nabla \log p_t(x|y)\|_2^2\mathrm{d}x \\
&\leq \int p_t^2\|\mathcal{T}_{\text{score}}(x, y, t) - f_3(x, y, t)\|_2^2\mathrm{d}x + \int p_t^2\|f_3(x, y, t) - \nabla \log p_t(x|y)\|_2^2\mathrm{d}x \\
&\hphantom{\leq} \qquad\qquad\qquad\qquad\qquad\qquad\qquad\qquad\qquad\qquad\qquad\quad \text{(By triangle inequality)} \\
&\leq \int \|\mathcal{T}_{\text{score}}(x, y, t) - f_3(x, y, t)\|_2^2\mathrm{d}x + \int p_t^2\|f_3(x, y, t) - \nabla \log p_t(x|y)\|_2^2\mathrm{d}x \\
&\hphantom{\leq} \qquad\qquad\qquad\qquad\qquad\qquad\qquad\qquad\qquad\qquad\qquad\quad \text{(By } p_t(x|y) \in [0, 1]) \\
&\lesssim \int \|\mathcal{T}_{\text{score}}(x, y, t) - f_3(x, y, t)\|_2^2\mathrm{d}x + \frac{B^2}{\sigma_t^2}N^{-2\beta}(\log N)^{k_1+1}\int \mathrm{d}x &&\text{(By (I.5))} \\
&\lesssim \int \|\mathcal{T}_{\text{score}}(x, y, t) - f_3(x, y, t)\|_2^2\mathrm{d}x + \frac{B^2}{\sigma_t^4}N^{-2\beta}(\log N)^{\frac{d_x}{2}+k_1+1} &&\text{(By } \|x\|_\infty \leq C_x\sqrt{\log N})
\end{aligned}
$$

$$\leq \frac{B^2}{\sigma_t^4} N^{-2\beta} (\log N)^{\frac{d_x}{2}+k_1+1}. \qquad\qquad\qquad (\text{ By Lemma I.6 })$$

Furthermore, the parameter bounds in the transformer network class follows Lemma H.6.

This completes the proof. $\qquad\qquad\qquad\qquad\qquad\qquad\qquad\qquad\qquad\qquad\qquad \square$

### I.2    MAIN PROOF OF THEOREM 3.2

Similar to the proof of Theorem 3.1, we implement the truncation due to the unboundedness of $x$.

**Lemma I.8** (Lemma B.2 of (Fu et al., 2024b)). Assume Assumption 3.2. For any $R_3 > 1$, it holds

$$\int_{\|x\|_\infty \geq R_3} p_t(x|y)\mathrm{d}x \lesssim R_3 \exp(-C_2' R_2^2).$$

Moreover, it holds

$$\int_{\|x\|_\infty \geq R_3} \|\nabla \log p_t(x|y)\|_2^2 p_t(x|y)\mathrm{d}x \lesssim \frac{1}{\sigma_t^2} R_3^3 \exp(-C_2' R_3^3),$$

where $C_2' = C_2/(2\max\{1, C_2\})$.

Unlike results under Assumption 3.1, the explicit form of $p_t(x|y)$ in (I.1) along with the upper and lower bound on the joint distribution Lemma I.2 allow us to skip the threshold $\epsilon_{\text{low}}$ as in Lemma H.8.

We state the formal version of Theorem 3.2.

**Theorem I.1** (Score Approximation, Formal Version of Theorem 3.2). Assume Assumption 3.2. For any precision parameter $0 < \epsilon < 1$ and smoothness parameter $\beta > 0$, let $\epsilon \leq \mathcal{O}(N^{-\beta})$ for some $N \in \mathbb{N}$. Let $C_\alpha$ and $C_\sigma$ be some positive absolute constants. For any $y \in [0,1]^{d_y}$ and $t \in [N^{-C_\sigma}, C_\alpha \log N]$, there exists a $\mathcal{T}_{\text{score}}(x, y, t) \in \mathcal{T}_R^{h,s,r}$ such that

$$\int_{\mathbb{R}^{d_x}} \|\mathcal{T}_{\text{score}}(x, y, t) - \nabla \log p_t(x|y)\|_2^2 \cdot p_t(x|y)\mathrm{d}x = \mathcal{O}\left(\frac{B^2}{\sigma_t^2} \cdot N^{-2\beta} \cdot (\log N)^{\beta+1}\right).$$

Notably, for $\epsilon = \mathcal{O}(N^{-\beta})$, the approximation error has the upper bound $\widetilde{\mathcal{O}}(\epsilon^2/\sigma_t^2)$. The parameter bounds in the transformer network class satisfy

$$C_{KQ}, C_{KQ}^{2,\infty} = O\big(N^{4\beta d+2\beta}(\log N)^{4d+2}\big); C_{OV}, C_{OV}^{2,\infty} = O\big(N^{-\beta}\big);$$

$$C_F, C_F^{2,\infty} = O\big(N^\beta(\log N)^{\frac{d_x+\beta+3}{2}}\big); \ C_E = O(1); \ C_\mathcal{T} = O(\sqrt{\log N}/\sigma_t),$$

where $O(\cdot)$ hides all polynomial factors depending on $d_x, d, L, \beta, C_1, C_2$.

*Proof of Theorem 3.2.* We apply Lemma I.7 and Lemma I.8.

Specifiaclly, we take $C_x = \sqrt{\frac{2\beta}{C_2'}}$ and $R_3 = C_x\sqrt{\log N}$.

Next, recall transformer parameter bounds in Lemma I.7. We have $\|\mathcal{T}_{\text{score}}\|_2 \leq \sqrt{\log N}/\sigma_t$.
Therefore,

$$\int_{\mathbb{R}^{d_x}} \|\mathcal{T}_{\text{score}} - \nabla \log p_t\|_2^2 \cdot p_t \mathrm{d}x \qquad\qquad\qquad (\text{By expanding } \ell_2 \text{ norm})$$

$$\leq \underbrace{\int_{\|x\|_\infty > \sqrt{\frac{2\beta}{C_2'} \log N}} \left( 2\|\mathcal{T}_{\text{score}}\|_2^2 + 2\|\nabla \log p_t\|_2^2 \right) p_t \mathrm{d}x}_{\text{(A.1)}} + \underbrace{\int_{\|x\|_\infty \leq \sqrt{\frac{2\beta}{C_2'} \log N}} \left( \|\mathcal{T}_{\text{score}} - \nabla \log p_t\|_2^2 \right) \cdot p_t \mathrm{d}x,}_{\text{(A.2)}}$$

$$\left( \text{By } \ell_2 \text{ bound on } \mathcal{T}_{\text{score}} \text{ and Lemma I.7} \right)$$

$$\lesssim \int_{\|x\|_\infty > \sqrt{\frac{2\beta}{C_2'} \log N}} \left( \frac{2 \cdot \log N}{\sigma_t^2} + 2\|\nabla \log p_t\|_2^2 \right) \cdot p_t \mathrm{d}x + \frac{B^2}{\sigma_t^2} N^{-2\beta} (\log N)^{k_1+1},$$

$$\left( \text{By Lemma I.8} \right)$$

$$\lesssim 2d_x \frac{\sqrt{\log N}}{\sigma_t^2} \left( \frac{2\beta}{C_2'} \log N \right)^{\frac{1}{2}} N^{-2\beta} + \frac{2}{\sigma_t^2} \left( \frac{2\beta}{C_2'} \log N \right)^{\frac{3}{2}} N^{-2\beta} + \frac{B^2}{\sigma_t^2} N^{-2\beta} (\log N)^{k_1+1},$$

$$\left( \text{By dropping lower order term} \right)$$

$$\lesssim \frac{B^2}{\sigma_t^2} N^{-2\beta} (\log N)^{\beta+1}.$$

This completes the proof. $\qquad\square$

## J  Proof of the Estimation Results for Conditional DiTs

**Overview of Proof Strategy of Theorem 3.3.**

**Step 0.  Preliminaries.** We introduce the mixed risk that accounts for risk with the distribution of the mask signal in Definition J.1. We restate the loss function and the score matching technique in Definition J.2.

**Step 1.  Truncate the Domain of the Risk.** We truncate the domain of the loss function in order to obtain finite covering number of transformer network class. Precise definition of the truncated loss function class is in Definition J.4. We bound the error from the truncation from the assumed light tail condition in Lemma J.1.

**Step 2.  Derive the Covering Number of Transformer Network.** We introduce the covering number of a given function class in Definition J.5. We provide lemma detailing the calculation of the covering number for transformer architecture in Lemma J.2. We derive the covering numbers under the respective parameter configurations for our two previous main results in Lemma J.3.

**Step 3.  Bound the True Risk on Truncated Domain.** With the previous steps, we present the upper-bound of the mixed risk in Lemma J.4.

**Overview of Proof Strategy of Theorem 3.4.** We decompose the total variation into three components and we bound the separately.

**Step 1.** We bound the total variation distance between the true distributions evaluated at $t = 0$ and early-stopping time $t = t_0$.

**Step 2.** We bound the total variation between the true distribution at $t_0$ and the reverse process distribution using the true score function.

**Step 3.** We bound the total variation between the reverse process distributions using the true and estimated score functions at $t_0$.

**Organization.** Appendix J.1 includes auxiliary lemmas for supporting our proof of Theorem 3.3. Appendix J.2 includes the main proof of Theorem 3.3. Appendix J.3 includes auxiliary lemmas for supporting our proof of Theorem 3.4. Appendix J.4 includes the main proof of Theorem 3.4.

### J.1  Auxiliary Lemmas for Theorem 3.3

**Step 0: Preliminary Framework.** We evaluate the quality of the estimator $s_W$ through the risk:

$$\mathcal{R}(s_W) \coloneqq \int_{t_0}^{T} \frac{1}{T - t_0} \mathbb{E}_{x_t, y} \| s_W(x_t, y, t) - \nabla \log p_t(x_t | y) \|_2^2 \mathrm{d}t. \tag{J.1}$$

**Definition J.1** (Mixed Risk). The risk (J.1) considers label $y$ throughout whole the diffusion process. We refer to it as the conditional score risk. In contrast, we have the mixed risk $\mathcal{R}_m$ that accounts for the distribution of the mask signal $\tau = \{\emptyset, \mathrm{id}\}$ with $P(\tau = \emptyset) = P(\tau = \mathrm{id}) = 0.5$:

$$\mathcal{R}_m(s_W) \coloneqq \int_{t_0}^{T} \frac{1}{T - t_0} \mathbb{E}_{(x_t, y, \tau)} \left[ \| s_W(x_t, \tau y, t) - \nabla \log p_t(x_t | \tau y) \|_2^2 \right] \mathrm{d}t, \tag{J.2}$$

**Remark J.1.** Given score estimator $\widehat{s}$ trained from empirical loss (F.8), the conditional score risk is upper-bounded by twice of the mixed risk. That is, we have $\mathcal{R}(\widehat{s}) \leq 2\mathcal{R}_m(\widehat{s})$ by observing

$$\mathcal{R}_m(\widehat{s}) = \frac{1}{2} \int_{t_0}^{T} \frac{1}{T - t_0} \mathbb{E}_{x_t} \left[ \| \widehat{s}(x_t, \emptyset, t) - \nabla \log p_t(x_t) \|_2^2 \right] \mathrm{d}t + \frac{1}{2} \mathcal{R}(\widehat{s}).$$

**Definition J.2** (Loss Function and Score Matching). Let $x = x_t | x_0$ denote the random variable following Gaussian distribution $N(\alpha_t x_0, \sigma_t^2 I_{d_x})$, we define loss function and score matching loss:

$$\ell(x, y; s_W) := \int_{T_0}^{T} \frac{1}{T - T_0} \mathbb{E}_{\tau, x} \left[ \| s_W(x_t, \tau y, t) - \nabla \log p_t(x_t | x_0) \|_2^2 \right] dt,$$

$$\mathcal{L}(s_W) := \int_{t_0}^{T} \frac{1}{T - t_0} \mathbb{E}_{x_0, y} \left[ \mathbb{E}_{\tau, x} \left[ \| s_W(x_t, \tau y, t) - \nabla \log p_t(x_t | x_0) \|_2^2 \right] \right] dt.$$

**Remark J.2.** Given i.i.d samples $\{x_{0,i}, y_i\}_{i=1}^n$, we write $\ell(x_i, y_i; s_W)$ with the understanding that $x_i = x_t | x_{0,i}$. When context is clear, we use $\ell(x_i, y_i; s_W)$ and $\ell(x_{0,i}, y_i; s_W)$; $\{x_{0,i}, y_i\}_{i=1}^n$ and $\{x_i, y_i\}_{i=1}^n$ interchangeably.

**Remark J.3.** By (Vincent, 2011), $\mathcal{L}(s_W)$ and $\mathcal{R}_m(s_W)$ differ by a constant that is inconsequential to the minimization. Therefore, minimizing the mixed risk is equivalent to minimizing the score matching loss

**Definition J.3** (Empirical Risk). Consider a score estimator $s_W \in \mathcal{T}_R^{h,s,r}$. Recall the definition of empirical loss: $\widehat{\mathcal{L}}(s_W) = \sum_{i=1}^n \frac{1}{n} \ell(x_i, y_i; s_W)$. Let $s^\circ := \nabla \log p_t(x|y)$, we define empirical risk:

$$\widehat{\mathcal{R}}_m(s_W) := \widehat{\mathcal{L}}(s_W) - \widehat{\mathcal{L}}(s^\circ) = \sum_{i=1}^n \frac{1}{n} \ell(x_i, y_i; s_W) - \sum_{i=1}^n \frac{1}{n} \ell(x_i, y_i; s^\circ).$$

**Remark J.4.** The key distinction between $\mathcal{R}_m$ and $\mathcal{L}$ lies in their formulations. Specifically, $\mathcal{R}_m$ measures the expected difference between $s_W$ and the ground truth $\nabla \log p_t(x|y)$ with respect to $(x_t, y, \tau)$. In contrast, the score matching loss $\mathcal{L}$ provides an explicit calculation based on the sample $\{x_{0,i}, y_i\}_{i=1}^n$. With the tower property of conditional expectation, $\mathcal{L}$ measures the expected difference between $s_W$ and $\nabla \log p_t(x|x_0)$ first with respect to $(x_t|x_0, \tau)$, and then with respect to $x_0$.

**Remark J.5.** Observe (I): $s^\circ = \nabla \log p_t(x|y)$ is the ground truth of score function with $\mathcal{R}_m(s^\circ) = 0$, and (II): By (Vincent, 2011), $\mathcal{R}_m$ and $\mathcal{L}$ differ by a constant. Based on (I) and (II), we define the empirical risk $\widehat{\mathcal{R}}_m$ using the score matching loss as an intermediary: $\mathcal{R}_m(s_W) = \mathcal{R}_m(s_W) - \mathcal{R}_m(s^\circ) = \mathcal{L}(s_W) - \mathcal{L}(s^\circ)$. This leads to the definition of the empirical risk $\widehat{\mathcal{R}}_m$ as a practical approximation of the true risk difference $\mathcal{R}_m(s_W) - \mathcal{R}_m(s^\circ)$.

**Remark J.6.** For any score estimator $s_W \in \mathcal{T}_R^{h,s,r}$ obtained from the training with i.i.d. samples $\{x_i, y_i\}_{i=1}^n$, it holds $\mathbb{E}_{\{x_i, y_i\}_{i=1}^n}[\widehat{\mathcal{R}}_m(s_W)] = \mathcal{R}_m(s_W)$. This follows from direct calculation with Definition J.3 and the i.i.d. assumption.

**Step 1: Domain Truncation of the Risk.** We define the loss function with truncated domain. This is essential for obtaining finite covering number for transformer network class.

**Definition J.4** (Truncated Loss). We define the truncated domain of the score function by $\mathcal{D} := [-R_\mathcal{T}, R_\mathcal{T}]^{d_x} \times [0,1]^{d_y} \cup \emptyset$. Given loss function $\ell(x, y; s_W)$, we define the truncated loss:

$$\ell^{\text{trunc}}(x, y; s_W) := \ell(x, y; s_W) \mathbb{1}\{\|x\|_\infty \leq R_\mathcal{T}\}. \tag{J.3}$$

We define $\mathcal{L}^{\text{trunc}}(s_W) := \mathcal{L}(s_W) \mathbb{1}\{\|x\|_\infty \leq R_\mathcal{T}\}$, $\mathcal{R}_m^{\text{trunc}}(s_W) := \mathcal{R}_m(s_W) \mathbb{1}\{\|x\|_\infty \leq R_\mathcal{T}\}$ and $\widehat{\mathcal{R}}_m^{\text{trunc}}(s_W) := \widehat{\mathcal{R}}_m(s_W) \mathbb{1}\{\|x\|_\infty \leq R_\mathcal{T}\}$. We define the function class of the truncated loss by

$$\mathcal{S}(R_\mathcal{T}) := \{\ell(\cdot, \cdot; s_W) : \mathcal{D} \to \mathbb{R} \mid s_W \in \mathcal{T}_R^{h,s,r}\}. \tag{J.4}$$

Next, we introduce the following lemma dealing with the error bound for the truncation of the loss.

**Lemma J.1** (Truncation Error, Lemma D.1 of (Fu et al., 2024b)). Consider the truncated loss $\ell^{\text{trunc}}(x, y; s_W)$ and $t \in [n^{-\mathcal{O}(1)}, \mathcal{O}(\log n)]$. Under Assumption 3.1, we have $|\ell(x, y; s_W)| \lesssim 1/t_0$.

Consider the parameter configuration in Theorem 3.1, it holds:

$$\mathbb{E}_{x,y}\left[\left|\ell(x,y,t) - \ell^{\text{trunc}}(x,y,s)\right|\right] \lesssim \exp\left(-C_2 R_{\mathcal{T}}^2\right) R_{\mathcal{T}}\left(\frac{1}{t_0}\right).$$

Moreover, under Assumption 3.2, we have $|\ell(x,y;s_W)| \lesssim \log(1/t_0)$. Consider the parameter configuration in Theorem I.1, it holds:

$$\mathbb{E}_{x,y}\left[\left|\ell(x,y,t) - \ell^{\text{trunc}}(x,y,s)\right|\right] \lesssim \exp\left(-C_2 R_{\mathcal{T}}^2\right) R_{\mathcal{T}} \log\left(\frac{1}{t_0}\right).$$

**Step 2: Covering Number of Transformer Network Class.** We begin with the definition.

**Definition J.5** (Covering Number). Given a function class $\mathcal{F}$ and a data distribution $P$. Sample n data points $\{X_i\}_{i=1}^n$ from $P$, then the covering number $\mathcal{N}(\epsilon, \mathcal{F}, \{X_i\}_{i=1}^n, \|\cdot\|)$ is the smallest size of a collection (a cover) $\mathcal{C} \subset \mathcal{F}$ such that for any $f \in \mathcal{F}$, there exist $\widehat{f} \in \mathcal{C}$ satisfying

$$\max_i \left\|f(X_i) - \widehat{f}(X_i)\right\| \leq \epsilon.$$

Further, we define the covering number with respect to the data distribution as

$$\mathcal{N}(\epsilon, \mathcal{F}, \|\cdot\|) = \sup_{\{X_i\}_{i=1}^n \sim P} \mathcal{N}(\epsilon, \mathcal{F}, \{X_i\}_{i=1}^n, \|\cdot\|).$$

Next, we introduce the following lemma that provides results for the calculation of the covering number for transformer networks.

**Lemma J.2** (Modified from Theorem A.17 of (Edelman et al., 2022)).

Let $\mathcal{T}_R^{h,s,r}(C_{\mathcal{T}}, C_Q^{2,\infty}, C_Q, C_K^{2,\infty}, C_K, C_V^{2,\infty}, C_V, C_O^{2,\infty}, C_O, C_E, C_{f_1}^{2,\infty}, C_{f_1}, C_{f_2}^{2,\infty}, C_{f_2}, L_{\mathcal{T}})$

represent the class of functions of one transformer block satisfying the norm bound for matrix and Lipsichitz property for feed-forward layers. Then for all data point $\|X\|_{2,\infty} \leq R_{\mathcal{T}}$ we have

$$\log \mathcal{N}(\epsilon_c, \mathcal{T}_R^{h,s,r}, \|\cdot\|_2)$$
$$\leq \frac{\log(n L_{\mathcal{T}})}{\epsilon_c^2} \cdot \left(\alpha^{\frac{2}{3}}\left(d^{\frac{2}{3}}\left(C_F^{2,\infty}\right)^{\frac{4}{3}} + d^{\frac{2}{3}}\left(2(C_F)^2 C_{OV} C_{KQ}^{2,\infty}\right)^{\frac{2}{3}} + 2\left((C_F)^2 C_{OV}^{2,\infty}\right)^{\frac{2}{3}}\right)\right)^3,$$

where $\alpha := (C_F)^2 C_{OV}(1 + 4C_{KQ})(R_{\mathcal{T}} + C_E)$.

Then, we derive the covering number under transformer weights configuration in Theorem 3.1 and Theorem I.1.

**Lemma J.3** (Covering Number for $\mathcal{S}(R_{\mathcal{T}})$). Consider $\epsilon_c > 0$ and $\|x\|_\infty \leq R_{\mathcal{T}}$. Given sample $\{x_i, y_i\}_{i=1}^n$, the $\epsilon_c$-covering number for $\mathcal{S}(R_{\mathcal{T}})$ with respect to $\|\cdot\|_{L_\infty}$ under Theorem 3.1 and Theorem I.1 satisfy

$$\log \mathcal{N}\left(\epsilon_c, \mathcal{S}(R_{\mathcal{T}}), \|\cdot\|_\infty\right) \lesssim \frac{\log n}{\epsilon_c^2} N^{\nu_1} (\log N)^{\nu_2} (R_{\mathcal{T}})^2,$$

where $\nu_1 = 16\beta d + 12\beta$ and $\nu_2 = 20 d_x + 4\beta + 18$

*Proof.* Applying Lemma J.2, we have

$$\log \mathcal{N}(\epsilon_c, \mathcal{T}_R^{h,s,r}, \|\cdot\|_2)$$

$$
\le \frac{\log n}{\epsilon_c^2} \cdot \alpha^2 \left( \underbrace{2 \left((C_F)^2 C_{OV}^{2,\infty}\right)^{\frac{2}{3}}}_{\text{(I)}} + \underbrace{(d^{\frac{2}{3}} \left(C_F^{2,\infty}\right)^{\frac{4}{3}}}_{\text{(II)}} + \underbrace{d^{\frac{2}{3}} \left(2(C_F)^2 C_{OV} C_{KQ}^{2,\infty}\right)^{\frac{2}{3}}}_{\text{(III)}} \right)^3, \qquad \text{(J.5)}
$$

where $\alpha := (C_F)^2 C_{OV}(1 + 4C_{KQ})(R_{\mathcal{T}} + C_E)$.

Note that we drop $L_{\mathcal{T}}$ because it is inconsequential under either Assumption 3.1 or Assumption 3.2.

- **Step A: Covering Number of Transformer Network Configuration.**

  Recall that the parameter bounds in Theorem 3.1 follows

  $$
  C_{KQ}, C_{KQ}^{2,\infty} = O\big(N^{4\beta d+2\beta}(\log N)^{4d+2}\big); C_{OV}, C_{OV}^{2,\infty} = O(N^{-\beta});
  $$
  $$
  C_F, C_F^{2,\infty} = O\big(N^{\beta}(\log N)^{\frac{d_x+\beta+3}{2}}\big); C_E = O(1); C_{\mathcal{T}} = O(\sqrt{\log N}/\sigma_t^2),
  $$

  Among three terms, it is obvious that (III) dominates the (II) and (I). Therefore,

  $$
  (C_F)^2 C_{OV} C_{KQ}^{2,\infty}
  $$
  $$
  = O(\underbrace{N^{4\beta}(\log N)^{2d_x+2\beta+4}}_{(C_F)^4} \underbrace{N^{-2\beta}}_{(C_{OV})^2} \underbrace{N^{8\beta d+4\beta}(\log N)^{8d_x+4}}_{(C_{KQ}^{2,\infty})^2})
  $$
  $$
  = O(N^{8\beta d+6\beta}(\log N)^{10d_x+2\beta+8}).
  $$

  Therefore,

  $$
  \log \mathcal{N}(\epsilon_c, \mathcal{T}_R^{h,s,r}, \|\cdot\|_2) \lesssim \frac{\alpha^2 \log(nL_{\mathcal{T}})}{\epsilon_c^2}(N^{8\beta d+6\beta}(\log N)^{10d_x+2\beta+8}).
  $$

  By Lemma J.2, we have

  $$
  \alpha
  $$
  $$
  := (C_F)^2 C_{OV}(1 + 4C_{KQ})(R_{\mathcal{T}} + C_E)
  $$
  $$
  \lesssim \underbrace{N^{2\beta}(\log N)^{d_x+\beta+2}}_{(C_F)^2} \underbrace{N^{-\beta}}_{(C_{OV})} \underbrace{N^{4\beta d+2\beta}(\log N)^{4d_x+2}}_{(C_{KQ})}(R_{\mathcal{T}} + C_E) \qquad \text{(By the definition of } \alpha\text{)}
  $$
  $$
  = O(R_{\mathcal{T}} N^{4\beta d+3\beta}(\log N)^{5d_x+\beta+4}).
  $$

  Altogether, we have

  $$
  \log \mathcal{N}(\epsilon_c, \mathcal{T}_R^{h,s,r}, \|\cdot\|_2) \lesssim \frac{\log(nL_{\mathcal{T}})}{\epsilon_c^2} R_{\mathcal{T}}^2 N^{16\beta d+12\beta}(\log N)^{20d_x+4\beta+16}.
  $$

  Further, by $\|\cdot\|_\infty \le \|\cdot\|_2$, we have

  $$
  \log \mathcal{N}(\epsilon_c, \mathcal{T}_R^{h,s,r}, \|\cdot\|_\infty) \lesssim \frac{\log(nL_{\mathcal{T}})}{\epsilon_c^2} R_{\mathcal{T}}^2 N^{16\beta d+12\beta}(\log N)^{20d_x+4\beta+16}. \qquad \text{(J.6)}
  $$

  Furthermore, the same bounds hold for results under Theorem I.1.

- **Step B: Covering Number under Domain Truncation.**

  We extend the result to the covering number for $\mathcal{S}(R_{\mathcal{T}})$ defined in (J.4).

For score estimators $s_1(x, y, t), s_2(x, y, t) \in \mathcal{T}_R^{h,s,r}$ such that $\|s_1 - s_2\|_{L_\infty, \mathcal{D}} \leq \epsilon$, by lemma D.3 in Fu et al. (2024b), the difference between the loss $\ell(\cdot, \cdot, s_1)$ and $\ell(\cdot, \cdot, s_2)$ in $L_\infty$ is bounded by

$$|\ell(\cdot, \cdot, s_1) - \ell(\cdot, \cdot, s_2)| \lesssim \epsilon \log N. \tag{J.7}$$

By replacing $\epsilon_c$ with $\epsilon_c / \log N$ in (J.6), we obtain the log-covering number

$$\log \mathcal{N}\left(\epsilon_c, \mathcal{S}(R_\mathcal{T}), \|\cdot\|_\infty\right) \lesssim \frac{\log(nL_\mathcal{T})}{\epsilon_c^2} R_\mathcal{T}^2 N^{\nu_1} (\log N)^{\nu_2}.$$

where $\nu_1 = 16\beta d + 12\beta$ and $\nu_2 = 20 d_x + 4\beta + 18$.

This completes the proof. $\qquad\square$

**Step 3: Bound the True Risk on Truncated Domain.** We begin with the definition.

**Definition J.6.** Let $s^\circ := \nabla \log p_t(x|y)$ denote the ground truth of score function for simplicity. Given i.i.d samples $\{x_i, y_i\}_{i=1}^n$ and a score estimator $s_W \in \mathcal{T}_R^{h,s,r}$, we define the difference function:

$$\Delta_n(s_W, s^\circ) := \left| \mathbb{E}_{\{x_i, y_i\}_{i=1}^n} \left[ \widehat{\mathcal{R}}_m^{\text{trunc}}(s_W) - \mathcal{R}_m^{\text{trunc}}(s_W) \right] \right|.$$

**Remark J.7.** Note that the difference function $\Delta_n(s_W, s^\circ)$ measures the expected difference between the truncated empirical risk and the truncated mixed risk with respect to the training sample. Since the true risk is unattainable, we construct $\Delta_n(s_W, s^\circ)$ serving as an intermediate that allows us to derive the upper-bound on the mixed risk. Surprisingly, we are able to handle the upper-bound of the difference function, presented in Lemma J.4.

**Definition J.7.** Given the truncated loss function class $\mathcal{S}(R_\mathcal{T})$, we define its $\epsilon_c$-covering with the minimum cardinality in the $L^\infty$ metric as $\mathcal{L}_\mathcal{N} := \{\ell_1, \ell_2, \ldots, \ell_\mathcal{N}\}$. Moreover, we define $\ell_J \in \mathcal{L}_\mathcal{N}$ with random variable $J$. By definition, there exist $\ell_J \in \mathcal{L}_\mathcal{N}$ such that $\|\ell_J - \ell(x_i, y_i; s_W)\|_\infty \leq \epsilon_c$.

Note that Lemma J.3 provides the upper-bound on the $\epsilon_c$-covering number of $\mathcal{S}(R_\mathcal{T})$ for score estimator trained from transformer network class. Next, we bound the difference function.

**Lemma J.4** (Bound on Difference Function). Consider i.i.d training samples $\{x_{0,i}, y_i\}_{i=1}^n$ and score estimator $\widehat{s}$ from (2.1). Under Assumption 3.1 and parameter configuration in Theorem 3.1, it holds:

$$\Delta_n(\widehat{s}, s^\circ) \lesssim \mathbb{E}_{\{x_i, y_i\}_{i=1}^n} \left[ \widehat{\mathcal{R}}_m(\widehat{s}) \right] + \frac{1}{t_0} \left( R_\mathcal{T} \exp\left(-C_2 R_\mathcal{T}^2\right) + \frac{1}{n} \log \mathcal{N} \right) + 7\epsilon_c,$$

where $\mathcal{N}(\epsilon_c, \mathcal{T}_R^{h,s,r}, \|\cdot\|_2)$ is the covering number of transformer network class. Moreover, Under Assumption 3.2 and parameter configuration in Theorem I.1, it holds:

$$\Delta_n(\widehat{s}, s^\circ) \lesssim \mathbb{E}_{\{x_i, y_i\}_{i=1}^n} \left[ \widehat{\mathcal{R}}_m(\widehat{s}) \right] + \log \frac{1}{t_0} \left( R_\mathcal{T} \exp\left(-C_2 R_\mathcal{T}^2\right) + \frac{1}{n} \log \mathcal{N} \right) + 7\epsilon_c.$$

*Proof.* In this proof, we let $z_i := (x_{0,i}, y_i)$, $\widehat{\ell}(z_i) := \ell^{\text{trunc}}(z_i; \widehat{s})$ and $\ell^\circ(z_i) := \ell^{\text{trunc}}(z_i; s^\circ)$. For simplicity, we use $\kappa = 1/t_0$ for the case in Theorem 3.1 and $\kappa = \log(1/t_0)$ for the case in Theorem I.1.

- **Step A: Rewrite the true risk.**

  To derive the upper-bound of the true risk, we introduce a different set of i.i.d samples $\{x'_{0,i}, y'_i\}_{i=1}^n$ independent of the training data drawn from the same distribution.

This allows us to rewrite the true risk as:

$$\mathcal{R}_m(\widehat{s}) - \mathcal{R}_m(s^\circ) = \mathcal{L}(\widehat{s}) - \mathcal{L}(s^\circ) = \mathbb{E}_{\{x_i', y_i'\}_{i=1}^n} \left[ \frac{1}{n} \sum_{i=1}^n \left( \ell(x_i', y_i', \widehat{s}) - \ell(x_i', y_i', s^\circ) \right) \right]. \quad \text{(J.8)}$$

With (J.8), we rewrite the difference function:

$$\Delta_n(\widehat{s}, s^\circ) = \left| \frac{1}{n} \mathbb{E}_{\{z_i, z_i'\}_{i=1}^n} \left[ \sum_{i=1}^n \left( \left( \widehat{\ell}(z_i) - \ell^\circ(z_i) \right) - \left( \widehat{\ell}(z_i') - \ell^\circ(z_i') \right) \right) \right] \right|. \quad \text{(J.9)}$$

- **Step B: Introduce the $\epsilon_c$-covering.**

  Before further decomposing (J.9), we introduce three definitions.

  - $\omega_J(z) := \ell_J(z) - \ell^\circ(z)$ and $\widehat{\omega}(z) := \widehat{\ell}(z) - \ell^\circ(z)$.

  - $\Omega := \max\limits_{1 \leq J \leq \mathcal{N}} \left| \sum\limits_{i=1}^n \frac{\omega_J(z_i) - \omega_J(z_i')}{h_J} \right|$.

  - $h_J := \max\{\mathcal{A}, \sqrt{\mathbb{E}_{z'}[\ell_J(z') - \ell^\circ(z')]}\}$ with constant $\mathcal{A}$ to be chosen later.

  With $h_j$, $\omega_j$ and $\Omega$, we start bounding (J.9) by writing

$$\Delta_n(\widehat{s}, s^\circ) = \left| \frac{1}{n} \mathbb{E}_{\{z_i, z_i'\}_{i=1}^n} \left[ \sum_{i=1}^n \left( \left( \widehat{\ell}(z_i) - \ell^\circ(z_i) \right) - \left( \widehat{\ell}(z_i') - \ell^\circ(z_i') \right) \right) \right] \right|$$

$$\leq \left| \frac{1}{n} \mathbb{E}_{\{z_i, z_i'\}_{i=1}^n} \left[ \sum_{i=1}^n \left( \omega_J(z_i) - \omega_J(z_i') \right) \right] \right| + 2\epsilon_c \qquad \text{(By Replacing } \widehat{\ell} \text{ with } \ell_J \text{)}$$

$$\leq \frac{1}{n} \mathbb{E}_{\{z_i, z_i'\}_{i=1}^n} [h_J \Omega] + 2\epsilon_c \qquad \text{(By introducing } \Omega \text{ and } h_J \text{)}$$

$$\leq \frac{1}{n} \sqrt{\mathbb{E}_{\{z_i, z_i'\}_{i=1}^n}[h_J^2] \mathbb{E}_{\{z_i, z_i'\}_{i=1}^n}[\Omega^2]} + 2\epsilon_c \qquad \text{(By Cauchy-Schwarz inequality )}$$

$$\leq \frac{1}{n} \left( \frac{n}{2} \mathbb{E}_{\{z_i, z_i'\}_{i=1}^n}[h_J^2] + \frac{1}{2n} \mathbb{E}_{\{z_i, z_i'\}_{i=1}^n}[\Omega^2] \right) + 2\epsilon_c \qquad \text{(By AM-GM inequality)}$$

$$= \underbrace{\frac{1}{2} \mathbb{E}_{\{z_i, z_i'\}_{i=1}^n}[h_J^2]}_{\text{(I)}} + \underbrace{\frac{1}{2n^2} \mathbb{E}_{\{z_i, z_i'\}_{i=1}^n}[\Omega^2]}_{\text{(II)}} + 2\epsilon_c. \quad \text{(J.10)}$$

- **Step B.1: Bounding (I).**

  By the definition of $h_J$,

$$\mathbb{E}_{\{z_i, z_i'\}_{i=1}^n}[h_J^2] \leq \mathcal{A}^2 + \mathbb{E}_{\{z_i, z_i'\}_{i=1}^n} \left[ \mathbb{E}_{z'}[\omega_J^2(z)] \right]$$

$$\leq \mathcal{A}^2 + \mathbb{E}_{z'}[\widehat{\omega}^2(z')] + 2\epsilon_c$$

$$= \mathcal{A}^2 + \mathbb{E}_{\{z_i\}_{i=1}^n} \left[ \mathcal{R}_m^{\text{trunc}}(\widehat{s}) \right] + 2\epsilon_c. \quad \text{(J.11)}$$

- **Step B.2: Bounding (II).**

  By Lemma J.1, we have $|\ell(z; s_W)| \lesssim \kappa$, and by the definition of $\Omega^2$, we write

$$\mathbb{E}_{\{z_i, z_i'\}_{i=1}^n} \left[ \sum_{i=1}^n \left( \frac{\omega_J(z_i) - \omega_J(z_i')}{h_J} \right)^2 \right] \leq \sum_{i=1}^n \mathbb{E}_{\{z_i, z_i'\}_{i=1}^n} \left[ \left( \frac{\omega_J(z_i)}{h_J} \right)^2 + \left( \frac{\omega_J(z_J')}{h_J} \right)^2 \right]$$

$$\text{(By the independence between } z_i \text{ and } z_i')$$

$$\leq \kappa \sum_{i=1}^{n} \mathbb{E}_{\{z_i, z_i'\}_{i=1}^{n}} \left[ \frac{\omega_J^2(z_i)}{h_J} + \frac{\omega_J^2(z_i')}{h_J} \right]$$

$$\leq 2n\kappa.$$

We use the Bernstein's inequality and the following two facts

* (1) $\left| \frac{\omega_J(z_i) - \omega_J(z_i')}{h_J} \right| \leq \kappa/\mathcal{A}$.

* (2) $\sum_{i=1}^{n} \frac{\omega_J(z_i) - \omega_J(z_i')}{h_J}$ is centered.

For any $J$ and $\omega \geq 0$, it holds

$$P\left( \left( \sum_{i=1}^{n} \frac{\omega_J(z_i) - \omega_J(z_i')}{h_J} \right)^2 \geq \omega \right) = 2P\left( \sum_{i=1}^{n} \frac{\omega_J(z_i) - \omega_j(z_i')}{h_j} \geq \sqrt{\omega} \right)$$

$$\leq 2\exp\left( -\frac{\omega/2}{\kappa\left(2n + \frac{\sqrt{\omega}}{3\mathcal{A}}\right)} \right).$$

Therefore,

$$P\left(\Omega^2 \geq \omega\right) \leq \sum_{J=1}^{\mathcal{N}} P\left( \left( \sum_{i=1}^{n} \frac{\omega_J(z_i) - \omega_J(z_i')}{h_J} \right)^2 \geq \omega \right) \leq 2\mathcal{N}\exp\left( -\frac{\omega/2}{\kappa\left(2n + \frac{\sqrt{\omega}}{3\mathcal{A}}\right)} \right).$$

For some $\omega_0 > 0$, we bound $\Omega^2$ by

$$\mathbb{E}_{\{z_i, z_i^n\}_{i=1}^{n}}\left[\Omega^2\right] = \int_0^{\omega_0} P\left(\Omega^2 \geq \omega\right) d\omega + \int_{\omega_0}^{\infty} P\left(\Omega^2 \geq \omega\right) d\omega, \qquad \text{(By integral identity)}$$

$$\leq \omega_0 + \int_{\omega_0}^{\infty} 2\mathcal{N}\exp\left( -\frac{\omega/2}{\kappa\left(2n + \frac{\sqrt{\omega}}{3\mathcal{A}}\right)} \right) d\omega,$$

$$\leq \omega_0 + 2\mathcal{N}\int_{\omega_0}^{\infty} \left\{ \exp\left(-\frac{\omega}{8n\kappa}\right) + \exp\left(-\frac{3\mathcal{A}\sqrt{\omega}}{4\kappa}\right) \right\} d\omega,$$

$$\leq \omega_0 + 2\mathcal{N}\left\{ 8n\kappa\exp\left(-\frac{\omega_0}{8n\kappa}\right) + \left(\frac{8\kappa\sqrt{\omega_0}}{3\mathcal{A}} + \frac{32\kappa}{9\mathcal{A}^2}\right)\exp\left(-\frac{3\mathcal{A}\sqrt{\omega_0}}{4\kappa}\right) \right\}.$$

Taking $\mathcal{A} = \sqrt{\omega_0}/6n$ and $\omega_0 = 8n\kappa\log\mathcal{N}$, we have

$$\mathbb{E}_{\{z_i, z_i^n\}_{i=1}^{n}}[\Omega^2] \leq n\kappa\log\mathcal{N}. \tag{J.12}$$

* **Step C: Combine (I) and (II).**

  Combining (J.11) and (J.12), it holds:

$$\Delta_n(\widehat{s}, s^\circ) \leq \frac{1}{2}\mathbb{E}_{\{z_i, z_i'\}_{i=1}^{n}}[h_J^2] + \frac{1}{2n^2}\mathbb{E}_{\{z_i, z_i'\}_{i=1}^{n}}[\Omega^2] + 2\epsilon_c$$

$$\lesssim \frac{1}{2}\mathbb{E}_{\{z_i\}_{i=1}^{n}}\left[\mathcal{R}_m^{\text{trunc}}(\widehat{s})\right] + \frac{\kappa}{2n}\log\mathcal{N} + \frac{7}{2}\epsilon_c.$$

Recall Definition J.6 and multiply the above inequality with 2, we have

$$\mathbb{E}_{\{z_i\}_{i=1}^n}\left[\mathcal{R}_m^{\text{trunc}}(\widehat{s})\right] \lesssim 2\mathbb{E}_{\{z_i\}_{i=1}^n}\left[\widehat{\mathcal{R}}_m^{\text{trunc}}(\widehat{s})\right] + \frac{\kappa}{n}\log\mathcal{N} + 7\epsilon_c.$$

Therefore,

$$\Delta_n(\widehat{s}, s^\circ) \lesssim \mathbb{E}_{\{z_i\}_{i=1}^n}\left[\widehat{\mathcal{R}}_m^{\text{trunc}}(\widehat{s})\right] + \frac{\kappa}{n}\log\mathcal{N} + 7\epsilon_c \qquad\qquad \text{(By Lemma J.1)}$$

$$\lesssim \mathbb{E}_{\{x_i, y_i\}_{i=1}^n}\left[\widehat{\mathcal{R}}_m(\widehat{s})\right] + \kappa\left(R_\mathcal{T}\exp\left(-C_2 R_\mathcal{T}^2\right) + \frac{1}{n}\log\mathcal{N}\right) + 7\epsilon_c,$$

This completes the proof.  □

## J.2 PROOF OF THEOREM 3.3

*Proof of Theorem 3.3.* Recall Definition J.1. We bound the mixed risk with following three steps.

- **Step A: Decompose the mixed risk.**

  Let $s^\circ(x, y, t) = \nabla\log p_t(x|y)$ be the ground truth. If $y = \emptyset$, we set $s^\circ(x, y, t) = \nabla\log p_t(x)$.

  Recall Definition J.3 and Lemma J.4.

  Let $\{x_i', y_i'\}_{i=1}^n \sim P_0(x, y)$ be a different set of i.i.d. samples independent of the training samples.

  Next, we rewrite the mixed risk

  $$\mathcal{R}_m(\widehat{s}) = \mathbb{E}_{\{x_i', y_i'\}_{i=1}^n}\left[\frac{1}{n}\sum_{i=1}^n\left(\ell(x_i', y_i', \widehat{s}) - \ell(x_i', y_i', s^\circ)\right)\right] = \mathbb{E}_{\{x_i', y_i'\}_{i=1}^n}\left[\widehat{\mathcal{R}}_m'(\widehat{s})\right],$$

  where we use $\widehat{\mathcal{R}}_m'(\widehat{s})$ to denote the empirical risk of the score estimator $\widehat{s}$ trained from $\{x_i', y_i'\}_{i=1}^n$.

  Next, the decomposition of $\mathbb{E}_{\{x_i, y_i\}_{i=1}^n}[\mathcal{R}_m(\widehat{s})]$ follows

  $$\mathbb{E}_{\{x_i, y_i\}_{i=1}^n}[\mathcal{R}_m(\widehat{s})] = \underbrace{\mathbb{E}_{\{x_i, y_i\}_{i=1}^n}\left[\mathbb{E}_{\{x_i', y_i'\}_{i=1}^n}\left[\widehat{\mathcal{R}}_m'(\widehat{s}) - \widehat{\mathcal{R}}_m'^{\text{trunc}}(\widehat{s})\right]\right]}_{\text{(I)}}$$

  $$+ \underbrace{\mathbb{E}_{\{x_i, y_i\}_{i=1}^n}\left[\mathbb{E}_{\{x_i', y_i'\}_{i=1}^n}\left[\widehat{\mathcal{R}}_m'^{\text{trunc}}(\widehat{s}) - \widehat{\mathcal{R}}_m^{\text{trunc}}(\widehat{s})\right]\right]}_{\text{(II)}}$$

  $$+ \underbrace{\mathbb{E}_{\{x_i, y_i\}_{i=1}^n}\left[\widehat{\mathcal{R}}_m^{\text{trunc}}(\widehat{s}) - \widehat{\mathcal{R}}_m(\widehat{s})\right]}_{\text{(III)}} + \underbrace{\mathbb{E}_{\{x_i, y_i\}_{i=1}^n}\left[\widehat{\mathcal{R}}_m(\widehat{s})\right]}_{\text{(IV)}}$$

- **Step B: Derive Upper Bounds.**

  Recall Lemma J.4.

  Let $\kappa = 1/t_0$ for results in Theorem 3.1 and $\kappa = \log(1/t_0)$ for results in Theorem I.1.

  By Lemma J.1, we have both (I), (III) $\lesssim \kappa\exp\left(-C_2 R_\mathcal{T}^2\right)R_\mathcal{T}$.

  By Lemma J.4, we have (II) $\lesssim$ (IV) $+ \kappa\left(R_\mathcal{T}\exp\left(-C_2 R_\mathcal{T}^2\right) + \frac{1}{n}\log\mathcal{N}\right) + 7\epsilon_c$,

  Next, we we bound (IV) by

  $$\text{(IV)} = \mathbb{E}_{\{z_i\}_{i=1}^n}\left[\widehat{\mathcal{R}}(\widehat{s})\right] \leq \mathbb{E}_{\{z_i\}_{i=1}^n}\left[\widehat{\mathcal{R}}_m(s)\right] = \mathcal{R}_m(s).$$

  Therefore, (IV) $\leq \min_{s_W\in\mathcal{T}_R^{h,s,r}}\mathcal{R}_m(s)$.

The inequality holds because $\widehat{s}$ is the minimizer of the empirical risk.

Combining these bounds, it holds

$$\mathbb{E}_{\{x_i,y_i\}_{i=1}^n}[\mathcal{R}_m(\widehat{s})] \leq 2 \min_{s_W \in \mathcal{T}_R^{h,s,r}} \int_{t_0}^T \frac{1}{T-t_0} \mathbb{E}_{x_t,y,\tau}\left[\|s(x_t,\tau y,t) - \nabla \log p_t(x_t|\tau y)\|_2^2\right] dt$$
$$+ \mathcal{O}\left(\frac{\kappa}{n}\log\mathcal{N}\right) + \mathcal{O}(\exp(-C_2 R_\mathcal{T}^2)\kappa) + \mathcal{O}(\epsilon_c). \tag{J.13}$$

Next, we take $R_\mathcal{T} = \sqrt{\frac{(C_\sigma + 2\beta)\log N}{C_2}}$.

Since $\kappa \lesssim \log(1/t_0) \lesssim 1/t_0 = N^{C_\sigma}$, it holds

$$\mathbb{E}_{\{x_i,y_i\}_{i=1}^n}[\mathcal{R}_m(\widehat{s})] \leq 2 \min_{s \in \mathcal{T}_R^{h,s,r}} \int_{t_0}^T \frac{1}{T-t_0} \mathbb{E}_{\tau,x_t,y}\left[\|s(x,\tau y,t) - \nabla \log p_t(x|y)\|_2^2\right] dt$$
$$+ \mathcal{O}\left(\frac{\kappa}{n}\log\mathcal{N}\right) + \mathcal{O}\left(N^{-2\beta}\right) + \mathcal{O}(\epsilon_c). \tag{J.14}$$

To apply the previous approximation theorems (Theorem 3.1 and Theorem I.1) to the first term on the RHS of (J.13), we rewrite the expectation as

$$2\mathbb{E}_{x_t,y,\tau}\left[\|s(x_t,\tau y,t) - \nabla \log p_t(x_t|\tau y)\|_2^2\right] \tag{J.15}$$
$$= \int_{\mathbb{R}^{d_x}} \|s(x,\emptyset,t) - \nabla \log p_t(x|y)\|_2^2 p_t(x)\mathrm{d}x + \mathbb{E}_y\left[\int_{\mathbb{R}^{d_x}} \|s(x,y,t) - \nabla \log p_t(x|y)\|_2^2 p_t(x|y)\mathrm{d}x\right].$$

Since $p_t(x)$ satisfies the subgaussian property as well, the previous result of the conditional score estimation applies to its unconditional counterpart by removing the label throughout the process.

- **Step C: Combine All Bounds.**

  First, we derive final bounds for results under Assumption 3.1.

  **Result under Assumption 3.1**

  Let $\gamma_1, \gamma_2 \in (0,1)$ be arbitrary real numbers.

  By taking $N = n^{\frac{\gamma_1}{\nu_1}}$ and $\epsilon_c = n^{-\gamma_2}$, we rewrite (J.14)

  $$\mathbb{E}_{\{z_i\}_{i=1}^n}[\mathcal{R}_m(\widehat{s})]$$
  $$\lesssim \underbrace{\mathcal{O}\left(N^{-\beta}(\log N)^{d_x+\frac{\beta}{2}+1}\right)}_{(i)} + \underbrace{\mathcal{O}\left(N^{-2\beta}\right)}_{(ii)} + \underbrace{\mathcal{O}\left(\frac{1}{t_0}\cdot n^{-1}\cdot N^{\nu_1}\cdot(\log N)^{\nu_2}\cdot\epsilon_c^{-2}\right)}_{(iii)} + \underbrace{\mathcal{O}(\epsilon_c)}_{(iv)}$$

  (By Theorem 3.1)

  $$\lesssim \mathcal{O}\left(n^{-\frac{\beta\cdot\gamma_1}{\nu_1}}(\log n)^{d_x+\frac{\beta}{2}+1}\right) + \mathcal{O}\left(n^{\frac{C_\sigma\gamma_1}{\nu_1}}\cdot n^{-1}\cdot n^{\gamma_1}\cdot(\log n)^{\nu_2}\cdot n^{2\gamma_2}\right) + \mathcal{O}\left(n^{-\gamma_2}\right)$$
  $$\lesssim \mathcal{O}\left(n^{-\min\left(\frac{\beta\gamma_1}{\nu_1}, 1-\frac{C_\sigma\cdot\gamma_1}{\nu_1}-\gamma_1-2\gamma_2, \gamma_2\right)}\cdot(\log n)^{\nu_2}\right),$$

  where $\nu_1 = 16\beta d + 12\beta$ and $\nu_2 = 20d_x + 4\beta + 18$.

  We find the optimal upper-bound by the following choice of $\gamma_1$ and $\gamma_2$.

  For any $\gamma_1, \gamma_2 \in (0,1)$ satisfying,

  $$\gamma_1 + 2\gamma_2 + \frac{C_\sigma \cdot \gamma_1}{\nu_1} < 1,$$

we consider

$$\min\left\{\frac{\beta\gamma_1}{\nu_1}, 1 - \gamma_1 - 2\gamma_2 - \frac{C_\sigma\gamma_1}{\nu_1}, \gamma_2\right\}.$$

To simplify, we set

$$\frac{\beta\gamma_1}{\nu_1} = 1 - \gamma_1 - 2\gamma_2 - \frac{C_\sigma\gamma_1}{\nu_1} = \gamma_2,$$

and hence

$$\frac{\beta\gamma_1}{\nu_1} = \gamma_2, \quad 1 - \gamma_1 - 2\gamma_2 - \frac{C_\sigma\gamma_1}{\nu_1} = \gamma_2.$$

Rearranging and solving for $\gamma_1, \gamma_2$, we obtain

$$\gamma_1 = \frac{\nu_1}{\nu_1 + C_\sigma + 3\beta}, \quad \gamma_2 = \frac{\beta}{\nu_1 + C_\sigma + 3\beta},$$

so that the three arguments in the $\min\{\cdot\}$ all coincide.

Therefore,

$$\min\left\{\frac{\beta\gamma_1}{\nu_1}, 1 - \gamma_1 - 2\gamma_2 - \frac{C_\sigma\gamma_1}{\nu_1}, \gamma_2\right\} = \gamma_2 = \frac{\beta}{\nu_1 + C_\sigma + 3\beta}.$$

Next, we ensure condition $\gamma_1 + 2\gamma_2 + \frac{C_\sigma\gamma_1}{\nu_1} < 1$.

Since

$$\frac{\nu_1}{\nu_1 + C_\sigma + 3\beta} + \frac{2\beta}{\nu_1 + C_\sigma + 3\beta} + \frac{C_\sigma}{\nu_1 + C_\sigma + 3\beta} = \frac{\nu_1 + C_\sigma + 2\beta}{\nu_1 + C_\sigma + 3\beta} < 1,$$

the condition holds for all $d_x, \beta$ and $C_\sigma$.

Next, we derive final bounds for results under Assumption 3.2.

**Result under Assumption 3.2**

Let $\gamma_3, \gamma_4 \in (0, 1)$ be arbitrary real numbers.

By taking $N = n^{\gamma_3/\nu_1}$ and $\epsilon_c = n^{-\gamma_4}$, we rewrite (J.14)

$$\mathbb{E}_{\{z_i\}_{i=1}^n}[\mathcal{R}_m(\widehat{s})] \qquad\qquad\qquad\qquad\qquad\qquad\qquad (\text{By Theorem I.1})$$

$$\lesssim \underbrace{\mathcal{O}\left(N^{-2\beta}(\log N)^{\beta+1}\right)}_{(\mathbf{i})} + \underbrace{\mathcal{O}\left(N^{-2\beta}\right)}_{(\mathbf{ii})} + \underbrace{\mathcal{O}\left(\frac{1}{t_0} \cdot n^{-1} \cdot N^{\nu_1} \cdot (\log N)^{\nu_2} \cdot \epsilon_c^{-2}\right)}_{(\mathbf{iii})} + \underbrace{\mathcal{O}\left(\epsilon_c\right)}_{(\mathbf{iv})}$$

$$\qquad\qquad\qquad\qquad\qquad\qquad\qquad\qquad\qquad (\text{By dropping lower order term } (\mathbf{ii}))$$

$$\lesssim \mathcal{O}\left(n^{-\frac{2\beta\cdot\gamma_3}{\nu_1}} \cdot (\log n)^{\beta+1}\right) + \mathcal{O}\left(\log n \cdot n^{-1} \cdot n^{\gamma_3} \cdot (\log n)^{12} \cdot n^{2\gamma_4}\right) + \mathcal{O}\left(n^{-\gamma_4}\right)$$

$$\lesssim \mathcal{O}\left(n^{-\min\left\{\frac{2\beta\gamma_3}{\nu_1}, 1-\gamma_3-2\gamma_4, \gamma_4\right\}} \cdot (\log n)^{\max(13, \beta+2)}\right).$$

where $\nu_3 = \frac{4(12\beta d_x + 31\beta d + 6\beta)}{d} + \frac{12C_\alpha \cdot (12d_x + 25d + 6)}{d} + 72C_\sigma$.

We find the optimal upper-bound by the following choice of $\gamma_1$ and $\gamma_2$.

For any $\gamma_1, \gamma_2 \in (0, 1)$ satisfying,

$$\gamma_3 + 2\gamma_4 < 1,$$

we consider

$$\min \left\{ \frac{2\beta\gamma_3}{\nu_1}, 1 - \gamma_3 - 2\gamma_4, \gamma_4 \right\}$$

To simplify, we set

$$\frac{2\beta \cdot \gamma_3}{\nu_1} = 1 - \gamma_3 - 2\gamma_4 = \gamma_4,$$

and get

$$\gamma_3 = \frac{\nu_1}{\nu_1 + 6\beta}, \quad \gamma_4 = \frac{2\beta}{\nu_1 + 6\beta}$$

with $\gamma_3 + 2\gamma_4 < 1$.

This completes the proof. $\square$

### J.3 AUXILIARY LEMMAS FOR THEOREM 3.4

We begin with following two lemmas that serves as the key components for the proof of Theorem 3.4.

**Lemma J.5** (Lemma D.4 of Fu et al. (2024b), Proposition D.1 of Oko et al. (2023) and Chen et al. (2022)). Consider probability distribution $P_0$ and two stochastic processes $Y = \{Y_t\}_{t \in [0,T]}$ and $Y' = \{Y'_t\}_{t \in [0,T]}$ with distribution $P_t$ and $P'_t$ respectively, satisfying the following SDE:

$$\mathrm{d}Y_t = b(Y_t, t)\mathrm{d}t + \mathrm{d}W_t \quad Y_0 \sim P_0,$$
$$\mathrm{d}Y'_t = b'(Y'_t, t)\mathrm{d}t + \mathrm{d}W_t \quad Y'_0 \sim P_0.$$

If the condition $\int_x P_t(x)\|(b - b')(x, t)\|\mathrm{d}x \leq C$ holds for any $t \in [0, T]$ and constant $C$, it holds

$$\mathrm{KL}(P_T \,||\, P'_T) = \int_0^T \frac{1}{2} \int_x P_t(x)\|(b - b')(x, t)\|\mathrm{d}x\mathrm{d}t.$$

Moreover, we need the following lemma to bound to total variation.

**Lemma J.6** (Lemma D.5 of Fu et al. (2024b)). Assume Assumption 3.1. For any $y \in [0, 1]^{d_y}$ it holds

$$\mathrm{TV}\left(P_0(\cdot|y), P_{t_0}(\cdot|y)\right) = \mathcal{O}\left(\sqrt{t_0} \log^{\frac{d_x+1}{2}}\left(\frac{1}{t_0}\right)\right).$$

### J.4 MAIN PROOF OF THEOREM 3.4

*Proof of Theorem 3.4.* Given label $y$, let $\widehat{P}_{t_0}(\cdot|y)$ denote the data distribution with early-stopped time $t_0$ generated by the reverse process with the score estimator $\widehat{s}$ from transformer network class.

We define the reverse process starting with standard Gaussian.

$$\mathrm{d}\widetilde{X}_t^{\leftarrow} = \left[\frac{1}{2}\mathrm{d}\widetilde{X}_t^{\leftarrow} + \nabla \log p_{T-t}(\widetilde{X}_t^{\leftarrow}|y)\right]\mathrm{d}t + \mathrm{d}\overline{W}_t, \quad \widetilde{X}_0^{\leftarrow} \sim N(0, I_{d_x}). \quad (\mathrm{J}.16)$$

We denote the distribution of $\widetilde{X}_t^{\leftarrow}$ conditioned on the label $y$ as $\widetilde{P}_{T-t}(\cdot|y)$.

Next, we decompose the total variation into three parts

$$\mathrm{TV}\left(P(\cdot|y), \widehat{P}_{t_0}(\cdot|y)\right)$$
$$\lesssim \underbrace{\mathrm{TV}(P(\cdot|y), P_{t_0}(\cdot|y))}_{(\mathrm{I})} + \underbrace{\mathrm{TV}\left(P_{t_0}(\cdot|y), \widetilde{P}_{t_0}(\cdot|y)\right)}_{(\mathrm{II})} + \underbrace{\mathrm{TV}\left(\widetilde{P}_{t_0}(\cdot|y), \widehat{P}_{t_0}(\cdot|y)\right)}_{(\mathrm{III})},$$

where we introduce distribution $\widetilde{P}_{t_0}$ at time $t_0$ as an intermediary defined in (J.16).

- **Step A: Bounding (I).**

  By Lemma J.6, we bound term (I) by

  $$\mathrm{TV}\left(P(\cdot|y), \widetilde{P}_{t_0}(\cdot|y)\right) = \mathcal{O}\left(\sqrt{t_0} \log^{\frac{d_x+1}{2}}\left(\frac{1}{t_0}\right)\right).$$

- **Step B: Bounding (II).**

  By Data Processing Inequality and Pinsker's Inequality (Canonne, 2022, Lemma 2), we write

  $$\mathrm{TV}\left(P_{t_0}(\cdot|y), \widetilde{P}_{t_0}(\cdot|y)\right) \lesssim \sqrt{\mathrm{KL}(P_{t_0}(\cdot|y) \,||\, \widetilde{P}_{t_0}(\cdot|y))},$$

$$\lesssim \sqrt{\mathrm{KL}(P_T(\cdot|y) \,\|\, N(0, I_{d_x}))},$$
$$\lesssim \sqrt{\mathrm{KL}(P(\cdot|y) \,\|\, N(0, I_{d_x}))} \exp(-T). \qquad (\mathrm{J}.17)$$

Next, we bound term (II) by

$$\mathrm{TV}\left(P_{t_0}(\cdot|y), \widetilde{P}_{t_0}(\cdot|y)\right) \lesssim \sqrt{\mathrm{KL}(P(\cdot|y) \,\|\, N(0, I_{d_x}))} \exp(-T), \qquad (\text{By (J.17)})$$
$$\lesssim \exp(-T).$$

- **Step C: Bounding (III).**

  By Lemma J.5 and (J.17), we bound term (III) by

  $$\mathrm{TV}\left(\widetilde{P}_{t_0}(\cdot|y), \widehat{P}_{t_0}(\cdot|y)\right) \lesssim \sqrt{\int_{t_0}^{T} \frac{1}{2} \int_x p_t(x|y) \|\widehat{s}(x, y, t) - \nabla \log p_t(x|y)\|^2 \mathrm{d}x \mathrm{d}t}.$$

By incorporating three steps and taking expectation with respect to $y$, we obtain the upper-bound

$$\mathbb{E}_y\left[\mathrm{TV}\left(P(\cdot|y), \widehat{P}_{t_0}(\cdot|y)\right)\right]$$
$$\lesssim \sqrt{t_0} \log^{\frac{d_x+1}{2}}\left(\frac{1}{t_0}\right) + \exp(-T) + \sqrt{\int_{t_0}^{T} \frac{1}{2} \int_x p_t(x|y) \|\widehat{s}(x, y, t) - \nabla \log p_t(x|y)\|^2 \mathrm{d}x \mathrm{d}t},$$
$$\text{(By Jensen's inequality)}$$
$$\lesssim \underbrace{\sqrt{t_0} \log^{\frac{d_x+1}{2}}\left(\frac{1}{t_0}\right)}_{(i)} + \underbrace{\exp(-T)}_{(ii)} + \underbrace{\sqrt{\frac{T}{2} \mathcal{R}(\widehat{s})}}_{(iii)}.$$

- **Result under Assumption 3.1.**

  Recall **Step C.1** in the proof of Theorem 3.3.

  We set $N = n^{\gamma_1/\nu_1}$ and $\epsilon_c = n^{-\gamma_2}$ for all $\gamma_1, \gamma_2 \in (0, 1)$ satisfying $\gamma_1 + 2\gamma_2 + \frac{C_\sigma \cdot \gamma_1}{\nu_1} < 1$.

  Therefore,

  $$\mathbb{E}_y\left[\mathrm{TV}\left(P(\cdot|y), \widehat{P}_{t_0}(\cdot|y)\right)\right]$$
  $$\lesssim \underbrace{N^{-\frac{C_\sigma}{2}} \log^{\frac{d_x+1}{2}}(N^{-C_\sigma})}_{(i)} + \underbrace{N^{-C_\alpha}}_{(ii)} \qquad (\text{By Theorem 3.3})$$
  $$+ \underbrace{\left(C_\alpha \cdot \log N \cdot n^{-\min\left\{\frac{\beta \cdot \gamma_1}{\nu_1}, 1-\gamma_1-2\gamma_2-\frac{C_\sigma \cdot \gamma_1}{\nu_1}, \gamma_2\right\}} \cdot (\log n)^{\nu_2+2}\right)^{\frac{1}{2}}}_{(iii)}$$
  $$\lesssim \underbrace{n^{-\frac{C_\sigma \cdot \gamma_1}{2\nu_1}} \cdot \log^{\frac{d_x+1}{2}}(n)}_{(i)} + \underbrace{n^{-\frac{C_\alpha \cdot \gamma_1}{\nu_1}}}_{(ii)} \qquad (\text{By } \nu_2 > d_x + 1)$$
  $$+ \underbrace{n^{-\min\left\{\frac{\beta \cdot \gamma_1}{\nu_1}, 1-\gamma_1-2\gamma_2-\frac{C_\sigma \cdot \gamma_1}{\nu_1}, \gamma_2\right\}} \cdot (\log n)^{\frac{\nu_2}{2}+\frac{3}{2}}}_{(iii)}$$
  $$\lesssim n^{-\omega} \cdot (\log n)^{\frac{\nu_2}{2}+\frac{3}{2}},$$

where $\omega = \min\left\{ \frac{C_\sigma \cdot \gamma_1}{2\nu_1}, \frac{C_\alpha \cdot \gamma_1}{\nu_1}, \frac{\beta \cdot \gamma_1}{\nu_1}, 1 - \gamma_1 - 2\gamma_2 - \frac{C_\sigma \cdot \gamma_1}{\nu_1}, \gamma_2 \right\}$.

Moreover, recall that the following choice of $\gamma_1$ and $\gamma_2$ leads to the tightest upper-bound

$$\gamma_1 = \frac{\nu_1}{\nu_1 + C_\sigma + 3\beta}, \quad \gamma_2 = \frac{\beta}{\nu_1 + C_\sigma + 3\beta}.$$

Therefore,

$$\omega = \min\left\{ \frac{C_\sigma}{2(\nu_1 + C_\sigma + 3\beta)}, \frac{C_\alpha}{\nu_1 + C_\sigma + 3\beta}, \frac{\beta}{\nu_1 + C_\sigma + 3\beta} \right\}.$$

This completes the first part of the proof.

- **Result under Assumption 3.2.**

  Recall **Step C.2** in the proof of Theorem 3.3.

  We set $N = n^{\gamma_3/\nu_1}$ and $\epsilon_c = n^{-\gamma_4}$ for all $\gamma_3, \gamma_4 \in (0,1)$ satisfying $\gamma_3 + 2\gamma_4 < 1$.

  Therefore, by Theorem 3.3,

$$\mathbb{E}_y\left[ \mathbf{TV}\left( P(\cdot|y), \widehat{P}_{t_0}(\cdot|y) \right) \right]$$
$$\lesssim \underbrace{N^{-\frac{C_\sigma}{2}} \log^{\frac{d_x+1}{2}}(N^{-C_\sigma})}_{(i)} + \underbrace{N^{-C_\alpha}}_{(ii)}$$
$$+ \underbrace{\left( C_\alpha \cdot \log N \cdot n^{-\min\left\{ \frac{2\beta \cdot \gamma_3}{\nu_1}, 1 - \gamma_3 - 2\gamma_4, \gamma_4 \right\}} \cdot (\log n)^{\max\{13, \beta+2\}} \right)^{\frac{1}{2}}}_{(iii)}$$
$$\lesssim \underbrace{n^{-\frac{C_\sigma \cdot \gamma_3}{2\nu_1}} \cdot \log^{\frac{d_x+1}{2}}(n)}_{(i)} + \underbrace{n^{-\frac{C_\alpha \cdot \gamma_3}{\nu_1}}}_{(ii)}$$
$$+ \underbrace{n^{-\min\left\{ \frac{2\beta \cdot \gamma_3}{\nu_1}, 1 - \gamma_3 - 2\gamma_4, \gamma_4 \right\}} \cdot (\log n)^{\nu_2}}_{(iii)},$$
$$\lesssim n^{-\phi}(\log n)^{\frac{\nu_2}{2} + \frac{3}{2}}, \tag{J.18}$$

where $\phi = \min\left\{ \frac{C_\sigma \cdot \gamma_3}{2\nu_1}, \frac{C_\alpha \cdot \gamma_3}{\nu_1}, \frac{2\beta \cdot \gamma_3}{\nu_1}, 1 - \gamma_3 - 2\gamma_4, \gamma_4 \right\}$.

Moreover, recall that the following choice of $\gamma_3$ and $\gamma_4$ leads to the tightest upper-bound

$$\gamma_3 = \frac{\nu_1}{\nu_1 + 6\beta}, \quad \gamma_4 = \frac{2\beta}{\nu_1 + 6\beta}.$$

Therefore,

$$\phi = \min\left\{ \frac{C_\sigma}{2(\nu_1 + 6\beta)}, \frac{C_\alpha}{\nu_1 + 6\beta}, \frac{2\beta}{\nu_1 + 6\beta} \right\}.$$

This completes the second part of the proof.

$\square$

## J.5 PROOF OF THEOREM 3.5

Recall Lemma 3.1, we have the minimax optimal rate

$$\inf_{\widehat{\mu}} \sup_{p \in \mathcal{P}} \mathbb{E}_{\{x_i\}_{i=1}^n} \left[ \mathrm{TV}(\widehat{\mu}, \mathrm{P}) \right] \geq \Omega(n^{-\frac{\beta}{d_x+2\beta}})$$

Further, recall Theorem 3.4, we have

$$\mathbb{E}_{\{x_i, y_i\}_{i=1}^n} \left[ \mathbb{E}_y \left[ \mathrm{TV} \left( \widehat{P}_{t_0}(\cdot|y), P_0(\cdot|y) \right) \right] \right] = \mathcal{O} \left( n^{-\phi} (\log n)^{\frac{\nu_2}{2}+\frac{3}{2}} \right),$$

where $\min \left\{ \frac{C_\sigma}{2(\nu_1+6\beta)}, \frac{C_\alpha}{\nu_1+6\beta}, \frac{\beta}{\nu_1+6\beta} \right\}$. Therefore, unconditional diffusion transformers achieve minimax optimality under the setting $C_\sigma = 2C_\alpha = 2\beta$ and $\frac{\beta}{\nu_1+6\beta} = \frac{\beta}{d_x+2\beta}$.

This completes the proof.

