# OpenReview forum: "On Statistical Rates of Conditional Diffusion Transformers: Approximation,  Estimation and Minimax Optimality"
_ICLR.cc/2025/Conference — ICLR 2025 Poster_

### Official Review · Reviewer_1NAW · 2024-11-02

**Soundness:** 3
**Presentation:** 2
**Contribution:** 2
**Rating:** 5
**Confidence:** 3

**Summary:**

This paper mathematically investigates the approximation and estimation properties of conditional diffusion Transformers (DiTs) with classifier-free guidance. Theoretical connections between conditional DiTs and their unconditional latent variants are established for data distributions with certain smoothness.

**Strengths:**

1. This paper seems mathematically sound.
2. The settings studied in this work are abundant and quite complex.
3. The roadmaps of proofs are clear.

**Weaknesses:**

1. It seems that some estimates scale poorly in the asymptotic sense, particularly in high-dimensional and long sequences settings (see details in the "Questions" section).
2. Without verifications by numerical simulations, one cannot guarantee that the derived approximation and estimation estimates can be achieved in practice, i.e. whether these theoretical constructions can be found through concrete training dynamics. Is it possible to numerically verify (partial of) the dependence in these mathematically derived upper bounds?

**Questions:**

1. For the score approximation:
- $\sigma_t^2 \to 0$ when $t \to 0$, leading to infinitely large upper bounds (Table 1).
- $\epsilon^{1/d} \to 1$ when $d \to \infty$, leading to $O(1)$ upper bounds (Table 1) that are vacuous in high-dimensional settings.
- The upper bound is *exponentially* large given high-dimensional inputs, due to the multiplicative term $(\log N)^{d_x}$ (Theorem 3.1).
- The parameter bounds of Transformers are also *exponentially* large given long input sequences (with fixed dimensions), since $\||W_{\{Q,K\}}\||=O(N^{L/d})$ (Theorem 3.1).

2. For the score estimation:
- In Theorem 3.3, why does the upper bound deriving from Assumption 3.1 (weaker) have a much better dependence of $\log n$ ($(\log n)^{-dL}$ v.s. $(\log n)^{\Omega(1)}$) over the upper bound deriving from Assumption 3.2 (stronger)? Also, the former error term $(\log n)^{-dL}$ decreases exponentially fast as the data dimension or sequence length increases, which seems unreasonable.
- In Theorem 3.3, for the upper bound deriving from Assumption 3.2, $\nu_3=O(L/d)$, which gives $N=n^{\Omega(d/L)}$, $t_0 \sim O(n^{-d/L})$ and $\log (1/t_0) = \Omega(d/L\cdot\log n)$. This term can be merged into the subsequent polynomial dependences in data dimensions and sequence lengths (i.e. $d^{14}L^4$), and hence introduces no singularity. However, the power of $1/n$ is approximately $1/(\nu_3\cdot d)=\Omega(1/L)$, leading to the vacuous sample complexity estimate given long input sequences.

3. For the distribution estimation:
- In Theorem 3.4, there are the same issues raised in the second point of the score estimation, since the upper bounds in Theorem 3.4 and Theorem 3.3 are in similar forms.

4. It seems that there is no characterization of the model capacity dependence. That is, all the provided estimates do not contain (at least, explicitly) error terms related to $h, s, r$.

5. Minor issues:
- The notation $s$ denotes both the head size and Hölder smoothness (Definition 3.1).
- In the caption of Table 1, "... where $L$, $\tilde{L}$ are sequence length of transformer inputs", length ->lengths.

**Details Of Ethics Concerns:**

There are no ethics concerns.

---

> ### Author Response · Authors · 2024-11-18
> **Response 1**
>
> ### Thank you for your review.
> ### We have addressed all your comments and questions in this and the following responses.
> ### The draft has been updated accordingly, with changes highlighted in blue in the latest revision.
>
> ---
>
> > `W1.` It seems that some estimates scale poorly in the asymptotic sense, particularly in high-dimensional and long-sequence settings
>
> Thanks for pointing this out.
>
> We have fixed some typos and finer details of our proof. The rates are now updated. We believe we have address the asymptotic issues. Please see the responses below.
>
> Here we quote the updated Table 1 for your reference:
>
> ### **Summary of Theoretical Results**
>
> The initial data is $d_x$-dimensional, and the condition is $d_y$-dimensional. For latent DiT, the latent variable is $d_0$-dimensional. $\sigma_{t}^{2} = 1 - e^{-t}$ is the denoising scheduler. The sample size is $n$, and $0 < \epsilon < 1$ represents the score approximation error. While we report asymptotics for large $d_x, d_0$, we reintroduce the $n$ dependence in the estimation results to emphasize sample complexity convergence.
>
> | **Assumption** | **Score Approximation** | **Score Estimation** | **Dist. Estimation (Total Variation Distance)** | **Minimax Optimality** |
> |----------------|-------------------------|---------------------|--------------------------------------------------|------------------------|
> | Generic Hölder Smooth Data Dist.        | $\mathcal{O}\left((\log(\frac{1}{\epsilon}))^{d_x}/{\sigma_t^4}\right)$ | $n^{-\Theta(1/d_x)} \cdot (\log n)^{\mathcal{O}(d_x)}$ | $n^{-\Theta(1/d_x)} \cdot (\log n)^{\mathcal{O}(d_x)}$ | ✘ |
> | Stronger Hölder Smooth Data Dist.       | $(\log(\frac{1}{\epsilon}))^{\mathcal{O}(1)}/{\sigma_t^2}$ | $n^{-\Theta(1/d_x^2)} \cdot (\log n)^{\mathcal{O}(1)}$ | $n^{-\Theta(1/d_x)} \cdot (\log n)^{\mathcal{O}(1)}$ | ✔ |
> | Latent Subspace + Generic Hölder Smooth Data Dist. | $\mathcal{O}\left((\log(\frac{1}{\epsilon}))^{d_0}/{\sigma_t^4}\right)$ | $n^{-\Theta(1/d_0)} \cdot (\log n)^{\mathcal{O}(d_0)}$ | $n^{-\Theta(1/d_0)} \cdot (\log n)^{\mathcal{O}(d_0)}$ | ✘ |
> | Latent Subspace + Stronger Hölder Smooth Data Dist. | $(\log(\frac{1}{\epsilon}))^{\mathcal{O}(1)}/{\sigma_t^2}$ | $n^{-\Theta(1/d_0^2)} \cdot (\log n)^{\mathcal{O}(1)}$ | $n^{-\Theta(1/d_0)} \cdot (\log n)^{\mathcal{O}(1)}$ | ✔ |
>
>
> > `W2.` Without verifications by numerical simulations, one cannot guarantee that the derived approximation and estimation estimates can be achieved in practice, i.e., whether these theoretical constructions can be found through concrete training dynamics. Is it possible to numerically verify (partially) the dependence in these mathematically derived upper bounds?
>
>
> Thanks for the comment.
>
> In response, **we have included empirical results in Appendix D of the revised manuscript.**
>
> Specifically, we train a conditional diffusion transformer model on the CIFAR10 to verify three points:
>
> - **(E1)** the influence of input data dimension $d_x$ on the testing loss in Theorem 3.3.
> - **(E2)** the influence of input data dimension $d_x$ on the parameter norm bounds in Theorem 3.1.
> - **(E3)** the influence of backward timestamp $t_0$ on the testing loss in Theorem 3.3.
>
> Our numerical results align with theory. Please refer to the empirical results in Appendix D of the latest revision for details.
>
> We also quote the results in below for your reference:
>
> ---
>
> ### **(E1) & (E2): Influence of Input Data Dimension $d_x$ on the Testing Loss and Parameter Norm Bounds at Backward Timestamp $t_0 = 5$**
>
> The testing loss and parameter norm bounds ($\|W_O\|$ and $\|W_V\|$) increase with an increasing $d_x$. These results are consistent with the findings in **Theorems 3.1 and 3.3**.
>
> | **Input Data Dim.** $d_x$           | $32 \cdot 32 $     | $48 \cdot 48 $     | $64 \cdot 64 $     | $80 \cdot 80$     |
> |-------------------------------------|---------------------------|---------------------------|---------------------------|---------------------------|
> | **Testing Loss**                    | 0.9321                    | 0.9356                    | 0.9364                    | 0.9476                    |
> | **$\|W_O\|_{2,\infty}$**            | 1.6074                    | 1.6332                    | 1.6789                    | 1.6886                    |
> | **$\|W_V\|_{2,\infty}$**            | 2.1513                    | 2.1767                    | 2.1858                    | 2.1994                    |
>
> ---
>
> ### **(E3) Influence of Backward Timestamp $t_0$ on the Testing Loss**
>
> The testing loss increases with increasing $t_0$. This is consistent with the result in **Theorem 3.3**.
>
> | **Testing Loss**       | $t_0 = 5$ | $t_0 = 4$ | $t_0 = 3$ | $t_0 = 2$ | $t_0 = 1$ |
> |------------------------|-----------|-----------|-----------|-----------|-----------|
> | $32 \cdot 32 = 1,024$  | 0.9321    | 0.9329    | 0.9335    | 0.9350    | 0.9361    |
> | $48 \cdot 48 = 2,304$  | 0.9356    | 0.9357    | 0.9360    | 0.9363    | 0.9367    |
>
> ---

---

> ### Author Response · Authors · 2024-11-18
> **Response 2 (Question 1: score approximation)**
>
> > `Q1-1.`  [Approximation, Thm 3.1] - $\sigma_t^2 \to 0$ when $t \to 0$, leading to infinitely large upper bounds (Table 1).
>
>
> Thanks for your question.
>
> **There is an early stopping time $t_0$ introduced to prevent the score function from blowing up.**
> This is common practice in both experimental techniques [Dhariwal21, Song20] and theoretical analysis [Fu24, Chen23].
>
> **We had mentioned this in `line 116` of the submitted draft.** We understand that the submitted draft wasn’t apparent. We have made modifications to highlight it. See `line 115` of the latest revision.
>
> [Fu24] Unveil Conditional Diffusion Models with Classifier-free Guidance: A Sharp Statistical Theory.
>
> [Chen23] Score Approximation, Estimation and Distribution Recovery of Diffusion Models on Low-dimensional Data. ICML 2023.
>
> [Dhariwal21] Diffusion Models Beat GANs on Image Synthesis. NeurIPS 2021.
>
> [Song20] Improved Techniques for Training Score-based Generative Models. NeurIPS 2020.
>
> > `Q1-2.`   [Approximation, Thm 3.1] - $\epsilon^{1/d} \to 1$ when $d \to \infty$, leading to $O(1)$ upper bounds (Table 1) that are vacuous in high-dimensional settings.
>
>
> Thank you for the question. We assume the reviewer is referring to the approximation results in the second column of Table 1.
>
> In response, **we remind the reviewer that Table 1 of the submitted version omits all log factors.** Theorems 3.1 and 3.2 show that the $\log N$ term dominates in the high-$d_x$ region.
>
> **In fact, one would obtain an $O(\log(1/\epsilon)^d)$ asymptotic for Theorem 3.1 and a $\log(1/\epsilon)^{O(1)}$ asymptotic for Theorem 3.2.** This implies that DiTs, under a stronger Hölder assumption, mitigate the curse of dimensionality in score approximation. The bounds are not vacuous.
>
> We acknowledge that the submitted Table 1 did not emphasize asymptotic behaviors. We have corrected these issues and updated the draft accordingly. Please refer to the revised Table 1 and the updated draft for details.
>
> > `Q1-3`  [Approximation, Thm 3.1]  - The upper bound is exponentially large given high-dimensional inputs, due to the multiplicative term $(\log N)^{d_x}$ (Theorem 3.1).
>
> Yes, you are correct. Theorem 3.1 (DiT score approximation under a weak Hölder assumption) suffers from high-dimensional input. However, as discussed above,
> this is expected, as practical DiTs (those without strong assumptions) naturally adapt to a latent design to avoid high-dimensional input [Bao23, Peebles22].
>
> Moreover, Theorem 3.2 mitigates this issue with a stronger assumption.
>
> [Bao23] All are worth words: A Vit Backbone for Diffusion Models, CVPR 2023.
>
> [Peebles22] Scalable Diffusion Models with Transformers, ICCV 2022.
>
> > `Q1-4.` [Approximation, Thm 3.1]   - The parameter bounds of Transformers are also exponentially large given long input sequences (with fixed dimensions), since $\| W_{Q,K} \| = O\left( N^{L/d} \right)$ (Theorem 3.1).
>
>
> Thanks for pointing this out. We apologize for the typo.
>
> We have updated the parameters’ norm bounds. **There is no longer exponential dependence on input length.**

---

> ### Author Response · Authors · 2024-11-18
> **Reponse 3 (Question 2:  score estimation)**
>
> > `Q2-1`   - In Theorem 3.3, why does the upper bound derived from Assumption 3.1 (weaker) have a much better dependence of $\log n$ $\left( (\log n)^{-dL} \right)$ vs. $\left( \log n \right)^{\Omega(1)}$ over the upper bound derived from Assumption 3.2 (stronger)?
>
> Thanks for pointing this out.
>
> We have acknowledged **there was a typo of asymptotics of $d_x$ for weak Holder assumption (Assumption 3.1)**. The correct dependence is $\log^{d_x}$ instead of $\log^{-d_x}$ for score estimation under assumption 3.1. Please see revised Table 1 for corrected asymptotics.
>
> We remark that **this typo arises from the change of dominance between the $n$ and $\log n$ terms in our analysis.**
>
> In response, in Appendix K.3 of the latest revision, **we have identified a sharp transition of the dominance between the $N$ and $\log N$ terms at $d_x=\Omega(\frac{\log N}{\log \log N})$.** This leads to two distinct asymptotic behaviors of the estimation error.
>
> Consequently, we provide separate analyses for the 2 cases:
>
> * High dim region ($d_x=\Omega (\frac{\log N}{\log \log N})$) [`Thm 3.3, 3.4` of the latest revision]
> * Low dim region ($d_x=o(\frac{\log N}{\log \log N}$) [`Corollary 3.3.1, 3.4.1` of the latest revision]
>
> Since the high-dim region is more practical, we modified Thm 3.1, 3.2, 3.3, 4.1, 4.2, 4.3, and Table 1 with corresponding results and left the low dim results to corollary 3.3.1 and 3.4.1.
>
> **We remark that the poor asymptotics of $d_x$ in the previous submission were due to results in the low-dimensional region (where we shouldn’t consider asymptotics of $d_x$ at all).**
>
> > `Q2-2` Also, the former error term $(\log n)^{-dL}$ decreases exponentially fast as the data dimension or sequence length increases, which seems unreasonable.
>
> Yes, it is not reasonable for above explained reasons.
>
> As discussed in above question, it’s the result of low-$d_x$ region where $n$ term dominates. Please see the newly added Corollary 3.3.1 in the latest revision for details.
>
> > `Q2-3`   - In Theorem 3.3, for the upper bound derived from Assumption 3.2, $\nu_3 = O(L/d)$, which gives $N = n^{\Omega(d/L)}$, $t_0 \sim O(n^{-d/L})$, and $\log(1/t_0) = \Omega(d/L \cdot \log n)$. This term can be merged into the subsequent polynomial dependences in data dimensions and sequence lengths (i.e., $d^{14} L^4$), and hence introduces no singularity. However, the power of $1/n$ is approximately $1/(\nu_3 \cdot d) = \Omega(1/L)$, leading to the vacuous sample complexity estimate given long input sequences.
>
>
> **We have updated Thm 3.3 with appropriate rates for high-dimensional considerations.**
>
> * The updated score estimation rate under Assumption 3.1 (weak Holder) is $n^{-\Theta(1/d_x)} \cdot (\log n)^{\mathcal{O}(d_x)}$
>
> * The updated score estimation rate under Assumption 3.2 (strong Holder) is $n^{-\Theta(1/d_x^2)} \cdot (\log n)^{\mathcal{O}(1)}$
>
> We can easily see that both cases are not vacuous when $d_x$ is large.
> Again, this suggests DiT with a stronger Holder assumption mitigates the curse of dimension.

---

> ### Author Response · Authors · 2024-11-18
> **Response 4 (Question 3-5)**
>
> > `Q3.` **For the distribution estimation:** - In Theorem 3.4, there are the same issues raised in the second point of the score estimation, since the upper bounds in Theorem 3.4 and Theorem 3.3 are in similar forms.
>
> You are correct. The same issue propagated from Thm 3.3 to Thm 3.4 in the submitted draft. As explained above, we have fixed this issue and modified the draft accordingly. Thank you for your attention to the details!
>
>
> > `Q4.` It seems that there is no characterization of the model capacity dependence. That is, all the provided estimates do not contain (at least, explicitly) error terms related to $h, s, r$.
>
>
> Throughout this paper, we use the transformer network class $\mathcal{T}^{h,s,r}$, i.e., h=1, s=1, r=4. We use the universal approximation of an adapted one-layer single-head self-attention transformer network (see Figure 1) for approximation analysis. We remark that this is a special case of our considered function class $\mathcal{T}^{h,s,r}$.
>
> We have added a new Corollary H.2.1 to highlight this. **The choice is meant to show that DiT with the simplest transformer configuration (with 1-head, 1-layer attention) is capable of executing the discussed approximation and estimation.**
>
> We remark that this configuration is minimally sufficient but not necessary. More complex configurations can also achieve transformer universality and hence score approximation, as reported in [Hu24, Kajitsuka24, Yun20].
>
> The significance of this setting for analysis is to show — **even with this simple transformer network, it’s still possible to to get minimax optimal distribution estimation.**
>
> [Hu24] On Statistical Rates and Provably Efficient Criteria of Latent Diffusion Transformers (DiTs), NeurIPS 2024.
>
> [Kajitsuka24] Are Transformers with One Layer Self-Attention Using Low-Rank Weight Matrices Universal Approximators? ICLR 2024
>
> [Yun20] Are Transformers universal approximators of sequence-to-sequence functions? ICLR 2020
>
> > `Q5-1.` **Minor issues:**   - The notation $s$ denotes both the head size and Hölder smoothness (Definition 3.1).
>
> Thanks for pointing this out. We agree it’s an ambiguous notation design. **We have changed the Holder smoothness from $s$ to $k_1$.** The choices of $k_1$ is for expanding Eqn (3.1) with $k_1$- and $k_2$-order Taylor polynomials.
>
> > `Q5-2.` **Minor issues:**   - In the caption of Table 1, "... where $L, \tilde{L}$ are sequence length of transformer inputs", "length" $\to$ "lengths".
>
> Thank you for pointing that out. We have fixed this.
>
> Moreover, we have conducted 3 more rounds of proofreading and fixed all identified typos and grammatical errors.
>
>
> ---
>
> Thank you for your review and attention to detail. We hope the revisions and clarifications address your concerns and highlight the value of our work. We look forward to any further feedback and discussion.

---

> > ### Comment · Reviewer_1NAW · 2024-12-03
> > **Response to authors**
> >
> > Thanks for the detailed clarifications. I appreciate authors' hard efforts to provide further analysis and numerical verifications  under varied settings. However, I still have the following concerns:
> > 1. Given that so many terms and bounds are different from the original version, I am not convinced that the updated new bounds are validate. It seems that these major revisions are not due to simply minor typos. How do these changes happen?
> > 2. The authors make the claim that "DiT with a stronger Holder assumption mitigates the curse of dimension". This is right, but there are still terms like $n^{-1/d}$. This is the typical form of CoD, since $n^{-1/d} \le 0.1$ with $0.1$ as a tolerance implies $n \ge 10^d$. That is, the sample complexity is exponentially large for *relatively* large (not need to be that large) $d$.
> >     - We agree that the dimension-dependence of practical DiTs may scale poorly as input dimensions increase, but it is not supposed to be *exponentially* large as the theoretical bounds here.
> > 3. Minor typos in the experiments: "The testing loss increases with increasing $t_0$", increasing -> decreasing.

---

> ### Author Response · Authors · 2024-11-24
> **A Gentle Reminder**
>
> Dear Reviewer,
>
> As the rebuttal phase nears its end, we wanted to gently remind you of our responses.
>
> We have addressed all your comments and questions with the utmost care and hope our efforts meet your expectations. If you have any remaining concerns, we would be happy to discuss them further. Otherwise, we kindly invite you to consider raising your score if you find our updates satisfactory.
>
> We deeply appreciate the time and effort you have dedicated to reviewing our paper, especially given its substantial length and complexity! Regardless of the final outcome, we want to express our heartfelt gratitude for your thoughtful feedback and contributions to improving our work.
>
> Thank you!

---

> ### Author Response · Authors · 2024-12-02
> **Another Gentle Reminder**
>
> Dear Reviewer 1NAW,
>
> As the rebuttal phase nears its end, we would greatly appreciate your further feedback on our rebuttal.
>
> In above rebuttal, we have made every effort to address all your concerns and additional questions in our responses.
>
> If you feel your concerns have been adequately addressed, with utmost respect, we invite you to consider a score adjustment. If not, we hope to make the most of the remaining two days to provide further clarifications.
>
> Thank you for your time and consideration!
>
> Best,
>
> Authors

---

> ### Author Response · Authors · 2024-12-03
> **Response to Additional Comments**
>
> >`Comment 1.` Given that so many terms and bounds are different from the original version, I am not convinced that the updated new bounds are validated. It seems that these major revisions are not due to simply minor typos. How do these changes happen?
>
> Thank you for your question. The changes are due to the following two reasons, and we have ensured that the updated version is correct.
>
> **Miscalculation of the granularity $D$**:
>
> -  The first reason is an error in Lemma H.4, where the order of the granularity $D$ was miscalculated.
> - This error propagated to Lemma H.5, affecting the matrix norm bounds in the self-attention and feed-forward layers.
> - Consequently, the norm bounds in the main theorems (Theorems 3.3, 3.4, E.3, and 4.2) also changed, as they were derived using results from Lemma H.5.
>
> **Overlooked transition of transformer approximations in data dimension**:
>
> - The second reason is related to the choice of transformers for approximating various functions.
> - To ensure bounds sufficient for all cases, we must select the largest bound as the final approximation bound (Appendix K.3).
> - During this process, we overlooked a critical transition between high-dimensional and low-dimensional data inputs in Theorem 3.1. Specifically, the norm bounds differ when $d_x$ (input dimensionality) transitions around the threshold $\Theta(\log N / \log \log N)$.
> - This oversight led to an incorrect choice of the largest bound.
> - To address this, we added Appendix K.3–K.7 for analyses of high-dimensional and low/mild-dimensional cases and corrected the derived rates accordingly.
>
> >`Comment 2.` The authors make the claim that "DiT with a stronger Holder assumption mitigates the curse of dimension". This is right, but there are still terms like $n^{-1/d}$. This is the typical form of CoD, since $n^{-1/d} \leq 0.1$ as a tolerance implies $n \geq 10^d$. That is, the sample complexity is exponentially large for relatively large (not need to be that large) d. We agree that the dimension-dependence of practical DiTs may scale poorly as input dimensions increase, but it is not supposed to be exponentially large as the theoretical bounds here.
>
> Thanks for your comment. Here are some clarifications:
>
> - We acknowledge that there is still a curse of dimensionality for the conditional DiT under a stronger Hölder assumption.
> - However, in practice, the latent conditional DiT is commonly used. We observe the dependence on the hidden dimension $d_0$​ rather than the input dimension $d_x$​, where $d_0$​ is a model hyperparameter that we set ourselves. This solves the curse of dimensionality in practical applications.
>
> >`Comment 3.` Minor typos in the experiments: "The testing loss increases with increasing $t_0$", increasing -> decreasing.
>
> Thank you for pointing this out. We have updated it to the latest version.

---

### Official Review · Reviewer_GBgr · 2024-11-03

**Soundness:** 3
**Presentation:** 3
**Contribution:** 3
**Rating:** 6
**Confidence:** 3

**Summary:**

This paper provide a theoretical analysis for statistical rates of conditional diffusion transformer, mainly focusing on the score approximation, score estimation and distribution estimation.

**Strengths:**

1. This paper is well-written, which provides detailed theoretical analysis and clear explanations. Additionally, the main theorems are presented concisely and effectively, and the discussions in remarks are particularly helpful for interpreting the results.
2. Comparing with the previous work ([1]), the theoretical analysis alleviates the double exponential factor and achieves minimax optimal statistical rates for DiTs under Hölder smoothness data assumption.

[1]. Hu J Y C, Wu W, Li Z, et al. On statistical rates and provably efficient criteria of latent diffusion transformers (dits)[J]. arXiv preprint arXiv:2407.01079, 2024.

**Weaknesses:**

1. As a follow-up work of [1], maybe the contribution mainly focuses on polishing and extending the previous results, which may lead to a lack of novelty. Could you provide some new insight points about it?
2. With the good rates theoretically,  could this work provide some implementations in practice?

[1]. Hu J Y C, Wu W, Li Z, et al. On statistical rates and provably efficient criteria of latent diffusion transformers (dits)[J]. arXiv preprint arXiv:2407.01079, 2024.

**Questions:**

See weakness.

---

> ### Author Response · Authors · 2024-11-18
> **Response 1**
>
> ### Thank you for your review.
> ### We have addressed all your comments and questions in this and the following responses.
> ### The draft has been updated accordingly, with changes highlighted in blue in the latest revision.
>
> ---
>
> > `W1.` As a follow-up work of [1], maybe the contribution mainly focuses on polishing and extending the previous results, which may lead to a lack of novelty. Could you provide some new insight points about it?
>
>
> Thanks for the question. We believe our work goes beyond [Hu2024] in 4 aspects:
> 1. **New Technique:** We discretize the input domains into infinitesimal grids and perform a term-by-term Taylor expansion on the conditional diffusion score function. This allows for the fine-grained use of transformers’ universal approximation capabilities (`line 254-280`).
> 2. **New Universal Approximation:** We provide a modified universal approximation (Theorem H.2) of the single-layer self-attention transformers (modified from [Kajitsuka24]) to avoid the need for dense layers required in [Hu24, Yun20].
> 3. **New Insights:** Leveraging our advanced techniques from 1. and the improved universal approximation from 2., we show that both conditional DiTs and their latent variants lead to the minimax optimality of unconditional DiTs under identified settings. Additionally, latent conditional DiTs attain lower bounds in both approximation and estimation compared to conditional DiTs, offering practical insights for developing more efficient and precise DiT models.
> 4. **Comprehensive Analysis of Interaction among DiTs with Common Assumptions:** By building on insights from 1. to 3. and considering different data assumptions (Assumption 3.1 and 3.2), **we conduct a unified and comprehensive analysis of DiT.** This provides practical guidance and demonstrates the effectiveness of DiT across various data properties.
>
> Our primary emphasis is on the technical advancements and a comprehensive understanding of DiTs' approximation and estimation capabilities. We believe these represent a significant step beyond mere incremental improvements.
>
> [Hu24] On Statistical Rates and Provably Efficient Criteria of Latent Diffusion Transformers (DiTs), NeurIPS 2024.
>
> [Kajitsuka24] Are Transformers with One Layer Self-Attention Using Low-Rank Weight Matrices Universal Approximators?
>
> [Yun20] Are Transformers universal approximators of sequence-to-sequence functions? ICLR 2020
>
> > `W2.` With the good rates theoretically, could this work provide some implementations in practice?
>
>
> Thanks for the comment.
> Here are two examples of practical implementation guidelines from our work:
>
> 1. **Use the latent conditional DiT to bypass the challenges associated with high input data dimensions.** By comparing Theorem 4.1 with Theorems 3.1 and 3.2 (score approximation), Theorem E.3 with Theorem 3.3 (score estimation), and Theorem 4.2 with Theorem 3.4 (distribution estimation), it shows that the results of the latent conditional DiT depend on the latent variable dimension $d_0$, rather than the input data dimension $d_x$​. This shows the utility of using latent conditional DiT to address challenges linked to high data dimensionality.
>
> 2. **Set the termination timestamp $T$ in the forward process as $\mathcal{O}(\log n)$ ($n$ is the number of training samples).** The results in Theorems 3.3 and E.3 support setting $T = \mathcal{O}(\log n)$. This leads to minimax optimal estimation rates in Corollary 3.4.2 and Remark 4.3. This provides guidance on setting the hyperparameter $T$ in practice: setting $T = \mathcal{O}(\log n)/d_x$ is sufficient, and $T$ does not need to be too large. A larger value is detrimental as it increases the number of denoising steps in the sampling process.
>
> In summary, our results imply design choices to obtain sharper estimation and approximation for DiTs.
>
> ---
>
> Thank you for your time and valuable feedback. We hope that above clarifications address the reviewer's concerns. We are happy to engage further discussions!

---

> > ### Comment · Reviewer_GBgr · 2024-11-25
> > **Response to authors**
> >
> > Thanks for addressing my concerns. That makes sense. I have raised the score to 6.

---

> > > ### Author Response · Authors · 2024-11-25
> > >
> > > Thank you!
> > >
> > > We are glad our revisions and responses meet your expectations.
> > >
> > > If there is anything further we can clarify or improve to strengthen your support and encourage a higher score, please let us know, and we will address it promptly.
> > >
> > > If not, thank you again for your review and constructive comments. They have been invaluable in improving our work!

---

> ### Author Response · Authors · 2024-11-24
> **A Gentle Reminder**
>
> Dear Reviewer,
>
> As the rebuttal phase nears its end, we wanted to gently remind you of our responses.
>
> We have addressed all your comments and questions with the utmost care and hope our efforts meet your expectations. If you have any remaining concerns, we would be happy to discuss them further. Otherwise, we kindly invite you to consider raising your score if you find our updates satisfactory.
>
> We deeply appreciate the time and effort you have dedicated to reviewing our paper, especially given its substantial length and complexity! Regardless of the final outcome, we want to express our heartfelt gratitude for your thoughtful feedback and contributions to improving our work.
>
> Thank you!

---

### Official Review · Reviewer_mSoC · 2024-11-04

**Soundness:** 4
**Presentation:** 3
**Contribution:** 3
**Rating:** 8
**Confidence:** 2

**Summary:**

The paper studies the statistical understanding of the conditional diffusion transformers (DiTs) with classifier-free guidance. The authors focus on understanding the approximation and estimation rates of these models under various data conditions, particularly in "in-context" settings. They show that both conditional DiTs and their latent variants can achieve minimax optimality, similar to unconditional DiTs, in certain scenarios. The analysis is based on discretizing input domains into small grids and applying Taylor expansions to the conditional diffusion score function, assuming data follows Hölder smoothness. This approach allows for a more refined piecewise approximation, leveraging transformers' universal approximation capacity. The study also extends to latent conditional DiTs, showing that these models offer better approximation and estimation bounds than standard conditional DiTs, especially under linear latent subspace assumptions.

**Strengths:**

- The paper provides a detailed theoretical framework for analyzing the approximation and estimation rates of conditional diffusion transformers (DiTs) under various data assumptions.
- The work integrates classifier-free guidance into the analysis of conditional DiTs.
-  The paper provides rigorous results on score approximation and distribution estimation, including sample complexity bounds for score estimation which lead to minimax optimal estimation results.
- The work integrates classifier-free guidance into the analysis of conditional DiTs.
- Beyond theoretical contributions, the findings offer practical guidance for configuring transformer-based diffusion models to achieve optimal performance.

**Weaknesses:**

A general weakness of this paper is its reliance on a lengthy and detailed appendix, which is not mandatory for reviewers to read. While the results are strong and represent a valuable contribution to the field, the extensive appendix makes it challenging to validate the findings within a limited review timeframe.

- The analysis relies significantly on Hölder smoothness assumptions, which may not apply to all types of data. This limitation restricts the generalizability of the theoretical results to datasets that satisfy these specific conditions. Additionally, the assumption of isotropic smoothness across input variables does not always hold, particularly in cases where the condition is discrete or irregular.

- The study does not address the high-dimensional structural dependence of the data. It is overly restrictive to assume that the ambient dimensions of practical data distributions are small and that the rates of dependence relative to the ambient dimension are sufficiently slow.

- The paper lacks empirical validation that connects the theoretical rates and dimensionality to practical datasets. For instance, it would be beneficial to clarify what smoothness refers to in the context of the given dataset. Furthermore, it is unclear how one might establish practical relevance for the proposed rates and validate them empirically.

**Questions:**

In addition to the weaknesses outlined, please consider the following:

- Can the work be extended to a manifold setup where the rates depend on the intrinsic dimension of the data, similar to the approach taken by Oko et al. (2023)?

- Regarding the assumptions made, if one has a function with significantly less restrictive conditions than those presented in this study, what advantage does using a transformer provide over traditional statistical methods for achieving a minimax rate? This question is particularly relevant from a theoretical perspective, which is the primary focus of this paper.

---

> ### Author Response · Authors · 2024-11-18
> **Response 1**
>
> ### Thank you for your review.
> ### We have addressed all your comments and questions in this and the following responses.
> ### The draft has been updated accordingly, with changes highlighted in blue in the latest revision.
> ### Please refer to the updated PDF for details.
>
> ---
>
> > `W1.` A general weakness of this paper is its reliance on a lengthy and detailed appendix, which is not mandatory for reviewers to read. While the results are strong and represent a valuable contribution to the field, the extensive appendix makes it challenging to validate the findings within a limited review timeframe.
>
>
> Thanks for the comment and suggestion. Due to the comprehensive theoretical work we provide under various data assumptions, the proof in the appendix is indeed lengthy. However, in the revised version, we have organized the content to be more precise and concise and removed redundant or similar proofs.
>
> > `W2.` The analysis relies significantly on Hölder smoothness assumptions, which may not apply to all types of data. This limitation restricts the generalizability of the theoretical results to datasets that satisfy these specific conditions. Additionally, the assumption of isotropic smoothness across input variables does not always hold, particularly in cases where the condition is discrete or irregular.
>
> Thanks for your comment. Here are some clarifications.
>
> The results in Sections 3 and 4 indeed depend on the Hölder smoothness assumptions. As stated in the introduction, **our analysis for these sections is not intended to apply to all types of data.** Our primary goals are
>
> 1. to obtain **tight statistical results** and
> 2. to conduct a **comprehensive study** of how conditional DiTs interact with various **common assumptions**.
>
> We remark that the considered assumptions are common in literature [Wibisono24,Fu24,Hu24,Oko23]. Please see related work discussions for details.
>
> Moreover, **our results in Appendix F apply to all types of data.** In Appendix F, we extend the analysis from [Hu24] to the conditional DiT setting and provide an improved version. By setting the matrix $U$ in Assumption F.1 as the identity matrix and following the proof in Appendix F, we are able to derive the approximation and estimation errors for all types of data.
>
> [Wibisono24] Optimal score estimation via empirical bayes smoothing. arXiv preprint arXiv:2402.07747 (2024).
>
> [Hu24] On Statistical Rates and Provably Efficient Criteria of Latent Diffusion Transformers (DiTs), NeurIPS 2024.
>
> [Oko23] Diffusion models are minimax optimal distribution estimators. ICML 2023
>
> [Fu24] Unveil Conditional Diffusion Models with Classifier-free Guidance: A Sharp Statistical Theory
>
> > `W3.` The study does not address the high-dimensional structural dependence of the data. It is overly restrictive to assume that the ambient dimensions of practical data distributions are small and that the rates of dependence relative to the ambient dimension are sufficiently slow.
>
> Thanks for the comment.
> We believe there may be a slight oversight. **Our analysis of latent DiTs addresses the high-dimensional structural dependence of the data.** It is clear that latent design bring exponential improvement w.r.t. $d_x/d_0$.
>
> Specifically, the results in Theorems 4.1, 4.2, and E.3 show that the approximation and estimation error bounds of latent conditional DiTs depend on the latent variable dimension $d_0$​, not $d_x$​. **By controlling $d_0$​ to prevent it from becoming too large, we overcome the challenges associated with high-dimensional input data.** Additionally, this aligns with the de facto use of latent conditional DiTs in practice [Bao23, Peebles22].
>
> Moreover, our results indicate that, under the stronger Holder assumption, the high-dimensional asymptotics are better. In particular,
> * the score approximation results are **dimension-free** (`Table 1` of the latest revision)
> * Moreover, while the estimation results under the weak Hölder assumption depend on  $e^{d_x}$, **the estimation results with the stronger Hölder assumption eliminate this exponential dependence on $e^{d_x}$.**
>
> These indicate that DiTs have better scalability on strongly Hölder-smooth data.
>
> [Bao23] All are worth words: A Vit Backbone for Diffusion Models, CVPR 2023.
>
> [Peebles22] Scalable Diffusion Models with Transformers, ICCV 2022.

---

> ### Author Response · Authors · 2024-11-18
> **Response 2**
>
> > `W4.` The paper lacks empirical validation that connects the theoretical rates and dimensionality to practical datasets. For instance, it would be beneficial to clarify what smoothness refers to in the context of the given dataset. Furthermore, it is unclear how one might establish practical relevance for the proposed rates and validate them empirically.
>
> Thanks for the comment.
> In response **empirical validation**, **we have included empirical results in Appendix D of the revised manuscript.**
>
> Specifically, we train a conditional diffusion transformer model on the CIFAR10 to verify three points:
>
> - **(E1)** the influence of input data dimension $d_x$ on the testing loss in Theorem 3.3.
> - **(E2)** the influence of input data dimension $d_x$ on the parameter norm bounds in Theorem 3.1.
> - **(E3)** the influence of backward timestamp $t_0$ on the testing loss in Theorem 3.3.
>
> Our numerical results align with theory. Please refer to the empirical results in Appendix D of the latest revision for details.
>
> In response **data smoothness, defining the smoothness of the given dataset (e.g., CIFAR10) is challenging because it requires knowledge of the dataset's exact distribution.** However, we can generate a dataset with a specified level of smoothness. For example, by constructing a density function that satisfies Assumption 3.1, we can sample a dataset according to this density function. The sampled dataset then exhibits the specified smoothness.
>
> ---
>
> We also quote the results in below for your reference:
>
>
> ### **(E1) & (E2): Influence of Input Data Dimension $d_x$ on the Testing Loss and Parameter Norm Bounds at Backward Timestamp $t_0 = 5$**
>
> The testing loss and parameter norm bounds ($\|W_O\|$ and $\|W_V\|$) increase with an increasing $d_x$. These results are consistent with the findings in **Theorems 3.1 and 3.3**.
>
> | **Input Data Dim.** $d_x$           | $32 \cdot 32 $     | $48 \cdot 48 $     | $64 \cdot 64 $     | $80 \cdot 80$     |
> |-------------------------------------|---------------------------|---------------------------|---------------------------|---------------------------|
> | **Testing Loss**                    | 0.9321                    | 0.9356                    | 0.9364                    | 0.9476                    |
> | **$\|W_O\|_{2,\infty}$**            | 1.6074                    | 1.6332                    | 1.6789                    | 1.6886                    |
> | **$\|W_V\|_{2,\infty}$**            | 2.1513                    | 2.1767                    | 2.1858                    | 2.1994                    |
>
> ---
>
> ### **(E3) Influence of Backward Timestamp $t_0$ on the Testing Loss**
>
> The testing loss increases with increasing $t_0$. This is consistent with the result in **Theorem 3.3**.
>
> | **Testing Loss**       | $t_0 = 5$ | $t_0 = 4$ | $t_0 = 3$ | $t_0 = 2$ | $t_0 = 1$ |
> |------------------------|-----------|-----------|-----------|-----------|-----------|
> | $32 \cdot 32 = 1,024$  | 0.9321    | 0.9329    | 0.9335    | 0.9350    | 0.9361    |
> | $48 \cdot 48 = 2,304$  | 0.9356    | 0.9357    | 0.9360    | 0.9363    | 0.9367    |
>
> ---

---

> ### Author Response · Authors · 2024-11-18
> **Response 3**
>
> > `Q1.` Can the work be extended to a manifold setup where the rates depend on the intrinsic dimension of the data, similar to the approach taken by Oko et al. (2023)?
>
> Thanks for your question. Here are some clarifications.
>
> **We have already included the results for the manifold setup in the original submission (Appendix F of the latest revision.)**  We provide the score approximation error in Theorem F.1, score estimation error in Theorem F.2, and distribution estimation error in Theorem F.3. All of these results demonstrate that latent DiTs have the potential to overcome the challenges associated with high-dimensional input data.
>
> > `Q2.` Regarding the assumptions made, if one has a function with significantly less restrictive conditions than those presented in this study, what advantage does using a transformer provide over traditional statistical methods for achieving a minimax rate? This question is particularly relevant from a theoretical perspective, which is the primary focus of this paper.
>
>
> Thanks for your question. Here are some clarifications.
>
> By setting the matrix $U$ in Assumption F.1 as the identity matrix and following the proof in Appendix F, we are able to derive the approximation and estimation errors for settings with less restrictive conditions.
>
> In this setting, deriving a minimax rate proves challenging due to the difficulty in establishing a minimum rate under less restrictive conditions. However, **under Assumption 3.2, we are able to derive the minimax rates for both DiTs and latent DiTs.**
>
> ---
>
> We sincerely appreciate the time and effort that Reviewer mSoC has invested in reviewing our paper. We have taken all comments into careful consideration and have made corresponding revisions to address the concerns raised.
>
>  Please do not hesitate to let us know if there are any other aspects of our work that you would like us to clarify.

---

> ### Author Response · Authors · 2024-11-24
> **A Gentle Reminder**
>
> Dear Reviewer,
>
> As the rebuttal phase nears its end, we wanted to gently remind you of our responses.
>
> We have addressed all your comments and questions with the utmost care and hope our efforts meet your expectations. If you have any remaining concerns, we would be happy to discuss them further. Otherwise, we kindly invite you to consider raising your score if you find our updates satisfactory.
>
> We deeply appreciate the time and effort you have dedicated to reviewing our paper, especially given its substantial length and complexity! Regardless of the final outcome, we want to express our heartfelt gratitude for your thoughtful feedback and contributions to improving our work.
>
> Thank you!

---

> > ### Comment · Reviewer_mSoC · 2024-11-25
> > **Reviewer's response**
> >
> > Thank you for your time and effort. I will thoroughly review the updated draft and take it into consideration before finalizing my score at the end of the review period.

---

> > > ### Author Response · Authors · 2024-11-25
> > >
> > > Thank you for letting us know!
> > >
> > > Since the rebuttal period has been extended by a week, please let us know if there is anything further we can clarify or improve to strengthen your support and encourage a higher score. We will address it promptly.
> > >
> > > Thank you again for reviewing our work and providing constructive comments—they have been invaluable in improving our manuscript!

---

> > > > ### Comment · Reviewer_mSoC · 2024-12-02
> > > >
> > > > Dear Authors,
> > > >
> > > > Thank you for your thoughtful and thorough responses to my concerns and questions. After carefully reviewing your clarifications and the revised paper, which I believe is in significantly better shape, I have decided to raise my score.
> > > >
> > > > I appreciate the effort you have put into addressing the issues, and I believe the manuscript has been strengthened as a result.
> > > >
> > > > Sincerely,
> > > > Reviewer mSoC

---

> ### Author Response · Authors · 2024-12-02
>
> We are very happy to see that our revisions and responses have met your satisfaction.
>
> Thank you again for reviewing our work and constructive comments!

---

### Official Review · Reviewer_WTrU · 2024-11-04

**Soundness:** 3
**Presentation:** 3
**Contribution:** 2
**Rating:** 6
**Confidence:** 3

**Summary:**

This paper studies the statistical rates of approximation and estimation for conditional diffusion transformers (DiTs) with classifier-free guidance. It provides a comprehensive analysis under four common data assumptions, showing that both conditional and latent variants of DiTs can achieve minimax optimality. It uses a modified universal approximation of the single-layer self-attention transformer model to circumvent the need for dense layers, which results in tighter error bounds for both score and distribution estimation.

**Strengths:**

The paper is original in its theoretical approach to studying conditional diffusion transformers, offering new insights into their performance and limitations. The quality of the theoretical analysis is high, with clear outlines and complete proofs. The clear and thorough exploration of statistical limits and estimation procedures is a key strength.

**Weaknesses:**

1. Some results seem to suffer from the curse of dimensionality, which seems not a practical bound.
2. There is an absence of empirical results to complement the theoretical analyses.
3. More practical implications of the assumptions could be added.

**Questions:**

**1.** On the top of page 3, should the denominator of $\nu_3$ be $d_0+d_y$ instead of $d_x + d_y$?

**2.** The bound in Theorem 3.1, the first bound of Theorem 4.1, and the first bound of Theorem 4.2 seem to depend exponentially on $d_x$ or $d_0$, which seems to suffer from CoD. Is the bound tight here?

**3.** Is there any practical example that matches the assumption 3.2? If so, I’d suggest adding a brief paragraph to discuss this.

**4.** In the first bound of Theorem 3.3 and Theorem 3.4, the bound seems to be decaying in a rate of $exp(-d_x)$, is this a typo?

**5.** The second bound of Theorem 3.3 seems to be not sensitive to $d_y$. The second bound of Theorem 3.4 depends on $d_x$ and $d_y$ only through $d_x+d_y$, and is not sensitive to it. Similarly, the second bound of Theorem 4.2 only depends on $d_0$ and $d_y$ only through $d_0+d_y$. Is there any intuition behind this?

**6.** There aren't any empirical results in the current version. I’d suggest adding a few practical examples to illustrate the validation of the bounds.

---

> ### Author Response · Authors · 2024-11-18
> **Response 1**
>
> ### Thank you for your review.
>
> ### We have addressed all your comments and questions in this and the following responses.
>
> ### The draft has been updated accordingly, with changes highlighted in **blue** in the latest revision.
>
> ### Please refer to the updated PDF for details.
>
> ---
>
> > `W1.` Some results seem to suffer from the curse of dimensionality, which seems not a practical bound.
>
> Thanks for pointing this out. Here are some clarifications:
>
> 1. Yes, you are correct. **However, we remark that this aligns with practice.**
>     In fact, **it is well-known that DiT cannot handle raw high-dimensional data well.**
>     Practitioners always use the latent conditional DiT and control $d_0$ to keep the input dimension manageable.
>
>
> 2. Moreover, the results for latent DiTs in Section 4, Appendix E, and Appendix F show that both approximation and estimation errors depend on the latent dimension $d_0$ rather than the input dimension $d_x$. By controlling $d_0$ to keep it manageable, we address the challenges posed by high-dimensional input data.
>
>     This demonstrates that **latent conditional DiTs can effectively bypass the curse of dimensionality, which aligns with the practical use of latent conditional diffusion transformers [Peebles22].**
>
> 3. Additionally, our results indicate that **under the stronger Hölder assumption, high-dimensional asymptotics improve.** Specifically,
>     - **score approximation becomes dimension-free**, and
>     - while estimation results under the weak Hölder assumption depend on $e^{d_x}$, the **stronger assumption removes this exponential dependence on input dimension**.
>
>     These results also suggest better scalability for DiTs on strongly Hölder-smooth data, and even more scalable with latent design (Sec 4, E, F).
>
> We hope these address your concern.
>
> > `W2.` There is an absence of empirical results to complement the theoretical analyses.
>
> > `Q6.` There aren't any empirical results in the current version. I'd suggest adding a few practical examples to illustrate the validation of the bounds.
>
> Thanks for the suggestion. We agree to include empirical results to complement the theoretical analyses.
>
> In response, **we have included empirical results in Appendix D of the revised manuscript.**
>
> Specifically, we train a conditional diffusion transformer model on the CIFAR10 to verify three points:
>
> - **(E1)** the influence of input data dimension $d_x$ on the testing loss in Theorem 3.3.
> - **(E2)** the influence of input data dimension $d_x$ on the parameter norm bounds in Theorem 3.1.
> - **(E3)** the influence of backward timestamp $t_0$ on the testing loss in Theorem 3.3.
>
> Please refer to the empirical results in Appendix D of the latest revision for details.
>
> We also quote the results in below for your reference:
>
> ---
>
> ### **(E1) & (E2): Influence of Input Data Dimension $d_x$ on the Testing Loss and Parameter Norm Bounds at Backward Timestamp $t_0 = 5$**
>
> The testing loss and parameter norm bounds ($\|W_O\|$ and $\|W_V\|$) increase with an increasing $d_x$. These results are consistent with the findings in **Theorems 3.1 and 3.3**.
>
> | **Input Data Dim.** $d_x$           | $32 \cdot 32 $     | $48 \cdot 48 $     | $64 \cdot 64 $     | $80 \cdot 80$     |
> |-------------------------------------|---------------------------|---------------------------|---------------------------|---------------------------|
> | **Testing Loss**                    | 0.9321                    | 0.9356                    | 0.9364                    | 0.9476                    |
> | **$\|W_O\|_{2,\infty}$**            | 1.6074                    | 1.6332                    | 1.6789                    | 1.6886                    |
> | **$\|W_V\|_{2,\infty}$**            | 2.1513                    | 2.1767                    | 2.1858                    | 2.1994                    |
>
> ---
>
> ### **(E3) Influence of Backward Timestamp $t_0$ on the Testing Loss**
>
> The testing loss increases with increasing $t_0$. This is consistent with the result in **Theorem 3.3**.
>
> | **Testing Loss**       | $t_0 = 5$ | $t_0 = 4$ | $t_0 = 3$ | $t_0 = 2$ | $t_0 = 1$ |
> |------------------------|-----------|-----------|-----------|-----------|-----------|
> | $32 \cdot 32 = 1,024$  | 0.9321    | 0.9329    | 0.9335    | 0.9350    | 0.9361    |
> | $48 \cdot 48 = 2,304$  | 0.9356    | 0.9357    | 0.9360    | 0.9363    | 0.9367    |
>
> ---

---

> > ### Comment · Reviewer_WTrU · 2024-11-26
> > **Thank you**
> >
> > Dear authors,
> >
> > Thank you for answering my questions and addressing my concerns. I've updated my score accordingly.

---

> > > ### Author Response · Authors · 2024-11-26
> > >
> > > Thank you!
> > >
> > > We are glad our revisions and responses meet your expectations.
> > >
> > > If there is anything further we can clarify or improve to strengthen your support and encourage a higher score, please let us know, and we will address it promptly.
> > >
> > > If not, thank you again for your review and constructive comments. They have been invaluable in improving our work!

---

> ### Author Response · Authors · 2024-11-18
> **Response 2**
>
> > `W3.` More practical implications of the assumptions could be added.
>
> > `Q3.` Is there any practical example that matches the assumption 3.2? If so, I'd suggest adding a brief paragraph to discuss this.
>
> Thanks for your suggestion. Here are two common examples. We will include this discussion in the latest revision.
>
> * **Diffusion Models in Image Generation:** When modeling conditional distributions of images given attributes (e.g., generating images based on class labels), **the considered Holder assumptions hold if the data distribution around these attributes is smooth and decays.**
>
>     In diffusion-based generative models, the data distribution often decays smoothly in high-dimensional space. The assumption that the density function decays exponentially reflects the natural behavior of image data, where pixels or features far from a central region or manifold are less likely. **This is commonly observed in images with blank boundaries.**
>
> * **Physical Systems with Gaussian-Like Decay:** This applies to cases where the spatial distribution of a physical quantity, such as temperature, is smooth and governed by **diffusion equations with exponential decay.**
>
>     In physics-based diffusion models, like those simulating the spread of particles or heat in a medium (e.g., stars in galaxies for astrophysics applications), **the conditional density typically decays exponentially with distance from a central region.**
>
> > `Q1.` On the top of page 3, should the denominator of $ \nu_3 $ be $ d_0 + d_y $ instead of $ d_x + d_y $?
>
> Thanks for the question. We believe the reviewer "WTrU" refers to $\nu_3$ at the top of page 2.
>
> In response, **there is no typo in $\nu_3$.** $\nu_3$ comes from computing the covering number of the transformer we used to facilitate score approximation.
>
> The term $\nu_3$ appears in the estimation error of conditional DiTs when considering the **input data dimension $d_x$** without assuming a low-dimensional latent subspace.
>
> In contrast, $d_0$ represents the **latent variable dimension** and is related to the estimation error of latent conditional DiTs. Therefore, the denominator of  $\tilde{\nu_3}$ should be $d_0+d_y$, while the denominator of $\nu_3$ should be $d_x+d_y$.
>
> In addition, we have updated Table 1 with only asymptotics for better clarity.
>
> > `Q2.` The bound in Theorem 3.1, the first bound of Theorem 4.1, and the first bound of Theorem 4.2 seem to depend exponentially on $ d_x $ or $ d_0 $, which seems to suffer from CoD. Is the bound tight here?
>
> Thanks for your question. Here are some clarifications.
>
> Yes, you are correct. The approximation rates scale poorly under the weak Holder assumption due to the exponential dependence.
>
> However, we remark that this aligns with practice.
> * In fact, **it is well-known that DiT cannot handle raw high-dimensional data well [Bao23, Peebles22].** Practitioners always use the latent conditional DiT and control $d_0$ to keep the input dimension manageable.
>
> * From our results, it is clear that **latent design bring exponential improvement w.r.t. $d_x/d_0$**. In this way, they address the CoD caused by the large $d_x$ [Bao23, Peebles22]. Additionally, our results also indicate that latent conditional DiTs are capable of overcoming the challenges associated with the curse of dimensionality.
>
> [Bao23] All are worth words: A Vit Backbone for Diffusion Models, CVPR 2023.
>
> [Peebles22] Scalable Diffusion Models with Transformers, ICCV 2022.
>
> > `Q4.` In the first bound of Theorem 3.3 and Theorem 3.4, the bound seems to be decaying at a rate of $ \exp(-d_x) $, is this a typo?
>
> Thanks for pointing this out. **It is indeed a typo.** We have fixed it.
>
> **In the revised version, the first bounds in Theorems 3.3 and 3.4 now show an increasing trend at a rate of $\exp(+d_x)$ from the $\log n$ dependence.**
>
> The typo arises from the change of dominance between the $\log n$ and $n$ terms in our analysis.
>
> In response, **in Appendix K.3 of the latest revision , we have identified a sharp transition of the dominance between the $N$ and $\log N$ terms at $d_x=\Omega(\frac{\log N}{\log \log N})$.** This leads to two distinct asymptotic behaviors of the estimation results.
>
> Consequently, we provide separate analyses for the 2 cases:
>
> * High dim region ($d_x=\Omega (\frac{\log N}{\log \log N})$) [`Thm 3.3, 3.4` of the latest revision]
> * Low dim region ($d_x=o (\frac{\log N}{\log \log N})$) [`Corollary 3.3.1, 3.4.1` of the latest revision]
>
> Since the high-dim region is more practical, we modified Thm 3.1, 3.2, 3.3, 4.1, 4.2, 4.3, and Table 1 with corresponding results and left the low dim results to corollary 3.3.1 and 3.4.1.
>
> We remark that the poor asymptotics of $d_x$ in the previous submission was due to results in the low-dimensional region (where we shouldn’t consider asymptotics of $d_x$ at all).

---

> ### Author Response · Authors · 2024-11-18
> **Response 3**
>
> > `Q5.` The second bound of Theorem 3.3 seems not to be sensitive to $ d_y $. The second bound of Theorem 3.4 depends on $ d_x $ and $ d_y $ only through $ d_x + d_y $ and is not sensitive to it. Similarly, the second bound of Theorem 4.2 only depends on $ d_0 $ and $ d_y $ through $ d_0 + d_y $. Is there any intuition behind this?
>
> Thanks for your question. Here are some clarifications.
>
> This dependence aligns with the intuition related to the model inputs of the conditional DiT, as illustrated in Figure 1. Specifically, we follow the ``in-context conditioning” conditional DiT network [Peebles22]. The network concatenates the embeddings of the input data $x$, input label $y$, and the timestamp $t$. The model then processes the concatenated embedding. Therefore, the network establishes the dependence on $d_x$ and $d_y$ through $d_x+d_y$. The intuition for $d_0+d_y$ is similar.
>
> [Peebles22] Scalable Diffusion Models with Transformers, ICCV 2022.
>
> ---
>
> We hope that the revisions and clarifications address the reviewer's concerns. We look forward to further feedback and discussion.

---

> ### Author Response · Authors · 2024-11-24
> **A Gentle Reminder**
>
> Dear Reviewer,
>
> As the rebuttal phase nears its end, we wanted to gently remind you of our responses.
>
> We have addressed all your comments and questions with the utmost care and hope our efforts meet your expectations. If you have any remaining concerns, we would be happy to discuss them further. Otherwise, we kindly invite you to consider raising your score if you find our updates satisfactory.
>
> We deeply appreciate the time and effort you have dedicated to reviewing our paper, especially given its substantial length and complexity! Regardless of the final outcome, we want to express our heartfelt gratitude for your thoughtful feedback and contributions to improving our work.
>
> Thank you!

---

### Author Response · Authors · 2024-11-18
**Global Response**

Dear Reviewers,

Thank you for your insightful feedback and the time invested in reviewing our work. We have addressed your comments and question with highest care and the updated manuscript.

Following your suggestions, we have improved the paper's readability through extensive proofreading, detailed proof polishing, and typo corrections. These revisions include additional explanations, paragraphs and sections to help readers understand our techniques and results.

Notably, we add experiments to confirm our theoretical findings. Specifically, we verify the relation in our theoretical results in terms of input data dimension $d_x$, norm bounds of parameters in DiT, and testing loss (score estimation error). Additionally, we clarify the analysis of the approximation rate and include the results under the high-dimensional and low-, mild-dimensional regions.

---

## **Major Revisions**

* **Newly Added `Appendix D`:** Numerical results to validate our theoretical findings. [`WTrU`, `mSoC`]

* **Newly Added  `Appendix K.3-7`:** High-dimensional and low-, mild-dimensional analysis with approximation and estimation [`1NAW`]

* **Updated `Thm3.1, 3.2, 3.3, 3.4`:** Update rates without $e^{d_x}$ dependence and vacuous asymptotics. See updated Table 1 quoted below for a quick summary.  [`WTrU`, `1NAW`]

## **Minor Revisions**

* Proofreading the manuscript and fixing all identified typos and grammatical errors identified by reviewers and authors. [`WTrU`, `mSoC`, `GBgr`, `1NAW`]
* Restructuring and Improving the readability of all proof. [`mSoC`]

---

We hope these revisions address the reviewers' concerns and improve the overall quality of our paper.

---

### **Updated `Table 1`: Summary of Theoretical Results**

The initial data is $d_x$-dimensional, and the condition is $d_y$-dimensional. For latent DiT, the latent variable is $d_0$-dimensional. $\sigma_{t}^{2} = 1 - e^{-t}$ is the denoising scheduler. The sample size is $n$, and $0 < \epsilon < 1$ represents the score approximation error. While we report asymptotics for large $d_x, d_0$, we reintroduce the $n$ dependence in the estimation results to emphasize sample complexity convergence.

| **Assumption** | **Score Approximation** | **Score Estimation** | **Dist. Estimation (Total Variation Distance)** | **Minimax Optimality** |
|----------------|-------------------------|---------------------|--------------------------------------------------|------------------------|
| Generic Hölder Smooth Data Dist.        | $\mathcal{O}\left((\log(\frac{1}{\epsilon}))^{d_x}/{\sigma_t^4}\right)$ | $n^{-\Theta(1/d_x)} \cdot (\log n)^{\mathcal{O}(d_x)}$ | $n^{-\Theta(1/d_x)} \cdot (\log n)^{\mathcal{O}(d_x)}$ | ✘ |
| Stronger Hölder Smooth Data Dist.       | $(\log(\frac{1}{\epsilon}))^{\mathcal{O}(1)}/{\sigma_t^2}$ | $n^{-\Theta(1/d_x^2)} \cdot (\log n)^{\mathcal{O}(1)}$ | $n^{-\Theta(1/d_x)} \cdot (\log n)^{\mathcal{O}(1)}$ | ✔ |
| Latent Subspace + Generic Hölder Smooth Data Dist. | $\mathcal{O}\left((\log(\frac{1}{\epsilon}))^{d_0}/{\sigma_t^4}\right)$ | $n^{-\Theta(1/d_0)} \cdot (\log n)^{\mathcal{O}(d_0)}$ | $n^{-\Theta(1/d_0)} \cdot (\log n)^{\mathcal{O}(d_0)}$ | ✘ |
| Latent Subspace + Stronger Hölder Smooth Data Dist. | $(\log(\frac{1}{\epsilon}))^{\mathcal{O}(1)}/{\sigma_t^2}$ | $n^{-\Theta(1/d_0^2)} \cdot (\log n)^{\mathcal{O}(1)}$ | $n^{-\Theta(1/d_0)} \cdot (\log n)^{\mathcal{O}(1)}$ | ✔ |


---

---

> ### Author Response · Authors · 2024-11-18
> **Summary of Numerical Results**
>
> **We have included empirical results in Appendix D of the revised manuscript.**
>
> Specifically, we train a conditional diffusion transformer model on the CIFAR10 to verify three points:
>
> - **(E1)** the influence of input data dimension $d_x$ on the testing loss in Theorem 3.3.
> - **(E2)** the influence of input data dimension $d_x$ on the parameter norm bounds in Theorem 3.1.
> - **(E3)** the influence of backward timestamp $t_0$ on the testing loss in Theorem 3.3.
>
> Our numerical results align with theory. Please refer to the empirical results in Appendix D of the latest revision for details.
>
> We also quote the results in below for your reference:
>
> ---
>
> ### **`Table 3` (E1) & (E2): Influence of Input Data Dimension $d_x$ on the Testing Loss and Parameter Norm Bounds at Backward Timestamp $t_0 = 5$**
>
> The testing loss and parameter norm bounds ($\|W_O\|$ and $\|W_V\|$) increase with an increasing $d_x$. These results are consistent with the findings in **Theorems 3.1 and 3.3**.
>
> | **Input Data Dim.** $d_x$           | $32 \cdot 32 $     | $48 \cdot 48 $     | $64 \cdot 64 $     | $80 \cdot 80$     |
> |-------------------------------------|---------------------------|---------------------------|---------------------------|---------------------------|
> | **Testing Loss**                    | 0.9321                    | 0.9356                    | 0.9364                    | 0.9476                    |
> | **$\|W_O\|_{2,\infty}$**            | 1.6074                    | 1.6332                    | 1.6789                    | 1.6886                    |
> | **$\|W_V\|_{2,\infty}$**            | 2.1513                    | 2.1767                    | 2.1858                    | 2.1994                    |
>
> ---
>
> ### **`Table 4` (E3) Influence of Backward Timestamp $t_0$ on the Testing Loss**
>
> The testing loss increases with increasing $t_0$. This is consistent with the result in **Theorem 3.3**.
>
> | **Testing Loss**       | $t_0 = 5$ | $t_0 = 4$ | $t_0 = 3$ | $t_0 = 2$ | $t_0 = 1$ |
> |------------------------|-----------|-----------|-----------|-----------|-----------|
> | $32 \cdot 32 = 1,024$  | 0.9321    | 0.9329    | 0.9335    | 0.9350    | 0.9361    |
> | $48 \cdot 48 = 2,304$  | 0.9356    | 0.9357    | 0.9360    | 0.9363    | 0.9367    |
>
> ---

---

### Meta-Review · Area_Chair_7Cma · 2024-12-19

**Metareview:**

This paper studies the statistical rates of approximation and estimation of conditional diffusion transformers. It's a relatively novel setting for the theoretical analyses on diffusion models. During the rebuttal phase, the authors have addressed most of the questions raised by the reviewers. Although there are still questions about the assumptions and the curse of dimensionality, I feel that the novelty of the theoretical results are enough to warrant a publication at ICLR.

**Additional Comments On Reviewer Discussion:**

During the rebuttal phase, the authors have addressed most of the questions raised by the reviewers. There is an outstanding one about the curse of dimensionality. But I feel that the novelty of the theoretical results are enough to warrant a publication at ICLR.

---

### Decision · Program_Chairs · 2025-01-22

Accept (Poster)